# Diels–Alder Adducts of Morphinan-6,8-Dienes and Their Transformations [note 1]

**DOI:** 10.3390/molecules27092863

**Published:** 2022-04-30

**Authors:** János Marton, Anikó Fekete, Paul Cumming, Sándor Hosztafi, Pál Mikecz, Gjermund Henriksen

**Affiliations:** 1ABX Advanced Biochemical Compounds Biomedizinische Forschungsreagenzien GmbH, Heinrich-Glaeser-Strasse 10-14, D-01454 Radeberg, Germany; 2Department of Medical Imaging, Division of Nuclear Medicine and Translational Imaging, Faculty of Medicine, University of Debrecen, Nagyerdei krt. 98, H-4032 Debrecen, Hungary; fekete.aniko@science.unideb.hu (A.F.); mikecz.pal@med.unideb.hu (P.M.); 3Department of Nuclear Medicine, Bern University Hospital, Freiburgstraße 18, 3010 Bern, Switzerland; paul.cumming@insel.ch; 4School of Psychology and Counselling, Queensland University of Technology, Brisbane, QLD 4059, Australia; 5Institute of Pharmaceutical Chemistry, Semmelweis Medical University, Högyes Endre utca 9, H-1092 Budapest, Hungary; hosztafi.sandor@pharma.semmelweis-univ.hu; 6Norwegian Medical Cyclotron Centre Ltd., Sognsvannsveien 20, N-0372 Oslo, Norway; 7Institute of Basic Medical Sciences, University of Oslo, N-0317 Oslo, Norway; 8Institute of Physics, University of Oslo, Sem Sælands vei 24, N-0371 Oslo, Norway

**Keywords:** morphine alkaloids, opioid receptors, morphinan-6,8-dienes, Diels–Alder reaction, Grignard addition, *O*-demethylation, radiolabeling, positron emission tomography

## Abstract

6,14-ethenomorphinans are semisynthetic opiate derivatives containing an ethylene bridge between positions 6 and 14 in ring-C of the morphine skeleton that imparts a rigid molecular structure. These compounds represent an important family of opioid receptor ligands in which the 6,14-etheno bridged structural motif originates from a [4 + 2] cycloaddition of morphinan-6,8-dienes with dienophiles. Certain 6,14-ethenomorphinans having extremely high affinity for opioid receptors are often non-selective for opioid receptor subtypes, but this view is now undergoing some revision. The agonist 20*R*-etorphine and 20*R*-dihydroetorphine are several thousand times more potent analgesics than morphine, whereas diprenorphine is a high-affinity non-selective antagonist. The partial agonist buprenorphine is used as an analgesic in the management of post-operative pain or in substitution therapy for opiate addiction, sometimes in combination with the non-selective antagonist naloxone. In the context of the current opioid crisis, we communicated a summary of several decades of work toward generating opioid analgesics with lesser side effects or abuse potential. Our summary placed a focus on Diels–Alder reactions of morphinan-6,8-dienes and subsequent transformations of the cycloadducts. We also summarized the pharmacological aspects of radiolabeled 6,14-ethenomorphinans used in molecular imaging of opioid receptors.

## 1. Introduction

Opiates are the most widely investigated class of natural products [1]. The use of opium has been attested since antiquity and morphine (**1**, Figure 1) [2,3] has been essential in the treatment of pain since its first isolation from opium in 1804 by Friedrich Wilhelm Adam Sertürner, a Paderborn apothecary and pioneer in alkaloid chemistry. Morphine (**1**) and its 3-O-methyl derivative codeine (**2**) inhibit the cough reflex at several sites on the brainstem [4], with additional effects against construction of the upper airway [5], and opioids are widely used as cough suppressants and antitussive agents. Both morphine (**1**) and codeine (**2**) have been identified as endogenous compounds in human cerebrospinal fluid [6] and may play a physiological role in some aspect of pain perception [7] by acting at several subtypes of opioid receptors in the brain [8]. However, Papaver somniferum is an abundant natural source of alkaloids with the morphinan skeleton [9,10], including neopine (**3**), thebaine (**4**), and oripavine (**5**), all of which are useful starting materials for pharmaceutically important derivatives (e.g., from **3**: B/C-trans-morphinans, and from **4** and **5**: naltrexone, nalbuphine, naloxone, etorphine, buprenorphine, and diprenorphine).

The reaction of thebaine (**4**) with maleic anhydride was described [11,12] a few years after Gulland and Robinson’s [13] 1925 publication of the correct structure of thebaine (**4**). This coincided closely with the original report of Diels and Alder [14] on the reaction of conjugated dienes (such as cyclopentadiene or butadiene) with unsaturated anhydrides or acids (e.g., maleic anhydride or maleic acid) to form a new type of cycloadducts. The first compounds belonging to the 6,14-ethenomorphinan series were soon reported by German authors. In 1938, Sandermann [11] achieved the synthesis of the so-called “Diën-Anlagerungsverbindungen” of thebains with *p*-benzoquinone, 1,4-naphthoquinone, and maleic anhydride. For the thebaine-*p*-benzoquinone adduct, Sandermann managed to provide its correct structural planar projection. In the same year, Schöpf, Gottberg, and Petri [12] reported their extended investigations in the field of thebaine addition products, also with maleic anhydride and *p*-benzoquinone. At the time, they affirmed that their investigations had been performed seven years before the publication. Today’s reader may note with astonishment that the authors were able to solve the correct stereo-structure of the four theoretically possible thebaine-maleic anhydride cycloadducts without any help of computational chemistry, modelling, or X-ray crystallography. Shortly after World War II, the preparation of the thebaine acrolein adduct was reported by Kanevskaya and Mitryagina [15]. The Diels–Alder adducts of thebaine returned to the spotlight in the mid-1950s, when Bentley and Thomas [16] published on the dihydro derivative of the thebaine-*p*-benzoquinone adduct, showing it to be practically equipotent with the non-morphinan pethidine as an analgesic in rodents. In the early 1960s, Bentley and Hardy proceeded to publish results on orvinols with extremely high analgesic potency “to the unprecedented level of about 10,000 times that of morphine” [17], thus potentially obtaining pain relief at microgram doses in humans. 6,14-ethenomorphinans are often referred to as Bentley compounds in recognition of that group’s efforts and achievements in the field [18,19].

Opioid receptors (ORs) belong to the G-protein-coupled receptor family (GCPRs) [20,21,22] and are widely distributed in the central nervous system as well as in the periphery. As noted above [8], endogenous opioid ligands and exogenous compounds act in the nervous system via their binding to one or more of several classes of opioid receptors, often designated as μ-OR, δ-OR, κ-OR, and the nociceptin opioid receptor (NOR). These receptor subtypes have distinct distributions in brain and varying degrees of selectivity for opioid ligands. In 2012, the crystal structure of opioid receptor subtypes was determined by X-ray structure analysis in the presence of high-affinity ligands. For the case of μ-OR, Manglik et al. [23] used the irreversible μ-OR antagonist β-funaltrexamine (β-FNA). For δ-OR, Granier et al. [24] used the selective antagonist naltrindole (NTI). For κ-OR, Wu et al. [25] complexed human κ-OR with the selective antagonist JDTic. Finally, for NOP, Thompson et al. [26] used a complex with the peptide mimetic antagonist Banyu compound-24 (C-24). Taken together, this series of investigations has enabled far-reaching molecular docking studies that are of fundamental importance for structure-based drug design.

Molecular imaging of receptors is enabled by incorporation of radionuclides into the structure of molecules with lesser or greater degree of selectivity for their targets in the brain. Most commonly, opioid receptor ligands are prepared with cyclotron-generated carbon-11 (half-life of 20.34 min) or fluorine-18 (half-life of 109.8 min) for use in conjunction with positron emission tomography (PET). In some cases, radioligands containing gamma emitters such as iodine-123 (half-life of 13.22 h) are used in conjunction with single-photon computed tomography (SPECT), which presents logistic advantages at the expense of lower special resolution and imperfect quantitation. To date, there have been several hundred reports on the molecular imaging of opioid receptors in the human brain, most commonly with the non-selective antagonist ligand [^11^C]DPN (K_i_ (μ) = 0.07 nM, K_i_ (δ) = 0.23 nM, K_i_ (κ) = 0.02 nM)/[^18^F]FE-DPN (K_i_ (μ) = 0.24 nM, K_i_ (δ) = 8.00 nM, K_i_ (κ) = 0.20 nM), the μ,κ-OR-preferring antagonist [^18^F]FcyF, and the extremely high affinity μ-OR selective agonist carfentanil (K_i_ (μ) = 0.024 nM, K_i_ (δ) = 3.28 nM, K_i_ (κ) = 43.1 nM) [8,27,28,29]. While antagonist ligands at tracer doses are without pharmacological effect, the safe use of carfentanil PET calls for attention in maintaining ultra-high specific activity. These PET tracers have found use in investigating a wide range of topics including chronic pain syndromes [30], monitoring of methadone occupancy in patients treated for heroin addiction [31], the physiological basis of obesity [32,33], compulsive behaviors such as gambling and binge eating [34] and alcohol addiction [35], opioid modulation of cognitive dysfunction in the contexts of depression [36], and physiological correlates of high-intensity exercise [37]. While impressive in its scope and implications for human disease, this body of research is outperformed at least ten-fold by corresponding studies of brain dopamine; this might be remedied in part by the development of useful SPECT ligands for opioid receptor imaging, which would increase the accessibility of this line of medical research.

In this summary, we present an overview of the organic synthesis of 6,14-ethenomorphinans as well as the chemistry of their labelled derivatives.

## 2. Synthesis of 6,14-Ethenomorphinans

More than eight decades after Shöpf’s [12] first attempt to synthesize diene addition derivatives of thebaine (**4**), the investigation of 6,14-ethenomorphinans has remained one of the most promising fields of research on morphine alkaloids. In addition to the parent poppy alkaloids with conjugated diene system in ring-C, thebaine (**4**) [38], oripavine (**5**) [39], and various substituted morphinan-6,8-dienes have been synthesized and reacted with a remarkably large variety of dienophiles. The structures of selected morphinan-6,8-dienes are depicted in Figure 2 (and Table 1), together with the corresponding references [38,39,40,41,42,43,44,45,46,47,48,49,50,51,52,53,54,55,56,57,58,59,60,61,62,63,64,65,66,67,68] that first reported their synthesis.

### 2.1. Diels–Alder Reactions

Thebaine (**4**) and oripavine (**5**) molecules comprise an enol ether and a 6,8-diene function (1-methoxy-1,3-cyclohexadiene substructural unit) which, as an electron-rich diene, readily reacts with electron-deficient dienophiles [11,12] in an *endo*-HOMO(diene)–LUMO(dienophile) controlled [69,70] Diels–Alder reaction. Theoretically, the 6,8-diene system of the T-shaped thebaine **4**) or oripavine (**5**) molecule can be attacked from both faces (Figure 3), resulting in distinct isomers.

In an extreme example, the Diels–Alder reaction of morphinan-6,8-dienes with CH_2_=CH-X-type unsymmetrical monoacceptor-substituted dienophiles can afford eight isomers (Figure 4), i.e., four isomers from β-face addition (7α-X-6α,14α-etheno-, 7β-X-6α,14 α-etheno-, 8α-X-6α,14α-etheno-, and 8β-X-6α,14α-etheno-) and four from α-face addition (7αX-6β,14β-etheno-, 7βX-6β,14β-etheno-, 8αX-6β,14β-etheno-, and 8βX-6β,14β-etheno-) [71]. These eight products are potentially all distinct with respect to potency, selectivity, and agonist or antagonist properties.

The potential transitional states of thebaine (**4**) and/or oripavine (**5**) are depicted in Figure 3, and the possible stereoisomers are in Figure 4. Due to electronic and steric reasons, the β-face-*endo* approach is clearly favored in the case of **4** and **5**. The approach of the dienophile to the conjugated diene system of thebaine (**4**) in ring-C is hindered by the presence of the piperidine ring (D-ring) from the β-face (upper face) and the aromatic ring (A-ring) from the α-face (lower face).

#### 2.1.1. Nomenclature and Stereochemistry

In the final two decades of the 20th century, the Maat research group in Delft [54,72,73] has extensively studied the Diels–Alder reactions of miscellaneous morphinan-6,8-dienes with a great variety of dienophiles. Some of the afforded adducts proved to be highly active opioid receptor ligands. In the course of this work, the Maat group made great efforts to resolve inconsistencies and contradictions in the nomenclature of Bentley compounds, and to develop a new and more rational nomenclature. Unfortunately, the use of this rational nomenclature system is not widespread, leading to some confusion in the face of the great diversity of adducts.

In particular, there remains considerable confusion about the correct nomenclature of the Diels–Alder adducts of morphinandienes and their derivatives. Initially, the synthesized cycloadducts were simply named after their native components as for example “morphinandiene-dienophile-adduct” (e.g., *thebaine-chinone-adduct* [11], or more directly, *thebaine maleic anhydride* [12]). Bentley and associates [74] applied both trivial names and a systematic nomenclature. Due to their relative simplicity, many of the trivial names introduced by Bentley are still used today. For example, consider *thevinone* (**16a**) (**the**baine methyl **vin**yl ket**one**) and its secondary or tertiary alcohol derivatives, which are called *thevinols*, whereas those with a hydroxyl group in position three are known as *orvinols*, such as nepenthone (**16b**) and nepenthol. The Bentley group also introduced methods for systematic naming of the resultant heterocycles: 6,14-*endo*-etheno-6,7,8,14-tetrahydrothebaine (Figure 5A) or 6,14-*endo*-etheno-6,7,8,14-tetrahydrooripavine. In the case of α-face addition, the new substituent is oriented at the opposite side of the C-ring. 

Accordingly, such adducts were named 6,14-*exo*-etheno-6,7,8,14-tetrahydro-thebaines (Figure 5B). In cases with saturation of the 6,14-endo/exo-*etheno* bridge, the new derivatives were called 6,14-endo/exo-*ethano* compounds (Figure 5C). However, in the 1970s, the Chemical Abstract nomenclature introduced the name 6,14-ethenomorphinan (Figure 5A, β-face adduct). The descriptor “6β,14β” was added for differentiation of the α-face adducts, i.e., 6β,14β-ethenomorphinan (Figure 5B). According to the Chemical Abstracts nomenclature, the derivatives with a saturated 6,14-bridge (Figure 5C) have been designated as 18,19-dihydro-6,14-ethenomorphinans. Meanwhile, the Maat group at Delft continued to elaborate an arguably more reasonable and consequent nomenclature system [75].

By application of the Maat/Delft nomenclature [75], the 6,14-etheno bridge was added to the native structure of morphinan (B/C-*cis* morphinan, β-face addition), as well isomorphinan (B/C-*trans* morphinan, α-face addition), giving adducts which correctly have been called 6,14-ethenoisomorphinans (Figure 5A) and 6,14-ethenomorphinans (Figure 5B). Although application of prefixes (*endo*/*exo*) and descriptions (6α,14α/6β,14β) are redundant in this nomenclature system, widespread traditional use of clumsier naming systems continues to this day.

#### 2.1.2. Reactions of Δ^6,8^-morphinandienes

##### Thebaine

The cycloaddition of monosubstituted ethylene analogues to thebaine (**4**) occurs in a regio- and stereoselective manner on the β-face of the molecule, resulting exclusively in C-7 substituted derivatives [18,19,74]. The regioselectivity can be explained by the polarizing effect of the 6-OCH_3_ group on the diene system [48,49,74]. Thebaine (**4**) readily reacts with α,β-unsaturated ketones (alkyl- or aryl vinyl ketones), acrylic esters, and acrylonitrile (Figure 6).

Diels–Alder addition of methyl vinyl ketone to thebaine (**4**) resulted in thevinone (**16a**, 93%) as the main product and β-thevinone (**17a**, 0.5%) as a minor by-product [74]. The reaction of **4** with ethyl vinyl ketone (3.7 equiv., neat, 100 °C, 4 h) [74] gave the 7α-propionyl derivative (**16d**) with a yield of 80%. After applying 3.7 equiv. ethyl vinyl ketone as dienophile [76] in toluene (reflux, 6 h), the 7α-propionyl derivative (**16d**, 90%) and the 7β-isomer (**17d**, 1.5%) were isolated. When reacting thebaine (**4**) with acrolein (14 equiv., neat, reflux, 1 h, or 1.8 equiv., in benzene, 65–70 °C, 4 h), the reaction yielded only the 7α-formyl derivative (**16c**, >85% and 57% respectively). Using equimolar amounts of phenyl vinyl ketone as dienophile (benzene, reflux, 2 h), there was exclusive formation of the 7α-benzoyl derivative (**16b**, R = COPh, nepenthone, 70%) [77]. The regio- and stereoselective addition of ethyl acrylate (4.88 equiv. reflux, 6 h) to **4** resulted in the 7α- (**16f**) and 7β-esters (**17f**) in a ratio of 94:6. The cycloaddition of acrylonitrile to thebaine (**4**) yielded a 1:1 mixture of the 7α- (**16g**, R = CN) and 7β-nitriles (**17g**, R = CN). Testing of the ketones (**16**) as a subcutaneously administered analgesic [74] indicated greater potency than morphine (**1**) in the rat-tail pressure test.

##### *N*^17^-Substituted Northebaine Derivatives

Reacting thebaine **(4**) directly with nitroethene (CH_2_=CH-NO_2_) under various conditions (at temperatures between −78 °C and 110 °C) and in several solvents (benzene, CH_2_Cl_2_, boiling toluene) did not yield a Diels–Alder adduct because the dienophile polymerized. This polymerization was favored by the basic property of the thebaine piperidine nitrogen *N*^17^. To diminish the basicity of the alkaloid, a formyl group was introduced in position *N*^17^. When *N*^17^-formyl-northebaine (**9a**) was heated with the same dienophile (in benzene, boiling for 19 h), the cycloaddition took place successfully [40].

In this reaction, only one of the eight possible Diels–Alder stereoisomers (Figure 3 and Figure 4) was formed, namely the product of the β-face addition, 7α-nitro adduct (**18**, Figure 7) in almost quantitative yield. Structure elucidation performed by ^1^H- and ^13^C-NMR spectroscopy identified the product as 7α-nitro compound (**18**). Due to the hindered rotation of the *N*-formyl group, it occurred as a 1:1 mixture of E- and Z-isomers at 20 °C in the NMR solvent (CDCl_3_).

##### Morphinandienes with An Opened E-Ring

α-Face addition was observed in case of morphinan-6,8-dienes with an opened E-ring, for example, β-dihydrothebaine (**10a**) [48] and 6-demethoxy-β-dihydrothebaine (**10d**) derivatives [49]). Ghosh et al. [48] reported that the Diels–Alder reaction of β-dihydrothebaine-4-phenyl ether (**10c**) with methyl vinyl ketone resulted in an α-face addition product (6,14-exo-ethenomorphinan or 6β,14β-ethenomorphinan). On the other hand, Linders et al. [49] found that the cycloaddition of dienophiles to morphinan-6,8-dienes with an opened E-ring occurred on the α-face of the diene. For instance, 6-demethoxy-β-dihydrothebaine (**10d**, Figure 8), bearing a hydrogen atom in position six instead of OCH_3_ and possessing an “opened E-ring” (lacking the 4,5-epoxy bridge), gave upon Diels–Alder reaction with methyl vinyl ketone the 7β-acetyl-6β,14β (**19**) and 8β-acetyl-6β,14β (**20**) adducts (α-Face approach) in a ratio of 3:2, according to HPLC analysis of the product mixture.

Cycloaddition of methyl vinyl ketone as well as nitroethene to 4-*O*-acetyl-*N*^17^-formyl-6-demethoxy-β-dihydrothebaine (**10e**) was investigated by Linders et al. in 1989 [50]. In the first case, the 7β-acetyl-6β,14β-isomer (**21**, 17%, α-face approach, Figure 9) was the main isolated product and 8β-acetyl-6β,14β-isomer (**22**, 3%, α-face approach) was a minor by-product. On the other hand, in the reaction of **10e** with nitroethene (benzene, reflux 50 h), the 8β-nitro-6β,14β-adduct (**23**, 27%, α-face approach) was formed as the major product.

##### 6-Demethoxythebaine

Knipmeyer and Rapoport [51] demonstrated the importance of the 6-methoxy group on ring-C for regioselectivity of the Diels–Alder reactions of 6,8-morphinandienes. In the reaction of 6-deoxythebaine (only methyl group in position six, **11a**) with methyl vinyl ketone, three regio- and diastereomers were formed.

Linders et al. [78] studied the influence of *N*^17^ substituents on the diastereoselectivity of the Diels–Alder addition, comparing results of the cycloaddition of *N*^17^-methyl- and *N*^17^-formyl-morphinandienes (Figure 10). The application of microwave heating was at that time somewhat novel, and reportedly improved the efficacy and conversion rate of Diels–Alder reactions. In the reaction of 6-demethoxythebaine (**11b**) with methyl vinyl ketone, the main isomer formed was the 7α-acetyl-6α,14α-isomer (**24**, β-face approach). The secondary isolated product was the 8α-acetyl-6α,14α-isomer (**25**, β-face approach, 10%), and traces of the 8β-acetyl-6β,14β-adduct (**26**, α-face approach, 0.4%) were also isolated. There was a report of the successful application of a Lewis acid catalyst (AlCl_3_, at room temperature) in the addition reaction of methyl vinyl ketone with 6-demethoxythebaine (**11b**) [78]. Lewis acid catalysis of the Diels–Alder reactions of thebaine (**4**) with dienophiles had long been considered unfeasible because thebaine (**4**) itself undergoes rearrangement reactions, which can result in degradation products such as thebenine and neodihydrothebaine. Nonetheless, lithium tetrafluoroborate-catalyzed Diels–Alder reactions of thebaine (**4**) were performed later by Coop et al. [44,79]. Cycloaddition of methyl vinyl ketone to *N*^17^-formyl-6-demethoxy-northebaine (**11d**, Figure 10) resulted in two products in a 9:1 ratio—the major product 7α-acetyl-6α,14α-adduct (β-face approach, **27**) isolated in 78% yield.

It is noteworthy that the Diels–Alder reaction of *N*^17^-formyl-6-demethoxynorthebaine (**11d**) and nitroethene (reflux in benzene, for 60 h, 70% conversion) resulted in a complex product mixture [54,80] (Figure 11); three cycloadducts (**28**, **29**, **30**) were isolated by column chromatography. Two isolated products were results of β-face addition, namely 8α-nitro-6α,14α (**28**, 50%) and 7α-nitro-6α,14α-adducts (**29**, 10%). Removal of the *C*^6^-methoxy group allowed the α-face approach, and the formation of the 8β-nitro-6β,14β-isomer (**30**, 30%) was then observed.

##### 6-Halo-, 7-Halomorphinandienes and Pseudohalo Analogues of Thebaine

In 1995, Berényi et al. [81] investigated the cycloaddition reactions of 6-halomorphinandienes (6-fluoro-6-demethoxythebaine (**11e**), 6-chloro-6-demethoxy-thebaine (**11f**), and 6-bromo-6-demethoxythebaine (**11g**), Figure 12) with methyl vinyl ketone. The resulting derivatives of the β-face addition (**31** (85%), **32**, (96%), **33** (88%), respectively) were converted by subsequent reactions to the corresponding 6-halo-substituted thevinols and orvinols.

The pharmacological profiles of the 6-halogenated 20-methyl-thevinols and 20-methylorvinols as opioid agonists were thoroughly studied [81] using the guinea pig isolated ileum (GPI) bioassay in vitro and the mouse hot plate test in vivo. In GPI assay, 6-halogenated-20-methylorvinols displayed 17–30 times higher activity in comparison to normorphine. In the mouse hot plate tests, 6-fluoro-20-methylorvinol was the most potent analgesic from this series; these studies did not confirm the subtype specificity of the receptor ligands. The same research group reported the cycloaddition of 4-phenyl-4*H*-1,2,4-triazoline-3,5-dione (PTAD) to 6-thiocyanato-6-demethoxythebaine (**11h**), 6-isothiocyanato-6-demethoxythebaine (**11i**), 6-azido-6-demethoxythebaine (**11j**) [58], and 7-halomorphinandienes (**11k**–**l**) [82] (for details, see Section 2.4.2). β-Face addition was observed in each case, and the adducts (Figure 12, **34a**–**c**, **35a**–**b**) were isolated with yields in the range of 45–80%.

##### 5β-Alkylthebaines

In 1990, Woudenberg et al. [83] studied the steric impact of a methyl group in position five (as blocking substituent on the β-side of the diene) on cycloadditions. Reactions of 5β-methylthebaine (**12a**, Figure 13) with various dienophiles (ethyl acrylate, methyl vinyl ketone, and *p*-benzoquinone) were investigated.

Products from the β-face approach to the diene system were exclusively obtained in all cases (**36**, **37**, **38**). Ergo, the 5-methyl-substitution did not alter the course or orientation of the Diels–Alder reaction. However, the steric hindrance caused by the introduction of the 5-substituent led to a longer reaction time compared to the native diene thebaine (**4**).

With this in mind, Woudenberg et al. [64] synthesized 5β-methyl-6-demethoxythebaine (**13a**) via two separate reaction routes starting from 6-demethoxythebaine (**11b**) and 5β-methylthebaine (**12a**). To explore the course of the Diels–Alder reaction of morphinane-6,8-dienes, the reaction of **13a** was investigated with various dienophiles (Figure 14). 

Boiling **13a** under reflux for five days with asymmetric dienophiles resulted in the Diels–Alder reactions with 40% conversion for methyl vinyl ketone and 50% conversion for ethyl acrylate. In both cases, two cycloadducts were formed in a ratio of 1:1. Besides the 7α-substituted-6α,14α-derivatives (acetyl, ethoxycarbonyl, β-face approach, **39**, **41**), the corresponding 8β-substituted-6β,14β compounds (α-face approach, **40**, **42**) were also isolated in equal yields. The cycloaddition of **43** with a symmetrically substituted dienophile (maleic anhydride (MA), three equiv., reflux in toluene for ten days) also resulted in two products (**43** and **44**) in a ratio of 7:3 in favor of the β-face addition. The possibility of a successful α-face approach in the cases presented above has been explained by the combined effects of steric hindrance and electronic influences due to the 5β-methyl substituent and the absence of the 6-OCH_3_ group. In the case of 6-methoxy-morphinan-6,8-dienes (thebaine (**4**) and 5β-methylthebaine (**12a**)), they reacted with maleic anhydride, giving exclusive β-face addition.

In 1992, Woudenberg et al. [61] synthesized 5β- and 10α-alkyl-substituted-thebaine derivatives (**12b**, **12f**, **12e**, Figure 15) in an effort to understand the factors influencing the face selectivity of the Diels–Alder reactions of morphinane-6,8-dienes. 5β-Ethylthebaine (**12b**), 5β,10α-diethylthebaine (**12f**), and 10α-ethyl-5β-methylthebaine (**12e**) were reacted with ethyl acrylate (reflux, oil bath, 103 °C, 2–3 weeks, Figure 15). In all three cases, the cycloaddition occurred with a low conversion, and exclusively on the β-face of the diene, resulting in compounds **45a**–**c**. The much longer reaction time required for the Diels–Alder reaction of 5β-ethylthebaine (**12b**) compared to that of thebaine (**4**) was attributed to the steric influence of the 5-beta substituent, but the authors concluded that the 5β-substituent could not completely block access to the β-side of thebaine (**4**). In an extension of this work [61], 5β-ethyl-20-methylorvinol (**47a**) and 5β-metyl-10α-ethyl-20-methylorvinol (**47c**) were synthesized by Grignard addition with methylmagnesium bromide and following 3-*O*-demethylation (**45a**, **45c**).

In 1993, Woudenberg et al. [84] described the synthesis of morphinan derivatives doubly bridged at ring-C (Figure 16). In brief, thebaine-5β-methanol (**12d**) was first prepared from thebaine (**4**).

The anion, formed by deprotonation of thebaine (**4**) with butyl lithium, was treated with methyl chloroformate to yield methyl thebaine-5β-carboxylate (**48**). Reduction of the carboxylic acid methyl ester (**48**) with LiAlH_4_ gave thebaine-5β-methanol (**12d**). The latter compound was esterified with acryloyl chloride to thebaine-5β-methyl acrylate (**49a**), or with methacryloyl chloride to obtain thebaine-5β-methyl methacrylate (**49b**), both in good yields (94–95%). Intramolecular Diels–Alder reaction of the latter compounds was accomplished by heating under reflux (boiling in toluene, three weeks). In ring-C, doubly bridged derivatives were obtained: **50a** with 70% conversion and 48% isolated yield, as well as **50b** with 65% conversion and 57% yield. Here, the cycloaddition was only possible from the β-side of the diene.

Woudenberg et al. [62] performed the Diels–Alder reaction of thebaine-5β-methanol (**12d**, Figure 17) with ethyl acrylate, concluding that introduction of a hydroxymethyl group in the 5β position of thebaine had no influence on the face selectivity of the cycloaddition. Boiling **12d** with ethyl acrylate for one week resulted in exclusively the corresponding 7α-ethoxycarbonyl-6α,14α-etheno-adduct (**51**, 50% conversion, 41% isolated yield, β-face addition). 

Compound **51** was converted to 5β-hydroxymethyl-20-methylorvinol (**53**) via Grignard reaction with methylmagnesium iodide and following 3-*O*-demethylation (KOH, ethylene glycol, microwave heating, 6 h, 25%). In vitro binding studies indicated that introduction of a substituent (methyl, ethyl, or hydroxymethyl) in position 5β in the 20-methylorvinol series diminished the affinity for all the opioid receptor subtypes compared to the native 20-methylorvinol (MORV, proton in position 5β). 5β-Hydroxymethyl-20-methylorvinol (**53**) showed a higher affinity to δ- and κ-opioid receptor subtypes as compared to the 5β-methyl and 5β-ethyl analogues.

In 2005, Coop et al. [85] investigated the reactions of 5β-trimethylsilylthebaine (**12g**, Figure 18) with methyl vinyl ketone as well as with *p*-benzoquinone. Treatment of **12g** with methyl vinyl ketone (45 °C, 7 days) gave only unreacted starting material, indicating that the diene system was hindered both from the β- and α-sides. The Z-enamine (**54**) was formed instead of the desired Diels–Alder cycloadduct during the reaction of 5β-TMS-thebaine (**12g**) with the same dienophile when boiled in toluene. Compound **54** was stabilized through saturation of the Δ^9,10^ double bond by catalytical hydrogenation to **55**. The normal β-face adduct (**56**) was isolated in low yield (10%) when 5-β-TMS-thebaine (**12g**) was reacted with *p*-benzoquinone (45 °C, 7 days). Removal of the trimethylsilyl group with tetrabutylammonium fluoride resulted in the quinol (**57**).

##### Substituted 5β-methyl-6-demethoxythebaine Derivatives

Woudenberg and Maat [65] investigated the electronic influence of the 7-chloro-substituent on the course of the cycloaddition in the presence or absence of the 5β-methyl group in the 6-demethoxy series (Figure 19). Formation of four products was detected (HPLC, NMR) in the Diels–Alder reaction of 7-chloro-6-demethoxythebaine (**11k**) with ethyl acrylate (reflux for four days). The two major cycloadducts were identified as 18-chloro-7α-ethoxycarbonyl-6α,14α-isomer (**58a**, 32%, β-face approach) and 18-chloro-8α-ethoxycarbonyl-6α,14α-isomer (**59a**, 12%, β-face approach), with isolation of lesser amounts of a third product 18-chloro-8β-ethoxycarbonyl-6β,14β-isomer (**60a**, 10%, α-face approach). When 7-chloro-5β-methyl-6-demethoxythebaine (**13b**) was reacted with ethyl acrylate (reflux for two weeks), HPLC/NMR analysis revealed the formation of three products. Two major adducts were isolated by column chromatography: 18-chloro-5β-methyl-7α-ethoxycarbonyl-6α,14α-isomer (**58b**, 4%, β-face approach) and 18-chloro-5β-methyl-8β-ethoxycarbonyl-6β,14β-cycloadduct (**60b**, 32%, α-face approach). When 7-chloro-6-demethoxythebaine (**11k**) was reacted with maleic anhydrid as dienophile (three equiv., boiling in toluene, for four days), there was formation of only one cycloadduct (**61a**, β-face addition). Addition of maleic anhydride to 7-chloro-5β-methyl-6-demethoxythebaine (**13b**, boiling for two weeks in toluene) gave two adducts in a ratio of 4:1 favoring **61b** (β-face approach) over **62b** (α-face approach), as determined by HPLC.

Woudenberg et al. reported in 1993 [66] the cycloaddition of 7-methoxy-5β-methyl-6-demethoxythebaine (**13c**, Figure 20) prepared from 7α-hydroxy-6-*O*-acetyl-5β-methylneopine with ethyl acrylate and maleic anhydride. The cycloaddition of **13c** with ethyl acrylate (58 equiv., reflux, two weeks, 60% conversion) resulted exclusively in the 8β-ethoxycarbonyl-product (**63**) of the α-face addition in 62% yield based on converted material. The regio- and stereochemistry of the ethoxycarbonyl compound (**63**) was elucidated by NMR spectroscopy and single-crystal X-ray structure analysis. Preferred formation of the α-face adduct was explained by the electronic influence of the 7-methoxy group, the steric-hindrance of the 5α-methyl group, and consideration of the asynchronous mechanism of the Diels–Alder reaction of the diene (**13c**) and the asymmetric dienophile.

In the second case, the cycloaddition of three equivalents of the symmetric dienophile maleic anhydride to **13c** (toluene, reflux, four days) resulted in the β-face- (**64**) and α-face-adducts (**65**) in a ratio of 5:4. The formation of both adducts was interpreted by the reduced steric hindrance of the 5β substituent in the “C6-maleic anhydride” transition state (asynchronous mechanism), which allowed approach from the α-face as well as from the β-face.

##### 6-*O*-Alkylmorphinandienes

In 2010, Czakó et al. [67] described the synthesis of a series of 6-substituted-morphinan-6,8-dienes (**14a–d**, Figure 21). Compounds **14a–c** (R = ethoxy, propoxy, and cyclopropylmethyloxy) were prepared from codeinone (**66**), applying the method of Seki (ROH, *p*-toluenesulfonic acid, benzene, reflux, 3 h, azeotropic removal of water) [86]. 6-phenyl-6-demethoxy-thebaine (**14d**) was prepared from 6-bromo-6-demethoxythebaine (**11g**) [87] via the Suzuki–Miyaura cross-coupling. Adducts of the β-face addition (**67a–d**) were isolated from the product mixture of the Diels–Alder reaction of the morphinandienes (**14a–d**) with methyl vinyl ketone. Here, the stereostructure of the cycloadducts was confirmed by 1D NOE experiments. The prepared 6-substituted-6-demethoxythevinones (**67a–d**) were converted to their corresponding thevinol (**68a–d**) derivatives via Grignard addition with methylmagnesium bromide. 6-Substituted-6-demethoxy-orvinols (**69a–d**) were obtained from thevinols (**68a–d**) by 3-*O*-demethylation with KOH in diethylene glycol (210–220 °C). The orvinols (**69a–d**) showed three- to five-fold higher μ-OR affinities compared with the native 20-methylorvinol (MORV) as measured on rat brain membranes.

##### Heteroring-Fused Morphinandienes

The fusion of an aminothiazole scaffold to the morphinan skeleton has been implemented in various ways [88,89,90]. There have been reports of C-ring-fused aminothiazolomorphinans of oxycodone [91], dihydrocodeinone, and dihydromorphinone [92], as well as 2-aminothiazol-annulated morphinans at the 2,3 and 1,2 positions of the A-ring [88,89,90]. Berényi et al. [93] synthesized C-ring 6,7-fused 2′-substituted-thiazolomorphinandienes (**15a–c**, Figure 22) from 14-bromocodeinone (**70**) via the Hantsch thiazole synthesis with either thiocarbamide, thioacetamide, or thiobenzamide. The cycloaddition of methyl vinyl ketone to the resulting 2-methylthiazolo- (**15a**), 2-phenylthiazolo- (**15b**), and 2-aminothiazolo-morphinandienes (**15c**) was thoroughly investigated by Sipos et al. [68]. Two products of the β-face attack (**71a**, R = CH_3_, **71b**, R = Ph) were obtained in 72 and 64% yield, respectively. The structures of the ring-constrained thevinone analogues (**71a,b**) were verified by 2D NMR experiments [68]. In the case of 2′-aminothiazolo-morphinan-6,8-diene (**15c**, R = NH_2_), one molecule of the diene (**15c**) reacted with three methyl vinyl ketone molecules to yield **72** (R_1_ = R_2_ = (CH_2_)_2_COCH_3_).

##### Miscellaneous Cycloadducts

Hromatka et al. [94] unsuccessfully attempted to convert the 7α-carboxylic acid ethylester (“thebaine-ethyl acrylate adduct” (**16f**)) by aminolysis directly to 7α-carboxamides. Application of *N*,*N*-diethylacrylamide (CH_2_=CH-CO-NEt_2_) as dienophile in benzene (7 h, reflux) did not result in any conversion, but in xylol (24 h, reflux), the reaction yielded carboxamide (**73**, Figure 23) quantitatively. Thebaine (**4**) with acrylic acid morpholide (CH_2_=CH-CON(CH_2_CH_2_)O) gave the appropriate morpholinyl-substituted 7α-carboxamide (**74**) in similar yield.

Hromatka et al. [95] synthesized the thebaine-maleimide adduct (**75a**) and its *N*^1′^-substituted derivatives (Figure 24) via different routes. Thebaine (**4**) gave upon reaction with maleimide (DMF, reflux, 15 h) the adduct **75a** in 76% yield, versus nearly quantitative yield of **75b** in the case of *N*-benzylmaleimide (EtOH, reflux, 1 h). The reaction of **4** with maleamide in methanol (reflux, 72 h) resulted, instead of the anticipated 7α,8α-dicarboxamido derivative (“thebaine-maleamide adduct”), in the product **75a** (“thebaine-maleimide adduct”) in 86% yield.

Treatment of the thebaine-maleic acid dimethyl ester adduct (**78**) with benzylamine led to formation of compound **75b** in 63% yield. Saponification of thebaine-maleonitrile adduct (**77**) with hydrogen peroxide and 6M NaOH (reflux, 20 h) also gave **75a** with low (19%) yield. Numerous *N*^1^-substituted derivatives (Figure 24) were prepared by aminolysis of the *N*^1^-carbethoxymethyl-maleimide adduct (**75d**, R = CH_2_COOH) with a variety of amines (morpholine, *N*-methylpiperazine, piperidine, and benzylamine) in 80–83% yield. The *N*^1^-benzylpyrrolidine compound (**76**) was synthesized from **75b** (R = CH_2_Ph) by reduction with LiAlH_4_ (THF, reflux, 16 h).

Aiming to synthesize constrained buprenorphine analogues, Barton et al. [44] investigated the cycloaddition of thebaine (**4**), *N*^17^-cyclopropylmethyl-northebaine (**9d**), and non-basic *N*^17^-acyl derivatives, *N*^17^-fomyl- (**9a**) and *N*^17^-cyclopropylcarbonyl-northebaine (**9e**), with cycloalkenones (2-cyclopenten-1-one and 2-cyclohexen-1-one) (Figure 25). They also studied the influences of several reaction parameters (e.g., temperature, pressure) and the application of Lewis acid catalysis on the rate of the cycloaddition. There proved to be a marginal impact of temperature on the rate of the conversion of the selected morphinan-6,8-dienes (**9a**, **9d**, **9e**, Figure 25) with 2-cyclopenten-1-one. Performing the reactions in toluene in a sealed tube at 130 °C, 24 h resulted in only modest yields (8–36%) of adducts. Application of longer reaction times and higher temperature was inappropriate due to the decomposition of the starting dienes as well as the desired products.

One of the first examples of the use of Lewis acid catalysis in the field of Diels–Alder reactions of morphinandienes was described by Barton et al. in 1993 [44]. Successful application of lithium tetrafluoroborate as a novel Diels–Alder catalyst was reported in the cycloaddition of *N*^17^-cyclopropylcarbonyl-northebaine (**9e**) with 2-cyclopenten-1-one (neat, 14 days, RT, **79**, R = CPCO, X = CH_2_ 75%) as well as with 2-cyclohexen-1-one (neat, 28 days, RT, **79**, R = CPCO, X = (CH_2_)_2_, 39%). The reaction of *N*^17^-formyl-northebaine (**9a**) with 2-cyclopenten-1-one resulted in the corresponding 7,8-ring-constrained thevinone analogue in a yield of 25% (LiBF_4_, 7 days, RT).

Coop et al. [96] (Figure 25) reported in a subsequent paper the preparation of 7,8- and 7,7-ring-constrained analogues of thevinones, prepared from thebaine (**4**) as well *N*^17^-cyclopropylmethyl- (**9d**) and *N*^17^-cyclopropylcarbonyl-northebaine (**9e**) with 1-indenone or cycloalkanones. In the first approach, the morphinane-6,8-dienes (**9d**, **9e**) were reacted with 1-indenone, generated in situ from a sulfoxide precursor [2-(*p*-chlorophenylsulfinyl)-1-indanone], by refluxing in toluene. In the reaction of thebaine (**4**) or *N*^17^-cyclopropylmethyl-northebaine (**9d**) with 1-indenone, the formation of two adducts (**80**, **81**, R = CPM) was observed. Coop et al. [96] found that the nature of the *N*^17^ substituent had a significant impact on the ratio of products formed. When using thebaine (**4**) as starting material, a product ratio of 6:1 (**80**/**81**, R = CH_3_) was observed (toluene, reflux, 9 h, isolated yield: 65%). When reacting *N*^17^-cyclopropylmethyl-northebaine (**9d**) with 1-indenone, the corresponding product ratio was 16:1 (**80**/**81**, R = CPM, 10 h, 34%). In the case of a *N*^17^-cyclopropylcarbonyl substituent (**9e**), a single product **80** (R = CPCO) was isolated after 30 h reaction time in a yield of 79%.

In a second approach [96], thebaine (**4**) was treated with methylene cycloalkanones to yield 7,7-ring-constrained thevinone analogues (Figure 26). The dienophiles were generated in situ from the corresponding Mannich bases (liberated from methiodide salts (1.4 equiv.) with sodium carbonate (1.5 equiv.)) via rapid β-elimination in boiling toluene (140 °C, autoclave, 48 h). The dienophiles 2-methylene-cyclopentanone, 2-methylene-cyclobutanone, and 2-methylene-cyclohexanone were allowed to react with thebaine (**4**), resulting in the formation of two adducts. When reacting thebaine (**4**) with 2-methylene-cyclobutanone, there was a 1:1 product ratio of the 7α/7β spiro cycloadducts (**82**/**83**, X = CH_2_, reaction time two days, total isolated yield: 60%). When applying 2-methylene-cyclopentanone as dienophile, there was as a 3:4 7α/7β product ratio (**82**/**83**, X = (CH_2_)_2_), 6 days, yield: 54%). In the case of 2-methylene-cyclohexanone, even after two weeks of boiling in toluene, only traces of the desired product were detected by MS analysis. In the reaction of thebaine (**4**) with 2-methyleneindanone, there was a 4:3 7α/7β product ratio of the adducts (**84**, **85**, X = CH_2_), with an isolated total yield of 50%.

### 2.2. Grignard Addition

The Grignard reaction is considered one of the most important routes to functionalization of the 7-acyl type Diels–Alder adducts (**16**, **17**) of morphinanedienes [97]. However, the reactions of 7-acyl-6,14-ethenomorphinan derivatives with Grignard reagents are generally complicated and lead to the formation of product mixtures. The various products are formed as result of at least three processes [97]: (**1**) normal Grignard addition (Figure 27), which can result in two diastereomeric tertiary alcohols (where R_1_ and R_2_ represent different substituents); (**2**) Grignard reduction (Figure 28, β-hydrogen transfer; when possible) which can afford two diastereomeric secondary carbinols; (**3**) base-catalyzed rearrangement (Figure 29) caused by an excess of the Grignard reagent, giving a tertiary alcohol with a “[5,6,7]cyclopropane” ring and an opened E-ring.

The normal Grignard addition of the R_2_MgX reagent to 7-acyl-6,14-ethenomorphinans (**16a–d**) takes place with a high degree of stereoselectivity. The asymmetric induction is explicable via the formation of a six-membered chelate-type (**A**) transition state (i.e., Cram’s rule) during the Grignard addition [97]. The magnesium atom of the Grignard reagent can simultaneously coordinate the oxygen atoms of both the 6-methoxy and the 7α-acetyl groups to form the intermediate (**A**) (Figure 27). An attack of the R_2_ group of the R_2_MgX reagent is more feasible and less hindered from the β-face as comparted to the direction of the 6,14-bridge (α-face). It is important to be aware of that all of the main products of the normal Grignard addition belong to the same stereochemical series (**B**), formed according to the mechanism presented in Figure 27. By proper application of the Cahn Ingold Prelog conventions of nomenclature, the products are designated as 20*S*-*tert*-butyl-dihydrothevinol (**105b**, see buprenorphine synthesis in Section 2.3.1) and the other side as 20*R*-*n*-propyl-thevinol (20*R*-**90**, see dihydroetorphine synthesis) or 20*R*-phenethyl-thevinol (**86**, see PEO synthesis, Figure 30) according of the substituent priority rules. The missing stereoselectivity of the Grignard addition for derivatives without substituents in the six position here led to no enabling of chelate formation [51].

The reaction of 7α-thevinone (**16a**) with 2-phenylethylmagnesium bromide (Figure 30) serves as an example for the complexity of the Grignard reaction of 7-acyl-6,14-ethenomorphinans (**16**, **17**). For the synthesis of phenethyl-orvinol (PEO, **99**), the Grignard reaction of 7α-thevinone (**16a**) with six molar equivalents of 2-phenylethylmagnesium bromide was carried out [98]. The reaction took place with a high degree of stereoselectivity and resulted in the tertiary alcohol **86** with 20*R* absolute configuration as the main product (62%) [98]. The formation of the major by-product (**87**) here resulted from a base-catalyzed rearrangement (Figure 28 and Figure 30), which was followed by Grignard addition. The isolated yield of this by-product was 6.5%.

A β-hydrogen transfer (Figure 28) was also possible when applying 2-phenylethylmagnesium- or *n*-propylmagnesium bromide Grignard reagents. In the case described above, a mixture of secondary alcohols (20*S*-**88**/20*R*-**88**) was isolated as minor by-products (totaling 1.7%) [98]. The reaction mechanism of the idealized Grignard reduction is presented for this case in Figure 28.

The reaction mechanism of the formation of the major by-product (**87**) is depicted in Figure 29. The initial step is the removal of the C-7 proton [99], which is in α position to the 7-carbonyl group (**16a** → **A** → **B**, Figure 29). This removal occurs due to the influence of the Grignard reagent, which is used in excess as a strong base acting on the starting 7-α-ketone (**16a**). The next elementary step is the opening of the 4,5-epoxy-ring (**B** → **C**), which forms a phenolate anion and a new “[5,6,7]cycloropane” ring (**C**). A consequence of the above procedures, which compete with the normal Grignard addition (**16a** → **86**) of 2-phenylethylmagnesium bromide on 7α-carbonyl group of thevinone (**16a**), is that the C-7 is no longer asymmetric.

If a sufficient amount of the Grignard reagent is present, the resulting ketone (**C**, Figure 29) can react with it to the corresponding tertiary alcohol, and thus stabilize the cyclopropane ring; the product is consequently a tertiary alcohol (**87**), which contains a phenolic hydroxyl group in the four position and a “[5,6,7]cyclopropane” ring. The isolated yield of this by-product was 6.5% [98]. If the reagent is exhausted, the cyclopropane ring in the transition state (**C**) may be opened by attack at C-5, which would lead to the starting thevinone (**16a**). Provided that the attack takes place at C-6 (**C** → **D**), a rearranged product, isothevinone (**D**), may also be formed.

On comparing the ^1^H-NMR spectra of **87** with **86**, the absence of the 7β-proton signals is conspicuous, compared to the triplet at 2.02 ppm for the case of **86** [98]. A further difference is the multiplicity of the 8α-H signals, which occurs at 0.78 ppm as a *doublet* instead of the generally noted *doublet of doublets*. Both of these observations are consistent with the absence of the 7β-proton in **87**. In the ^1^H-NMR spectrum of **87**, another deuterable broad singlet appears at 5.80 ppm downfield to an exchangeable broad singlet at 4.92 ppm (20-OH). This 5.80 ppm signal belongs to the 4-hydroxyl group. There is a major difference in the chemical shifts for C-5 (98.8 and 32.0 ppm) and C-6 (84.1 and 71.4 ppm) as revealed by comparison of the ^13^C-NMR spectra of **86** with **87**. These observations are thus consistent with an opening of the E-ring. The structure of **87** was confirmed by acid-catalyzed dehydration, enol-ether hydrolysis, and rearrangement [99] to the 5,14-bridged thebainone derivative **89**.

The structure of **89** was further elucidated by the following observations: (a): The typical singlet of the 6-OCH_3_ group disappeared from the proton spectrum, being evident at 3.74 ppm in the case of **5b**; (b): C-6 appeared in the ^13^C spectrum of **89** at 196.6 ppm (and in **87** at 71.4 ppm). Further by-products of this reaction were the secondary alcohols 20*S*-**88** and 20*R*-**88**. In the case of 2-phenylethylmagnesium bromide Grignard reagent, a β-hydrogen transfer was also possible, which was responsible for the formation of these compounds (Grignard reduction). Here, the mixture consisting of the diastereomeric secondary alcohols was isolated in low yield (1.7%) [98]. The ^1^H-NMR spectrum here indicated a 5:1 20*S*-**88**/20*R*-**88** product ratio.

As a second example of Grignard difficulties, we mention the reaction of 7α-dihydrothevinone (**104**, Figure 31) with six equivalents of *n*-propylmagnesium bromide [100] under identical conditions as described above for the reaction of thevinone (**16a**) and 2-phenethylmagnesium bromide [98]. Four compounds were isolated from the complex product mixture: 20*R*-dihydroethorphine methyl ether (20*R*-**90**, 51%) as the main product, 20*S*-dihydrothevinol (**91**, 20%) as the main by-product, a small amount of 20*S*-dihydroetorphine methyl ether (20*S*-**90**, 4.2%), and finally the product of the base-catalyzed rearrangement (**92**, 7.6%). When reacting 7β-dihydrothevinone (**164**) with six molar equivalents of *n*-propylmagnesium bromide in toluene-diethylether, the normal Grignard adduct (**93**) was accompanied by secondary alcohol **94** (2%) and compound **92** (10%) [101]. It is noteworthy that the same base-catalyzed derivative **92** was isolated from the product mixture in both cases (7α-acetyl/7β-acetyl).

In the 7α-acyl series, the simplest cases of Grignard addition are those in which R_1_ and R_2_ are identical (Figure 27, compound (**B**)). Examples for this are the reaction of thevinone (**16a**) with methylmagnesium iodide or the addition of phenylmagnesium bromide to nepenthone (**16b**), and instances where the character of the reagent does not allow a β-hydrogen transfer. In such reactions, the target tertiary thevinols are reported as the sole products. Thus, an exclusive formation of the corresponding tertiary alcohol was observed in the reactions of 7α-thevinone (**16a**), [97] as well as *N*-substituted-7α-dihydronorthevinone derivatives (see Section 2.3.1) [102,103] with methylmagnesium iodide.

The reaction of 7α-acetyl compounds (e.g., 7α-dihydrothevinone (**104**) [97] or *N*-substituted-7α-dihydronorthevinone derivatives (**112a–e**, see Section 2.3.1) [103]) with *tert*-butylmagnesium chloride resulted in the desired 20*S* tertiary alcohol (yield 41–54%) main product and secondary alcohol (31–41%) by-products arising from the Grignard reduction.

Upon treatment of 7β-acyl-6,14-ethenomorphinans (e.g., 7β-thevinone (**17a**) [97] or *N*-substituted-7β-dihydronorthevinone derivatives [104]) with methylmagnesium iodide, there was formation of the expected tertiary alcohol (60–73%), accompanied by another tertiary carbinol (8–13%) with an opened E-ring. This side-product carried a C-4 phenolic hydroxyl group and a cyclopropane ring, both resulting from the attack of the strong nucleophilic Grignard reagent. The Grignard reactions in the 7β-acyl-series led to more complex product mixtures than in the 7α-series. Numerous major and minor by-products were isolated and characterized during the Grignard reaction step of the syntheses of 7β-diprenorphine (**165a**) [104], 7β-buprenorphine (**166**), 7β-etorphine (**167a**), and 7β-dihydroetorphine (**167b**) [101].

While opioid analgesics are essential for the medical control of pain, there is inevitably an association between analgesic potency and potential for abuse and dependence [105]. Among the many relevant factors for analgesic potency are effects at the several OR-types and the pharmacological distinctions between stereoisomers. For example, the µOR-selective agonist ohmefentanyl has three chiral centers and thus eight possible isomers, not all of which are agonists [106]. Treatment of cells expressing high µOR levels with the irreversible antagonist β-funaltrexamine reveals a reserve of receptors with distinct binding properties for the diasteromeric oxymorphone metabolites 6α- and 6β-oxymorphol [107]. A complete pharmacological evaluation of an opioid ligand structure might call for functional assessment of the activities of its isomers at each of the OR types. While the positive reinforcing properties of morphine are largely attributed to µOR signaling in specific brain regions, this pathway entails complex interactions with multiple cellular targets [108] and the additional involvement of δORs [109].

The primary reason for the historical interest in these diastereoselective reactions lies in its utilization for the preparation of pure 20*R* and 20*S* diastereoisomeric pair tertiary carbinols. For example, the treatment of thevinone (**16a**) with phenylmagnesium bromide results in 20*S*-phenyl-thevinol [74,110,111], while the addition of methyllitium [112] or methylmagnesium iodide [110] to nepenthone (**16c**) provides 20*R*-phenyl-thevinol. The absolute configuration of the C-20 chiral center is of high importance for the analgesic activity of the orvinol OR agonists. As such, there is an enormous range of analgesic potency of the diastereoisomers relative to morphine. For example, 20*R* etorphine is 3200-fold more potent, while its isomer 20*S* etorphine is only 40-fold more potent than morphine in the rat tail pressure test [111].

The pharmacological investigations of 7β-etorphines and their 18,19-dihydro analogues showed that they are OR agonists with similarly high affinities to the ORs as compared to the corresponding 7α-isomers [113]. In the etorphine series, the highest affinity was observed in the case of 7β-etorphine (K_i_ = 0.47 nM) in rat brain membrane fractions.

Recently, Husbands et al. [114] reported on the addition of a large variety of Grignard reagents (R = alkyl, aryl, alkyl-substituted-phenyl, halogeno-substituted-phenyl, 2-thienyl, 3-methyl-2-thienyl, 5-chloro-2-thienyl, and 3-thienyl) to *N*-cyclopropylmethyl-dihydronorthevinone (**112a**) as well as to its 18,19-didehydro analogue. Stereoselective addition of 2-pyridyl-lithium or 4-pyridyl-lithium to *N*-cyclopropylmethyl-dihydronorthevinone (**112a**) was explained by the Felkin–Ahn model.

### 2.3. Orvinols

For the Diels–Alder adduct of **the**baine (**4**) with methyl **vin**yl ket**one**, Bentley and associates coined the trivial name thevinone (**16a**). Consequently, secondary alcohols (e.g., **88**, **91**) or tertiary alcohols (e.g., **86**, **90**) derived from thevin**one** (**16a**) through reduction or Grignard addition, respectively, are called thevin**ol**s. Orvinols (e.g., **95–97**), which typically possess about ten times higher affinity to opioid receptors than thevinols, are prepared by 3-*O*-demethylation of the latter compounds. Their names can be deduced from morphine alkaloid oripavine (**5**) [115].

Orvinols [116], represented by diprenorphine (**95**, DPN), buprenorphine (**96**, BPN), etorphine (**97**), dihydroetorphine (**98**, DHE), and phenethyl-orvinol (**99**, PEO), are important semisynthetic C-ring-bridged morphinan derivatives (Figure 32). In the following section, we present a brief account of the pharmacological properties of their most important representatives.

*Diprenorphine* (**95**, DPN, *Revivon*^®^, M5050) is a potent non-selective opioid antagonist with partial kappa agonist effects [116], which is used to reverse the effects of etorphine or carfentanil in veterinary practice. Diprenorphine not only antagonizes the analgesic action of opioid agonists such as morphine (**1**) and etorphine (**97**) but can reverse the agonist action of such derivatives as pentazocine or levallorphan. Having a high receptor affinity, [^3^H]DPN has been used to identify and characterize ORs in vitro [117] and has been widely used (along with [^18^F]FE-DPN) in PET studies of ORs [8].

*Buprenorphine* (**96**, BPN, *Temgesic*^®^, *Buprenex*^®^, *Subutex*^®^) [116,118,119,120] is a strong analgesic with partial agonist activity at µ-opioid receptors and antagonist activity at κ- and δ-receptors. The *tert*-butyl group attached to position 20 increases the lipophilicity of the molecule, which tends to promote its entry into the brain. Injectable BPN is used clinically as a potent analgesic in the treatment of cancer-related and postoperative pain (since 1978) [121] and is now available as sublingual tablets (since 1982). Buprenorphine in conjunction with the OR antagonist naloxone combination (*Suboxone*^®^) is an effective and safe alternative to methadone in substitution therapy of opioid-dependent humans. In addition to effects at the three classical ORs, BPN is a full agonist at the ORL1 receptor [122].

*Etorphine* (**97**, *Immobilon*^®^, M99) is a non-selective OR agonist with extremely high potency, which is approved only in veterinary medicine for the immobilization of large wild animals [123]. Its agonist effect may be reversed by diprenorphine or naloxone [124].

*Dihydroetorphine* (**98**, DHE), the etorphine (**97**) analogue with a saturated 6,14-*endo*-ethano bridge, is known as one of the strongest semisynthetic 6,14-ethenomorphinan derivatives ever synthesized, up to 12,000 times more potent than morphine [111,125,126]. In the early 1970s, *phenethyl-orvinol* (**99**, PEO) played an important role in the development of a new opioid receptor model. Structure–activity relationship studies of thebaine Diels–Alder adducts suggested a modification of the Beckett-Casy opioid receptor model. These considerations led Lewis and Bentley and Cowan [127,128] to propose the presence of a second lipophilic binding site with allosteric effects on the classical binding interaction. Recently, thienorphine (**100**) [129] and the buprenorphine analogues BU08028 (**101**) [130], BU10119 (**102**), and BU10120 (**103**) [131,132,133] (Figure 32) were reported as orvinols with potential in the treatment of opiate addiction.

#### 2.3.1. Synthesis of Orvinols

The synthesis of orvinol derivatives (**95**–**103**) starting from thebaine (**4**) involves, in most cases, the following transformations: Diels–Alder reaction, Grignard addition, *N*-demethylation, *N*-alkylation of the secondary amine, and finally 3-*O*-demethylation. DPN (**95**), BPN (**96**), DHE (**98**), and PEO (**99**) can be prepared by the original method by Bentley [74,97,111] or one of the more recently developed multistep procedures [102,103] starting from the poppy alkaloid thebaine (**4**). Recently developed procedures [134,135,136] of the Hudlický group described the synthesis of buprenorphine (**96**) from oripavine (**5**) by the application of their own newly developed *N*-demethylation methods.

##### Diprenorphine, Buprenorphine, Etorphine, and Their Analogues

Bentley et al. [74,97,111] described the first synthesis of diprenorphine, buprenorphine, and etorphine in the 1960s. Diprenorphine (**95**) and buprenorphine (**96**) were prepared from thebaine in eight steps (Figure 33). Thevinone (**16a**) [74] was obtained in a regio- and stereoselective Diels–Alder addition of thebaine (**4**) with methyl vinyl ketone. The 6,14-endo-etheno bridge of **16a** was saturated by heterogeneous catalytic hydrogenation to obtain dihydrothevinone (**104**). Next, the Grignard reaction of the latter compound with methylmagnesium iodide or *tert*-butylmagnesium chloride gave the corresponding tertiary alcohols **105a–b**. *N*-Demethylation was performed by application of the von Braun method [137]. Reaction with cyanogen bromide followed by cyanamide hydrolysis afforded the secondary amines (**107a–b**). The *nor*-derivatives (**107a–b**) were transformed with cyclopropylcarbonyl chloride to the *N*^17^-acyl compounds (**108a–b**). Finally, reduction of the carbonyl group of **108a–b** with LiAlH_4_ and 3-*O*-demethylation of **109a–b** with KOH in diethylene glycol resulted in the desired orvinols (**95**, **96**).

The pathway to opioid dependence is a matter of intense study around the world; in the briefest terms, drug-liking can cascade to tolerance (the need for higher doses) and dependence (an inability to function without the drug), with the emergence of life-threatening symptoms upon precipitous withdrawal. These phenomena are subverted by environmental factors; heroin addicts can experience drug craving simply upon viewing visual cues such as the paraphernalia of drug use [138]. This likewise occurs in those with alcohol, nicotine, and cocaine dependence, and the extent of cue-evoked craving is a predictor of relapse risk. The neurophysiology of craving is poorly understood but certainly involves a form of conditioning. However, the primary step in the development of opioid dependence is best understood in terms of actions at the µORs in brain. Activation of µORs by an agonist such as morphine sends a signal to the receptive neuron promoting the intracellular formation of a second messenger molecule known as cyclic-AMP (cAMP), which then interacts with a variety of intracellular targets. The pharmacological potency of a given opioid ligand can be assessed in an assay measuring the rate of cAMP formation in vitro; such assays distinguish full agonists that maximally activate the cAMP pathway, partial agonists that incompletely stimulate the pathway, antagonists that simply block the binding of naturally occurring or exogenous opioid agonists, and even inverse agonists that inactivate a constitutive cAMP production. In silico analyses are imperfect predictors of the pharmacological action of receptor ligands [139] but can identify the key amino acid residues involved in ligand binding patterns [140].

In the early 1990s, it emerged that the partial agonist buprenorphine (**96**) could serve not only as an analgesic in the low-dose management of post-operative pain but also at higher doses as an alternative to methadone in the treatment of heroin and other opioid dependencies [141]. This application gave an immense impulse to methodological research for novel synthesis routes of buprenorpine (**96**). Currently, the global consumption of buprenorphine (**96**) has reached 5.5 tons per year [3]. In 1993, a research group of the Alkaloida Chemical Company Ltd. (Alkaloida Vegyészeti Gyár RT, Tiszavasvári, Hungary) developed a procedure (Figure 34) for the synthesis of diprenorphine (**95**) and buprenorphine (**96**) staring from *N*^17^-formyl-northebaine (**9a**) and likewise from *N*^17^-benzyl-northebaine (**9b**) [102]. Both morphinandienes (**9a**, **9b**) were reacted with methyl vinyl ketone to yield the corresponding Diels–Alder adducts (**110a**, **110b**). By treatment of *N*^17^-benzyl-northevinone (**110b**) under heterogeneous catalytic conditions (H_2_, EtOH, 10% Pd-C, 60 °C, 6 bar), there occurred saturation of the 6,14-*endo*-etheno bridge accompanied by the cleavage of the *N*^17^-benzyl group, thus resulting directly in dihydronorthevinone (**111**, 91%).

The Δ^18,19^ double bond of *N*^17^-formyl-northevinone (**110a**) was saturated under similar conditions to yield *N*^17^-formyl-dihydronorthevinone, which was transformed to dihydronorthevinone (**111**) by acidic hydrolysis (HCl/EtOH, reflux, 4 h). When *N*^17^-benzyl-northebaine (**110b**) was treated with acrolein, the 7α-formyl cycloadduct was obtained in 83% yield. This latter product was reacted with methylmagnesium iodide in toluene to obtain a mixture of 20*R* and 20*S* secondary alcohols. These diastereomeric alcohols were separated as tosylates by column chromatography. The mixture of secondary alcohols was then converted to *N*^17^-benzyl-northevinone (**110b**) by Swern oxidation. For the preparation of DPN (**95**) and BPN (**96**), dihydronorthevinone (**111**) was *N*-alkylated with cyclopropylmethyl bromide to yield *N*^17^-cycloropylmethyl-dihydronorthevinone (**112a**). This compound was reacted in separate reactions with methylmagnesium iodide as well as *tert*-butylmagnesium chloride in a Grignard reaction to yield the respective diprenorphine-3-*O*-methylether (**109a**) as well buprenorphine-3-*O*-methylether (**109f**). 3-*O*-Demethylation with potassium hydroxide in diethylene glycol at 210 °C provided the target orvinols in 62–63% yield.

In 1994, the Makleit research group reported the synthesis of numerous *N*^17^-substituted-diprenorphine (**113b–e**) and buprenorphine (**113g–j**) analogues (Figure 35) [103]. The Δ^18,19^ double bond of thevinone (**16a**) was saturated under heterogeneous catalytic conditions (H_2_, toluene, Pd-C, 55 °C, 6 bar). Dihydronorthevinone (**111**) was prepared by *N*-demethylation of dihydrothevinone (**104**) using diethyl azodicarboxylate (DEAD). Numerous *N*^17^-substituted-dihydronorthevinone derivatives (**112a–e**) were prepared from **111**.

Grignard addition of methylmagnesium iodide or *tert*-butylmagnesium chloride to the 7α-acetyl group resulted in the corresponding *N*^17^-substituted-20-methyl-thevinols (**109a–e**, 75–94%) or *N*^17^-substituted-20-*tert*-butyl-thevinols (**109f–j**, 41–54%). Formation of 20*S*-secondary alcohols as by-products (31–41%) was also detected by applying *tert*-butylmagnesium chloride. The thevinols (**109a–j**) were 3-*O*-demethylated (KOH, diethyleneglycol, 210 °C) to the target DPN (**113b–e**) as well as BPN (**113g–j**) analogues. Selective 3-*O*-demethylation of buprenorphine-3-*O*-methylether (**109f**) was also achieved with diisobutylaluminumhydrdide (DIBAL).

In 2004, a series of guanidinyl-substituted morphinan derivatives were reported by Jarrott et al. (Figure 36) [142]. They had applied three different synthetic approaches. In the first instance, unsubstituted *N*^17^-linked morphine guanidine was synthesized from morphine (**1**) via 3,6-di-*tert*-butyldimethylsilyloxy-morphine. The latter was converted with cyanogen bromide to the TBDMS-protected cyanamide derivative. The *N*-cyano normorphine compound was treated with methylchloroaluminum amide and the TBDMS-protecting group was removed in a 10:1 (*v/v*) acetonitrile-THF mixture with 40% aqueous HF to yield *N*^17^-(2-carboxamidino)-morphine.

The corresponding diprenorphine and buprenorphine analogues were prepared by applying similar procedures starting from 20-methyl-dihydronororvinol (**114a**, nordiprenorphine) as well as 20-*tert*-butyl-dihydronororvinol (**114b**, norbuprenorphine), respectively.

In their second approach (Figure 36), unsubstituted guanidines with a three-carbon spacer were synthesized from normorphine, nordiprenorphine (**114a**), and norbuprenorphine (**114b**). For the synthesis of the diprenorphine derivate, nordiprenorphine (**114a**) was reacted with acrylonitrile to obtain the *N*^17^-(2-cyanoethyl)-compound (**115a**). Next, the phenolic hydroxyl in position three (**115a**) was protected with a TBDMS group. The *N*^17^-cyanoethyl compound (**116**) was then reduced with LiAlH_4_ in diethylether to the *N*^17^-propylamino derivative (**117**), and the latter product was reacted with 1*H*-pyrazole-1-carboxamidine to provide **118**. The 3-*O*TBDMS derivative (**118**) was deprotected with NH_4_F in methanol to yield *N*^17^-[3-(2-carboxamidino)-aminopropyl]-nordiprenorphine (**119a**) with an overall yield of 31%. *N*^17^-[3-(2-carboxamidino)-aminopropyl]-norbuprenorphine (**119b**) was prepared analogously from norbuprenorphine (**114b**), but without protection of the phenolic hydroxyl group, in a three-step procedure with an overall yield of 46%. In their third approach, *N*-substituted guanidines (*N*-linked or with (CH_2_)_3_ spacer) and 3-*O*- and/or 6-*O*-substituted guanidines based on the above mentioned three morphinans were prepared in a similar manner. Not all synthetic opiates can cross the blood brain barrier (BBB), which is an essential property for central analgesic action. The Straub tail test, a behavioral response with rigid dorsiflexion of the tail, indicated lesser central action in mice compared to that of morphine (**1**). The ligands were further evaluated by rapid in vivo screening with the mouse phenylquinone-induced abdominal constriction test. Most of the compounds showed only moderate analgesic activity, but the diaryl-substituted guanidines with a (CH_2_)_3_ spacer or without spacer did display a promising effect on analgesia.

In 2006, Park et al. [143] reported the synthesis of novel buprenorphine analogues (**122a–b**, **123a–b**, **127a–d**, Figure 37). Here, the *tert*-butyl group of buprenorphine (**96**) in position 20 was replaced with cyclopropyl and cyclobutyl substituents. When dihydrothevinone (**104**) was reacted with cyclopropylmagnesium bromide, a mixture of 20*R* and 20*S* tertiary alcohols was formed in a 7:3 ratio favoring 20*R*, as determined by HPLC. After separation of the tertiary alcohols by column chromatography, the 20*R* diastereomer was *N*-demethylated by the von Braun method. 

Cyanamide (**124a–b**) was hydrolyzed, and the secondary amine (**125a–b**) was converted by *N*-alkylation to *N*^17^-cyclopropylmethyl as well as *N*^17^-cyclobutylmethyl derivatives (**126a–d**). Finally, 3-*O*-demethylation was achieved by reaction with KOH in ethylene glycol. 20*R*-cyclobutyl analogues were also prepared using analogous procedures, and the various prepared ligands were screened for analgesic activity and propensity for substance dependence and addictive activity in male mice of the DBA strain. In the behavioral test, 20*R*-cyclobutyl-dihydronororvinol (**122b**) had 100–554 times higher analgesic potency than buprenorphine (**96**) itself. Pharmacological investigation identified 20*R*-cyclobutyl-*N*^17^-cyclopropylmethyl-dihydronororvinol (**127c**) as a κ-OR selective partial agonist ligand with low abuse potential despite its high analgesic potency.

Maurer and Rapoport [42] elaborated syntheses of *N*^17^-*C*^8β^-(methano and ethano)-bridged conformationally constrained analogues of 20*R*-*n*-butylorvinol (homoetorphine) in a fourteen-step procedure (Figure 38) starting from *N*^17^-benzoyl-northebaine (**9c**).

For the preparation of *N*^17^-*C*^8β^-methano-bridged homoetorphine analogue, the starting diene (**9c**) was reacted with maleic anhydride in a Diels–Alder reaction to obtain the expected 7α-8α-adduct (**128**). Reduction of the anhydride (**128**) with lithium aluminum hydride resulted in the *N*^17^-benzyl-7α-8α-bis(hydroxymethyl) compound (**129**) with a yield of 64%. After selective protection of the 7α-hydroxymethyl with *tert*-butyl-dimethysilyl (TBDMS), the free 8α-hydroxymethy group of **130** was oxidized by the Swern method (oxalyl chloride, DMSO) to the 7α-formyl derivative (**131**). The latter was converted to an 8β-aldehyde (**132**) by silica gel-catalyzed epimerization, whereupon the 8β-formyl derivative (**132**) was reduced with NaBH_4_ to the 8β-hydroxymethyl compound (**133**). This compound was converted to mesyl ester, which reacted with the tertiary nitrogen yielding a quaternary compound (**134**), i.e., through a ring closure mechanism. The *N*^17^-benzyl group of the quaternary salt (**134**) was removed (ammonium formate/Pd-C in methanol), leading to a tertiary amine heptacycle (**135**). After removing the TBDMS protecting group, the 7α-hydroxymethyl group (**136**) was oxidized to aldehyde (**137**). The subsequent Grignard addition of methylmagnesium iodide to the C-20 carbonyl (**137**) resulted in the diastereomeric secondary alcohols (**138**) in an 11:1 ratio. Oxidation of the secondary alcohols (**138**) led to the formation of 7α-methyl ketone (**139**). Treatment of **139** with *n*-butylmagnesium bromide gave solely the 20*R*-tertiary alcohol thevinol derivative (**140**), which was 3-*O*-demethylated with NaSPr to obtain the target orvinol. *N*^17^-*C*^8β^ and *N*^17^-*C*^8α^-ethano-bridged analogues of 20*R*-butylorvinol were also synthesized and tested for analgesic activity by the tail withdrawal method after subcutaneous administration in Sprague-Dawley rats [42].

Surprisingly, *N*^17^-*C*^8β^-methano-homoetorphine (**141**) and *N*^17^-*C*^8β^-ethano-homoetorphine showed diminished agonist activities compared to morphine and homoetorphine, and the *N*^17^-*C*^8α^-ethano-bridged analogue showed an analgesic potency that was lower by a third compared to morphine. The authors concluded that the conformation of the *N*^17^ (lone pair orientation) in bridged morphinans was critical for opioid receptor binding and agonism. Fixing or locking the *N*^17^ electron lone-pair toward the A-ring side of the molecule, i.e., equatorial orientation assuming a piperidine chair conformation, resulted in the loss of analgesic activity [42].

In 1988, Linders et al. [72] reported the synthesis of 6-demethoxy-3-deoxyetorphine (**145**, Figure 39) starting from 6-demethoxythebaine (**11b**) in a five-step procedure. In brief, 6-demethoxythebaine (**11b**) was reacted with methyl vinyl ketone. The resulting 6-demethoxythevinone (**24**) was subjected to a Grignard addition with *n*-propylmagnesium bromide. 20-*n*-propyl-6-demethoxythevinol (**142**) was then 3-*O*-demethylated to 6-demethoxy-etorphine (**143**). For removal of the 3-hydroxyl group, 6-demethoxy-etorphine (**143**) was converted to its 3-(1-phenyltetrazol-5-yl) ether (**144**). Finally, the latter product was hydrogenolyzed (formic acid/10% Pd-C) to 6-demethoxy-3-deoxyetorphine (**145**). The synthesized etorphine analogue (**145**), which lacked the 3-OH and 6-OCH_3_ groups, was a strong agonist (ED_50_ = 0.05 mg/kg, ~116 x morphine) in the mouse tail-flick assay.

Furanocodides and furanomorphides are known rearrangement products of thevinols and orvinols, respectively [144]. Husbands et al. [145] synthesized ring-constrained analogues of diprenorphine (**95**) and buprenorphine (**96**). Buprenorphine-furanomorphide was prepared directly from buprenorphine (**96**) by acid-catalyzed rearrangement. The synthesis of 20-monomethyl-substituted analogues (**150a–b**, Figure 40) was started from diprenorphine methylether (**109a**), which yielded the 7α-isoprenyl derivative (**146**) upon treatment with formic acid.

The latter compound (**146**) was hydroborated and oxidized with hydrogen peroxide to obtain a mixture of the diastereomeric primary alcohols (**147a–b**). After separation by column chromatography, the respective alcohols (**147a** and **147b**) were mesylated, and these mesylates (**148a**, **148b**) were converted to furanocodides (**149a–b**) by ring closure through LiAlH_4_ treatment. The furanocodides were 3-*O*-demethylated with sodium propane thiolate to the corresponding furanomorphides (**150a–b**). The diprenorphine furanomorphide analogues displayed subnanomolar affinity for the μ-OR in guinea pig brain membranes (**150a**: K_i_ (μ) = 0.7 nM), **150b**: K_i_ (μ) = 0.3 nM,) [145].

Aiming to develop dual-acting κOR agonist and μOR partial agonist ligands as potential treatment agents for cocaine and other psychostimulant abuse, Greedy et al. [146] synthesized four series of 20-substituted tertiary and secondary alcohol analogues of buprenorphine (**96**) and isopentyl orvinol (M320 [147], R = *i*-pentyl) in 2013. For the synthesis of the first series, *N*^17^-cyclopropylmethyl-dihydronorthevinone (**112a**), which was prepared from thebaine (**4**) in two steps, was reacted with the corresponding RMgBr Grignard reagent (R = *i*Pr, *i*Bu, *i*Pentyl, *c*Pent, *c*Pent-CH_2_, *c*Pent-CH_2_CH_2_, *c*Hex, *c*HexCH_2_, *c*HexCH_2_CH_2_, Bn, PhCH_2_CH_2_, *n*Pr, *n*Pent, and *t*Bu) to obtain tertiary alcohols (**151**). The latter compounds were 3-*O*-demethylated with sodium propane thiolate in hexamethyl phosphoramide to the target orvinols (**152**, series-1, Figure 41) with the same relative stereochemistry as buprenorphine.

For preparation of the second series of tertiary alcohols with the opposite stereochemistry at C-20 (**157**), *N*^17^-cyclopropylmethyl-northebaine (**9d**) was chosen as the starting material, which was converted to the corresponding 7α-formyl Diels–Alder adduct with acroleine. Subsequently, the 6,14-etheno bridge of the adduct was reduced (H_2_, EtOH, 50 °C, 3.4 bar) in the presence of Pd-C catalyst. The 7α-formyl-6,14-*endo*-ethano (**153**) compound was converted to a mixture of secondary alcohols (**154**) by Grignard addition, which was then transformed to 7α-alkyl ketones (**155**) by Swern oxidation. Grignard reaction with methylmagnesium bromide and subsequent 3-*O*-demethylation resulted in the tertiary alcohols (**157**) with the opposite relative stereochemistry. Secondary alcohols (**159**, series-3, Figure 41) with the same relative stereochemistry as buprenorphine (**96**) were synthesized from the 7α-acyl derivatives (**155**) by reduction with LiAlH_4_ in THF (**158**) and subsequent 3-*O*-demethylation (**159**). Finally, a number of diastereomeric secondary alcohol orvinols (**160**, series-4) were prepared from the isolated major isomer secondary alcohol thevinol derivatives (**154**) by 3-*O*-demethylation.

Binding affinities of the synthesized ligands to μOR and κOR were determined [146] by displacement of the ambivalen ligand [^3^H]diprenorphine binding to C6-rat glioma cells expressing recombinant rat μOR and CHO cells expressing recombinant human κOR. Affinities (K_i_) for μOR ranged from 0.029 nM (series-3, R = phenethyl) to 2.7 nM (series-2, R = *n*-pentyl) with similarly high affinities for κOR (K_i_ = 0.015 nM/series-3, R = cyclohexylmethyl to K_i_ = 0.22 nM/series-2, R = cyclohexylmethyl), indicating a lack of μOR and κOR selectivity. Nonetheless, the κOR to μOR efficacy ratio could be controlled by the chain length of the substituent in position 20 (introduction of branched or ring systems into the chain where advantageous). Analogues of series-1, with *iso*-butyl (**152**, R = *i*Bu) and *iso*-pentyl (**152**, R = *i*Pe) substituents, were chosen for evaluation in vivo in the PPQ (abdominal stretch assay [148]). Compounds **152** (R = *i*Bu) and **152** (R = *i*Pe) were equally potent κOR agonists (ED_50_ [mg/kg (s.c.)] = 0.02) [146]. Their agonist effects could be fully reversed by the κOR-selective antagonist norBNI and partially reversed by the μOR-selective antagonist β-funaltrexamine (βFNA).

Quite recently, to investigate the substituent effect on the biological activity of buprenorphine analogues, novel *N*^17^-[3-(1-aryl)-1*H*-1,2,3-triazol-4-yl)methyl]-norbuprenorphine derivatives (**163a–c**, Figure 42) were synthesized by Faizi and associates [149]. 20-*tert*-Butyldihydrothevinol (**105b**) was prepared from thebaine (**4**) via thevinone (**16a**) and dihydrothevinone (**104**) as described by Bentley et al. [74,97].

For *N*-demethylation of the dihydrothevinol derivate (**105b**), a modified Polonovski reaction was used, which was first applied in the field of opiate alkaloids by Scammels et al. [150]. 3-*O*-Methyl-norbuprenorphine (**107b**) was prepared via the *N*-oxide (**161**) by employing stainless steel powder as an activating agent in isopropanol. *N*-Alkylation with propargyl bromide in THF occurred selectively and provided *N*^17^-propargyl-norbuprenorphine methyl ether [103] (**109j**) with 80% yield. The 1,2,3-triazole group was achieved by application of a copper(I)-catalyzed azide-alkyne cycloaddition (CuAAC) [151] among the *N*^17^-propargyl group and the corresponding azide (R-N_3_, R = Ph, 4-Et-Ph, 4-OMe-Ph, 3,4-di-OMe-Ph, 3,4-di-Cl-Ph, 4-F-Ph, benzyl, 4-F-benzyl, 4-Br-benzyl, and 4-methyl-benzyl). The target thevinols (**162**) were obtained with a yield of 65–95%. Anti-nociceptive effects of the synthesized ligands were tested in the mice tail-flick test [149]. The most active among the thevinol derivatives (**162a–c**) was 3-*O*-demethylated with sodium *n*-heptanthiolate in DMSO to the corresponding orvinols (**163a–c**). The analgesic activities (ED_50_) of the synthesized orvinols (**163a**, R = 4-ethylphenyl (14.8 mg/kg), **163b**, 4-hydroxylphenyl (16.2 mg/kg) and **163c**: 3,4-dichlorophenyl (15.8 mg/kg)) were similar to that of norbuprenorphine methyl ether (**107b**, 17.3 mg/kg) as well *N*^17^-propargyl-norbuprenorphine methyl ether (**109j**, 24.2 mg/kg), but they were somewhat less effective than buprenorphine (**96**, *N*^17^-cyclopropylmethyl, ED_50_ (8.67)).

#### 2.3.2. Synthesis of 7β-substituted 6,14-ethenomorphinans

##### Separation of 7α- and 7β-substituted 6,14-ethenomorphinan Derivatives

The regio- and stereoselective Diels–Alder cycloaddition of methyl vinyl ketone to thebaine (**4**) resulted solely in 7-acetyl isomers in a ratio of 98 (7α):2 (7β) (isolated yields: **16a** 93%, **17a** 1.5%, Figure 43) [74], but the addition of phenyl vinyl ketone to thebaine (**4**) provided exclusively the 7α-benzoyl-isomer (**16b**, nepenthone). When acrylonitrile was used as dienophile, formation of nearly equimolar ratios of the 7α- (**16g**) and 7β-nitriles (**17g**, for structures see Figure 6) was observed. Separation of the isomers **16a**/**17a** and **16g**/**17g** was performed by fractional crystallization [74] from methanol.

The research group of Makleit [104] achieved the pilot plant development of a commercial scale buprenorphine (**96**) synthesis. Their large-scale synthesis enabled the separation of 7α and 7β epimers both in the thevinone (**16a**/**17a**) and in the dihydrothevinone (**104**/**164**) phase of the synthesis. The separations of 7α-thevinone (**16a**) and 7β-thevinone (**17a**) as well as that of 7α-dihydrothevinone (**104**) and 7β-dihydrothevinone (**164**) were accomplished by fractional crystallization of their bitartrate salts [104]. In this manner, 92 g of 7β-dihydrothevinone (**164**) [104] was obtained starting from 8.3 kg thebaine (**4**).

##### Isomerization

Only few cases of epimerization of 7(8)-substituted-6,14-ethenomorphinans have been reported. Bentley et al. [152,153] attempted to establish the 7α/7β equilibrium mixtures by base-catalyzed reversible enolization of 7α-acyl derivatives. Applying sodium hydroxide as base in methanol, rearranged *iso*-derivatives (e.g., isothevinone [153], Figure 29, structure (**D**)) were obtained instead of the desired 7β-compounds. Maurer and Rapoport [42] reported the synthesis of *N*^17^-*C*^8β^-methano-homoetorphine (**141**, Figure 38). The 8α-formyl derivative (**131**) was converted to an 8β-aldehyde (**132**) by silica gel-catalyzed epimerization.

In order to enable the production of the starting material in sufficient quantity for the scheduled multistep syntheses in the 7β-series, extensive investigations were conducted regarding the epimerization of 7-acyl derivatives. Application of bases with a weak nucleophilic character (DBU, DBN, and MTBD) in dipolar aprotic solvents (acetonitrile, DMF) led to the desired 7α/7β equilibrium mixtures [76]. The separation of the isomers was achieved as described earlier [104]. Those investigations of isomerization were extended for a number of 7-acyl-6,14-ethenomorphinan derivatives (e.g., nepenthone (**16b**), dihydronepenthone (**383**, see in Section 2.7), and the thebaine-ethyl vinyl ketone adduct (**16d**). Using 7α-thevinone (**16a**) as substrate and adding 0.1 molar equivalent of 1,8-diazabicyclo-[5,4,0]undec-7-en base in acetonitrile (reflux, 33 h) formed an equilibrium mixture with a 7α/7β ratio of 87:13. Notably, when pure 7β-thevinone (**17a**) was subjected to isomerization under similar conditions (0.1 equiv. DBU, reflux, 24 h), there resulted a 66:13 7α/7β ratio, along with the additional formation of isothevinone (ca. 1%, Figure 29, structure **D**). 7α-Nepenthone (**16b**, 0.1 equiv., in DBU, 24 h, reflux) provided only 10% isonepenthone (**364**, see in Section 2.7), see also Figure 29, structure **D**, with the benzoyl group instead of acetyl in position seven), and no formation of 7β-nepentone (**17b**) was observed. When 7α-dihydronepenthone (**383**, 0.1 equiv. DBU, reflux, 24 h) was subjected to isomerization, the equilibrium mixture contained the 7α (**383**) and 7β benzoyl (7β-**383**) derivatives in a ratio of 88:12, with no discernible formation of isodihydronepenthone. When the 7α-formyl adduct (**16c**), prepared in the reaction of thebaine (**4**) and acrolein, was subjected to isomerization, the ratio of 7α/iso compounds was 17:83, without any trace of the 7β-isomer. These product ratios were determined from the ^1^H-NMR spectrum of the unseparated product mixtures.

Acid-catalyzed equilibrium enolization of thevinones has also been reported by application of aqueous hydrochloric acid [154] (e.g., 20% HCl, 60 °C, 16 h), perchloric acid [155], or TFA, as well as further achiral and chiral acids [156]. Treatment of dihydrothevinone (**104**) with HClO_4_ (70%, 90 °C, 16 h) resulted in the 7α/7β-isomers in a ratio of 58:42. Perchloric acid-induced epimerization [155] occurred without any rearrangement to 5,14-bridged morphinane derivatives.

In 2000, Derrick et al. [155], upon performing the Schmidt reaction of *N*^17^-cyclopropylmethyl-dihydronorthevinone (**112a**, HClO_4_, NaN_3_, 70 °C, 16 h, Figure 44), observed the formation of a 2:1 mixture of 7α- (**170**) and 7β-acetylamino (**171**) derivatives. In the initial step of the Schmidt reaction, the protonation of the ketone occurs, followed by the reaction with hydrazoic acid (HN_3_) generated in situ by adding NaN_3_ to the acidic reaction mixture. The formation of the 7β-isomer (**171**) was explained by acid-catalyzed epimerization of the starting 7α-acetyl (**112a**) derivative via the enol (**168**), which resulted in a mixture of 7α- (**112a**) and 7β-ketones (**169**). A competing reaction with hydrogen azide led to a mixture of the 7-acetylamino (7α (**170**)/7β (**171**) 67/33) derivatives.

Subsequently, a series of six 7α-acyl 6,14-ethenomorphinan derivatives (Figure 45, Table 2) were subjected to perchloric acid-induced epimerization [155], and their product ratios were determined from ^1^H-NMR spectra of the unseparated product mixtures. The chemical shift values of the characteristic 5β-H signals [d (^4^*J*_5β,18_, W)] were of diagnostic importance to determine the stereochemistry of the C-7 chiral center. The chemical shifts of the 5β-H-s for 7β-acyl derivatives exceeded those of the 7α-acyl compounds [112].

Upon reaction of thevinone (**16a**) with perchloric acid at 70 °C (4.4 h), 17% of 7β-isomer (**17a**) was formed together with 6-desmethyl-thevinone. This product ratio (7α/7β, 67/17) was extremely similar to that observed when using 20% aqueous hydrochloric acid (60 °C, 16 h) (7α/7β, 82/18) [154]. When the epimer 7β-tevinone (**17a**) was treated either with perchloric acid (7α/7β, 83/17) [155] or hydrochloric acid (ca. 80/20) [154], again, extremely similar product ratios were obtained for both acids. This latter case also proved the higher stability of the 7α-thevinone (**16a**) over its 7β-epimer (**17a**). Upon applying 7α-dihydronepenthone (7α-**383**) as substrate for the HClO_4_-catalyzed epimerization, there was some or significant formation of 7β-dihydronepenthone (7β-**383**, Table 2) (7α/7β, 56/44) [155]. In contrast to that scenario, nepenthone (**16b**), the Δ^18,19^ unsaturated analogue of dihydronepenthone (**383**), was itself extremely susceptible to decomposition under acidic conditions [77,153], showing numerous rearranged derivatives, i.e., neonepenthone-A (**368**), neonepenthone-B (**369**), flavonepenthone (**371**), and nepenthene (**372**, see also Section 2.7).

##### Synthesis of 7β-analogues of Diprenorphine, Buprenorphine, and Etorphine

Makleit et al., taking the advantages of their 7β-thevione isomers in large quantities, conducted extensive investigations [104] including the preparation of numerous new 7β-orvinol analogues. For example, 7β-diprenorphine (**165a**) and its *N*^17^-substituted analogues (Figure 46, **165b–f**) were prepared via three independent reaction routes [104] from 7β-dihydrothevinone (**164**).

Subsequently, 7β-buprenorphine (**166**, Figure 43 and Figure 47), 7β-etorphine (**167a**), and 7β-dihydroetorphine (**167b**, Figure 43 and Figure 48) were synthesized from 7β-thevinone (**17a**) as well as from 7β-dihydrothevinone (**164**) [101]. For the synthesis of 7β-buprenorphine (**166**), 7β-dihydrothevinone (**164**) was reacted with six equiv. *tert*-butylmagnesium chloride. The *tert*-alcohol (**178**) formed was *N*-demethylated via the classical von Braun method [137] (1. cyanogen bromide, CHCl_3_, (**179**) 2. KOH) to yield 7β-norbuprenorphine (**180**).

Subsequently, the secondary amine was alkylated with cyclopropylmethyl bromide to yield the 7β-*N*^17^-cyclopropylmethyl derivative (**181**), which in turn was 3-*O*-demethylated with KOH in diethylene glycol at 210 °C to provide 7β-buprenorphine (**166**) in an overall yield of 7.1% from the 7β-dihydrothevinone (**164**) starting material.

In another study, Makleit et al. [101] prepared 7β-etorphine analogues (**167a–b**, Figure 48) from the corresponding 7β-acetyl compounds (**17a**, **164**) via Grignard reaction with *n*-propylmagnesium bromide (six-fold excess) followed by 3-*O*-demethylation. The overall yield of 7β-etorphine (**167a**) from 7β-thevinone (**17a**) was 4.6%, and 7β-dihydroetorphine (**167b**) was prepared from 7β-dihydrothevinone (**164**) in 7.6%. The Grignard reactions in the 7β-series were apparently more complicated than those in the 7α-series and had unsatisfactory yields (10–34%) of the desired products. Here, the Grignard reactions in question resulted in the formation of secondary alcohols from Grignard reduction, which was further complicated by the occurrence of base-catalyzed rearrangement.

#### 2.3.3. On the 6,14-etheno Bridge Functionalized Analogues

Coop et al. [157,158,159] developed a synthetic strategy for the preparation of thevinone analogues functionalized on the 6,14-etheno bridge. In their first approach [157], thebaine anion was generated from thebaine (**4**) with two equivalents of butyl lithium in tetrahydrofurane in the presence of two equivalents of cation-complexing agent TMEDA. The subsequent reaction with the triphenylsilyl electrophile (10 equiv.) resulted exclusively in 7-triphenylsilylthebaine (**183**, Figure 49) in 64% yield. Diels–Alder reaction of the latter compound (**183**) with ten equivalents of 1,4-benzoquinone (toluene reflux, 4 h) resulted in the 18-triphenylsilyl-substituted adduct (**184**) in 95% yield. The structure of the isolated product was substantiated by means of NMR spectroscopy, and X-ray structure analysis confirmed it to be the quinol (**184**) form, presumably formed by aromatization of the normal adduct.

In a second approach [158], the methylacrylate adduct of thebaine (**16e**, Figure 50) was subjected to hydroboration reaction. Treatment of (**16e**) with the borane dimethylsulfid (BMS) complex (–70 °C, 5 h), followed by reaction with hydrogen peroxide and aqueous sodium hydroxide, resulted in two products containing secondary hydroxyl groups in position 18 (**185**) as well position 19 (**186**) on the 6,14-bridge. By reaction with BH_3_, the 7α-methoxycarbonyl group was also reduced to a 7α-hydroxymethyl group. The 18*R*-hydroxy (**185**, 46%) and 19*S*-hydroxy (**186**, 43%) isomers were separated by preparative TLC, and the structure of the 18*R*-hydroxy compound (**185**) was verified by X-ray analysis. Selective protection of the least hindered 7α-hydroxymethyl group of the 19-OH compound (**186**) via benzylation provided the 7α-benzyloxymethyl derivative (**188**).

The latter was oxidized with the Dess-Martin reagent to the corresponding 19-ketone (**190**). When the 18-OH derivative (**185**) was treated under the same benzylating conditions (BnBr, NaH, DMF, −60 °C), the 18*R*-benzyloxy-7α-hydroxymethyl compound (**187**) was formed exclusively. When reacting **187** with the Dess-Martin periodinane, the 7α-fomyl-18*R*-benzyloxy derivative (**189**) was obtained in a yield of 78%.

In 2007, Coop et al. [159] reported on the hydroboration reaction of thevinone (**16a**). Treatment of thevinone (**16a**) with borane tetrahydrofurane complex followed by reaction with hydrogen peroxide led to the formation of four of the 6,14-bridge-hydroxylated thevinols (**191**–**194**, R = CH_3_, Figure 51), which were separated by means of preparative HPLC. Hydroxylation on the Δ^18,19^ double bond and reduction of the 7α-acetyl group took place simultaneously. The latter reduction yielded 20*R* and 20*S* secondary alcohols, resulting in the formation of diastereomeric pairs from the C-18 and C-19 secondary alcohols. The attempted hydroboration of 20-methylthevinol led to the formation of 6-*O*-desmethyl-20-methylorvinol.

The lack of hydroxylation was explained by steric hindrance of the Δ^18,19^ double bond introduced by the two 20-methyl groups of the tertiary alcohol substrate. 3-*O*-Demethylation of the four isolated thevinols was performed with sodium propane thiolate. The binding properties of the prepared orvinols (**191**–**194**, R = H) were determined for cloned μ, κ, and δORs from CHO cell membranes in comparison with thienorphine (**100**), which showed reduced affinity compared to **100**.

#### 2.3.4. Synthesis of Thevinone Analogues by Methylation of Its Enolate

Lewis et al. [79] succeeded in developing a method for the conversion of thevinone (**16a**) to 20-methylthevinone (**194**) via methylation of thevinone enolate. The enolization was performed by treatment with lithium diisopropylamide (LDA) in tetrahydrofurane. Upon reacting the enolate with methyl iodide for 1 h, only 13% yield was observed. Extending the reaction time to 18 h led to somewhat higher yield (24%). Analogously, the formed 20-methylthevinone (**194**, Figure 52) was converted to 20,20-dimethyl-thevinone (**195**, 76%) and the latter to 20,20,20-trimethylthevinone (**196**, 59%).

The latter compound was identical to the *tert*-butyl vinyl ketone adduct (**196**) of thebaine [160]. *N*^17^-Cycopropylmethyl-northevinone (**310**) was also converted in the same manner to its 20-methyl- (**194**, R = CPM, 98%), 20,20-dimethyl- (**195**, R = CPM, 91%), and 20,20,20-trimethyl-analogues (**196**, R = CPM, 42%), and the method was also successfully extended to numerous 7,8-constrained- and 7,7-spiro-thevinone derivatives.

#### 2.3.5. *N*-demethylation and *O*-demethylation of 6,14-ethenomorphinans

##### *N*-demethylation

In diprenorphine (**95**), buprenorphine (**96**), and numerous semisynthetic morphinan derivatives (e.g., naltrexone, NTI, *nor*BNI), the *N*^17^-methyl group of the starting morphine alkaloid, thebaine (**4**) or oripavine (**5**), is formally replaced by a cyclopropylmethyl group (or by other functional groups such as *n*-propyl, allyl, dimethylallyl, and propargyl). For the introduction of the desired *N*^17^-substituent, the corresponding *N*^17^-methyl derivative must obviously be *N*-demethylated. The *N*-demethylation of morphine alkaloids and their semisynthetic derivatives can be achieved via several different procedures, including the von Braun method (reaction with BrCN) [137], the application of chloroformates [161] or azodicarboxylic esters [162], and the Polonovski reaction [150,163].

Northevinone (**393**, Appendix A) can be prepared from thevinone (**16a**) by treatment with azodicarboxylic esters (e.g., diethyl azodicarboxylate (DEAD) or diisopropyl azodicarboxylate (DIAD)) followed by the hydrolysis of the resulting *N*^17^-desmethyl-(*N*^17^,*N*’^17^-dialkoxycarbonyl-hydrazinomethyl)-thevinone adduct (Appendix A) with pyridine hydrochloride, yielding northevinone hydrochloride (**393**·HCl), formaldehyde, and dialkyl hydrazinedicarboxylate. This reaction has been utilized for the *N*-demethylation of dihydrothevinone (**104**) [103,146], nepenthone (**16b**), dihydronepenthone (**383**) [110], and C-7 aryl-substituted nepenthone (**421a–c**) derivatives [164].

The reaction of 20-phenethyl-thevinol (**86**) with diethyl azodicarboxylate in acetone led to an unexpected result [111]. Instead of the expected secondary amine, the formation of *N*^17^,*N*’^17^-methylenebis-(20-phenethyl-northevinol) was observed, with confirmation of its structure by NMR analysis. Hydolysis of *N*^17^,*N*’^17^-methylenebis-(20-phenethyl-northevinol) with acetic acid yielded the desired norderivative. Upon reaction with formaldehyde, the *nor*-derivative of the tertiary alcohols yielded *N*-hydroxymethyl-*nor*-compounds, leaving the resulting carbinolamine to react with another secondary amine to form the *N*,*N*’-methylenebis *nor*-derivative (Appendix A).

Treatment of thevinone (**16a**) or the C-20 tertiary alcohols with excess cyanogen bromide (reflux, 12 h) provided the *N*-cyanonorderivatives [111]. The obtained compounds were converted to the corresponding secondary amines in two steps (Appendix A): first, the *N*-cyano group was hydrolyzed with 2M hydrochloric (reflux, 2 h) to the *N*-aminocarbonyl compound, and second, reaction of the urea derivative with nitrous acid yielded the desired secondary amine (e.g., *N*^17^-northevinone (**393**)). The hydrolysis of *N*^17^-cyano-northevinone could not be performed with potassium hydroxide because its sensitivity to alkali would result in an isothevinone derivative (see Figure 29). In contrast to this, the *N*-cyano-*nor*-derivatives from the tertiary alcohols could be hydrolyzed with KOH at 160 °C in diethylene glycol or, alternatively, at 200–220 °C, leading to simultaneous hydrolysis and *O*-demethylation.

Scammells et al. [150] reported that morphinans can be converted to their norderivatives by the modified nonclassical Polonovski reaction. This method involves the preparation of the hydrochloride salt of the tertiary amine *N*-oxide followed by treatment of this salt with Fe(II)SO_4_, which yields the hydrochloride salt of the secondary amine. The formation of the parent *N*-methyl morphinan by deoxygenation presents a drawback of this procedure. After reaction, iron salts were removed by extraction with EDTA, whereupon normorphinan and the parent compound were separated by column chromatography. The hydrochloride salt of thevinone *N*-oxide was reacted in a Polonovski reaction in the presence of FeSO_4_, leading to isolation of northevinone (**393**) in 44% yield after column chromatography. More recently, Scammells et al. [165] found that treatment of a morphinan *N*-oxide hydrochloride salt with iron powder also yielded the expected *nor*-compounds. The re-formation of the deoxygenated tertiary amines (parent compounds) was also observed. The reaction of thevinone *N*-oxide hydrochloride in chloroform afforded 44% northevinone (**393**) and 26% thevinone (**16a**). When performing the reaction in isopropanol, the ratio of northevinone (**393**) to thevinone (**16a**) was 15:76.

More recently, the research group of Hudlický developed several new routes for the synthesis of buprenorphine (**96**). The application of palladium-catalyzed *N*-demethylation and -acylation protocol was reported [136] during the synthesis of buprenorpine (**96**), starting from 20*S*-*tert*-butyl-dihydrothevinol (**105b**). Other research showed that oripavine cyclopropylmethyl quaternary salt [134,135] can be *N*-demethylated with sodium *tert*-dodecanethiolate to *N*-cyclopropylmethyl-nororipavine. Machara et al. [135] reported an improved synthesis of buprenorphine (**96**) via a palladium-catalyzed *N*-demethylation and *N*-acylation reaction sequence. Here, the tertiary alcohol 20*S*-*tert*-butyl-dihydrothevinol (**105b**, Appendix A) was treated with Pd(OAc)_2_/Cu(OAc)_2_/Ac_2_O reagent in the presence of oxygen or air. Under these conditions, the acetamide was obtained in good yield. The acetamide was converted to the corresponding secondary amine by reaction with the Schwartz reagent (Cp_2_ZrHCl). *N*-Alkylation of the secondary amine with cyclopropylmethyl bromide yielded buprenorphine-3-*O*-methylether. Finally, *O*-demethylation was achieved with sodium dodecanethiolate, an approach serving as an alternative to the use of KOH in diethylene glycol. The *N*-demethylation and *N*-acylation protocol was also studied using cyclopropanecarboxylic anhydride instead of acetic anhydride (Appendix A). Here, the tertiary alcohol was converted to the *N*^17^-cyclopropylcarbonyl amide with Pd(OAc)_2_ catalyst in dioxane at 100 °C. The amide was reduced with lithium aluminum hydride in tetrahydrofuran resulting in 3-*O*-methyl buprenorphine (**109b**).

Werner et al. [134,136] elaborated another novel buprenorphine synthesis utilizing oripavine as the starting material (Appendix A). The reaction of oripavine (**5**) with methyl vinyl ketone resulted in orvinone, which was subsequently hydrogenated as the tartrate salt in an aqueous solution using Pd-C catalyst; the hydrogenation reaction was performed at 80 °C and 1 bar. The resulting dihydroorvinone was then reacted with ethyl chloroformate in order to protect the phenolic hydroxyl group. The subsequent reaction of the ethoxycarbonyl-protected orvinone with *tert*-butylmagnesium chloride resulted in the corresponding C-20 tertiary alcohol. The palladium-catalyzed *N*-demethylation and *N*-acylation reaction of the tertiary alcohol with cyclopropanecarboxylic anhydride provided the *N*-cyclopropylcarbonyl derivative (**109b**). Compound **109b** was reduced with sodium *bis*(2-methoxyethoxy)aluminum hydride (Red-Al, Vitride) in THF to afford buprenorphine (**96**).

Werner et al. [134,136] also reported another synthesis of buprenorphine via *N*-demethylation of oripavine quaternary salts (Appendix A). Oripavine (**5**) was converted to a mixture of diastereomeric quaternary salts (*S*/*R* = 2.6/1) with cyclopropylmethyl bromide at 80 °C in dimethyl formamide. The mixture of diastereomers was heated with *tert*-dodecanethiol as a nucleophilic reagent using sodium ethoxide as base in dimethyl sulfoxide at 80 °C. The desired *N*^17^-cyclopropylmethyl-nororipavine was obtained in high (77%) yield. This compound served as starting material for the synthetic sequence. However, the Grignard reaction of 3-ethoxycarbonyl-*N*^17^-cyclopropylmethyl-dihydronororvinone with *tert*-butylmagnesium chloride provided buprenorphine (**96**) in only 30% yield, and two by-products were also isolated. Despite these properties, the method presents the advantage of not requiring hazardous cyanogen bromide.

##### 3-*O*-Demethylation

An important and crucial transformation of the synthesis of orvinols is the 3-*O*-demethylation. The free 3-phenolic hydroxyl group was long thought to be essential for the pharmacological effect of morphinan ligands. Contrary to this expectation, a number of compounds with high OR binding affinities have been developed by the research group of Neumeyer [90,166,167] in the last two decades by bioisoteric replacement of the 3-phenolic hydroxyl group with an aminothiazole moiety, or by modification to oxazole, carbamate, urea, aminobenzyl, and aminophenyl derivatives. In general, the 3-*O*-demethylation of morphine alkaloids has proven difficult; the slow rate of demethylation call for long reaction times and harsh reaction conditions often lead to decomposition, tedious work-up, and poor chemical yields. In general, 3-*O*-demethylation of morphinans may be successfully carried out using one of a variety of reagents, such as pyridine hydrochloride, hydrobromic acid, boron tribromide, diphenylphosphine-*n*-BuLi (diphenylphosphide ion) [168], C_1-4_ alkyl or arylthiol-alkali alkoxide systems (thiolate anions) [169], hard acid-soft nucleophile systems (e.g., methane sulfonic acid-methionine) [170], potassium hydroxide [110,111], diisobutylaluminumhydride (DIBAL) [103,171], or lithium tri-*sec*-butylborohydride (L-selectride) [172,173]. The 3-*O*-demethylation of thevinol derivatives should be performed under basic conditions due to the acid-sensitivity of the substrates. Thevinols are prone to acid-catalyzed dehydration, enol ether hydrolysis, and rearrangement, e.g., to 5,14-bridged thebainone derivatives, anhydro-20-alkylthevinols, or 14-alkenylcodeinones [153]. Additionally, thevinols and orvinols can undergo acid-catalyzed rearrangement, with elimination of methanol and concurrent formation of a tetrahydrofuran ring at C6-C7 to yield furano [2′,3′:6,7]codides and furano [2′,3′:6,7]morphides, respectively [144,174].

Strong nucleophiles such as sodium propanethiolate in DMF or HMPA, as well potassium hydroxide in diethylene glycol (200–210 °C), are the most frequently applied reagents for the 3-*O*-demethylation of thevinols under harsh reaction conditions. 3-*O*-Demethylation of secondary or tertiary alcohols derived from thevinone (**16a**) or dihydrothevinone (**104**) was performed by heating the free bases with KOH in diethylene glycol at 200–220 °C [111]. This demethylation affected only the methoxy group at position three, whereas the 6-*O*-methyl group remained unchanged. In some cases, the 3-*O*-demethylation of tertiary alcohols (**391e–f**, **395e–f**, see in Section 2.7) deriving from nepenthone (**16b**) and dihydronepenthone (**383**) (if the basic nitrogen bears allyl, dimethylallyl or propargyl groups) resulted in low yields or failed due to extensive temperature-dependent decomposition when using the latter nucleophile (KOH) [110]. Diisobutylaluminum hydride [103] was also reported as an effective 3-*O*-demethylation reagent acting on 20-alkylthevinols. The research team of Rapoport [51,175] applied sodium propanethiolate (prepared from 1-propanethiol and sodium hydride) for the selective 3-*O*-demethylation of 6,14-ethenomorphinans in DMF. Husbands et al. [114,131,146,176] achieved 3-*O*-demethylation reactions of Bentley compounds with sodium propanethiolate in hexamethyphosphoramide at 120 °C.

L-selectride was first reported by Coop et al. [172,173] to be an efficient and generally applicable 3-*O*-demethylating agent for morphine alkaloids. The reaction calls for mild conditions, and the desired 3-phenolic compounds were obtained in higher yield and purity relative to that obtained by the utilization of other reagents. The L-selectride reagent was first successfully applied for the selective 3-*O*-demethylation of thebaine (**4**) to oripavine (**5**, 35%) [172], and later proved to be efficient for the conversions of oxycodoneindole to oxymorphindole and 20-methylthevinol to 20-methylorvinol (73%). Phenethyl-orvinol (**99**, PEO) was prepared from 3-*O*-methyl-PEO (**86**) in 60% yield using L-selectride, although the attempted transformation of 6-*O*-(2-fluoroethyl)-6-*O*-desmethyl-phenethyl-thevinol (FE-DPET) to the corresponding orvinol (FE-PEO) failed [177].

There are some examples of successful application of boron tribromide for selective 3-*O*-demethylation of non-acid-sensitive 6,14-ethenomorphinans (Appendix A, 6-demethoxythevinone (**24**) [178], 7α-amino- (**295**) [179], and 7α-aminomethyl-derivatives (**336**) [180]). Recently, Machara and Hudlický [181] provided an excellent survey on *N*^17^- and 3-*O*-demethylation methods of morphinans and other opiate-derived pharmaceuticals in their comprehensive review.

##### 6-*O*-Demethylation

Contrary to the 3-*O*-demethylation of the 6,14-ethenomorphinans, the 6-*O*-demethylation is less extensively investigated, and 6-*O*-desmethyl-orvinols had not been available before the 1980s. Nonetheless, 6-*O*-desmethyl-diprenorphine and 6-*O*-desmethyl-buprenorphine are known impurities of the diprenorphine and buprenorphine syntheses. 6-*O*-desmethyl orvinol derivatives are formed as minor by-products during the 3-*O*-demethylation of thevinols under harsh conditions (KOH, diethylene glycol 210 °C), together with other degradation products. In 1986, Kopcho and Schaeffer [182] reported the unexpectedly selective 6-*O*-demethylation of 7α-aldoxime- and 7α-aminomethyl-6,14-ethenomorphinan derivatives with 15 equiv. LiAlH_4_ in tetrahydrofuran containing halogenated co-solvent (CCl_4_, 4 equiv.). They postulated a six-membered cyclic aluminum complex to play an important role in this unusual *O*-dealkylation. Lever et al. extended this exploration advantageously for the syntheses of 6-*O*-desmethyl-diprenorphine [183] and 6-*O*-desmethyl-buprenorphine [184], compounds that are intermediates of the precursor synthesis for the molecular imaging PET tracers [6-*O*-methyl-^11^C]diprenorphine and [6-*O*-methyl-^11^C]buprenorphine, respectively. Luthra et al. [185] also synthesized the 6-*O*-desmethyl analogues of diprenorphine (**95**) and buprenorphine (**96**) as intermediates for the synthesis of PET precursors.

In 1999, Lewis et al. [186] reported the selective 6-*O*-demethylation of thevionols (e.g., the tertiary alcohol 20*R*-ethyl-thevinol and the secondary alcohols 20*S*- and 20*R*-thevinol) and orvinols (e.g., buprenorphine (**96**)). The substrates were reacted with six eq. LiAlH_4_ in a non-coordinating solvent such as toluene to obtain the corresponding 6-*O*-demethylated derivative in 31–81% yields. For the successful 6-*O*-demethylations, an OR (R = H, alkoxy) or an NH_2_ group was required in the 20 position. Shults et al. [187,188] investigated the *O*-demethylation reactions of thebaine-succinimide adducts with boron tribromide, in which demethylation of alkyl methyl ether (6-*O*-demethylation) and/or demethylation of aryl methyl ether (3-*O*-demethylation) was observed, depending on the character of the substrate. Coop et al. [159] reported on the unusual 6-*O*-demethylation of 20-methylthevinol through hydroboration (BH_3_/H_2_O_2_), whereas Husbands et al. studied the reactions of buprenorphine (**96**) with *N*-halosuccinimide derivatives in an acidic milieu [189]. Using 1.2 eq. NBS as reagent, bromination occurred in position two to obtain 2-bromobuprenorphine. An analogous reaction of **96** with NCS also resulted in the 6-*O*-demethylated derivative instead of the desired 2-chlorobuprenorphine.

The first biochemical investigations of 6-*O*-desmethyl-orvinols performed by Szücs et al. [190] showed an increased binding affinity of these ligands to μ-opioid receptors compared to the parent 6-*O*-methyl compounds; affinities were in the range: 0.01–0.21 nM (6-*O*-desmethyl-diprenorphine: K_i_ (μ-OR) = 0.03 nM, δ/μ = 44.6, κ/μ = 0.72); 6-*O*-desmethyl-buprenorphine: K_i_ (μ-OR) = 0.21, δ/μ = 71.9, κ/μ = 1.18; 6-O-desmethyl-dihydroetorphine: K_i_ (μ-OR) = 0.01 nM, δ/μ = 177.3, κ/μ = 56.5; and 6-O-desmethyl-phenethyl-orvinol: K_i_ (μ-OR) = 0.04 nM, δ/μ = 47.4, κ/μ = 0.5).

### 2.4. Hetero Diels–Alder (HDA) Reactions

#### 2.4.1. Nitrosocarbonyl Dienophiles

It was Kirby and Sweeny [191] who first reported the application of nitrosocarbonyl-alkanes and -arenes (RCONO) as dienophiles in a hetero Diels–Alder (HDA) reaction. These highly reactive transient species were synthesized by oxidative cleavage of hydroxamic acids.

When thebaine (**4**) was treated with benzohydroxamic acid (PhCONHOH) in the presence of tetraethylammonium periodate (Et_4_NIO_4_) in aqueous AcOH-NaOAc (pH ~ 6) solution, the nitroso intermediate (PhCONO) formed in situ reacted with the 6,8-diene system, and the hydroiodide salt of the corresponding HDA adduct (**197**, R = Ph, Figure 53) was isolated in 97% yield. 

When thebaine (**4**) was reacted with the HDA adducts (**202**) of 9,10-dimethylantracene (9,10-DMA, **201**) and nitrosocarbonylbenzene (generated in situ from benzoxyhydroxamic acid with Et_4_NIO_4_) or nitrosocarbonylmethane (from acetohydroxamic acid and Et_4_NIO_4_) in hot benzene, the thermo retro-Diels–Alder (rDA) cleavage regenerated the reactive species (**200**, RCONO, R = Me, Ph) and caused its intermolecular transfer into the thebaine 6,8-diene system [191,192]. Accordingly, the appropriate thebaine-nitroso adducts (**197**, R = Me, Ph) and 9,10-dimethylanthracene (**201**) were ultimately formed. Hydrolysis of **197** (R = Ph) with boiling HCl/MeOH resulted in 14-hydroxyaminocodeinone (**198**).

Formation of nitrosoformates (ROCON=O, R = *tert*-BuO, PhCH_2_) by heating carbonylnitrenes (ROCON_3_) in DMSO was proven indirectly [193]. When the nitrenes were thermally decomposed in the presence of thebaine (**4**) or 9,10-dimethylantracene (9,10-DMA, **201**), the corresponding HDA adducts were isolated from the reaction mixture. Thermal decomposition of benzyl azidoformate (PhCH_2_OCON_3_) at 130 °C in DMSO in the presence of thebaine (**4**) gave the hetero Diels–Alder cycloadduct **203** (R = PhCH_2_, Figure 54) in 66% yield. Upon reacting *N*-benzyloxycarbonylhydroxylamine (*N*-Z-hydroxylamine) with tetraethylammonium periodate in dichloromethane at 0 °C in the presence of thebaine (**4**), the same adduct (**203**, R = PhCH_2_O) was isolated in 76% yield.

Decomposition of *tert*-butyl azidoformate (*tert*-BuOCON_3_) in DMSO at 115 °C in the presence of thebaine (**4**) provided the hetero Diels–Alder adduct (**203**, R = *tert*-Bu) in 84% yield. When *N*-*tert*-butoxycarbonylhydroxylamine was oxidized with Et_4_NIO_4_, the adduct **203** (R = *tert*-Bu) was obtained in 74% yield.

Kirby et al. [194] reported the syntheses of thebaine-nitrosoarene cycloadducts (**204**, Figure 54). Following the β-face approach, the corresponding 6,14-*N*-aryl-epoxyimino adducts **204** (Ar = phenyl, 4-chlorophenyl, 3-methoxyphenyl, 4-nitrophenyl, 4-methylphenyl, 3-methyphenyl, and 4-dimethylaminophenyl) were prepared in yields in the range of 65–97%. According to NMR analysis, the synthesized hetero Diels–Alder adducts dissociated by retro-Diels–Alder reaction in 0.5 M CDCl_3_ solutions at 35 °C. The EWG substituents on the aromatic ring stabilized the adducts (e.g., no retro Diels–Alder reaction for the 4-nitro group was observed at 35 °C), and adducts with EDG substituents on the phenyl ring displayed higher dissociation ratios (Ph, 10%; 3-methylphenyl, 15%; 4-methylphenyl, 35%; and 4-methoxyphenyl, 45%) according to the ^1^H-NMR spectra. The thebaine-nitrosobenzene HDA adduct (**204**, Ar = Ph) was converted to 14-phenylamino-dihydrocodeinone (**206**) via 14-(*N*-hydroxyphenylamino)codeinone (**205**) by acidic hydrolysis followed by catalytic hydrogenation.

In 1980, Schwab [195] synthesized *N*^17^-methyl and *N*^17^-cyclopropylmethyl-14-(arylhydroxyamino)dihydrocodeinones (**206a–d**) starting from thebaine (**4**) and *N*^17^-cyclopropylmethyl-northebaine (**9d**, Figure 55). The morphinandienes (**4**, **9d**) were reacted with nitrosobenzene or 4-fluoro-nitrosobenzene in dichloromethane to readily (room temperature, 5 min) provide the HDA adducts (**204a–d**) with yields in the range of 57–91%.

The adducts (**204a–d**) were hydrolyzed with 1M HCl to obtain the corresponding 14-(arylhydroxyamino)codeinones (**205a–d**). The Δ^7,8^ double bond of these derivatives was saturated under heterogenous catalytic conditions to obtain the 14-(arylamino)dihydrocodeinones (**206a–d**, 22–57%). When 14-arylhydroxyamines (**205a–d**) were treated with NaOMe in methanol, they readily rearranged to 5,14-bridged thebainone derivatives (**207a–d**). 14-(Hydroxyamino)codeinone (**198**) was prepared directly from thebaine (**4**) with benzohydroxamic acid by oxidizing with Et_4_NIO_4_ and subsequent acidic hydrolysis in 47% yield. Pharmacological characterizations of compounds **205a–d** and **206a–d** were performed in the mouse tail-flick, writhing, and Straub tail assays [195]. Derivatives with *N*^17^-methyl substituent displayed about one-tenth to one-third of the analgesic potency of morphine. Compounds with an *N*^17^-cyclopropylmethyl group were μOR antagonists, with activity less than 2% that of naloxone.

In 1985, Corrie et al. [196] described a comprehensive investigation of C-nitrosocarbonyl compounds with numerous cyclic dienes. Nitrosocarbonylbenzene, generated in situ by oxidation of benzohydroxamic acid with Et_4_NIO_4_ in dichloromethane, in the additional presence of cyclopentadiene resulted in the Diels–Alder adduct 3-benzoyl-2,3-oxazabicyclo [2.2.1]hept-5-ene in 74% yield. Thermal retro-Diels–Alder reaction of this cyclopentadiene cycloadduct occurred in benzene at 60 °C. When performing the rDA reaction in the presence of equimolar amount of thebaine (**4**, benzene, 60 °C, 6.5 h), the HDA adduct (**197**, R = Ph, Figure 53) was obtained in 69% yield.

In 1985, Kirby et al. [197] reported their extensive investigations of C-nitrosoformate esters (*O*-nitrosocarbonyl compounds, ROCONO, R = benzyl, 2,2,2,-trichloroethyl, *tert*-butyl, 2-(toluene-4-sulfonyl)ethyl, Figure 56) with a large variety of dienes (1,3-butadiene, cyclopentadiene, ergosteryl acetate, 9,10-dimethylanthracene, 2,3-dimethylbutadiene, bicyclohexenyl, and thebaine). Thebaine 6,14-epoxyimino derivatives (**203**) were prepared with high yield (72–76%). Cleavage of the cycloadduct (**203**, R = *tert*-butyl) with methanolic hydrogen chloride resulted in 14-hydroxyaminocodeinone (**198**, 90%). Treatment of the *N*-tosylethyl compound (**203**, R = OCH_2_CH_2_Tos) with 1,5-diazabicylclo [4.3.0]non-5-ene (DBN) in benzene at room temperature for 1 h resulted in the epoxyimino derivative (**208**). Attempted crystallization of this acetal (**208**) failed due to quick rDA decomposition to thebaine (**4**) by loss of HNO. Thermal transfer of benzyl nitrosoformate to thebaine (**4**) from *N*-benzyloxycarbonyl-9,10-dihydro-9,10-dimethyl-9,10-epoxy-iminoantracene (**202**, PhCH_2_ group instead of Ph) in benzene (reflux, 5 h) yielded **203** (R = OCH_2_Ph) quantitatively.

Sheldrake and Soissons [198] generated a series of acyl nitroso dienophiles by oxidation of hydroxamic acids with benzyltrimethylammonium periodate (BnMe_3_NIO_4_). Thebaine (**4**) was reacted with the transient heterodienophiles to obtain regioselectively the corresponding 6*O*-14*N*-oxazine adducts (**203**, Figure 56) in good (41–77%) yields. By reduction of the synthesized HDA adducts (**203**) using two equivalents of the single-electron reducing agent samarium-(II) iodide, there was selective opening of the C-ring of the 4,5-epoxymorphinan structure, introduced by cleavage of the C5-C6 bond. The hexahydrobenzazocine-type product (**209**, R = Ph) was isolated with 77% yield.

In 2007, Miller et al. [199] investigated the iminonitroso Diels–Alder reaction of the diene system present in natural products with numerous C-nitroso dienophiles. For example, thebaine (**4**) was reacted with 6-methyl-2-nitrosopyridine (one equiv., CH_2_Cl_2_, 0 °C, 10 min) to obtain the corresponding adduct (**210**, Figure 56) in quantitative yield.

Kirby and McLean [200] presented an elegant procedure for the synthesis of 14-aminocodeinone (**214**, Figure 57) via an HDA adduct (**211**) of thebaine in a four-step synthesis. First, thebaine (**4**) was treated with 2,2,2-trichloroethyl nitrosoformate (prepared from 2,2,2-trichloroethyl *N*-hydroxycarbamate with sodium periodate) to yield a hetero Diels–Alder adduct (**211**) with an *N*^8^-Troc substituent and a 6,14-epoxyimino bridge in 82% yield. Next, the adduct **211** was converted into the ethylene acetal (**212**) by treatment with ethylene glycol containing hydrogen chloride. The cleavage of the *N*-Troc group was then performed by reduction with Zn/NH_4_Cl or ammonium carbonate in methanol to yield the ethylene ketal (**213**). Hydrolysis of the acetal (**213**) with methanolic hydrochloric acid led to 14-aminocodeinone (**214**).

Aiming to synthesize potent and orally active opioid antagonists with luteinizing hormone (LH)-stimulating properties (in the context of opioid regulation of sex steroid production), Révész et al. [201] prepared a series of *N*^17^-cyclopropylmethyl-14-alkylated-morphinans and their 4,5α-epoxy analogues (Figure 58). For the introduction of a 14-alkyl substituent into the morphinan skeleton, they established a new method. Regio- and stereoselective hetero Diels–Alder reaction (HDA) of *N*^17^-cyclopropylmethyl-northebaine (**9d**) with thioaldehydes (R-CHS, R = H, Me, Et, and Ph, generated in situ from thiosulfinates) resulted in the cycloadducts (**215a–d**) in 39–74% isolated yields. Treatment of the adducts (**215a–d**) with 48% hydrobromic acid resulted in rearrangement to 5,14-thiomethano-bridged-thebainone derivatives (**216a–d**) in high (84–90%) yields. Desulfurization with Raney-Ni (MeOH, reflux 23 h) was only possible for the hetero Diels–Alder adduct **215d** (R = Ph), which resulted in 14-benzyl-*N*^17^-cyclopropylmethyl-dihydrothebainone (**217**) in low isolated (25%) yield. Pharmacological screening in mice showed [201] that the synthesized ligands were primarily μ-OR antagonists with low κ-OR affinity but lacked the desired LH-stimulating properties. The authors concluded that μ-OR and κ-OR antagonistic properties are both necessary for potent LH stimulation.

In 1991, Kirby and Sclare [202] reported the synthesis of a series of 20*R*- and 20*S*-alkyl-8-thiathevinols (**219**, Figure 59). 8-Thiathevinone (**218**) was prepared in a hetero Diels–Alder reaction from thebaine (**4**) and 2-oxopropanethial (generated in situ from the Bunte salt: sodium *S*-(2-oxopropyl) thiosulfate, AcCH_2_SSO_3_Na, and triethylamine) in a yield of 42%. When thebaine (**4**) was reacted with the thiosulfonate, *S*-2-oxopropyl 4-methylbenzene sulfonothioate (AcCH_2_SSO_2_Tol, three equiv.), in benzene in the presence of triethylamine (three equiv.) and calcium chloride dihydrate (three equiv.), the cycloadduct (**218**) was obtained in 75% yield.

Grignard reaction of 8-thiathevinone (**218**) with alkylmagnesium bromides (R = Me, Et, *n*-Pr, *n*-Bu, *n*-Pe, Hex) did not proceeded stereoselectively, unlike the case of its carbon analogues (thevinone (**16a**) or dihydrothevinone (**104**)). Chromatographic separation resulted in the respective 20*R*- and 20*S*-thiathevinol (**219**) diastereomers. Reaction of 8-thiathevinone (**218**) with alkyllithium reagents resulted in only low yields (2–14%). Similar to the Grignard reactions in the thevinone series, the reactions were complicated by Grignard reduction and also by base-catalyzed rearrangements. Agonist potencies were determined [202] in vitro in GPI (guinea pig ileum preparations) relative to normorphine. Comparing the results with thiathevinols to those of the C-8 carbon analogues, it seemed that the former compounds were much less potent than the corresponding thevinols. The μOR potency of the 20-alkyl-thiathevinols (**219**) in vitro depended on the C-20 absolute configuration and on the 20-alkyl chain length. The highest potency was observed for 20*R*-pentyl-8-thiathevinol (**219**, R = pentyl), which was quasi equipotent with normorphine.

A research group of the Hannam University investigated the Diels–Alder reactions of thebaine (**4**) with various fluorine containing dienophiles [203] (Figure 60). The reaction with 2-fluoroacrolein (two equiv., benzene, 50 °C, 18h) resulted in the formation of the normal Diels–Alder adducts (75%, **220** and **221**), as well as a hetero Diels–Alder adduct (**223**). The 7β-fluoro-7α-formyl to 7α-fluoro-7β-formyl product ratio was 88:12, as determined by integration of the formyl signals in the ^1^H-NMR spectrum.

The authors were not successful in separating the stereoisomers by HPLC. There was no detectable formation of 8-substituted regioisomers of the normal Diels–Alder addition, but there was evidence for an anomalous reaction, i.e., a hetero Diels–Alder reaction. Thus, a by-product with a 1-fluoro-vinyl substituent in position 8α was isolated in 11% yield. To clarify the effect of the fluorine atom in hetero Diels–Alder reactions, thebaine (**4**) was allowed to react with trifluoroacetaldehyde (10 equiv., benzene, 50 °C, 24 h). This resulted in the sole formation of the hetero Diels–Alder adduct (**224**, 91%).

Jeong et al. described hetero Diels–Alder reactions of thebaine (**4**) with α-fluorinated aldehydes [204] (Figure 61). The reaction of **4** with perfluoroaldehydes (2,2,2-trifluoracetaldehyde, 2,2,3,3,3-pentafluoropropanal) in a sealed tube (benzene 50 °C, 24 h) resulted in the corresponding hetero Diels–Alder adducts in high yields (8α-CF_3_ (**224a**)/91% and 8α-CF_2_CF_3_ (**224b**)/86%). Heating of the thebaine-trifluoroacetaldehyde hetero Diels–Alder adduct (**224a**) in benzene at 50 °C led to the regeneration of thebaine (**4**, retro Diels–Alder reaction). The treatment of the hetero Diels–Alder adducts (**224a–b**) with conc. HCl-tetrahydrofuran (1:10, 25 °C, 7 h) afforded 14-substituted-codeinone derivatives (**225a–b**, 88% and 77%). The C-6 keto group of the codeinone derivative (**225a**, R = CF_3_) was reduced with sodium borohydride/cerium chloride (MeOH, 25 °C) to obtain a mixture of secondary alcohols. The formed products were separated by column chromatography to yield the 14-substituted-codeine (**226**, 70%) and 14-substituted-isocodeine (**227**, 22%) derivatives. The stereoselectivity of the reduction was explained by sterical differences governing the approach of the hydride ion from either the top or bottom face.

#### 2.4.2. Azadienophiles

Azadienophiles belong to the most reactive dienophiles. Beside a [4+2] cycloaddition of the azadienophile to the diene system of the ring-C, an attack of the aza-nitrogen to the *N*^17^-methyl group is also to be expected. Bentley and Hardy [111] reported that reaction of thebaine (**4**) with dimethyl azodicarboxylate did not result in the expected Diels–Alder cycloadduct. Merz and Pook [205], researchers of the pharmaceutical company Boehringer Ingelheim, were the first to systematically investigate reaction of thebaine (**4**) with azadienophiles, primarily with azodicarboxylic acid alkylesters. Diethylazodicarboxylate (DEAD) proved to be an effective reagent to convert *N*^17^-methyl-morphinans to their corresponding *N*^17^-desmethyl derivatives (e.g., thebaine (**4**) → northebaine (**230**)) [162,206]. Reaction of thebaine (**4**) with one equivalent of diethyl azodicarboxylate (DEAD) gave the product *N*^17^-(*N*,*N*’-diethoxycarbonyl-hydrazinomethyl)-*N*^17^-demethyl-thebaine (**228**, Figure 62), which yielded *N*^17^-northebaine (**230**) upon further treatment with aqueous NH_4_Cl or pyridine hydrochloride. Upon using two equivalents of DEAD and applying an acidic work-up, formation of 14,17-(*N*,*N*’-diethoxycarbonyl-hydrazomethyl)-norcodeinone (**229**) was observed. Merz and Pook suggested that the norcodeinone derivative (**229**) originated from the (hypothetical) Diels–Alder adduct, formed from (**228**). Following rearrangements and acid hydrolysis, the hexacyclic norcodeinone (**229**) was isolated in a yield of 57%.

In the early 1970s, Hromatka and Sengstschmied [207] investigated the reactions of thebaine (4) with cyclic azaoxo dienophiles (Figure 63). The dienophiles 4,4-diethyl-pyrazolidine-3,5-dione, 3-indazolone, PTAD, 1,4-dihydropyridazine-3,5-dione, and 1,4-dihydrophthalazine-1,4-dione were generated in situ from their hydrazo precursors by oxidation with lead tetraacetate (LTA) or with *tert*-butylhypochlorite. This resulted in the corresponding thebaine HDA adducts of the β-face approach (**231**–**234**, **235a**). The saturation of the Δ^18,19^ double bond of the adducts (**231**, **232**, **235a**) was feasible at room temperature and atmospheric pressure (H_2_, Pd-C, and EtOH).

In 1973, Giger et al. [208] noticed the acid sensitivity of the thebaine-PTAD adduct (**235a**, Figure 64) and its susceptibility to rapid rearrangement under mild conditions. Attempted preparation of the HCl addition salt of **235a** thus led to the formation of the 14-(1-(4-phenylimidazolidine-3,5-dione))-substituted-codeinone (**236**), recrystallization of which from ethanol resulted in a 5,14-bridged-thebainone (**237**) derivative.

In 1993, Pindur and Keilhofer [69] reinvestigated the reaction of thebaine (**4**) with numerous dienophiles. The cycloaddition of PTAD to thebaine (**4**) was successfully performed in dichloromethane at −70 °C with a yield of 55%. In 1996, Marton et al. [82] studied the cycloaddition of azadienophiles (diethyl azodicarboxylate and 4-phenyl-4*H*-1,2,4-triazoline-3,5-dione (PTAD, Cookson reagent)) to morphinandienes (Figure 65). Cycloaddition of 4,5α-epoxy-morphinan-6,8-dienes (thebaine (**4**), *N*^17^-formyl-northebaine (**9a**), 6-demethoxythebaine (**11b**), 7-chloro-6-demethoxythebaine (**11k**), and 7-bromo-6-demethoxythebaine (**11l**)) with PTAD at room temperature resulted in the corresponding β-face adducts in high yields (**235a** (80%), **235b** (79%), and **235c** (92%)) [82].

Reaction of morphinandienes with an opened ring-E (β-dihydrothebaine (**10a**), 4-acetoxy-β-dihydrothebaine (**10b**)) with PTAD led to the formation of α-face (exo) adducts [**238a**, (43%), **238b** (67%), Figure 66]. Formation of significant by-products (**239**, 23%, 7%) resulted from an electrophile substitution reaction of PTAD with the major exo adducts (**238a–b**). Retro-Diels–Alder reaction of **235a** and **235b** was also achieved in polar-aprotic solvents in the presence of bases with only slight nucleophilic character. Such results are among first examples for the retro-Diels–Alder reactions of morphinan-diene Diels–Alder adducts [82].

Berényi et al. [58] synthesized 6-azido-6-demethoxythebaine (**11j**, Figure 67), which proved to be heat-sensitive and susceptible to acidic conditions. In brief, 6β-azido-14-hydroxycodeine (**240**), prepared from thebaine (**4**) by applying the method of Makleit et al. [209] was reacted with phosphorus tribromide to yield 6β-azido-7β-bromodeoxyneopine (**241**). In the latter compound, the relative orientation of the 6α-hydrogen and 7β-bromo substituents is *trans*-diaxial, a geometry that favors the elimination of HBr. This occurred easily in the presence of KO*t*Bu in ethanol (room temperature, 30 min) to yield **11j** in 75% yield. The structure of the 6-azido-6,8-diene (**11j**) was verified from its ^1^H NMR and IR spectra. 6-Azido-6-demethoxythebaine (**11j**), which contains a vinyl azide substructural unit, is unstable and decomposes within a matter of days at room temperature. Its diene structure was additionally verified by a cycloaddition reaction. However, treatment of **11j** with methyl vinyl ketone at 80 °C led to uncontrollable decomposition, without giving any isolable cycloadditions product. Contrary to this, when **11j** was reacted with the highly reactive azadienophile PTAD at room temperature (20 min), the adduct of the β-face approach (**34c**) was isolated with 71% yield. The azidodiene (**11j**) was also transformed to 6-isothiocyanato-6-demethoxythebaine (**11i**) with CS_2_ and triphenylphosphine (reflux, 2 h, 51%). The latter compound was easily converted to the corresponding PTAD adduct (**34b**, 80%). Reacting 6-thiocyanato-6-demethoxythebaine (**11h**) with PTAD under similar conditions resulted in the product of the β-face addition (**34a**) in 72% yield.

#### 2.4.3. Thebaine HDA Adducts for the Synthesis of Oripavine and Heroin Vaccine Haptens

In the following section, we present two examples that highlight the potential medical and chemical applications of thebaine hetero Diels–Alder adducts in light of the current wave of opiate addiction. In 1935, Konowalowa et al. [115] achieved the first isolation of the morphine alkaloid oripavine (**5**) from *Papaver orientale*, a poppy species native to the South Caucasus and Transcaucasia. A direct conversion of thebaine (**4**) to oripavine (**5**) by selective 3-*O*-demethylation seemed unfeasible. Thebaine (**4**) contains a conjugated 6,8-diene system in ring-C, and the enol ether functionality in position six is known to be e sensitive to acidic media [39]. The preparation of oripavine (**5**) from thebaine (**4**) failed by application of commercially available, and for the 3-*O*-demethylation of morphinans, commonly used, reagents. Enol ether cleavage followed by rearrangements resulted in uncontrollable decomposition of the thebaine (**4**) molecule. Barton et al. [210] presented the application of 4-phenyl-4*H*-1,2,4-triazoline-3,5-dione (PTAD) for the protection of the ergosterol diene system. Analogously, the protection of the morpinan-6,8-diene (**4**) with PTAD [69,82,208] and the regeneration of the diene system (rDA reaction) proved to be possible [82], but the attempted 3-*O*-demethylation of the PTAD-protected thebaine (**235a**) was not feasible due to its sensitivity under acidic conditions [208]. Utilizing lithium-tri-*sec*-butyl-borohydride (L-selectride) as a demethylating agent, Coop et al. [172,173] reported the first successful conversion of thebaine (**4**) to oripavine (**5**).

Recently, Hudlický et al. [211] elaborated two independent methods (Figure 68) for the synthesis of oripavine (**5**) from thebaine (**4**). In the first approach, the conjugated 6,8-diene system of thebaine (**4**) was protected through its iron-tricarbonyl complex (**242**). To this end, thebaine (**4**) was reacted with iron pentacarbonyl under irradiation with ultraviolet light in benzene to achieve **242** in a yield of 95%. Surprisingly, the thebaine-iron tricarbonyl complex (**242**) was sufficiently stable to survive treatment with various *O*-demethylating agents without decomposition. BBr_3_, or BF_3_·SMe_2_ in dichloromethane, methionine or methansulfonic acid, also known as the push-pull system, and 9-iodoborabicyclononane (9-I-9-BBN) were successful applied for the 3-*O*-demethylation of **242**. The highest yield of the oripavine-iron tricarbonyl complex (**243**, 83%) was achieved with boron tribromide or boron trifluoride dimethylsulfide complexes. For removal of the iron tricarbonyl group of **243**, photolytic iron ligand exchange with acetonitrile (UV light, 40 °C, 2.5 h) provided oripavine (**5**) in 35% yield.

In a second line of experiments (Figure 68) [211], the conjugated diene system of thebaine (**4**) in ring-C was protected in the form of hetero Diels–Alder adducts (**244**, **245**, **246**). Dihydrothiopyrane adducts (**244**, **245**, **246**) were formed in the cycloaddition reaction of thebaine (**4**) with thioformyl cyanide in an approximate ratio of **244**:**245**:**246** = 1:4:3. The thioaldehyde was generated in situ from the Bunte salt (sodium *S*-(cyanomethyl) sulfothioate) with triethylamine in the presence of CaCl_2_ in benzene-methanol.

The hetero Diels–Alder reaction occurred on the β-face of the diene and formation of three cycloadducts were observed. The main products were the 7-substituted regioisomers, the 7β-cyano-6,14-epithiomethano- (**245**, 50%), and the 7α-cyano-6,14-epithiomethano-compounds (**246**, 38%), but formation of the 8α-cyano (**247**, 12%) minor by-product was also detected. 3-*O*-Demethylation of the 7-substituted regioisomers (**245**, **246**) was performed by applying the methods described above for the demethylation of thebaine-iron tricarbonyl complex (**242**), with yields of **247** and **248** ranging between 50 and 85%. The highest yield was attained by application of BBr_3_ (85%). For releasing of the 6,8-diene system by thermal retro-Diels–Alder reaction, a large excess of 2,3-dimethylbutadiene (20 equiv.) was utilized. Thioformyl cyanide, appearing transiently in the retro Diels–Alder reaction, was removed by a competitive hetero Diels–Alder reaction with 2,3-dimethylbutadiene. After treatment at 75 °C for 24 h in a sealed tube under argon atmosphere, oripavine (**5**) was obtained with 65% yield [211]. Alternatively, the thiomethano-adduct (**247**, **248**) was oxidized with *m*-chloroperbenzoic acid (*m*CPBA) to the corresponding sulfoxide, which was subjected to a retro Diels–Alder reaction resulting in oripavine (**5**, 78%) and the thioaldehyde. The transient sulfine was decomposed in boiling ethanol.

We noted in the introduction the clinical importance of substitution therapy for drug-dependent individuals with μOR partial agonists such as methadone or buprenorphine, sometimes in conjunction with naloxone. Vaccination and immunization present a promising alternate approach for treating substance dependence [212]. Here, an immunogenic carrier molecule (lipid, polysacharide, or protein such as tetanus toxoid (TT)) is conjugated to the surrogate of the target drug, the so-called hapten, which refers to a substance that can combine with a specific antibody but lacks intrinsic antigenicity. 

Haptens possess structural unit and functional groups (e.g., NH_2_, COOH, and SH) that facilitate their conjugation to an immunogenic carrier. Recently, Gutman et al. [212] reported the synthesis of C14-linked 4,5-epoxymorphinan-type analogues (Figure 69) as putative heroin vaccine haptens. To this end, the novel heroin haptens, 6,14-AmidoHap (**251**), 14-AmidoMorHap (**255**), and 14-AmidoHerHap (**256**) were synthesized from thebaine (**4**). The key intermediate of the synthesis was prepared in a hetero Diels–Alder reaction of thebaine (**4**) with 2,2,2-trichloroethyl nitrosoformate generated in situ from trichloroethyl *N*-hydroxycarbamate and sodium periodate [197]. The desired amino function in position 14 was introduced in the cycloadduct (**211**) in the first step of the synthesis. For preparation of 6,14-AmidoHap (**251**), the Δ^18,19^ double bond of the HDA adduct (**211**) was first saturated in a heterogenous catalytic reduction. The *N*-Troc group was cleaved, and (without isolation) the formed 14-aminodihydrocodeinone was further converted to 14-*N*HBoc-dihydrocodeinone (**249**). Reductive amination of the C-6 ketone (**249**) with titanium isopropoxide, NH_3_, and sodium borohydride led to the 6α-amino compound, which was acetylated with acetic anhydride to yield the acetamide (**250**). 3-*O*-Demethylation with boron tribromide and subsequent coupling with *S*-trityl protected thiopropanoic acid yielded the desired hapten (**251**) with an overall yield of 12% from thebaine (**4**). For preparation of 14-amidoMorHap (**255**) and 14-amidoHerHap (**256**), the *N*-Troc-substituted HDA adduct (**211**) was converted to 14-aminocodeinone (**214**) in a three-step procedure. The latter product was selectively reduced with sodium borohydride to 14-aminocodeine (**252**) quantitatively. The Δ^7,8^ double bond was saturated by heterogenous catalytic hydrogenation (5%Pd-C, 3.4 bar) in methanol.

Thereafter, 14-aminodihydocodeine (**253**) was converted to 14-aminodihydromorphine (**254**) by *O*-demethylation with BBr_3_. Coupling of **254** with 3-(tritylthio)propanoic acid using TBTU yielded 14-amidoMorHap (**255**), which was acetylated with acetic anhydride to 14-amidoHerHap (**256**). For preparation of the desired vaccine tetanus toxoid (TT) immunoconjugates, the primary surface amines (TT-NH_2_) were reacted with the heterobifunctional linker SM(PEG)_2_ to the corresponding TT-maleimides. Subsequently, the synthesized haptens were conjugated to the TT-linker applying the thiol-maleimide protocol. Vaccine efficacy studies in mice showed that immunization with the TT conjugate of 14-amidoHerHap (TT-**255**) partially attenuated the heroin-induced antinociception as shown behaviorally via the tail immersion test. The translation potential of this approach was the focus of a recent review [213], with consideration of the requirement for immunological activity against a broad spectrum of synthetic opioids [214,215].

### 2.5. Acetylenic Dienophiles

Morphinan-6,8-dienes can potentially react with acetylene derivatives via two different reaction routes. First, they can react in a [4+2] cycloaddition due to the properties of their conjugated diene system in ring-C, resulting in the normal 6,14-bridged Diels–Alder adducts. On the other hand, the *N*^17^-piperidine nitrogen of the morphinan derivatives can attack as a nucleophile on one of the acetylenic sp-carbons. The formed intermediate is stabilized by heterolytic cleavage of the *C*^9^-*N*^17^-σ-bond, in a process that is accompanied by the formation of products with an opened D-ring (piperidine). However, this process has not been observed in reactions of olefinic dienophiles with morphinan-6,8-dienes. Both types of products, the [4+2] cycloadducts and also the products of the nucleophilic attack of the *N*^17^ on the acetylenic sp-carbon, are frequently thermally unstable.

In 1963, Rapoport and Sheldrick [216] prepared some cycloadducts of thebaine (**4**) with acetylenic dienophiles. Thebaine (**4**) reacted easily with one equivalent of dimethyl acetylenedicarboxylate (DMAD) under relatively mild conditions (benzene, 50 °C, 1 h) and provided the adduct (**257**, Figure 70) in 90% yield. An examination of the thermal stability of the Diels–Alder adduct (**257**) revealed that boiling in di-*n*-butylether (b.p. 141 °C, 10–15 min) was sufficient for its conversion in 87% yield to a thermally rearranged isomer with the tetrahydrofuro-5*H*-[4,3,2-fg][3]benzazocine (**261**) structure.

When reacting the 18,19-dihydro derivative (**258**), which was readily prepared from the Diels–Alder adduct (**257**) by hydrogenation (H_2_ Pd-C, glacial acetic acid, 1h), no thermal rearrangement was observed under similar conditions. When thebaine (**4**) was reacted with propargylic acid ethyl ester (1.57 equiv. ethyl propiolate, EP, benzene, 50 °C, 8 h), the Diels–Alder adduct (**259**) was isolated in extremely low yield (5.8%). Thermal rearrangement of **259** occurred readily, as in the case of the DMA adduct (**257**) (di-*n*-butylether, 10–15 min), and yielded the corresponding benzazocine derivative.

Hayakawa et al. [217] re-examined the reaction of thebaine (**4**) with ethyl propiolate and methyl propiolate under still milder conditions by applying different solvents. When thebaine (**4**) was treated with 1.5 equiv. EP in acetonitrile (room temperature, 30 min) a novel type adduct (**260b**, Figure 70) was obtained in quantitative yield.

Similarly, the reaction of **4** with methyl propiolate gave the corresponding adduct (**260a**) in quantitative yield. In 1983, Hayakawa et al. [36] proceeded to investigate the addition reaction of acetylenic dienophiles to various morphinan-6,8-dienes (thebaine (**4**, Figure 71), 6-demethoxythebaine (**11b**, Figure 72), and 4-*O*-acetyl-β-dihydrothebaine (**10b**, Figure 72)) and neopinone dimethyl acetal (**269**, Figure 73). Ethyl propiolate (EP) methyl propiolate (MP) and dimethyl acetylenedicarboxylate (DMAD) were used as reagents. To elucidate the different factors governing the reactions, they undertook a systematic study by applying various solvents.

Treatment of thebaine (**4**) with dimethyl acetylenedicarboxylate (DMAD) or ethyl propiolate (EP) in benzene at 50 °C afforded the Diels–Alder-type adducts [**257a** (90%), **257b** (5.8%), Figure 71] [216], which, due to their thermal instability, easily underwent rearrangement to yield benzazocines (**261a–b**). Thebaine (**4**) reacted readily with methyl propiolate (MP) or ethyl propiolate (EP) in polar solvents to provide another type of adduct (**260a–b**) in high yields. For instance, **4** reacted readily with MP or EP in acetonitrile at room temperature within 30 min and gave the corresponding derivatives (**260a–b**) with almost quantitative yields (95% and 97%). Enol ether hydrolysis (cc. HCl, THF, room temperature, 3h) of the latter compounds resulted in the ketones **263a** and **263b**.

The investigations with 6-demethoxythebaine (**11b**, Figure 72) [47] assisted in the clarification of the role of the 6-*O*-methoxy group in the addition reactions of morphinan-6,8-dienes with acetylenic dienophiles. Reaction of 6-demethoxythebaine (**11b**) with dimethyl acetylenedicarboxylate (DMAD, benzene reflux, 12 h) resulted in the normal Diels–Alder adduct **264** in 72% yield. However, the product (**264**) showed similar thermal instability as seen with the thebaine-DMAD adduct (**257a**). Thermolysis of **264** in boiling xylene for 10 min led to **265** in quantitative yield. When 6-demethoxythebaine (**11b**) was treated with three equivalents of MP or reacted with DMAD in polar solvents, only the unreacted or unchanged starting diene (**11b**) was recovered. In contrast to thebaine (**4**), 6-demethoxythebaine (**11b**) did not display an ionic-type interaction with acetylenic reagents in polar solvents. Taken together, these results indicate an important role of the *C*^6^-methoxy group in the formation of *C*^9^-*N*^17^ bond-cleavaged products.

Next, 4-*O*-acetyl-β-dihydrothebaine (**10b**) [47], a morphinan-6,8-diene with opened E-ring, was subjected to addition reactions with MP and DMAD. 4-*O*-Acetyl-β-dihydro-thebaine (**10b**, Figure 72) reacted readily with acetylenic dienophiles at room temperature, but there was no formation of normal Diels–Alder adducts. Instead, *N*^17^-substituted-methine (**266a–b**)-type products (MP or DMAD in benzene or acetonitrile) were obtained. After carrying out the reaction in methanol, the “methanol-added” products (**267a**, MP, 91%, **267b**, DMAD, 86%) were isolated, the compounds of which were thermally unstable. In the case of **267a**, the thermal decomposition occurred even at low temperatures (chloroform, 50 °C, 5 min), and the methanol-added product **267b** rearranged upon heating at 70 °C. In both cases, the thermal treatment yielded 4-acetoxy-3,6-dimethoxy-phenantrene (**268**).

For determining the influence of the conjugated diene system on the reactivity of the morphinans, neopinone dimethyl ketal (**269**, Figure 73) [47] was chosen as a substrate. In the absence of the 6,8-diene system, only the nucleophilic attack of the alkaloid nitrogen was conceivable. When using a large excess of acetylenic reagents (MP or DMAD) (10–20 equiv., room temperature, 8–12 h), the *N*-substituted-methine-type (**270a–b**) compounds with an opened D-ring were obtained in modest (48–53%) yields. The authors concluded [47] that the 6,8-diene-system in ring-C had some influence on the nucleophilicity of the *N*^17^ nitrogen. When *N*^17^-northebaine (**230**) was treated with MP or DMAD, neither Diels–Alder reaction nor *C*^9^-*N*^17^-bond scission were observed. Instead, *N*^17^-substituted-northebaine derivatives (**271a–b**) resulting from a 1,2-addition were obtained in almost quantitative yields. In 1983, Singh et al. [218] reported the reactions of thebaine (**4**) with acetylenic dienophiles (MP, EP, DMAD, 3-butyn-2-one) in different solvents (THF, MeOH, *tert*-BuOH, trichloroethanol). Results confirmed the formation of unusual addition products (e.g., **260a**) instead of the normal Diels–Alder adducts. In 1993, Pindur et al. [69] carried out the reaction of thebaine (**4**) with the in situ generated aryne: benzyne (antranilic acid, trichloroacetic acid, THF, and isoamyl nitrite) in refluxing dichloromethane. The sole isolated stable product was a hexacyclic benzo[b]naphtha [2,1-e] azocine derivative (**272**, 6%, Figure 74).

In 1991, Jeong et al. performed the reaction of thebaine (**4**) with trifluoromethyl-substituted acetylenic dienophiles in polar or nonpolar solvents (Figure 75) [219]. The character of the solvent proved to play an important role in the formation of the “retro-Diels–Alder adduct (**276**)”. When thebaine (**4**) was allowed to react with trifluoropropyne (generated from 1,1,2-trichloro-3,3,3-trifluoropropene with activated zinc) in benzene or in CH_3_CN at room temperature for 18 h, there was no detectable formation of the Diels–Alder adduct.

Instead of the expected Diels–Alder adduct, compound **273** was obtained in 34% yield, versus 95% when reacted in in acetonitrile. When the same reaction was carried out in methanol, a “methanol-addition” product (**274**) was isolated with 60% yield. The treatment of thebaine (**4**) with hexafluoro-2-butyne (in benzene, 50 °C, 18vh) gave the Diels–Alder adduct (**275**, 38%) and equivalent amounts of a retro-Diels–Alder adduct (**276**, 39%). When using acetonitrile as solvent under the same conditions, the same products were obtained in lower yields (**275** 16%/**276** 24%). When the reaction was performed in methanol, only the “methanol-addition” product was obtained (**277**, 61%).

The research group of Moiseev [45] reported the reaction of morphinan-6,8-dienes with highly electron deficient acetylenes. Thebaine (**4**), *N*^17^-cyclopropylcarbonyl-nortebaine (**9e**) and *N*^17^-*tert*-butoxycarbonyl-northebaine (**9f**) were reacted with 1-trifluoro-acetylacetylene and 4-trimethylstannyl-1-trifluoroacetylacetylene. The interaction of these reaction partners could lead to a [4+2] cycloaddition or to the nucleophilic attack of the alkaloid nitrogen (*N*^17^) on the acetylenic carbon, depending on various factors, notably the nature of the diene (**4**, **9e**, **9f**), the character of the acetylenic dienophile, and the properties of the applied solvents.

The reaction of thebaine (**4**) with 1-trifluoroacteylacetylene resulted in low yields of the ketone (**278**, 16%, Figure 76) as a result of the nucleophilic attack, but traces of **279** was also detected (LC-MS). In order to hinder the nucleophilic attack of the *N*^17^ of **4** at the acetylenic carbon, Moiseev et al. [45] chose an acetylenic dienophile with a donating electron, and bulky trimethylstannyl group was chosen. Reacting **4** with 4-trimethyltin-1-trifluoroacetylacetylene, **278** was formed once again as the main isolated product (36%), along with small amounts (1.5%) of the enol ether hydrolysis (**279**) product. The product of the normal β-face [4+2] addition (**280**) was obtained in 2% yield, and traces of the destannylated Diels–Alder adduct (**281**) were noticed.

When the *N*^17^-acyl-northebaine derivatives, *N*^17^-cyclopropylcarbonyl-northebaine (**9e**, Figure 77) or *N*^17^-*tert*-butoxycarbonyl-northebaine (**9f**, Figure 78), were reacted with the above fluorinated acetylenic dienophiles, there was predominant formation of normal Diels–Alder adducts [45]. *N*^17^-Cyclopropylcarbonyl-northebaine (**9e**), upon reaction with diethyl acetylene dicarboxylate, resulted in the product of the [4+2] cycloaddition **282** in 77% yield. The same diene (**9e**) was reacted with trifluoroacetylacetylene (THF < 40 °C) to obtain the thermally unstable Diels–Alder adduct (**283**) in a yield of 11%. Upon boiling the conjugated enone (**283**) in methanol, compound **284** was formed in a Michael addition (15% yield).

Reacting **9e** with 4-trimethyltin-1-trifluoroacetylacetylene, in chlorobenzene (90 °C, 24 h) or THF (reflux, 24 h), yielded a 3*H*-furo-[4,3,2-fg][3]benzazocine derivative (**285**, 61% and 91%), whereas the formation of the normal Diels–Alder adduct was also reported (6% in chlorobenzene and 4% in THF). Treatment of *N*^17^-*tert*-butoxycarbonyl-northebaine (**9f**, Figure 78) with 4-trimethyltin-1-trifluoroacetylacetylene in chlorobenzene (66–75 °C, 4 h) resulted in a product mixture of Diels–Alder adduct (**287**, 14%) and a destannylated benzofurazocine (**288**, 72%). The same products were formed in boiling THF (6.5 h), with respective yields of 20% (**287**) and 74% benzazocine (**288**). The authors [45] assumed that the rearrangement of the stannylated Diels–Alder adduct (**287**) to the destannylated *N*^17^-Boc-benzazocine derivative (**288**) was achieved via the stannylated *N*^17^-Boc-benzazocine compound (**290**). The reaction of **9f** with trifluoroacetylacetylene readily (THF, 5–20 °C, for a few minutes) gave the normal Diels–Alder adduct (**289**, isolated yield: 47%) and the rearranged adduct (**288**) in a ratio of 11:3 (determined by ^19^F-NMR).

Recently, Sandulenko [220] summarized the reactions of morphinandienes (**4**, **10b**, **11b**, **9d–f**) with acetylenic dienophiles as well the synthetic potential of the formed products in a comprehensive review.

### 2.6. 7α-Amino and 7α-Aminomethyl Derivatives and Their Cyclic Analogues

In 1969, Bentley et al. [221] reported two independent reaction routes for the preparation of the 7α-amino-6,14-ethenomorphinan derivative (**295**, Figure 79). In the first approach, the procedure started with the 7α-ethoxycarbonyl compound (**16f**), which was prepared by reacting thebaine (**4**) with ethyl acrylate. The ester (**16f**) was reacted with hydrazine hydrate to obtain the acylhydrazide (**291**). That 7α-carbohydrazide (**291**) was converted with nitrous acid (generated with 15% hydrochloric acid and sodium nitrite) to acyl azide (**292**). The azide **292** was then heated for 10 min in the presence of benzyl alcohol (Curtius rearrangement via **293**) to yield the 7α-benzyloxycarbonyl-amino (7α-*N*Cbz, **294**) compound.

The benzyluretane (**294**) was converted into the 7α-amine (**295**) by boiling in ethanolic hydrogen chloride. During hydrogenolysis of the carbamate (**294**) under heterogenous catalytic conditions (H_2_, 10% Pd-C, EtOH), not only the Cbz protecting group cleaved but the Δ^18,19^ double bond was also saturated to form the corresponding 7α-amino-18,19-dihydro derivative. The second procedure utilized thevinone (**16a**) as staring material. The ketone (**16a**) was reacted with sodium azide and perchloric acid in a Schmidt reaction to yield the 7α-acetylamino derivative (**296**). The acetyl group was removed by hydrolysis with 5M hydrochloric acid to yield the target 7α-amino compound (**295**).

In 1984, a research team at the NIH [222] described the synthesis of potential irreversible ligands for the opioid receptors from the 7α-amino-6,14-ethenomorphinan derivative (**295**, Figure 80). The 6,14-etheno bridged-7α-amino compound (**295**) was prepared from thebaine (**4**) in three steps [221]. Next, selective 3-*O*-demethylation with boron tribromide resulted in the corresponding 7α-amino-oripavine derivative (**297**). The latter (**297**) was converted with thiophosgene to the 7α-isothiocyanato-derivative (**298**). Further 7α-acylamino-derivatives were prepared from (**297**): the 7α-bromoacetamido (**299**) derivative using bromoacetic anhydride and the 7α-methylfumaramido compound (**300**, fumaramidooripavine, FAO) by applying methylfumaroyl chloride. The antinociceptive activity of the target compounds was measured in the mouse hot-plate assay (ED_50_ [mmol/kg, sc] = 3.6 (**298**, NCS), 4.7 (**299**, bromoacetyl), 3.5 (**300**, FAO)). Despite these relatively potent nociceptive activities, the authors claimed that the target compounds were irreversibly binding ligands of the ORs. FAO (**300**) was found to be specific for δ-opioid receptors.

In order to convert the above agonists to OR antagonists, Lessor et al. [179] synthesized and characterized *N*^17^-substituted-7α-(acylamino)-oripavine derivatives (**306**–**309**, Figure 81), starting from the 7α-(acetylamino) derivative (**296**) [221]. First, **296** was *N*-demethylated using the von Braun method, in which with cyanogen bromide in chloroform (reflux, 16 h) resulted in cyanamide (**301**). This was selectively hydrolyzed under mildly acidic conditions (1M HCl, 90 °C), without affecting the 7α-(acetylamino) group, to the urea derivative (**302**). Subsequently, the urea (**302**) was converted into a secondary amine (**303**, *N*^17^-desmethyl- or *nor*- compound) with nitrous acid (generated with NaNO_2_ and 1M HCl) in a yield of 50% from (**296**). The *nor*-compound (**303**) was *N*-alkylated with the corresponding alkyl bromide (cyclopropylmethyl-, allyl-, propyl-) to obtain the appropriate *N*^17^-substituted-7α-acetylamino derivative (**304a–c**). Thereafter, the 7α-(acetylamino) group was hydrolyzed (3M hydrochloric acid, 110 °C) and the phenolic methyl ether was cleaved with boron tribromide (10 equiv.) in chloroform to obtain the corresponding *N*^17^-substituted-7α-amino-oripavine derivatives (**306a–c**). Finally, the 7α-amino group was acylated with methoxyfumaroyl chloride to the FAO analogues (**309a–c**) and with bromoacetyl chloride to **308a–c**.

The 7α-isocyanates (**307a–c**) were also prepared by treatment of the amines (**306a–c**) with thiophosgene. The binding affinities of the prepared compounds were determined in the rat brain membrane preparation in a competitive binding assay with [^3^H]dalamid (*D*-Ala^2^,Met^5^]enkephalinamide). The highest affinities were found for compounds with a cyclopropylmethyl *N*^17^ substituent (EC_50_ [nM] = 6–7). All of the target ligands were found to be potent narcotic antagonists in the mouse tail-flick assay.

In 1990, Klein et al. [223] subjected thevinone (**16a**) as well as *N*^17^-cyclopropylmethyl-northevinone (**310**) to reductive amination. The 7α-acetyl derivatives (**16a**, **310**) were transformed with sodium cyanoborohydride and ammonium acetate to a 1:1 mixture of diastereomeric primary amines (**311a–b**, **312a–b**). The 20*S* and 20*R* amines were separated by flash column chromatography. The stereochemistry of the C-20 chiral center of the amines (**311a**, **312a**), prepared from thevinone (**16a**), was determined indirectly on the base of attempted saturation (H_2_, 10% Pd-C, MeOH, 2 bar, 26 h) of the Δ^18,19^ double bond. The hydrogenation was successful only for one diastereomer (20*R*-**312a**), while the other epimer (20*S*-**311a**) did not react as expected due to the following mechanistic considerations. The configuration of the C-20 of the primary amines (**311a**, **312a**, Figure 82) was assigned provisionally on the base of the hypothesis that an intramolecular hydrogen bond exists between the C-20*R* amine (**312a**) and the oxygen of the 6-OCH_3_ group. In this case, the methyl group in position 20 was directed away from the 6,14-etheno bridge, which enabled the interaction between the Δ^18,19^ double bond and the surface of the catalyst.

The other C-20 amine (**311a**) was assigned the *S* configuration because the saturation of the double bond would be hampered by the C-20CH_3_ group, which in this case was turned towards the 6,14-etheno bridge. The stereochemistry of the *N*^17^-cyclopropylmethyl substituted primary amines (**311b**, **312b**) was assigned by comparison of their ^1^H-NMR spectra. Here, the chemical shifts of the protons in position 20 had diagnostic significance. The amines (**311a–b**, **312a–b**) were converted to their phenolic analogues (**313a–b**, **314a–b**) by 3-*O*-demethylation (KOH, diethylene glycol, 210 °C, 140 min). The latter compounds were transformed to the corresponding isothiocyanates (20*S*-NCS (**315a–b**) and 20*R*-NCS (**316a–b**)) by reaction with thiophosgene. The prepared compounds were tested by displacement of the κOR agonist [^3^H]bremazocine [224] (from a bovine striatal membrane preparation). The isothiocyanates with *N*^17^-methyl substituent showed an activity in the nanomolar range (IC_50_ = 6.2 nM for 20*S* (**315a**) and 63 nM for 20*R* (**316a**)) [223]. The corresponding isothiocyanates with *N*^17^-cyclopropylmethyl substituent displayed activities in the subnanomolar range (IC_50_: for 20*S* (**315b**): 0.32 nM and for 20*R* (**316b**): 0.76) in the membrane preparation against [^3^H]bremazocine [224].

In 1991, Berényi et al. [225,226,227] investigated the azidolysis of 20 tosylesters (**318a–g**, Figure 83) of different 6,14-ethenomorphinans. In brief, the morphinandienes thebaine (**4**), 6-demethoxythebaine (**11b**), and 6-chloro-6-demethoxythebaine (**11f**) were reacted with acrolein and methyl vinyl ketone to yield the corresponding 7α-formyl (**16c**) or 7α-acetyl (**16a**, **24**, **32**) adducts. The ketones (**16a**, **16c**, **24**, **32**) were reduced with sodium borohydride in methanol to the appropriate primary alcohol (**317a**) or to a mixture of diastereomeric secondary alcohols (**317b–g**). The tosylesters (**318a–g**) were prepared with tosyl chloride in pyridine at room temperature for 1–6 days. Subsequently, they were separated by column chromatography and the diastereomerically pure tosylates were subjected to azidolysis with sodium azide, (H_2_O, 100 °C, 1.5 h). Heating of the azides in *N*,*N*-dimethylformamide for 24 h resulted in a new class of morphinan derivatives (**320a–g**) possessing a 4-azatetracyclo-[4.4.0.0^3,8^.0^2,4^]-decane ring system. When applying longer reaction times for the azidolysis of the of tosylates (**318a–g**), the reaction gave direct formation of the aziridine derivatives (**318**) via the substitution and subsequent intramolecular cyclization. When using 20*R*-tosylates as starting materials, the 20*S*-azides formed by inversion were detected by TLC, but an isolation (and characterization by NMR spectroscopy) was only possible in the cases of the 7α-azidomethyl (**319a**) derivative and the 20*S*-7α-azidoethyl compound (20*S*-**319g**) due to the generally low stability of the azides. When 20*S*-tosylates (**318**) were subjected to azidolysis under identical conditions, 7-ethylidene derivatives were isolated as major products resulting from an elimination reaction.

The course of the reaction of the tosylesters depends on the configuration of the C-20 chiral carbon. The authors [225,226,227] explained the greater predisposition of the 20*S*-tosyltaes for elimination by the nearly antiperiplanar position of the tosyloxy group to the 7β-hydrogen, which is more advantageous for elimination.

In 2000, Derrick et al. [228] reported the syntheses of cinnamoyl derivatives of both 7α-amino-*N*^17^-cyclopropylmethyl- (**324a**) and 7α-aminomethyl-*N*^17^-cyclopropylmethyl-6,14-ethenomorphinans (**324b**, Figure 84). For the synthesis of the 7α-amino derivate (**324a**), *N*^17^-cyclopropylmethyl-northevinone (**310**) was converted in a Schmitt reaction to the 7α-acetaminoamino compound, which was hydrolyzed to the 7α-amino-*N*^17^-cyclopropylmethyl derivative (**321**). Saturation of the Δ^18,19^ double bond (H_2_/Pd-C, 2 bar) and 3-*O*-demethylation with KOH gave compound **322**. The 7α-amino-oripavine derivative (**323**) was converted to the cinnamide (**324a**) with cinnamoyl chloride in a yield of 87%. For the synthesis of the desired 7α-aminomethyl cinnamoyl derivative (**324b**), the authors attempted several synthetic routes before settling upon an effective indirect route. Here, the 7α-ethoxycarbonyl compound (**325**) was applied as starting point, which was prepared by reacting of *N*^17^-cycopropylmethyl-northebaine (**9d**) with ethyl acrylate. The ester (**325**) was hydrolyzed (2M HCl, steam bath, 3h) to the free acid, which was converted to the acid chloride with oxalyl chloride. The acid chloride was then reacted with benzylamine to the corresponding amide (**326**). The keto group of the amide was reduced with LiAlH_4_, and the resulting 7α-(benzylamino)methyl derivative was hydrogenated under heterogenous catalytic conditions (H_2_, 10% Pd-C, 2 bar, 45 °C, EtOH). Simultaneously with Δ^18,19^ double bond saturation, the benzyl protecting group was also cleaved. The ethano-amine derivative (**327**) was 3-*O*-demethylated with KOH in diethylene glycol (reflux, 8h) to obtain **328b** in 71% yield.

The free amine (**328b**) was acylated to the desired derivative (**324b**) with cinnamoyl chloride. The prepared cinnamoyl derivatives (**324a–b**) showed high affinity for opioid receptors (μ = δ > κ) in displacement binding assays in guinea pig brain membranes [228]. The 7α-(cinnamoyl)aminomethyl (**324b**) derivative was 20 to 70 times more potent as a μOR antagonist in the mouse vas deferens tissue assays than was the native 7α-(cinnamoyl)amino (**324a**) compound. The authors concluded that the irreversible binding and μ-antagonist activity was governed by the position of the PhCH=CH-CO group. A methylene spacer between the cinnamoylamino group and the C-7α carbon seems beneficial.

In 2002, the research group of Nagase [229] designed and synthesized a series of 6,14-ethenomorphinan-7α-carboxamide ligands targeting what the authors termed the ε (epsilon) OR conceived on the basis of the message-address concept [230,231,232]. The synthesized compounds were tested in rat vas deferens preparation for inhibition of the electrically-stimulated RVD contractions. One of the prepared compounds, TAN-821 (**333**), was the most potent agonist for the putative ε opioid receptor [229]. Its potency (IC_50_ = 72 nM) was comparable to that of β-endorphin (IC_50_ = 74 nM). TAN-821 (17-cyclopropyl-methyl-4,5α-epoxy-3,6-dihydroxy-6,14-*endo*-ethenomorphinan-7α-(*N*-methyl-*N*-phenethyl)-carboxamide, **333**) was synthesized in a five-step procedure from *N*^17^-cyclopropylmethyl-northebaine (**9d**, Figure 85). The diene was reacted in a Diels–Alder reaction with ethyl acrylate (reflux for 15h) to obtain the 7α-ethylcarboxylate (**329**) in 69% yield. The ethyl ester was hydrolyzed with 6M hydrochloric acid to the free acid. The 7α-carboxylic compound (**330**) was converted to the acid chloride (**331**) with oxalyl chloride. Next, the amidation of the crude acid chloride (**331**) with the corresponding amine resulted in carboxamide (**332**). The latter compound (**332**) was treated with BBr_3_ in CH_2_Cl_2_ to yield the 3,6-dihydroxy compound (TAN-821, **333**).

The εOR is a putative receptor class that represents a residual population of receptors that remains after the blockade of μ-, δ-, and κ-ORs and binds the endogenous opioid peptide β-endorphin [233]. Knockout studies suggest that εOR may not represent a distinct protein, but rather a state of classical ORs distinguished by their coupling to intracellular second messenger pathways [234]. The research group of Nagase [235] also designed and synthesized the first antagonist with selectivity for the putative εORs by modification of the εOR agonist ligand TAN-821 [235] and based on the accessory site concept [236,237]. Although 17-cyclopropylmethyl-4,5α-epoxy-6β,21-epoxymethano-3-hydroxy-6,14-*endo*ethenomorphinan-7α-(*N*-phenethyl)carboxamide (**335**, TAN-1014, Figure 85) possessed no agonistic effect on the rat vas deferens test, it antagonized the agonistic effect of β-endorphin (a 31 amino acid sequence derived from proopiomelanocortin) [235]. TAN-1014 (**335**), containing a 1,3-oxazinan-4-one substructure, was synthesized from the corresponding β-phenethyl-carboxamide derivative (**334**, “*N*^21^-desmethyl-TAN-821”) with paraformaldehyde by acid-catalyzed cyclization (1,4-dioxane, catalytic amount of sulphuric acid, reflux, 11 h). The 6,21-epoxymethano fragment or moiety of the molecule **335** was assumed to be a possible accessory pharmacophore.

In 2007, Rennison et al. [180] described the syntheses of a series of derivatives from 7α-aminomethyl-6,14-*endo*-ethano-tetrahydrooripavine (**339a–f**, Figure 86), where the 7α-amino-methyl group was acylated with aryl substituted cinnamoyl chlorides (R = H, 2-CH_3_, 2-Cl, 4-CH_3_, 4-Cl, 4-NO_2_). 7α-Aminomethyl-6,14-*endo*-ethano-tetrahydrothebaine (**337**) was prepared from the thebaine-acroleine adduct (**16c**) in three steps. The 7α-formyl compound (**16c**) was treated with hydroxylamine hydrochloride to afford an oxime, which was reduced with LiAlH_4_ leading to 7α-aminomethyl derivative (**336**). Reduction of the Δ^18,19^ double bond of the latter (**336**) under heterogenous catalytic conditions (H_2_, Pd-C, 40 bar, EtOH, 16 h) yielded the 7α-aminomethyl compound (**337**). Selective 3-*O*-demethylation of **337** using boron tribromide yielded 7α-aminomethyl-6,14-*endo*-ethano-tetrahydrooripavine (**338**). Acylation of the oripavine derivative (**338**) with substituted cinnamoyl chlorides yielded the target amides (**339a–f**). The others undertook a pharmacological characterization of the target ligands using competitive binding assays in recombinant human ORs. The oripavine derivatives (**339a–f**) had subnanomolar affinity (K_i_ = 0.14–0.25) for μ-ORs and also affinity in the nanomolar range for κ- an δ-ORs [180]. For determination of the OR functional activity, the authors used the in vitro [^35^S]GTPgS stimulation assay on recombinant ORs transfected into CHO cells. This assay marks agonism by promoting the exchange of GDP for [^35^S]GTPgS in the G-protein complex of ORs and many other classes G-protein-coupled receptors. The oripavine derivatives (**339a–f**) were found to be μ-OR partial agonists.

The same research team [180] reported the synthesis of thebaine Diels–Alder adducts containing a constrained pyrrolidine ring in the C^7^-C^8^ position (Figure 87). The addition of *N*-benzylmaleimide to thebaine (**4**) yielded *N*-benzyl-7α,8α-cycloadduct (1′-benzyl-2′,5′-dioxo-[7α,8α:3′,4′]pyrrolidino, **75b**), which was reduced with LiAlH_4_ to compound **76** containing a pyrrolidine ring in position C^7^-C^8^ [95]. Hydrogenolysis, performed under heterogenous catalytic conditions (H_2_, 10% Pd-C, EtOH, cc. HCl, RT, 2.7 bar, 5 days), led to compound **340** without affecting the Δ^18,19^ double bond. Selective 3-*O*-demethylation of the latter compound (**340**) was performed with boron tribromide affording **341** in 72% yield. The secondary amine (**341**) was *N*-acylated using the corresponding acid chlorides to the target compounds (**342a–b**) with a substituted cinnamoyl group on the pyrrolidine nitrogen. Biochemical characterization of the conformationally constrained cinnamoyl analogues (**342a–b**, Figure 87) showed similar binding affinities as K_i_ (in units of nM) (2-Me: μOR = 0.22, κOR = 0.52, δOR = 2.36; 4-Me: μOR = 0.48, κOR = 1.41, δOR = 4.83) to those of the 7α-(cinnamoyl)aminomethyl (**339b**, **339d**, Figure 86) (2-Me: μOR = 0.14, κOR = 1.82, δOR = 3.41; 4-Me: μOR = 0.23, κOR = 3.11, δOR = 6.53) derivatives [180]. In the functional in vitro assays, the compounds were partial agonists without any selectivity for OR subtypes. Compared to the 7α-(cinnamoyl)amionomethyl derivatives (**339b**, **339d**), the constrained analogues (**342a–b**) had considerably higher efficacy [180].

Lewis et al. [238] synthesized fumaroylamino-substituted 7α-amino (**343a**) and 7α-aminomethyl oripavine derivatives (**343b**, Figure 88) as analogues of the prototypic irreversibly binding OR ligand β-funaltrexamine (β-FNA). 7α-amino- (**323**) and 7α-aminomethyl-*N*^17^-cyclopropylmethyl-6,14-endoethanotetrahydro nororipavine (**328**) were acylated with monomethyl fumaric acid chloride in the presence of triethylamine in dichloromethane. Both compounds (**343a–b**) showed high affinity for all three classical OR subtypes; the K_i_ values (nM) in Hartely guinea pig brain membranes were as follows: 7α-amino (**343a**), μOR = 2.8, κOR = 1.9, δOR = 2.6; and 7α-aminomethyl (**343b**), μOR = 0.9, κOR = 2.8, δOR = 0.7) [238]. The corresponding antagonist activity of these compounds was determined in the mouse vas deferens assay, which indicated them to be potent μ-OR antagonists with subnanomolar K_e_ values. The 7α-fumaroylaminomethyl oripavine derivative (**343b**) proved to be a potent μOR antagonist with irreversible action.

Researchers at the Gazi University synthesized a series of 7α-(1,3,4-oxadiazol-2-yl) arenamine-substituted 6,14-ethenomorphinan (**345**, Figure 89) derivatives [239] as potential analgesics. The key intermediate of their syntheses was the 7α-carboxymethyl compound (**16e**) [74], which was prepared from thebaine (**4**) with methyl acrylate.

The ester was converted to 7α-carboxylic acid hydrazide (**291**) [221], which was reacted with a variety of aryl isocyanates in toluene (70 °C, 2 h) to obtain the corresponding 7α-carboxylic acid 2-[(phenylamino)carbonyl]hydrazides (**344**) in 69–88% yields. Cyclization of the hydrazides (**344**) via dehydration with phosphorus oxychloride (90 °C, 3 h) resulted in the target ethenomorphinans (**345**) in 53–78% yields.

The Gazi research group also synthesized numerous 7α-(5-(halophenyl))-1,3,4-oxadiazole-substituted 6,14-ethenomorphinan derivatives (**349**, Figure 89) from the 7α-carboxylic acid hydrazide (**291**) [240]. Here, they reacted the carbohydrazide (**291**) with an equimolar amount of the appropriate substituted benzoyl chloride to obtain diacylhydrazide derivatives (**346**) in 79–89% yield. Treatment of the latter compounds with phosphorus oxychloride (reflux, 3 h) led via cyclodehydration to 7α-oxadiazole-substituted 6,14-ethenomorphinans (**347**). Subsequently, cyanamides (**348**) were obtained from the 1,3,4-oxadiazoles (**347**) with cyanogen bromide in chloroform (reflux, 24 h) in 50–56% yield. Finally, the *N*-cyano compounds (**348**) were reacted with sodium azide in DMF (reflux, 4 h) to *N*^17^-(tetrazol-1*H*-5-yl)-7α-(5-phenyl-1,3,4-oxadiazole)-substituted 6,14-ethenomorphinan derivatives (**349**) in 50–65% yield.

Applying similar preparative methods, Yavuz et al. [241] synthesized the 7α-(1,3,4-thiadiazole) analogues (Figure 90) of the aforementioned oxadiazole ligands. The carbohydrazide (**291**) was reacted with substituted phenylisotiocyanates to yield acylthiosemicarbazide derivatives (**350**, Y = S). Subsequently, the latter compounds were subjected to cyclization by treatment with H_3_PO_4_ (90 °C, 3 h). The obtained 7α-(1,3,4-thiadiazole)-6,14-ethenomorphinan derivatives (**351**, Y = S) were converted to *N*^17^-(tetrazol-1*H*-5-yl)-7α-(1,3,4-thiadiazole) compounds (**353**) in a similar manner as described above for the oxadiazole ligands, namely by cyanamide synthesis using BrCN and subsequent reaction with NaN_3_ and NH_4_Cl in DMF. The analgesic activity of the target 6,14-ethenomorphinans [241] incorporating the oxadiazole or thiadiazole moiety, all of which contained an *O*-methyl group in position three of the morphinan A ring, were evaluated by rat tail-flick and rat hot plate methods. The most active compound (**353**, Y = O, R = NHPh) of the series was found to be more potent than morphine (**1**).

In 2018, Nagase and Yamamoto [242] described an eight-step transformation of *N*^17^-cyclopropylmethyl-northebaine (**9d**, Figure 91) into novel type 7-carboxamido-6,14-ethenomorphinan derivatives with a Δ^7,8^ double bond, having a κOR agonist activity. The 7-keto-6,14-ethenomorphinan intermediate (**356**) were prepared applying the method of Lewis et al. [243,244]. In 1971, Lewis et al. [243] described the cycloaddition of 2-chloroacrylonitrile (2.1 equiv.) to thebaine (**4**) in boiling benzene. The formation of a 4:1 mixture of the 7α-cyano-7β-chloro and 7α-chloro-7β-cyano Diels–Alder adducts was observed.

The epimers were separated by fractional crystallization of the crude product from methanol in 43% (7α-CN) and 6.5% (7β-CN) yield. In a following paper, the same group [244], first achieved the synthesis of 6,14-*endo*-etheno-7-oxo-tetrahydrothebaine by reacting the thebaine 2-chloroacrylonitrile adducts with aqueous sodium hydroxide (EtOH, reflux, 16 h). Nagase and Yamamoto adapted this protocol [242] to react *N*^17^-cyclopropylmethyl-northebaine (**9d**) [245] with 2-chloroacrylonitrile (as a masked ketene) in toluene to yield a mixture of the epimeric Diels–Alder adducts (**354**, **355**). Without separation the isomers, they were treated with 1M sodium hydroxide in ethanol to obtain the single ketone **356**. In the next step, the Δ^18,19^-bond was saturated in such a manner that the cyclopropyl-group remained intact. The 7-keto-6,14-*endo*-ethano derivate (**357**) was converted almost quantitatively to the 7-*O*-triflyl-7,8-didehydro compound (**358**) with *N*-phenyl-bis(trifluoromethansulfonimide) in THF in the presence of potassium bis(trimethylsilyl)amide (KHMDS). The triflate (**358**) was treated with 2,4,6-trichlorophenyl formate in toluene in the presence of palladium acetate, 4,5-bis(diphenylphosphino)-9,9-dimethylxanthene, and triethylamine. Hydrolysis of the 2,4,6-trichlorophenyl ester under basic conditions (NaOH) resulted in the sodium salt of the 7-carboxylic acid derivative (**360**). The free acid of **360** was coupled with *N*-benzyl-*N*-methylamine in tetrahydrofurane in the presence of triethylamine and DMAP with a yield of 86%. Finally, 3-*O*-demethylation of **361** with boron tribromide in dichloromethane quantitatively yielded the desired ligand (**362**). Competitive binding against subtype selective radioligands [^3^H]DAMGO (μ-OR), [^3^H]DPDPE (δ-OR), and [^3^H]U69,593 (κ-OR) in brain membranes (prepared from CHO cells) was performed to identify specificities of the products [242]. The *N*-methyl-*N*-benzyl-7-carboxamide (**362**) derivative showed extraordinarily strong affinity (for the κ-OR (K_i_ 2.1 pM) and good selectivity (μ/κ = 8.1, δ/κ = 5.8). The [^35^S]GTPγS binding tests described above showed **362** to be a potent κ-OR agonist (ED_50_ 2.8 pM), with high selectivity, ED_50_ (μ/κ) = 1000, ED_50_ (δ/κ) = 600).

In 2020, Krüll et al. [246] reported a study inspired by the high affinity (K_i_ (μ-OR) = 0.7 nM) of the cinnamoyl derivative of the 7α-aminomethyl-*N*^17^-cyclopropylmethyl-6,14-ethenomorphinans (**324b**, Figure 84) previously synthesized by Derrick et al. in 2000 [228]. They explored the possibility of the bioisosteric replacement of the cinnamide substructure with phenylazocarboxamides by combining both structural scaffolds. The 7α-aminomethyl compound (**328**, Figure 92) was coupled with the corresponding *tert*-butyl phenylazocarboxylates to obtain the azocarboxamide ligands with a methoxy group in position three in 42–87% yield. Applying *tert*-butyl 4-fluorophenylazocarboxylate as the azo compound, the product *N*-(4-fluorophenyl)azocarboxamide (**363**, X = N, R = F) was obtained in 87% yield. 

The highest binding affinities (HEK 2935 cells) were found for the 4-fluoro (**363**, X = N, R = F, K_i_ (μ-OR) = 1.5 nM, K_i_ (κ-OR) = 10 nM, and K_i_ (δ-OR) = 36 nM, thus a 7-fold selectivity over κ-OR = 6.7 and a 24-fold selectivity over δ-OR) and 4-bromo azocarboxamide derivatives (**363**, X = N, R = Br, K_i_ (μ-OR) = 1.3 nM, K_i_ (κ-OR) = 10 nM, and K_i_ (δ-OR) = 28 nM, thus an 8-fold selectivity over κ-OR and 22-fold over δ-OR) [246]. Evaluation of **363** (X = N, R = F) in functional assays at μ-ORs showed it to have weak partial agonist properties. On the basis of radioligand binding studies with human ORs and functional assays, the authors concluded that the bioisosteric replacement of the cinnamide unit was realizable without loss of binding affinity or subtype selectivity [246]. Radiosynthesis of the ^18^F-labeled azocarboxamide (**363**, X = N, R = ^18^F) was also performed (see Section 3.4 [^18^F]-labeled 6,14-ethenomorphinans) via nucleophilic substitution of ^18^F-labeled *tert*-butyl phenylazocarboxylate with the 7α-aminomethyl compound (**328**).

### 2.7. Nepenthone Derivatives and 7α-phenyl-6,14-ethenomorphinan Analogues

Bentley and Ball [77,247] were the first to report the reaction of thebaine (**4**) with phenyl vinyl ketone. Instead of the laborious and cumbersome systematic naming of the formed adduct (**16b**, 7α-benzoyl-4,5α-epoxy-17-methyl-3,6-dimethoxy-6,14-ethenomorpinan, Figure 93), they assigned the trivial name «*nepenthone*» to the new Diels–Alder adduct from Greek nēpenthés (νηπενθέζ), referring to the sedative reported in the fourth book of Homer’s Odyssey, which was probably a mixture of opium in sweet wine:
“The Zeus’s daughter Helen thought of something else.Into the mixing-bowl from which they drank their wineshe slipped a drug, heart’s-ease, dissolving anger,magic to make us all forget our pains.No one who drank it deeply, mulled in wine,could let a tear roll down his cheeks that day,not even if his mother should die, his father die,not even if right before his eyes some enemy brought downa brother or darling son with a sharp bronze blade.” (*Homer, Odyssey, 4, 219–227*) (translation: Fagles, R.)

The attempted isomerization of the 7α-benzoyl compound (**16b**, nepenthone) into the 7β-isomer (**17b**) using sodium hydroxide in methanol (reflux, 10 min) surprisingly resulted in isonepenthone (**364**) in 75% yield [152]. The mechanism of this base-catalyzed rearrangement was discussed in detail in Section 2.2 above (Figure 29). As isonepenthone (**364**) is structurally comprehensible or resolvable as a mixed acetal of an α,β-unsaturated ketone, it is extremely sensitive to acidic hydrolysis. Treatment of **364** with 5% hydrochloric acid or with cold acetic acid led to Ψ-nepenthone (**367**) with an opened E-ring and a phenolic hydroxyl group in position 4.

Nepenthone (**16b**) was also found to be susceptible to acidic conditions [77,153]. Although stable to boiling in 5% hydrochloric acid, treating **16b** with a 1:1 mixture of concentrated hydrochloric acid–glacial acetic acid heated at 100 °C gave two rearranged products: neonepenthone-A (**368**, 25%, isolated yield, Figure 94) and neonepenthone-B (**369**, 10% isolated yield). Their structures of **368** and **369** were deduced by infrared spectroscopy and elemental analysis.

Reduction of nepenthone (**16b**) under Meerwein-Ponndorf conditions in 2-propanol or with sodium borohydride in methanol gave the secondary alcohol nepenthol (**370**). Treatment of nepenthol (**370**) with a mixture of concentrated HCl and glacial acetic acid yielded flavonepenthone (**371**) in 84% yield. When nepenthol (**370**) was heated with formic acid for 16 h, a 14-alkenylcodeinone derivative, nepenthene (**372**), was obtained in 76% yield, which was convertible to flavonepenthone (**371**) with conc. HCl-AcOH.

Lewis and Readhead [243] synthesized 7-methy-*epi*-nepenthone (**375**, Figure 95) from thebaine in a three-step procedure in 1971. In the Diels–Alder cycloaddition of thebaine (**4**) with methacrylonitrile, the 7β-cyano-7α-methyl adduct (**373**) formed exclusively [248]. Treatment of the adduct (**373**) with an equimolar amount of phenylmagnesium bromide resulted in the imine (**374**), which was hydrolyzed with aqueous acetic acid to 7-methyl-*epi*-nepenthone (**375**). Grignard addition of methylmagnesium iodide to the ketone (**375**) gave the tertiary alcohol (**376**) with 20*S* absolute configuration. 7-Methyl-*epi*-nepenthone (**375**) was converted to 7-methyl-*epi*-20*R*-nepenthol (**377**) with sodium borohydride in 2-ethoxyethanol.

According to the rat tail pressure test, the 7β-benzimidoyl derivative (**374**) and 7-methyl-*epi*-nepenthol (**375**) were five times more potent analgesics than morphine (**1**) [243]. The 20*S* tertiary alcohol (**376**) showed only one third of the potency of morphine, but the 20*R* secondary alcohol (**377**) was 40 times more potent than morphine (**1**) [249].

Husbands and Lewis [250] developed a series of morphinan derivatives with cyclic imine and pyrrolidine structural units containing a constrained phenyl group (Figure 96). Thebaine (**4**) with 1,1-disubstituted ethylene and methacrylonitrile [243,249] yielded the 7β-cyano-7α-methyl adduct (**373**) with 40% yield. The 7β-nitrile (**373**) was treated with three equivalents of phenylmagnesium bromide in benzene to yield the anhydro-(5β-amino-7β-benzoyl)-7α-methylterahydrothebainol 6-*O*-methyl ether (**378**). The latter was 25 times more potent than morphine (**1**) in the rat tail pressure analgesia test [243]. The sterically hindered 4-OH group of the compound **378** was alkylated with bromobenzene (reflux, 3 days) and the formed 4-*O*-phenylether (**379**) was reduced with large excess of sodium (10 equiv.) in liquid ammonia to the pyrrolidine (**380**).

Reductive methylation of **380** with CH_2_O and NaCNBH_3_ resulted in the *N*-methyl-pyrrolidine derivative (**381**). 3-*O*-Demethylation of **381** with sodium propane thiolate in HMPA (110 °C, 3 h) led to the corresponding target phenol (**382**). Ligand displacement studies were performed in Hartley guinea pig membranes using [^3^H]DAMGO for μ-OR, [^3^H]U69,593 for κ_1_-OR, [^3^H]bremazocine for κ_2_-OR, and [^3^H]Cl-DPDPE for δ-OR. Here, the authors considered κ-ORs of the non-analgesic subtype κ_1_-OR and the anti-hyperalgesic and anti-allodynic κ_2_-OR; there is only one gene for κ-OR, so the subtypes were likely due to differential coupling to second messenger systems [251], as noted above for the putative ε-OR. The tested ligands displayed subnanomolar affinity for μ-OR and lesser affinity for κ- and δ-ORs. In this series, compound **378** with the free 4-hydroxyl group showed some μ-OR selectivity (μ/δ = 9.7, μ/κ_1_ = 67, μ/κ_2_ = 1278). The authors concluded that a fixation of the position of the bulky phenyl group did not confer any selectivity in OR binding assays, as likewise seen for the target compound with free 3-hydroxyl group (**382**) (μ/δ = 1.9, μ/κ_1_ = 1.4, μ/κ_2_ = 25).

In the 1950s, Bentley and Ball [77] were unable to obtain saturation of the Δ^18,19^ double bond of nepenthone (**16b**). That obstacle was explained by the stereochemical structure of the substrate, whereby the 7α-benzoyl group shields the 6,14-*endo*-etheno bridge and consequently prohibits an interaction between the double bond and the catalyst surface. Others returned to the catalytic hydrogenation of nepenthone (**16b**, Figure 97) in the mid-1990s [252]. Catalytic hydrogenation was tested under various conditions of temperature, pressure, reaction time, and catalyst. While there were no reactions at room temperature and atmospheric pressure, saturation of the double bond did occur with increasing temperature and pressure.

However, due to the unintended reduction of the 20-C=O carbonyl group, there was simultaneous formation of a stereochemically homogeneous secondary alcohol (20*S*-**384**) as a by-product. The best result was achieved by applying Pd-C in ethanol (27 °C, 5 bar, 9 h). Besides dihydronepenthone (**383**, 69%) and 20*S*-dihydronepenthol (20*S*-**384**, 24%), nepenthone (**16b**, 7%) was also present in the product mixture, based on the ^1^H-NMR spectrum. The application of Pd-C/PtCl_4_ (60 °C, 6 bar, 11 h) led to the sole formation of 20*S*-**384**. The reduction of dihydronepenthone (**383**) with sodium borohydride yielded the same 20*S*-dihydronepenthol (20*S*-**384**). For determination of the configurations of the by-product nepenthol (20*S*-**384**), as well for comparative NMR measurements, the authors also prepared the 20*R*-epimer (20*R*-**384**) in a multistep synthesis from the 7α-formyl derivative (**16c**); the latter compound was prepared from thebaine (**4**) with acroleine. The 7α-formyl-6,14-endoetheno derivative (**16c**) was reacted in a Grignard reaction with phenylmagnesium bromide, which gave a 1:1 diastereomeric mixture of 20*S* and 20*R*-nepenthol (**385**). This mixture of the secondary alcohols (20*RS*-**385**) was reduced (H_2_, Pd-C, and EtOH, 55 °C, 3 bar) to the corresponding mixture of diastereomeric 6,14-*endo*-ethano analogues (20*RS*-**384**). Next, the pure 20*R*-**384** was separated by repeated fractional crystallization. The absolute configuration of the diastereomeric secondary alcohols (20*S*-**384** and 20*R*-**384**) was determined by NOE difference measurements. For preparation of pure dihydronepenthone (**383**) in gram-scale, the mixture of dihydronepenthone (**383**) and 20*S*-dihydronepenthol (20*S*-**384**) was subjected to Swern oxidation.

In 1999, Micskei et al. [253] investigated the reactions of nepenthone (**16b**, Figure 98) with chromium(II) reagents in a neutral aqueous medium (biomimetic synthesis) in order to study the chemoselectivity of the reduction. Nepenthone (**16b**) possesses two theoretically reducible functional groups: the Δ^18,19^ double bond and the C-20 carbonyl group. When using [Cr(H_2_O)_6_]^2+^, no reaction was observed, but when applying the [Cr(IDA)(H_2_O)_3_] complex, there was reduction of the oxo group of nepenthone (**16b**, soluble in water in bitartrate form). The organometallic species A (Figure 99, structure **A**) contains two H_2_O molecules in the first coordination sphere, which can react with the organochromium bond as an internal electrophile.

As such, a fully chemoselective reduction (pH = 6.1) occurred with the formation of 20*R*-nepenthol (20*R*-**385**). To determine the absolute configuration of C-20, NOEDIF measurements and 2D NMR experiments (COSY) were performed. When nepenthone (**16b**) was reacted with [Cr(EDTA)]^2−^ (pH = 5.5) simultaneously with the carbonyl reduction, there was an unexpected opening of the C^6,7^-bond, thus forming a morphinan derivative (**386**) with a enol methylether structure and containing a new fused pentacycle. 

Applying the coordinated EDTA ligand, in the proposed structure **B** (Figure 99), the hydrophobic skeleton extrudes the H_2_O molecules from the first coordination sphere of the organometallic intermediate. In this case, protonation of the Cr ← C bond is not feasible, but the benzylic carbanion in postition-C20 attacks the C-19 carbon of the *endo*-eteno bridge. In the final step of a concerted electron shift, the C-7 carbanion was stabilized by protonation. The structure of compound (**386**) was elucidated by NMR spectroscopy and X-ray crystallography. When **386** was boiled with 3% hydrochloric acid (3 h), enol ether hydrolysis and water elimination occurred, and **387** was isolated in 94% yield.

By exploiting the mechanistic advantages of the Grignard addition, the research group of Makleit [110] synthesized a series of *N*^17^-substituted (20*R*)-(**392a–l**) and (20*S*)-phenyl-6,14-ethenomorphinan derivatives (**396a–l**, Figure 100). Here, nepenthone (**16b**), thevinone (**16a**) and their 18,19-dihydro derivatives dihydronepenthone (**389**) and dihydrothevinone (**104**) were *N*-demethylated with diethyl azodicarboxylate. *N*^17^-Substituted derivatives (**390a–l**, **394a–l**) were prepared by alkylation of the corresponding *N*^17^-desmethyl compounds (**388**, **389**, **393**, **111**). The 7α-acyl derivatives (**390a–l**, **394a–l**) were in turn reacted with the appropriate Grignard reagent. The diastereoselective Grignard reaction of 7α-benzoyl derivatives (nepenthone (**16b**) and dihydronepenthone (**383**) and their *N*^17^-substituted versions (**390a–l**)) with methylmagnesium iodide resulted in 20*R*-phenyl tertiary alcohols (**391a–l**). The addition of phenylmagnesium bromide to 7α-acetyl compounds (thevinone (**16a**) and dihydrothevinone (**104**) and their *N*^17^-substituted derivatives (**394a–l**)) gave 20*S*-phenyl products. Treatment of the 3-*O*-methylethers (**391a–l**, **395a–l**) with KOH in diethylene glycol at 210–220 °C gave the desired *N*^17^-substituted-nororipavine derivatives (**392a–l**, **396a–l**).

The biochemical investigations of compounds **392a–l** and **396a–l** showed significant μ-OR selectivity in some cases (*N*^17^-*n*-propyl-20*R*-phenyl (**392c**) and *N*^17^-methyl-20*S*-phenyl-dihydroorvinol (**397b**) (dihydrothevinone (**104**) → 20*S*-phenyl-dihydrothevinol (**397a**) → 20*S*-phenyl-dihydroorvinol (**397b**)). Notably, two compounds of this series were identified as lead compounds in later investigations [114,164]. *N*^17^-Cyclopropyl-20*S*-phenyl-dihydronororvinol (**396g**, BU127) has shown high affinity for μ- δ- and κ-ORs [14,119] (K_i_ values in nM, μ-OR = 0.71, δ-OR = 1.9 nM, and κ-OR = 0.49) and moderate affinity for the less frequently considered nociceptine receptors (K_i_ (NOP) = 42.3 nM). BU127 was tested in vitro and in vivo as a ligand, showing affinity and efficacy for NOP receptors comparable to that of buprenorphine (**96**). The Husbands group selected BU127 as lead compound for their research on buprenorphine analogues [114,119]. The second compound of this series also showing notable characteristics was *N*^17^-cyclopropylmethyl-nornepenthone (**390a**, SLL-020ACP [164]). This proved to a selective κ-OR agonist and was selected as the lead compound for further design of κ-OR ligands with little affinity for the classical ORs.

In 1999, the synthesis of dihydroflavonepenthone analogues (**398**–**399**, Figure 101) was reported by Bermejo et al. [254]. As starting point for the synthesis, they selected the tertiary alcohol 20*S*-phenyl-dihydrothevinol (**397a**), which was prepared applying the method of Bentley et al. [97] (thebaine (**4**) → thevinone (**16a**) → dihydrothevinone (**104**) → **397a**). Treatment of **397a** with concentrated hydrochloric acid resulted in a mixture of the *Z* and *E* dihydrothebainone ((*Z*)**-398a**, (*E*)**-399a**) derivatives in a *Z*:*E* ratio of 1:7. When 20*S*-phenyldihydroorvinol (**397b**, R = H) was subjected to acid-catalyzed rearrangement, the isomers were formed in a *Z*/*E* ratio of 1:9 ((*Z*)-**398b**/(*E*)-**399b**, Figure 101). The mixtures were separated by column chromatography, and the structure of the geometric isomers was verified by NMR spectroscopy (NOE experiments).

The prepared ligands were evaluated in competitive binding assays with transformed CHO cell expressing human ORs, using the radioligands [^3^H]DAMGO (μ-OR), [^3^H]U69,593 (κ-OR), and [^3^H]Cl-DPDPE (δ-OR) [254]. The prepared flavonepenthone analogues displayed moderate to high affinities for all three ORs: K_i_ (μ) = 3.6–29.6 nM, K_i_ (κ) = 0.1–5.3 nM, and K_i_ (δ) = 9.3–115 nM, and surprisingly possessed a certain selectivity for κ-ORs. The *E*-alpha-methylbenzylidene derivative ((*E*)-**399a**, K_i_ (μ-OR) = 4.5 nM, K_i_ (κ-OR) = 0.1 nM, and K_i_ (δ-OR) = 5.6 nM) had considerably higher affinity than the corresponding *Z*-isomer ((Z)-**398a**, K_i_ (μ-OR) = 30 nM, K_i_ (κ-OR) = 5.3 nM, and K_i_ (δ-OR) = 115 nM). Of the series, (*E*)-**399a** displayed the highest affinity and the greatest selectivity for κ-ORs (κ/μ = 45, κ/δ = 56).

Diverse pharmacokinetic and pharmacodynamic factors account for the tradeoff between efficacy and abuse potential. For example, the µOR agonist loperamide effectively blocks gut motility in the treatment of diarrhea but is actively extruded at the blood brain barrier (BBB), thus having little abuse potential, unless taken at large doses [255]. Enzymatic deacetylation converts the prodrug heroin to its more active metabolites 6-monoacetylmorphine and morphine [256]; it follows that the net effect of heroin is the composite of at least three molecules with differing plasma kinetics and pharmacological potency. Theoretically, an inhibitor of plasma butyrylcholinesterase in plasma [257] might disable the metabolism of heroin, rendering it less potent. Tolerance to repeated doses of opiates can occur in conjunction with a paradoxical up-regulation of µORs in the spinal cord [258], and *post-mortem* assays of brain from heroin-addicted individuals has revealed a loss of activity of one of the enzymes producing cAMP [259], consistent with a desensitization of µORs. These responses mark the phenomenon of tolerance, whereby a dependent user can consume opioids at doses likely to be fatal to inexperienced users. The agonist-induced desensitization, a property shared with many other classes of neurotransmitter receptors, accounts for the withdrawal syndrome, since endogenous opioid peptides no longer suffice to activate brain receptors that have adapted to an artificial excess. We might liken this to economic inflation; if too much money is printed, its buying power declines, and the restitution of normality entails a painful and prolonged adaption period. Similarly, precipitous opioid withdrawal is a life-threatening condition calling for medical treatment, which is increasingly based on partial agonists such as buprenorphine [260].

Buprenorphine (**96**, K_i_ (μ-OR) = 1.5 nM, K_i_ (κ-OR) = 2.5 nM, K_i_ (δ-OR) = 6.1 nM, and K_i_ (NOP) = 77) in combination with antagonists such as naloxone or naltrexone constitutes an effective and safe alternative to methadone in the substitution therapy of opiate-addicted humans. Its efficacy may derive from the net mix of selectivity and partial agonism. In order to promote the favorable NOP-agonist and κ-OR antagonist effects, the μ-OR agonist effect of buprenorphine must be essentially nullified. This is achieved by combining buprenorphine (**96**, 4 mg/day) with naloxone (50 mg/day) [261], which has approximately ten-fold μ/κ and μ/δ selectivity and little affinity for NOP. Given this theoretical principle, Kumar et al. [114] prepared a series of C20-aryl-substituted buprenorphine analogues in 2014 with the aim to generate novel orvinol-type OR-ligands with zero or very low efficacy for the μ- and κ-ORs, while retaining a buprenorphine-like profile for NOP receptors (partial agonist). *N*^17^-Cyclopropylmethyl-dihydronorthevinone (**112a**) was subjected to addition with large variety of Grignard reagents to yield the corresponding *N*^17^-cyclopropylmethyl-20*S*-aryl-thevinols (**395g**, **400a–l**, Figure 102).

For preparation of their 20*R*-phenyl analogues, the 7-α-formyl derivative, which was prepared from *N*^17^-Cyclopropylmethyl-northebaine (**9d**) and acroleine, was reacted with phenylmagnesium bromide to obtain a 1:1 mixture of 20*R* and 20*S* secondary alcohols. The 7α-benzoyl compound was prepared by Swern oxidation of the above derivatives. Grignard reaction of the latter with methylmagnesium iodide yielded *N*^17^-cyclopropylmethyl-20*R*-phenyl-thevinol. When applying pyridyl Grignard reagents, there was no addition reaction, and consequently, 2-pyridyl as well as 4-pyridyl-lithium were used. This procedure yielded thevinols with the opposite (20*R*-) configuration. The target 20-aryl-orvinols (**401a–l**) were prepared by 3-*O*-demethylation with the strong nucleophile propanethiolate anion in a highly polar aprotic solvent (HMPA) or with L-selectride in THF. The lead compound, BU127 (20*S*-phenyl-dihydroorvinol (**396g**, R = Ph)), showed high affinity but low efficacy, and it had antagonist properties at μ- and κ-ORs (K_i_ (μ-OR) = 0.71 nM, K_i_ (κ-OR) = 0.49 nM, and K_i_ (δ-OR) = 1.9 nM). BU127 had slightly higher affinity for NOP receptors (K_i_ (NOP) = 43.2 nM) compared to buprenorphine [114].

In 2015, Cueva et al. [131] aimed to synthesize OR ligands as analgesics with reduced side effects, which would mimic the pharmacological characteristics of the buprenorphine-naltexone combination as used in addiction treatment. The 20-methyl substituent of the native orvinol derivatives was moved to the 7β position and a bulky aryl substituent was introduced in position 20 to enhance the activity for NOP receptors while maintaining κ- and μ-OR antagonism.

For the synthesis of 7β-methyl-*N*^17^-cyclopropylmethyl-nororvinol analogues (**407**, **409**, Figure 103), *N*^17^-cyclopropylcarbonyl-northebaine (**9e**) was chosen as the starting material. In a LiBF_4_-catalyzed [44] Diels–Alder reaction with methacrolein, **9e** was successfully converted to a 1.4:1 mixture of the 7α-methyl-7β-formyl (**402**) and 7α-formyl-7β-methyl (**403**) isomers. After separation of the isomers by column chromatography, the 7α-formyl-7β-methyl (**403**) derivative was reacted with various arylmagnesium halides in the presence of *n*Bu_4_NBr in THF (reflux, 48 h) to obtain the secondary alcohols (**404**) with 20*S* absolute configuration. Reduction of the carbonyl group of **404** with LiAlH_4_ in THF resulted in the *N*^17^-cyclopropylmethyl derivative (**408**). 3-*O*-Demethylation of the latter compounds with sodium propanethiolate in HMPA yielded the 20*S*-aryl-7β-methyl-*N*^17^-cyclopropylmethyl-nororvinol (**409**) derivatives. For preparation of orvinols with opposite stereochemistry at C-20, the secondary alcohols **404** were first converted in a Swern oxidation to the corresponding 7β-methyl-7α-aroyl derivatives (**405**). When reacting the latter compound (**405**) with LiAlH_4_ in THF, both CO groups were reduced: the *N*^17^-cyclopropylcarbonyl to *N*^17^-cyclopropylmethyl and the 7α-aroyl groups stereoselectively to the 20*R*-secondary alcohol (**406**). 3-*O*-Demethylation of **406** resulted in 20*R*-aryl-7β-methyl-*N*^17^-cyclopropylmethyl-nororvinols (**407**).

For preparation of the 18,19-dihydro analogues of 20-aryl-7β-methyl-*N*^17^-cyclopropylmethyl-nororvinols (**407**, **409**, Figure 103), as a first step, the 7α-formyl-7β-methyl-6,14-*endo*-ethano derivative (**403**) was reduced to **410** (H_2_, Pd-C, 6.9 bar, Figure 104) [131]. The target compounds (**414**, **416**) were synthesized applying the same preparative methods as described above for the Δ^18,19^ unsaturated derivatives. Of the various derivatives prepared, the lead compound was 7β-methyl-*N*^17^-cyclopropylmethyl-20*R*-dihydronepenthol (**414**, Aryl = phenyl, K_i_ (μ) = 0.1 nM, K_i_ (κ) = 0.04 nM, K_i_ (δ) = 0.25 nM, and K_i_ (NOP) = 80 nM), thus essentially showing subtype affinities to that of buprenorphine (**96**).

In 2016, the research group of Moiseev [262] reported the synthesis of 20-trifluoromethyl-6,14-ethenomorphinan analogues (Figure 105). A direct conversion of thebaine (**4**) with trifluoromethy vinyl ketone to 21,21,21-trifluorothevinone (**418**) failed because of the low stability of the dienophile. The authors elaborated on the synthesis of 21,21,21-trifluoromethyl-thevinone (**418**) from the acroleine adduct (**16c**) of thebaine (**4**). The reaction of the 7α-formyl compound (**16c**) with trifluoromethyltrimethyl silane (TMSCF_3_, Ruppert-Prakash reagent) in the presence of tetrabutlyammonium fluoride yielded a mixture of 20*R* (major isomer) and 20*S* (minor isomer) diastereomeric secondary alcohols (**417**) in a ratio of 17:1 (determined by ^19^F-NMR analysis). The mixture of the secondary alcohols (**417**) was converted by the Swern oxidation (oxalyl chloride/dimethyl sulfoxide system) to the fluorinated thevinone (**418**). The Grignard reaction of this ketone (**418**) with phenylmagnesium bromide gave a mixture of fluorinated 20*S*- (**420**) and 20*R*-phenylthevinols (**419**) in a 4:3 ratio according to ^19^F-NMR spectral data. The epimers were separated by column chromatography and characterized using 1D and 2D-NMR methods.

The same group synthesized and analyzed numerous *N*-substituted (e.g., allyl, cyclopropylmethyl-) fluorinated thevinols and dihydrothevinols by tandem HRMS in the electrospray ionization mode (ESI) [263]. Recently, Zelentsova et al. [264] elaborated on a simple method for determination of the C-20 absolute configuration of 20-aryl-21,21,21-trifluoro-thevinols and orvinols using simple one-dimensional ^19^F NMR spectroscopy.

About twenty years after the findings by the Makleit group [110] that nepenthones are active ligands of opioid receptors, the research group of Li et al. at the Fudan University Shanghai proceeded to develop a series of novel nepenthone derivatives (Figure 106). In 2017, they reported on the syntheses and pharmacological evaluation of their nepenthone analogues [164]. They had prepared nitrophenyl vinyl ketones from the corresponding nitrobenzaldehydes (*o*-, *m*-, *p*-) in two steps. The formyl group was then reacted with vinyl magnesium bromide in a Grignard reaction in THF at −65 °C. The formed secondary vinyl alcohols were oxidized by Jones reagent (CrO_3_, H_2_SO_4_, acetone, <10 °C) to yield the α,β-unsaturated ketones. Diels–Alder reaction of thebaine (**4**) with the appropriate dienophile resulted in the novel nitro-nepenthone analogues (**421a–c**). Specifically, nitro-*N*^17^-nornepenthones (**423a–c**) were prepared from **421a–c** by treatment with diethyl azodicarboxylate followed by hydrolysis with 1M HCl. Alkylation of the *nor*-compounds (**423a–c**) with cyclopropylmethyl bromide yielded nitro-*N*^17^-cyclopropyl-methyl-nornepenthones (**424a–c**). The nitro derivatives (**421a–c**, **424a–c**) were reduced using hydrazine hydrate (Raney-Ni, ethanol, 60 °C) to amino-*N*^17^-nornepenthones (**422a–c**, **425a–c**).

Nepenthone (**16b**), *N*^17^-cyclopropylmethyl-nornepenthone (**390a**) and their novel nitro (**421a–c**, **424a–c**) and amino (**422a–c**, **425a–c**) analogues were pharmacologically characterized (in vitro and in vivo). *N*^17^-Cyclopropylmethyl-nornepenthone (**390a**, SLL-020ACP) [110,164] was found to be the lead κ-OR agonist compound with the highest affinity for κ-ORs (K_i_ (κ) = 0.4 nM), and with an excellent selectivity (μ/κ = 339, δ/κ = 2034; GR103545 (an arylacetamidopiperazine derivate): K_i_ (κ-OR) = 0.02 nM, K_i_ (μ-OR) = 16.2 nM, and K_i_ (δ-OR) = 536 nM and μ/κ = 810, δ/κ = 26,800).

Based on the above results regarding *N*^17^-cyclopropylmethyl-nornepenthone (**390a**, SLL-020ACP) [164] and owing to the results of the Husbands group findings concerning 7β-methyl orvinol analogues [131], the same Shanghai research group synthesized numerous 7β-methyl-7α-substituted-*N*^17^-cyclopropylmethyl-dihydro nornepenthone derivatives (**430**, Figure 107) [265] to identify novel κ-OR antagonists. Cueva et al. [131] identified an *N*^17^-cyclopropylmethyl-20-phenyl-7β-methyl-orvinol derivative (**414**, Aryl = phenyl K_i_ (μ-OR) = 0.1 nM, K_i_ (κ-OR) = 0.04 nM, K_i_ (δ-OR) = 0.25 nM, and K_i_ (NOP) = 80 nM) as the lead compound for the development of an analgesic with potential for a reduced side effect profile and efficacy as therapy against relapse to opiate addiction. Sun et al. [265] attempted to reverse the κ-OR agonist activity of SLL-02ACP (**390a**) based on the hybridization concept achieved by introduction of a 7β-methyl substituent.

*N*^17^-Cyclopropylcarbonyl-northebaine (**9e**, Figure 107) was chosen as the starting material, which could be prepared from thebaine (**4**) in two steps via *N*-demethylation and *N*-alkylation with cyclopropylmethyl bromide. First, the diene (**9e**) was reacted with metacrolein in the presence of LiBF_4_ catalyst [44] to form the 7α-formyl-7β-methyl Diels–Alder adduct (**403**) in low yield (25%). Saturation of the Δ^18,19^ double bond (H_2_, Pd-C, EtOAc) led to the 6,14-*endo*-ethano derivative (**426**, 90%). The formyl group and the cyclopropylcarbonyl group of the latter compound (**426**) were simultaneously reduced with LiAlH_4_ in THF to the primary alcohol and to the cyclopropylmethyl group giving **427** in 80% yield. Swern oxidation of the latter gave the 7α-formyl-7β-methyl compound (**428**) in 95% yield. The aldehyde (**428**) was converted with the corresponding aryl halides (*n*BuLi, hexane, THF) to the secondary alcohols (**429**, 21–50%), which were thereafter oxidized with 1,1,1-tris(acetoxy)-1,1-dihydro-1,2-benziiodoxol-3-(1*H*)-one (DMP, Dess-Martin periodinane) to the target nepenthone derivatives (**430**, 17–80%). Numerous esters (**431**) were also prepared from the primary alcohol (**427**) with the corresponding acyl chlorides in the presence of triethyl amine in dichloromethane (room temperature, 3 h, 18–83%). The new compounds were assayed for binding to ORs. The intermediates primary alcohol (**427**) and 7α-formyl-7β-methyl compound (**428**) showed low nanomolar affinities for κ-ORs and a weak selectivity for κ- over μ-ORs [265]. The introduction of a 7β-methyl substituent in the nepenthone series led to a loss of potency and subtype selectivity. Surprisingly, 7β-methyl-*N*^17^-cyclopropylmethyl-nornepenthone (**430**, R_1_ = phenyl, SLL-603) was found to be a full κ-OR agonist (K_i_ (μ-OR) = 304 nM, K_i_ (κ-OR) = 33.2 nM, and K_i_ (δ-OR) = 345 nM).

A series of *N*^17^-cyclopropylmethyl-7α-phenyl-6,14-ethenomorphinans, as potential κ-opioid receptor ligands, were synthesized by Xiao et al. [266] from *N*^17^-cyclopropylmethyl-northebaine (**9d**, Figure 108). 

The 7α-(4-aminophenyl) derivative (**422c**, *p*NH_2_, SLL-004C, Figure 106) was chosen as a primary scaffold to this investigation [266]. First, the message component [230,231] was optimized by introduction of the *N*^17^-cyclopropymethyl substituent and saturation of the Δ^18,19^ double bond. For optimizing the potential address component of the OR-ligand, the 7α-(4-aminophenyl) group of (**433**) was substituted with a large variety of groups (Figure 108, compound **434**) resulting in *secondary amines* (R = *N*HMe, *N*HEt, *N*HnPr, *N*CPM, *N*CH_2_Ph, *N*,*N*’-dimethylaminoethyl, *N*,*N*’-dimethylaminopropyl, (2-acetamido)amino, and [2-(ethoxycarbonyl)ethyl]amino), *tertiary amines* (R = pyrroidinyl, piperidinyl, piperazinyl, *N*’-CPM-piperazinyl, *N*’-acetamido, *N*’-benzoyl-piperazinyl, and *N*’-p-tosyl-piperazinyl), and *amides* (R = (*N*’-formyl)amino, (*N*’-acetyl)amino-, (*N*’-phenylacetyl)amino, and (*N*’-benzoyl)amino). The lead compound of the series [266] was the highly selective and potent κ-OR agonist SLL-039 (**434**, R = NHCOPh, K_i_ (μ-OR) = 321 nM, K_i_ (κ-OR) = 0.47 nM, K_i_ (δ-OR) = 133.7 nM, and κ/μ = 683, κ/δ = 284).

The same research group [267] designed and synthesized 7α-arylacetamidyl phenyl-*N*^17^-cyclopropylmethyl-6,14-ethenomorphinan derivatives (**435**, Figure 108) by acylation of the 7α-(4-aminophenyl) (**433**) key intermediate. As their rationale for design, the authors linked their own developed morphinan (**434**, R = NHCOPh, SLL-039 [266]) with known κ-OR selective arylacetamides (e.g., U50488H, GR103545). The target ligands were prepared from **433** with various acids (Q = Ph, 3,4-dichlorophenyl, 4-CF_3_-phenyl, 3-benzothiophenyl, 3-thiophenyl, 2-thiophenyl, 3-piridinyl, 4-CN-phenyl, 4-NO_2_-phenyl, 3,4-dimethylphenyl, 4-CH_3_-phenyl, 3-CH_3_-phenyl, 4-OCH_3_-phenyl, 3-OCH_3_-phenyl, 3-NO_2_-phenyl, 3-CN-phenyl, and 3,4-dimethyphenyl) in the presence of DIPEA (2.2 equiv.) and HATU (1.7 equiv.). Among this series, the 7α-(3,4-dimethylphenylacetamidyl) derivative (**435**) displayed nanomolar affinity and relatively high selectivity for the κ-ORs (K_i_ (μ-OR) = 114 nM, K_i_ (κ-OR) = 1.8 nM, K_i_ (δ-OR) = 293 nM, and κ/μ = 63, κ/δ = 162). The compound **435** with the native 3,4-dichlorophenylacetamidyl substituent showed only moderate affinity for κ-ORs (K_i_ (μ) = 69 nM, K_i_ (κ) = 215 nM, K_i_ (δ) = 199 nM), essentially without subtype selectivity (κ/μ = 0.3, κ/δ = 0.9).

Recently, He et al. [268] described a three-step transformation of *N*^17^-cyclopropylmethyl-northebaine (**9d**) to 7α-(*m*-substituted-phenyl)-*N*^17^-cyclopropyl-methyl-6,14-ethenomorphinan derivatives (**437a–s**, **438**, Figure 109), which were designed based on the previously developed κ-OR ligands, SLL-020ACP (**390a**) [143], and SLL-039 (**434**, R = NHCOPh) [266], of the Fudan group. Diels–Alder reaction of **9d** with 3-nitrostyrene in xylene afforded the 7α-(*m*-nitro-phenyl)-6,14-*endo*-etheno adduct (**424b**) in 46% yield [164]. Hydrogenation of **424b** under heterogenous catalytic conditions resulted in the 7α-(*m*-amino-phenyl)-6,14-*endo*-ethano derivative (**436**) with 92% yield. The aniline key intermediate (**436**) was converted in separate reactions to the target 7α-(*m*-substituted-phenyl)-derivatives (**437a–s**, Figure 109). Binding affinities to μ- δ- and κ-ORs were determined by ligand competition experiments ([^3^H]DAMGO (μ-OR), [^3^H]DPDPE (δ-OR) and [^3^H]U69,593 (κ-OR)) [268]. From this series, SLL-1206 (**437j**, R = OCH_3_, K_i_ (μ-OR) = 1880 nM, K_i_ (δ-OR) = 440 nM, K_i_ (κ-OR) = 7.1 nM, and subtype selectivity: κ/μ = 264, κ/δ = 62) was identified as the lead compound based on its advantageous in vivo and in vitro characteristics. The oripavine analogue of this compound (**438**, R = H) displayed an increased affinity (K_i_ (μ) = 0.294 nM, K_i_ (δ) = 0.072 nM, and K_i_ (κ) = 0.016 nM) with loss of subtype selectivity to κ-OR (κ/μ = 18.4, κ/δ = 4.5).

### 2.8. 1-Substituted 6,14-ethenomorphinans

Recently, the Shults research group [188] thoroughly investigated the feasibility of the introduction of various pyrimidine substituents in the position one of the morphinan aromatic A-ring. From a pharmacologically convergent point of view, the pharmacophore 2-substituted 4-arylpyrimidine group with adenosine receptor antagonist properties was combined with a 6,14-ethenomorphinan scaffold. Thebaine-*N*-phenylmaleimide Diels–Alder adduct (**75i**) [43] was chosen as the starting material. Iodination of compound **75i** was performed as described earlier by Bauman et al. [269] with *N*-iodosuccinimide (NIS) in trifluoroacetic acid. The 1-iodo derivative (**439**, Figure 110) was subjected to a Sonogashira cross-coupling with trimethylsilylacetylene to obtain the corresponding 1-(trimethylsilenylethynyl) compound, which was in turn de-silylated with Bu_4_NF in dichloromethane to yield the 1-ethynyl-6,14-ethenomorphinan compound (**440**) [270]. The latter (**440**) was reacted under Sonogashira conditions with aryl acid chlorides to obtain the alkynones (**441**) in good yields (60–77%).

Cyclocondensation of the alkynones (**441**) with acetamidine-, guanidine-, benzamidine-, and 4-nitrobenzamidine-hydrochloride in acetonitrile afforded the corresponding 1-pyrimidino-6,14-ethenomorphinan derivative (**442**). The reduction of the succinimides (**442**) with sodium borohydride in THF took place selectively and resulted in 1-(6-(aryl)pyrimidin-4-yl)-7α,8α-fused 2′α-hydoxylactam-type 6,14-ethenomorphinans (**443**).

*O*-Demethylation of selected succinimides (**442**, R = F, Br, R_1_ = Me, 4-NO_2_-C_6_H_4_) with an excess of boron tribromide in CHCl_3_ occurred both in 3-OCH_3_ and 6-OCH_3_ positions, as described earlier in the field of thebaine-maleimide adducts by Shults et al. [187], to obtain the respective C-bridged 6-*O*-demethyl-oripavine derivatives in 50–90% isolated yield. The analgesic activity of selected compounds (**440**, **442**, R_1_ = CH_3_, X = F, Br, OCH_3_) was evaluated in the rat tail-flick test after intraperitoneal (i.p.) administration [188]. The authors concluded that the antinociceptive potency of the 1-pyrimidin-4-yl derivatives (**442**, R_1_ = CH_3_, X = F, Br, and OCH_3_) depended heavily on the character of the substituent introduced in position six of the pyrimidine ring. It is noteworthy that only compound **443** (R_1_ = CH_3_, R = 4-bromophenyl), which has a 4-bromophenyl substituent in position six of the pyrimidine nucleus, was highly effective as an analgesic at a dose of 1 mg/kg i.o. (tail-flick latency, 8.7 ± 1.3, 41.3% MPE).

Binding characteristics of prominent 6,14-ethenomorphinans are presented in Table 3.

## 3. Radiochemistry

In this final section, we summarize the advances in the radiosynthesis of labeled orvinols over the past few decades. Radiolabeled non-selective 6,14-ethenomorphinans are widely used in brain imaging because they are excellent radiotracers for in vivo visualization and quantification of ORs by PET [28]. However, some non-orvinol-based PET radioligands are also known, such as the µ-OR selective agonist [^11^C]carfentanil [280], the δ-OR selective antagonist *N1′*-[^11^C]-methylnaltrindole (N*1′*-[^11^C]Me-NTI) [281], the µ/κ-OR selective antagonist [^18^F]cyclofoxy ([^18^F]FcyF) [282], and the κ-OR selective agonist arylacetamidopiperazine derivative [^11^C]GR103545 [283]. The applied strategies to label orvinols with positron (for PET) and gamma emitting (for SPECT) radionuclides are presented in Figure 111. There are a number of detailed reviews of the orvinol-based radioligands used for OR imaging [8,27,28,29,284,285,286,287]. Agonist potency and molar activity of the tracer are key issues in the context of molecular imaging, as some tracers are of such potency that even microgram quantities may exert pharmacological effects.

### 3.1. Tritium Labelled Orvinol Derivatives

As noted above, many assays of receptor specificity employ competition against selective ligands labeled with tritium, a weak beta-emitter. Lewis et al. [288,289] developed radiolabeling methods for the preparation of tritium-labeled orvinols to study the binding of these derivatives to ORs in brain tissue sections by means of autoradiography in vitro. One significant procedure for the radiosynthesis of [15,16-^3^H]-etorphine was the catalytic hydrogenation of the enamine bond of the 15,16-didehydro precursor by tritium gas [289]. Diprenorphine binds with high affinity to µ-, δ-, and κ-ORs, and therefore has been designated as the universal ligand for in vitro and in vivo studies of OR pharmacology. Accordingly, [^3^H]diprenorphine was synthetized by a method similar to that described above for visualizing ORs by autoradiography at Johns Hopkins University [290,291] in the 1970s. That research served as the basis for the subsequent development of [^11^C]diprenorphine as a PET ligand. 15,16-didehydro-buprenorphine was also tritiated (by hydrogenation with ^3^H_2_) at positions 15 and 16 in a similar manner, using PdO/BaSO_4_ as catalysts, and resulting in a specific molar radioactivity of 2.35 GBq/µmol [292].

### 3.2. [^125^I]Iodinated Orvinols

The gamma-emitting tracer 2-[^125^I]iodo-buprenorphine (**445b**, Figure 112) was synthesized by Debrabandere et al. [293] as the first iodinated orvinol obtained by the oxidative chloramin-T method; its initial application was for screening of urine samples from drug users by means of radioimmunoassay (RIA).

2-[^125^I]Iodo-diprenorphine (**445a**) and 2-[^125^I]iodo-etorphine (**446**) were later developed by Tafani et al. [294]. The appropriate orvinols were iodinated by electrophilic substitution on the A-ring using [^125^I]NaI and 1,3,4,6-tetrachloro-3α,6α-diphenyl-glycoluril (iodogen) as an oxidizing agent, and the binding properties of the labeled 6,14-ethenomorphinans were compared with that of the μ-OR selective [^125^I]iodo-NH_2_-carfentanil in mouse brain. (Figure 112) Musachio and Lever [295] reported the synthesis of radioiodinated SPECT tracers, namely *N*-[^125^I/^123^I]iodoallyl-nordiprenorphine (**448**) and 6-*O*-[^125^I/^123^I]iodoallyl-6-*O*-desmethyl-diprenorphine (**450**).

These compounds were prepared from the corresponding 3-(tri-*n*-butylstannyl) prop-2-enyl derivatives (**447**, **449**, Figure 112) via radioiododestannylation using radioactive NaI and chloramine-T in methanolic acetic acid at ambient temperature. In another report, the 6-*O*-[^123^I]iodoallyl-6-*O*-desmethyl-diprenorphine ([^123^I]-*O*-IA-DPN, **450**) was utilized in ex vivo autoradiographic studies, where the tracer uptake was measured *post-mortem* [296]. Their specific binding to ORs in mouse brain was verified with naltrexone blockade [296]. Furthermore, this radioiodinated ligand was evaluated by SPECT imaging, showing selective localization in regions of living baboon brain known to have high OR densities, i.e., striatum, thalamus, and temporal cortex. In 1995, Wang et al. developed the synthesis of a novel *O*-radioiodoallyl analogue at the tertiary hydroxyl position in the 7α-side chain [297]. The 7α-*O*-[^125^I]iodoallyl-diprenorphine (7α-*O*-[^125^I]IA-DPN, **452**) was prepared from [*E*]-7α-*O*-[((3-tri-*n*-butylstannyl)prop-2-enyl)oxy]-diprenorphine (**451**, Figure 112) by radioiododestannylation. Later, the in vitro OR-binding study of 7α-*O*-[^125^I]IA-DPN (**452**) revealed its high affinity (K_i_ = 0.4 nM) for mouse brain membranes. The in vivo experiments showed 63% specific binding of the radioiodinated orvinol ligand in brain. OR labelling was confirmed by ex vivo autoradiographic analysis [298].

### 3.3. Carbon-11 Labeled Orvinol Derivatives for PET

The incorporation of positron-emitting ^11^C isotope (half-life of 20.34 min) into an orvinol derivative was first developed by Mazière et al. [299]. Noretorphine (**453**, Figure 113A) was reacted with [^11^C]formaldehyde in the presence of acetic acid in acetonitrile and reduced by sodium cyanoborohydride at 70 °C for 5 min. The [*N*-methyl-^11^C]-etorphine (**455**) was obtained with a molar activity of 29.6 GBq/µmol in a total synthesis time of 25 min. Burns et al. collected evidence for a feasible radiosynthesis of ^11^C-labeled diprenorphine [300]. Studying the addition of [^11^C]methyllithium to 7-α-acetyl-6,14-ethenomorphinan derivatives, 20-[^11^C]methyl-thevinol was prepared using thevinone (**16a**) as precursor. Furthermore, the Burns group accomplished the synthesis of *N*-cyclopropylmethyl-dihydronororvinone from thebaine in 11 steps as a potential precursor for the synthesis of 20-[^11^C]methyl-diprenorphine.

In the late 1980s, initial studies from the Hammersmith group aimed to identify suitable radiolabeling positions and techniques suitable for providing the ^11^C-labeled orvinols in appropriate yields and purity, with consideration of the temporal constraints of working with this short-lived radionuclide. Luthra et al. carried out the synthesis of new type ^11^C-labeled orvinol derivatives, namely [*N*-cyclopropylmethyl-^11^C]diprenorphine (**457a**) (Figure 113B) [301] and [*N*-cyclopropylmethyl-^11^C]buprenorphine (**457b**) [302], using a similar method. In the first step, [carboxy-^11^C]cyclopropanecarbonyl chloride was prepared with cyclotron-produced [^11^C]CO_2_ via Grignard reaction and was then reacted with nordiprenorphine (*nor*-DPN (**114a**)) and norbuprenorphine (*nor*-BPN (**114b**)). Finally, the amides formed were reduced with lithium aluminum hydride. The radiochemical yields (decay-corrected from [^11^C]CO_2_) were between 5 and 20% for [*N*-cyclopropylmethyl-^11^C]DPN (**457a**) and about 20% for [*N*-cyclopropylmethyl-^11^C]BPN (**457b**). The overall preparation time was 53 min for [*N*-cyclopropylmethyl-^11^C]DPN (**457a**) and 57 min for [*N*-cyclopropylmethyl-^11^C]BPN (**457b**), corresponding to the three half-lives. The biodistribution of these labeled compounds was studied by PET imaging [303], in which radiometabolite analysis indicated the rapid formation of labeled metabolites in baboon plasma and mouse brain, which is an unfavorable property for quantitative PET analysis. A comparison of their in vivo properties suggested that ^11^C-labeled diprenorphine (**457a**) was a more promising radioligand for OR PET imaging, since it resulted in a significantly higher striatum to cerebellum ratio than [*N*-cyclopropylmethyl-^11^C]BPN (**457b**).

The introduction of more facile labeling chemistry via alkylation of the 6-*O*-methyl position led to the development of further methods for the radiosynthesis of ^11^C-labeled diprenorphine (**460a**, Figure 114A) [183] and buprenorphine (**460b**) 184]. This was enabled by the fundamental work of Kopcho and Schaeffer [182] in the field of the selective 6-*O*-demethylation of 6,14-ethenomorphinans.

The labeling of the six position led to radiotracers that were metabolically more stable compared to [*N*-cyclopropylmethyl-^11^C]DPN (**457a**) and [*N*-cyclopropylmethyl-^11^C]BPN (**457b**), both of which were rapidly metabolized via *N*-dealkylation, thus losing radioactive moiety but yielding potentially brain-penetrating metabolites. For the radiosynthesis of both 6-*O*-^11^C-methylated radioligands, a two-step procedure was applied by Lever et al. [183,184], in which the 3-*O*-silylated precursor (**458a–b**) was reacted with [^11^C]methyl iodide in DMF in the presence of NaH, followed by removal of the 3-*O*-*tert*-butyldimethylsilyl group with 1 M HCl. However, in the course of the preparation of [6-*O*-methyl-^11^C]buprenorphine (**460b**), [3-*O*-methyl-^11^C]-6-*O*-desmethyl-buprenorphine(*iso*-[^11^C]BPN) was detected as a by-product (<3%, inseparable by reverse phase HPLC) of the alkylation reaction. The applied precursor contained a *tert*-butyl group at C-20 position, which resulted in steric hindrance for alkylation of the 6-hydroxy position and also resulted in desilylation and subsequent 3-*O*-methylation of the resulting triol.

To overcome this difficulty, Luthra et al. [185] developed a refined method via selection of a more base-stable protection group for 6-hydroxy position, which gave solely the 6-*O*-labeled product. Their examination of the base stability of several different protecting groups showed the trityl group to be the most suitable protecting group for the 3-hydroxy position, due to its base stability and facile removal under mild conditions. Accordingly, [6-*O*-methyl-^11^C]diprenorphine (**460a**) and [6-*O*-methyl-^11^C]buprenorphine, **460b**) were synthesized from the 3-*O*-trityl-protected precursors (TDDPN (**461a**), TDBPN (**461b**), Figure 114A) by ^11^C-methylation using NaH, where the following step was 3-*O*-detritylation by acid hydrolysis. Galynker et al. [304] accomplished the radiosynthesis of **460b** using a modification of the method of Luthra et al. [185], and investigated its distribution in baboon brain by PET. The tracer retention was consistent with the known distribution of OR subtypes in humans and non-human primates (striatum > thalamus > cingulate gyrus > frontal cortex > parietal cortex > occipital cortex > cerebellum). [6-*O*-Methyl-^11^C]diprenorphine (**460a**) was applied by Koepp et al. [305] in humans with reading-induced epilepsy. Here, their aim was to undertake a competitive binding assay in vivo to measure focal dynamic competition from endogenous opioid peptides.

In 2013, a research group of the University of Manchester modified the above-mentioned method [306] in compliance with good manufacturing practice for routine production of **460a** suitable for human administration (Figure 114A). In the first step of this radiosynthesis, NaOH was used as the base instead of NaH to deprotonate the 6-hydroxy group of the 3-*O*-trityl-6-*O*-desmethyl-diprenorphine precursor (TDDPN, **461a**), which enabled the subsequent alkylation of the obtained alkoxide derivative with [^11^C]methyl iodide. This labeling step was followed by acid-catalyzed detritylation. 

This optimized radiochemical method enabled the production of **460a** with 97.0% radiochemical purity and an average radiochemical yield of 32 ± 5.8% based on 80 syntheses (decay corrected to the time of [^11^C]methyl iodide incorporation), and with 10 times higher molar activity (mean 242 GBq/µmol) compared to previous attempts [185]. Obtaining high molar activity is less an issue for antagonist ligands but is still desirable from the perspective of minimal dosing. They also accomplished the synthesis of **460a** with [^11^C]methyl triflate. However, the greater reactivity of this methylating agent reduced the selectivity of the alkylation towards the 6-hydroxy position, resulting in lower radiochemical yield.

In 2008, Marton et al. reported the preliminary results of the development of new radiolabeled phenethyl-orvinol derivatives, of which one was [6-*O*-methyl-^11^C]phenethyl-orvinol ([^11^C]PEO, **465**, Figure 114B) [307]. They later reported the detailed methods for the synthesis of the precursor, namely 3-*O*-trityl-6-*O*-desmethyl-phenethyl-orvinol (TDPEO, **463**), and the radiosynthesis of [^11^C]PEO (**465**), as well reporting on its ex vivo studies biodistribution and in vivo evaluation by PET in rodent brains [98]. The carbon-11 labeling was achieved by ^11^C methylation of TDPEO (**463**) in the presence of NaH in DMF at 90 °C for 5 min, which was by deprotection in 1M HCl at 90 °C for 2 min to obtain [^11^C]PEO (**465**). The brain uptake kinetics and regional binding distribution of this compound were studied in rats, showing a similar pattern to that reported for [^11^C]DPN (**460a**) and other PET tracers used for brain imaging of ORs. For the no-carrier-added [^11^C]PEO (**465**) synthesis, a selective retention of radioactivity was seen in the striatum, thalamus, and the frontal cortex, indicating the composite of μ- and κ-ORs. Pretreatment with 1 mg/kg naloxone completely blocked the binding of the carbon-11 labeled PEO derivatives in these regions, relative to the non-binding reference regions such as cerebellum (which is nearly devoid of ORs). Preliminary PET investigation on rodents [307] had also shown specific uptake in the spinal region, which is seldom quantified due to its small cross-sectional area, despite the importance of OR-signaling in the spinal grey matter in the context of analgesia. However, they did not undertake blocking studies to confirm the pharmacological identity of the spinal binding. Table 4 summarizes the important radiochemical parameters of the radiolabeling of ^11^C-labeled orvinols.

### 3.4. ^18^F-Labeled 6,14-ethenomorphinans

In the early 1990s, the favorable nuclear properties of fluorine-18 with respect to longer half-life (109.8 min) and inherently better spatial resolution of source maps motivated researchers to develop ^18^F-labeled orvinol analogues. While ^11^C-PET studies must be completed within 80 min or less, more prolonged ^18^F-PET recordings (as long as four hours) enable measurement of tracers, which are relatively slow to attain binding equilibrium in the brain. Chesis et al. developed a two-step procedure for the radiofluorination of diprenorphine derivatives at the position *N*-17 by ^18^F-fluoroalkylation of nordiprenorphine (**114a**, Figure 115) [309]. Nordiprenorphine (**114a**) was reacted with I(CH_2_)_n_^18^F, which had been previously prepared from iodoalkyl triflates with [^18^F](*n*-Bu)_4_NF. The ex vivo distribution of the synthetized *N*^17^-(2-[^18^F]fluoroethyl)- (**466a**), *N*^17^-(3-[^18^F]fluoropropyl)- (**466b**), and *N*^17^-((*S*)-3-[^18^F]fluoro-2-methylpropyl)-nordiprenorphine (**466c**) was determined in rats, showing accord with the known distribution of ORs. In addition, *N*^17^-(3-[^18^F]Fluoropropyl)-norbuprenorphine (*N*-(3-[^18^F]FP)-*nor*-BPN) was prepared by Bai et al. [310] with *N*-alkylation of norbuprenorphine (**114b**) using 1-[^18^F]fluoro-3-iodopropane. The binding properties of *N*^17^-(3-[^18^F]fluoropropyl)-*nor*-buprenorphine and *N*^17^-(3-[^18^F]fluoropropyl)-diprenorphine were investigated for ORs by means of ex vivo distribution in rodent brains and PET studies in baboons [310]. The absolute uptake of *N*^17^-(3-[^18^F]FP)-*nor*-BPN in rat brains was only half of that of *N*^17^-(3-[^18^F]FP-*nor*-DPN). PET studies of *N*^17^-(3-[^18^F]FP)-*nor*-DPN in baboons revealed its specific binding to ORs in the striatum relative to the cerebellum. However, compared to [^11^C]DPN the *N*^17^-(^18^F-fluoroalkylated)-orvinols, it proved to be less suitable for PET imaging than the parent compound due to lower specific binding in vivo and an unsuitable metabolite profile [310].

Subsequently, Wester et al. [308] accomplished the synthesis and biologic evaluation of another type of ^18^F-labeled 6,14-ethenomorphinan derivative, namely 6-*O*-(2-[^18^F]fluoroethyl)-6-*O*-desmethyl-diprenorphine ([^18^F]FE-DPN, **468**, Figure 115, K_i_ (μ-OR) = 0.24 nM, K_i_ (δ-OR) = 8.0 nM, and K_i_ (κ-OR) = 0.2 nM, all relative to standard in vitro competitive binding assays). Shifting the ^18^F-labeling to the six position resulted in a metabolically more stable radioligand with [^18^F]FE-DPN (**468**) having similar pharmacokinetics to that of [6-*O*-methyl-^11^C]DPN (**460a**).

The radiosynthesis of [^18^F]FE-DPN (**468**) was carried out with a three-step procedure consisting of ^18^F-fluorination of ethylene glycol-l,2-ditosylate, ^18^F-fluoroethylation of the 6-hydroxy position of 3-*O*-trityl-6-*O*-desmethyl-diprenorphine (TDDPN, **461a**), and acidic deprotection of the resulting compound. Subsequently, [^18^F]FE-DPN (**468**) has proven to be an appropriate PET probe for the investigation of OR status in clinical trials of pain signaling and processing, psychological studies, and tracer kinetic modeling [311,312,313].

In 2012, Marton and Henriksen reported the synthesis of a full agonist OR radiotracer, namely 6-*O*-(2-[^18^F]fluoroethyl)-6-*O*-desmethyl-phenethyl-orvinol ([^18^F]FE-PEO, **470**) [177]. In that report, they also described practical synthetic routes for the preparation of the 2-(*p*-toluenesulfonyloxy)ethoxy precursor (TE-TDPEO, **469**, Figure 115) and the reference standard (FE-PEO). This novel 6-(2-(tosyloxy)ethoxy) precursor (**469**) allows a two-step one-pot synthesis of [^18^F]FE-PEO (**470**) using direct nucleophilic radiofluorination in a radiochemical yield of 35 ± 8% and a molar activity of 55–128 GBq/µmol (1490–3458 mCi/μmol) at end of synthesis. On the basis of these promising results, further examination of [^18^F]FE-PEO (**470**) extended to its optimized radiosynthesis, in vitro characterization, and in vivo evaluation in Lister hooded rats [314]. ^18^F-FE-PEO (**470**) was synthesized from (TE-TDPEO, **469**) via an automated process using the GMP-compliant TRACERlab FX F-N radiosynthesis module with a yield of 28 ±15% and a molar activity of 52–224 GBq/µmol. [^18^F]FE-PEO (**470**) was the first ^18^F-labeled opioid receptor *agonist* with high-affinity, κ/µ subtype selectivity, and a slow pharmacokinetic profile that enabled good quantitation.

Later, Schoultz et al. [274] described the synthesis of 6-*O*-(2-[^18^F]fluoroethyl)-6-*O*-desmethyl-buprenorphine ([^18^F]FE-BPN, **471**, Figure 115) using a slightly modified three-step procedure. In addition, they applied this same method to the radiosynthesis of the previously mentioned [^18^F]FE-DPN (**468**) [308,315] and [^18^F]FE-PEO (**470**) [177]. Thus, a triplet of 6-*O*-^18^F-fluoroethylated derivatives was for the first time prepared on a standard automated radiosynthesis module with a three-step two-pot synthesis using the respective 3-*O*-trityl-6-*O*-desmethyl precursors (TDDPN (**461a**), TDBPN (**461b**), and TDPEO (**463**)). The production protocol began with the radiosynthesis of 2-[^18^F]fluoroethyl-tosylate ([^18^F]FEOTos), and proceeded via ^18^F-fluoroalkylation of TDDPN (**461a**), TDBPN (**461b**), and TDPEO (**463**). These latter reactions were performed in DMF and in the presence of NaH at 100 °C for 10 min in each case. Next, the resulting mixtures were cooled to 40 °C and stirred with 2 M HCl in ethanol for 5 min to deprotect the 3-*O*-position of the morphinan ring by the removal of the trityl group. Furthermore, the authors evaluated the in vivo characteristics of the three [^18^F]fluoroethylated orvinol radiotracers (**468**, **470**, **471**) by brain imaging with PET in rats [274]. The results of these studies corresponded well to the previously reported pharmacological properties of structurally analogous carbon-11 labeled compounds ([^11^C]DPN (**460a**) [185], [^11^C]BPN (**460b**) [304], and [^11^C]PEO (**465**) [98]). Based on this study, these fluorine-18 radioligands provide the full palette of agonist-antagonist characteristics: [^18^F]FE-DPN (**468**) (antagonist), [^18^F]FE-BPN (**471**) (mixed agonist-antagonist), and [^18^F]FE-PEO (**470**) (agonist)]. This repertoire allows for systematic investigations of the effect of intrinsic OR activity on ligand-exchange kinetics in PET imaging protocols [316].

Recently, Krüll et al. [246] accomplished the radiosynthesis of a buprenorphine-derived 4-[^18^F]fluoro-phenylazocarboxamide (**472**, Figure 116) as a novel type of fluorine-18 labeled buprenorphine analogue for the visualization of µ-ORs by PET. First, *tert*-butyl 4-fluoro-phenyl-azocarboxylate was prepared from the corresponding quaternary ammonium triflate using a conventional radiofluorination reaction in acetonitrile at 85 °C for 30 s. Next, the resulting azo ester was reacted with the primary amino group of the 7α-aminomethyl-6,14-ethenomorphinan derivative (**328a**) in the presence of cesium carbonate to yield the desired tracer (**472**). Recently, the chemistry of [^18^F]fluorinated morphinan OR ligands were reviewed by the Moiseev group [317].

Radiosynthesis of most currently available radiolabeled OR ligands are demanding on time and labor, often resulting in relatively low radiochemical yields, as shown in Table 4 and Table 5. This does not favor repeated examinations with carbon-11 tracers or the commercial distribution of fluorine-18 tracers from a production site. Consequently, radiolabeled OR ligands with more effective radiosyntheses are still desirable.

## 4. Conclusions and Outlook

The most recent pandemic of opioid dependence is related to the credulity of medical practitioners regarding claims that a slow-release formulation of an opiate should produce pain relief with minimal risk of abuse potential. This proved not to be the case; the enormous efforts in synthesizing the diversity of ligands described in this review may have been motivated by the aspiration to produce an effective analgesic that does not lead to dependence. Current pharmacological approaches for the treatment of opioid addiction emphasize replacement with partial agonists such as methadone or buprenorphine [318], sometimes in conjunction with the antagonist naloxone [319]. Despite its partial agonist action, buprenorphine is an effective analgesic [320], as is methadone, albeit to a lesser extent [321]. In consideration of all these issues, the search for a “magic bullet”, that is to say, an opioid agonist that relieves pain without having abuse potential may seem a quixotic ambition.

In this survey, we provided an overview of the organic chemical investigations of more than eight decades in the field of 6,14-ethenomorphinans. The present review also summarized the pharmacological aspects of labeled Bentley compounds applied in molecular imaging of ORs. There is no summary currently available in the literature that covers both the organic synthesis of C-ring bridged morphinans as well as the chemistry of their labeled derivatives. Therefore, we felt the need to present them together.

Future investigations may lead to the development of new 6,14-ethenomorphinans with subtype selectivity and labeled Bentley-type OR ligands via more effective radiochemical procedures. Novel radiotracers can strongly contribute to OR imaging in small animals, thus supporting the investigations in the field of pain—as well as in addiction— research.

## Figures and Tables

**Figure 1 molecules-27-02863-f001:**
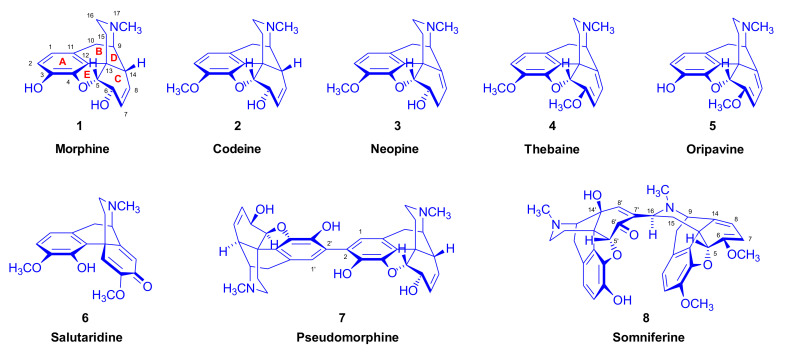
Chemical structure of selected poppy alkaloids with morphinan scaffold.

**Figure 2 molecules-27-02863-f002:**
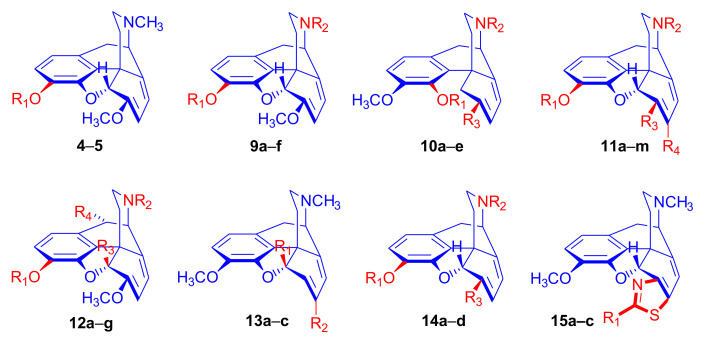
Structures of selected morphinan-6,8-dienes.

**Figure 3 molecules-27-02863-f003:**
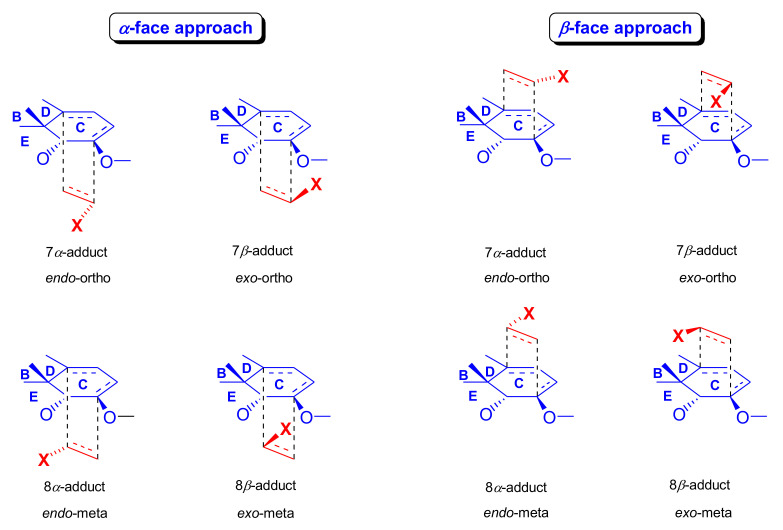
Potential transitional states of morphinan-6,8-diene and CH_2_=CH-X-type asymmetrical dienophile.

**Figure 4 molecules-27-02863-f004:**
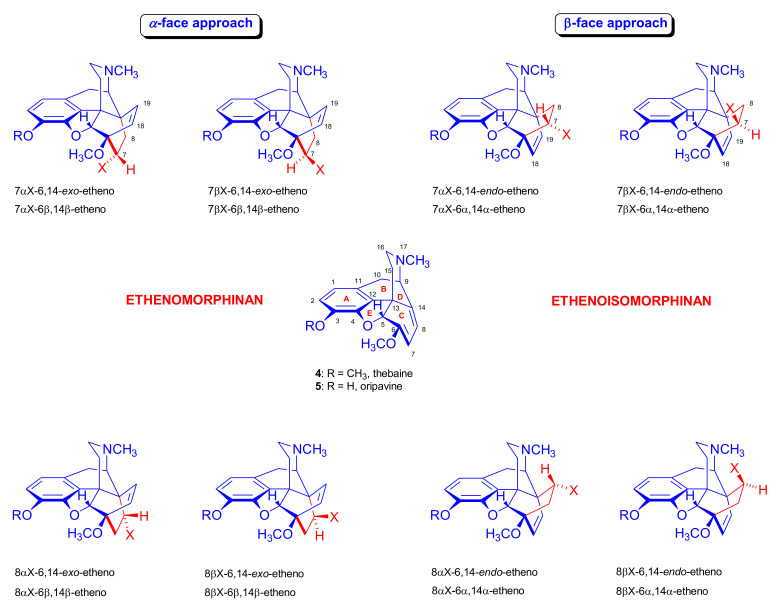
The eight theoretically possible isomers from the Diels–Alder reaction of thebaine (**4**) and/or oripavine (**5**) with CH_2_=CH-X type of molecule as the asymmetrical dienophile.

**Figure 5 molecules-27-02863-f005:**
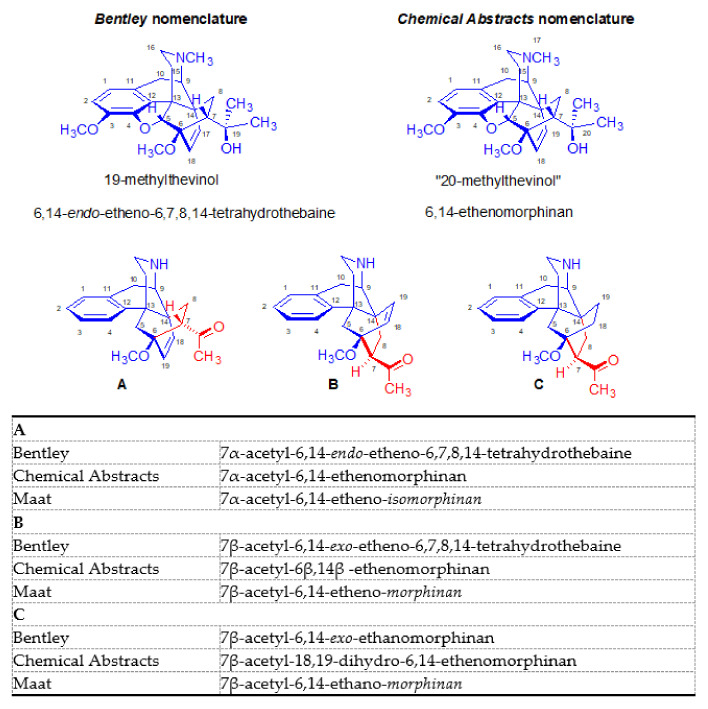
Nomenclature systems and numbering of Diels–Alder adducts of morphinan-6,8-dienes [75].

**Figure 6 molecules-27-02863-f006:**
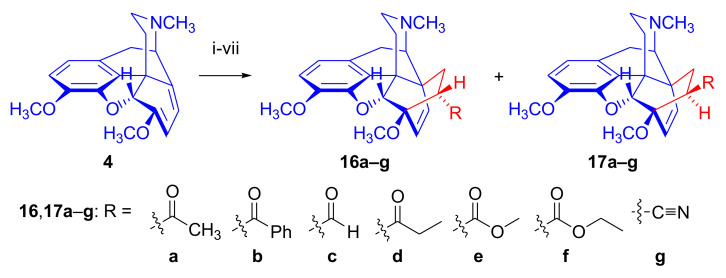
Reaction of thebaine with unsymmetric dienophiles. *Reagents and conditions*: (i): 11.5 equiv. methyl vinyl ketone, reflux, 1 h, 93% (7α) and 0.5% (7β); (ii): 1 equiv. phenyl vinyl ketone, benzene, reflux, 2 h, 70% (7α); (iii): (A): 14 equiv. acrolein, neat, reflux, 1 h, >85%, or (B): 1.8 equiv. acrolein, benzene, 65–70 °C, 4 h, 57%; (iv): 3.7 equiv. ethyl vinyl ketone, neat, 100 °C, 4 h, 80%; (v): 17 equiv. methyl acrylate, reflux, 6 h; (vi): 4.88 equiv. ethyl acrylate, neat, reflux, 6 h, 94% (7α) and 6% (7β); and (vii): 16.4 equiv. acrylonitrile, neat, reflux, 3 h, ca. 1:1 mixture of 7α/7β isomers.

**Figure 7 molecules-27-02863-f007:**
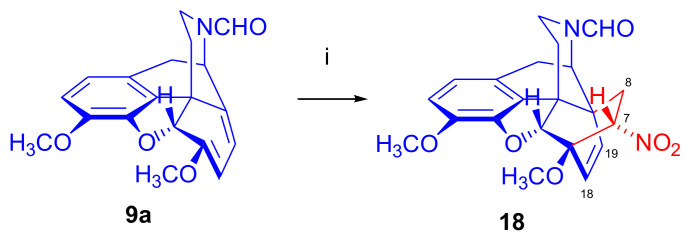
Reaction of *N*^17^-formyl-northebaine with nitroethene. *Reagents and conditions*: (i): nitroethene, benzene, boiling, 19 h.

**Figure 8 molecules-27-02863-f008:**
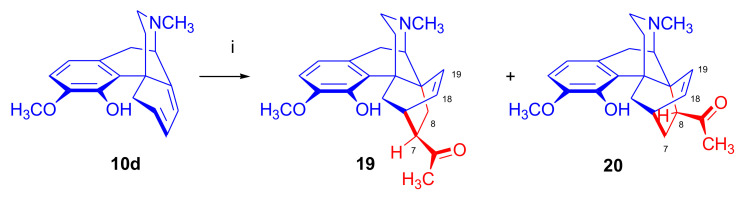
Cycloaddition of methyl vinyl ketone to 6-demethoxy-β-dihydrothebaine. *Reagents and conditions*: (i): methyl vinyl ketone, benzene, reflux, 24 h.

**Figure 9 molecules-27-02863-f009:**
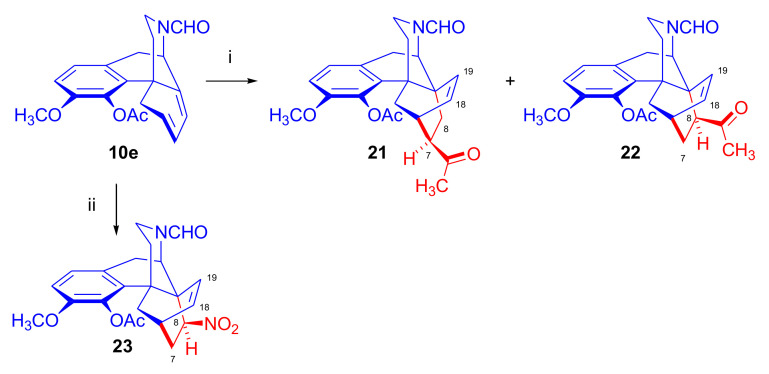
Cycloaddition reactions of 4-*O*-acetyl-6-demethoxy-*N*^17^-formyl-β-dihydrothebaine with methyl vinyl ketone as well as nitroethene. *Reagents and conditions*: (i): methyl vinyl ketone, microwave heating, 20 h, (ii): nitroethene, benzene, reflux, 50 h.

**Figure 10 molecules-27-02863-f010:**
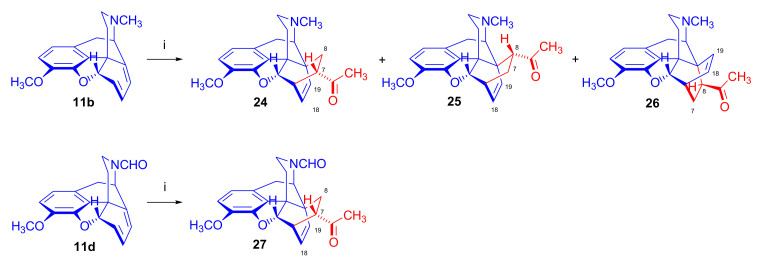
Cycloaddition of methyl vinyl ketone to 6-demethoxythebaine as well as to *N*^17^-formyl-6-demethoxythebaine. *Reagents and conditions*: (i): methyl vinyl ketone, heating, microwave, 4 h.

**Figure 11 molecules-27-02863-f011:**
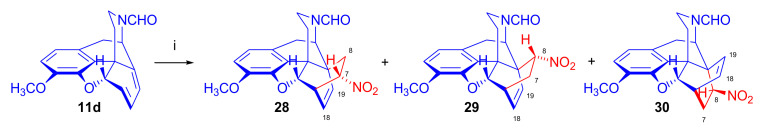
Reaction of *N*^17^-formyl-6-demethoxynorthebaine with nitroethene. *Reagents and conditions*: (i): nitroethene, benzene, reflux, 50 h.

**Figure 12 molecules-27-02863-f012:**
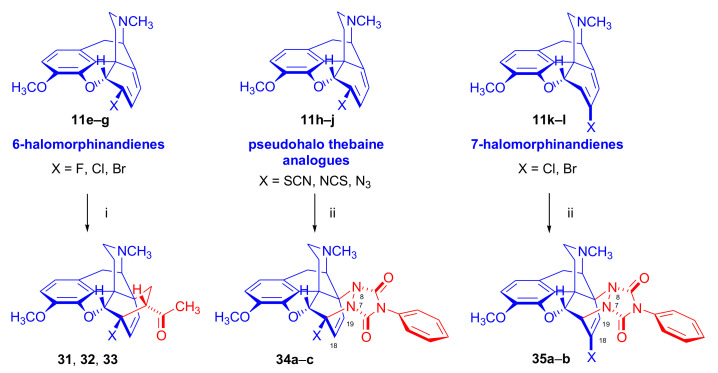
Cycloadducts of 6-halomorphinanedienes, 7-halomorphinandienes, and 6-pseudohalo thebaine analogues. *Reagents and conditions*: (i): methyl vinyl ketone, toluene, reflux, 24 h, 85–96%; (ii): PTAD, room temperature, 20 min.

**Figure 13 molecules-27-02863-f013:**
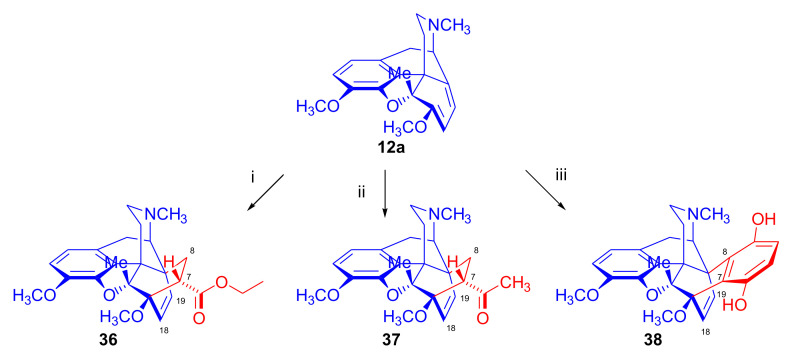
Cycloaddition reactions of 5β-methylthebaine. *Reagents and conditions*: (i): ethyl acrylate, reflux, 96 h, (ii): methyl vinyl ketone, reflux, 48 h, and (iii): *p*-benzoquinone, toluene, 90–100 °C, 24 h.

**Figure 14 molecules-27-02863-f014:**
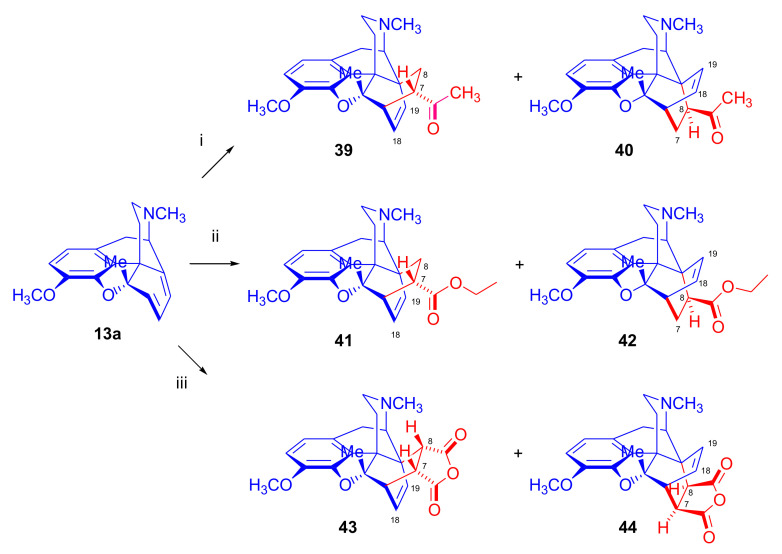
Cycloaddition reactions of 6-demethoxy-5β-methylthebaine. *Reagents and conditions*: (i): methyl vinyl ketone, reflux, five days (ii): ethyl acrylate, reflux, five days, and (iii): maleic anhydride, toluene, reflux, ten days.

**Figure 15 molecules-27-02863-f015:**
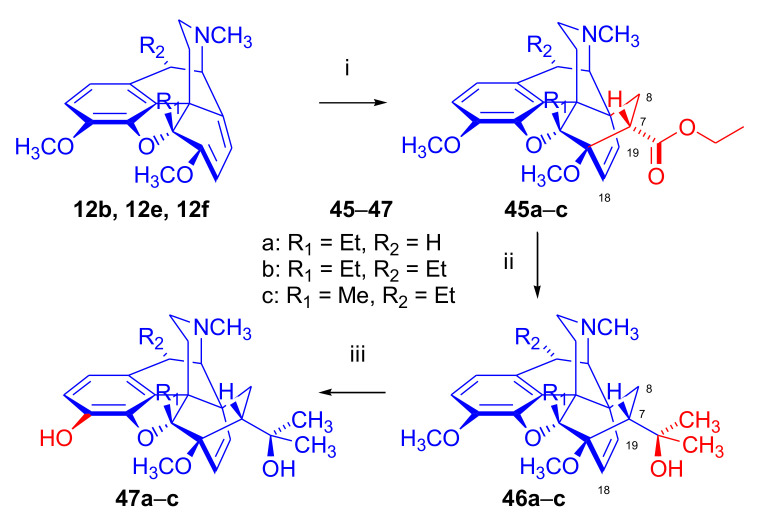
Synthesis of 5β- and 10α-substituted 20-methylorvinol derivatives. *Reagents and conditions*: (i): ethyl acrylate, (ii): methylmagnesium bromide, and (iii): KOH, diethylene glycol.

**Figure 16 molecules-27-02863-f016:**
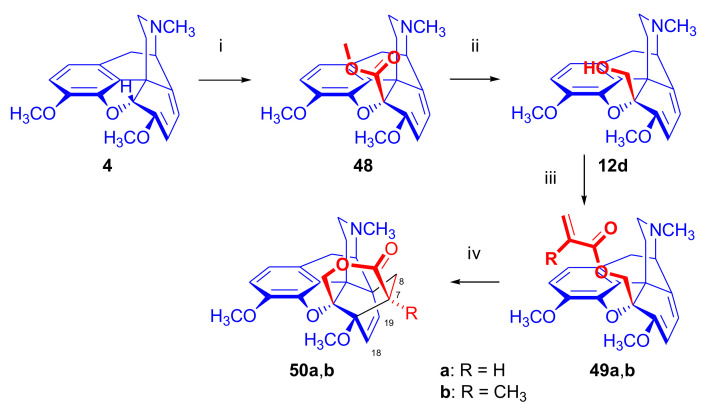
Synthesis of morphinans doubly bridged at ring-C. *Reagents and conditions*: (i): (1) butyl lithium, (2) methyl chloroformate, (ii): LiAlH_4_, (iii): (a) acryloyl chloride or (b) methacryloyl chloride, and (iv): toluene, reflux, three weeks.

**Figure 17 molecules-27-02863-f017:**
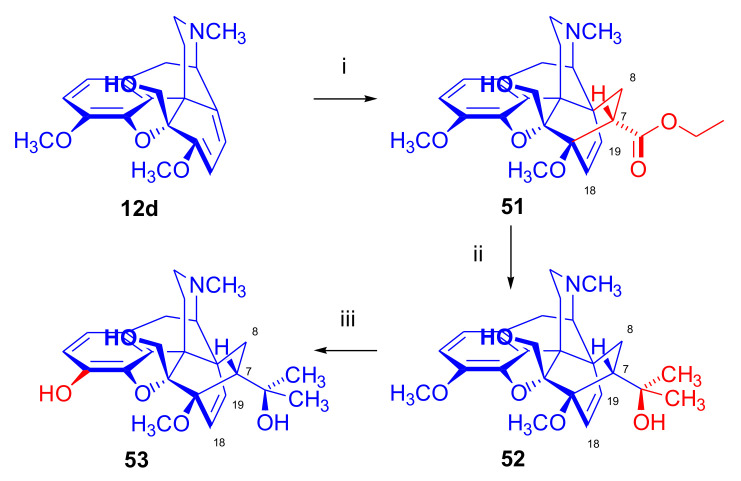
Synthesis of the 5β-methanol analogue of 20-methylorvinols. *Reagents and conditions*: (i): ethyl acrylate, reflux, one week, (ii): methylmagnesium iodide, benzene, diethyl ether, and (iii): KOH, ethylene glycol, microwave heating, 6 h.

**Figure 18 molecules-27-02863-f018:**
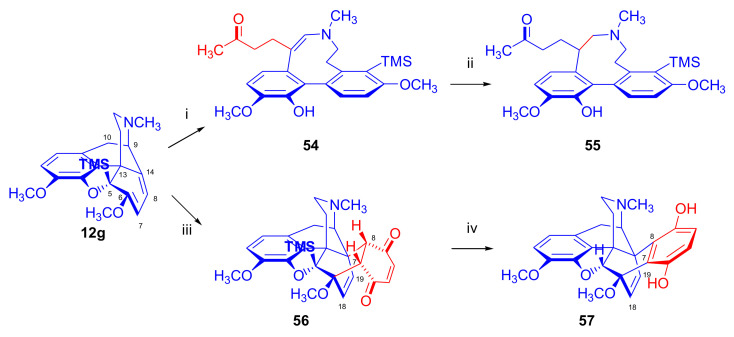
Reaction of 5β-trimethylsilylthebaine with methyl vinyl ketone and *p*-benzoquinione. *Reagents and conditions*: (i): methyl vinyl ketone, toluene, reflux, (ii): H_2_, Pd-C, (iii): *p*-benzoquinone, 45 °C, 7 days, and (iv): tetrabutylammonium fluoride.

**Figure 19 molecules-27-02863-f019:**
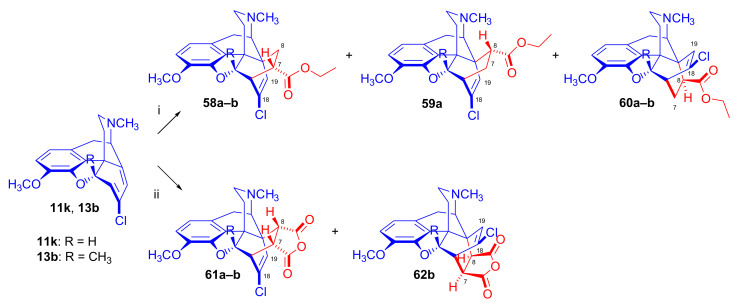
Cycloaddition reactions of 7-chloro-6-demethoxythebaine and 7-chloro-5β-methyl-6- demethoxythebaine. *Reagents and conditions*: (i): ethyl acrylate, reflux, four days (**11k**)/two weeks (**13b**), (ii): maleic anhydrid, toluene, reflux, one day (**11k**)/two weeks (**13b**).

**Figure 20 molecules-27-02863-f020:**
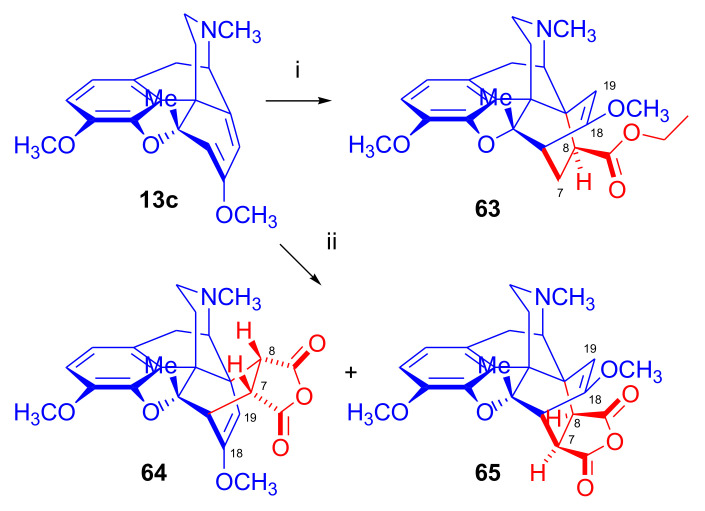
Reactions of 7-methoxy-5β-methyl-6-demethoxythebaine. *Reagents and conditions*: (i): ethyl acrylate, reflux, 2 weeks, 62%; (ii) three equiv. maleic anhydride, toluene, reflux, 4 days.

**Figure 21 molecules-27-02863-f021:**
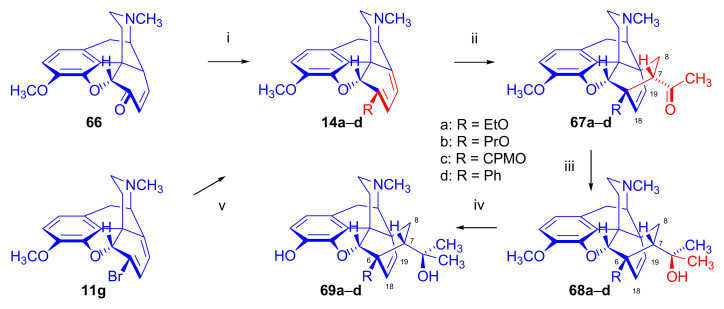
Synthesis of 6-*O*-substituted-20-methylorvinol derivatives. *Reagents and conditions*: (i): ROH, *p*-toluenesulfonic acid, benzene, 3 h, reflux, (ii): PhB(OH)_2_, Pd(PPh_3_)_4_, Ba(OH)_2_·8H_2_O, (iii): methyl vinyl ketone, reflux, 4 h, (iv): methylmagnesium bromide, THF, and (v): KOH, ethylene glycol-water, reflux, 8 h.

**Figure 22 molecules-27-02863-f022:**
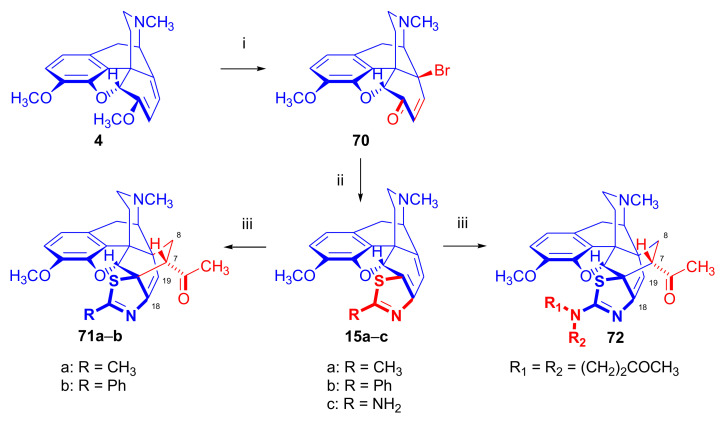
Synthesis of C-ring-fused thiazolothevinone analogues. *Reagents and conditions*: (i): *N*-bromosuccinimide, acetone–water 2:1 (*v/v*); (ii): RCSNH_2_, DMF, reflux, 30 min, 27–63%; and (iii): methyl vinyl ketone, 100 °C, 4 h, 64–87%.

**Figure 23 molecules-27-02863-f023:**
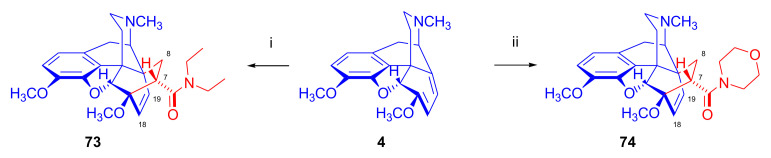
Reaction of thebaine with acrylicacidamides. *Reagents and conditions*: (i): acrylic acid diethylamide, xylol, reflux, 24 h, quant.; (ii): acrylic acid morpholide, xylol, reflux, 24 h, quant.

**Figure 24 molecules-27-02863-f024:**
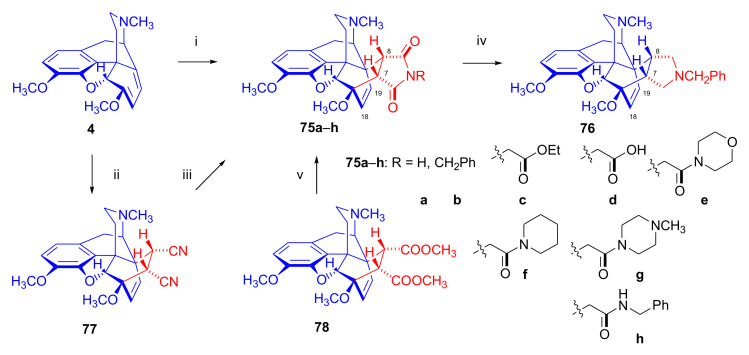
Thebaine-maleimide and *N*^1^-substituted maleimide adducts. *Reagents and conditions*: (i): R = H, maleimide, DMF, reflux, 15 h, 76% or maleamide, methanol, reflux, 72 h, 86%; R = CH_2_Ph, *N*-benzylmaleimide, ethanol, reflux, 1 h, quant.; (ii): maleonitril, toluene, reflux, 8 h, 80%; (iii): (1) H_2_O_2_, ethanol, (2) 6M NaOH, reflux, 20 h, 19%; (iv): LiAlH_4_, THF, reflux, 16 h; and (v): benzylamine, reflux, 3 h, 63%.

**Figure 25 molecules-27-02863-f025:**
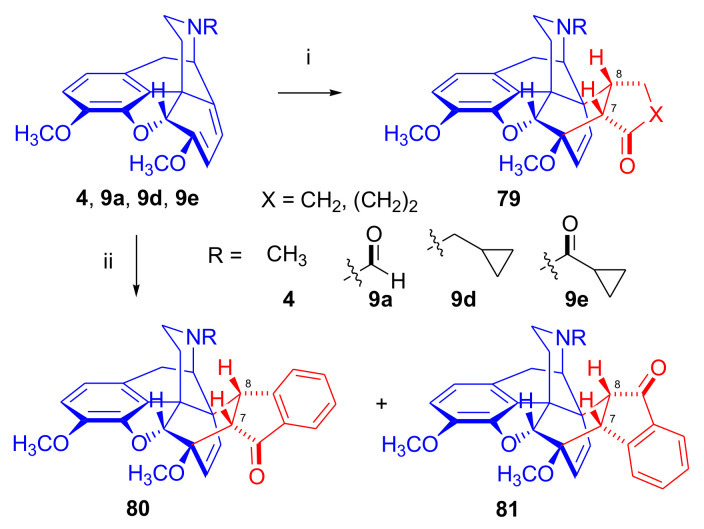
Diels–Alder reactions of morphinan-6,8-dienes with cycloalkenones and with 1-indenone. *Reagents and conditions*: (i): 2-cyclopenten-1-one or 2-cyclohexen-1-one, (A) toluene, 130 °C, sealed tube, 24 h; or (B) LiBF_4_, neat, room temperature, 7–28 days, (ii): 2-(*p*-chlorophenylsulfinyl)-1-indanone, toluene, reflux, 9–30 h.

**Figure 26 molecules-27-02863-f026:**
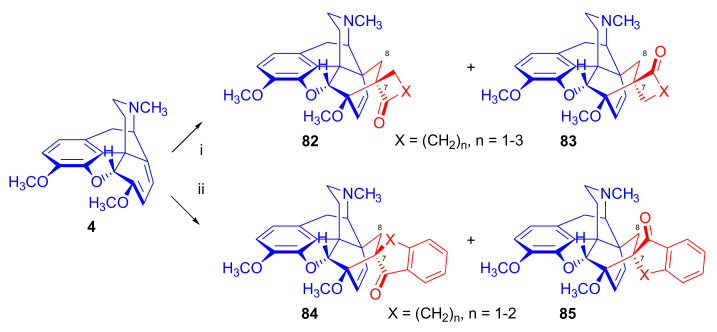
Spiro-7,7-disubstituted-6,14-ethenomorphinans. *Reagents and conditions*: (i) 1.4 equiv. Mannich salt, 1.5 equiv. sodium carbonate, toluene, reflux, 2–14 days; (ii): 2-methyleneindan-1-one.

**Figure 27 molecules-27-02863-f027:**
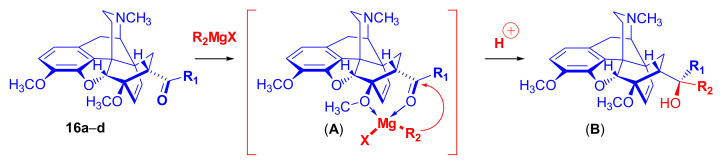
Mechanism of the normal Grignard addition of 7α-acyl-6,14-ethenomorphinans.

**Figure 28 molecules-27-02863-f028:**
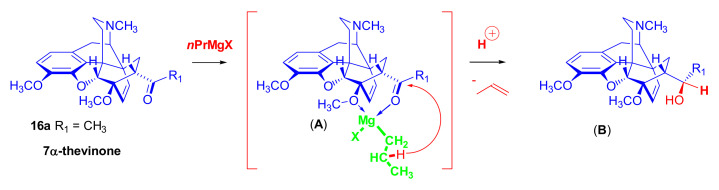
Mechanism of the Grignard reduction of 7α-acyl-6,14-ethenomorphinans.

**Figure 29 molecules-27-02863-f029:**
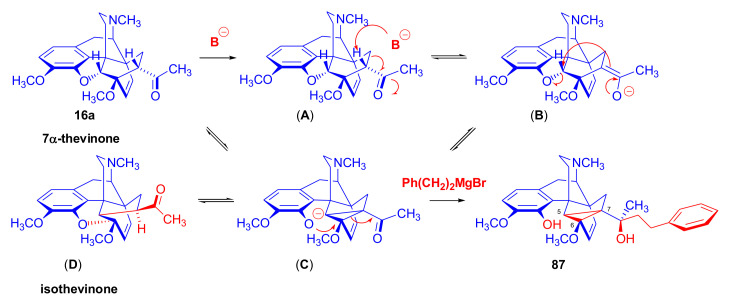
Base-catalyzed rearrangement in the reaction of thevinone (**16a**) with 2-phenethylmagnesium bromide.

**Figure 30 molecules-27-02863-f030:**
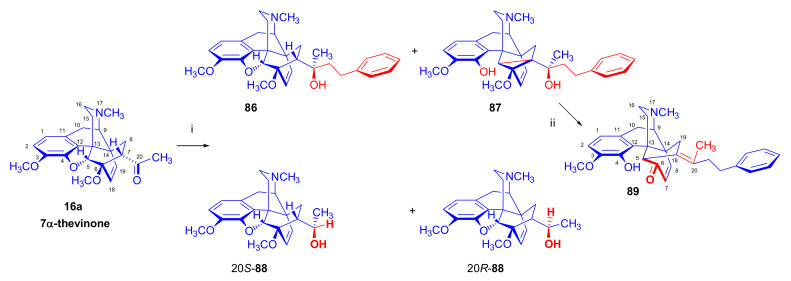
Reaction of thevinone (**16a**) with 2-phenylethylmagnesium bromide. *Reagents and conditions*: (i): six equiv. PhCH_2_CH_2_MgBr, toluene-THF, 2 h; (ii): 1M HCl, reflux, 4 h.

**Figure 31 molecules-27-02863-f031:**
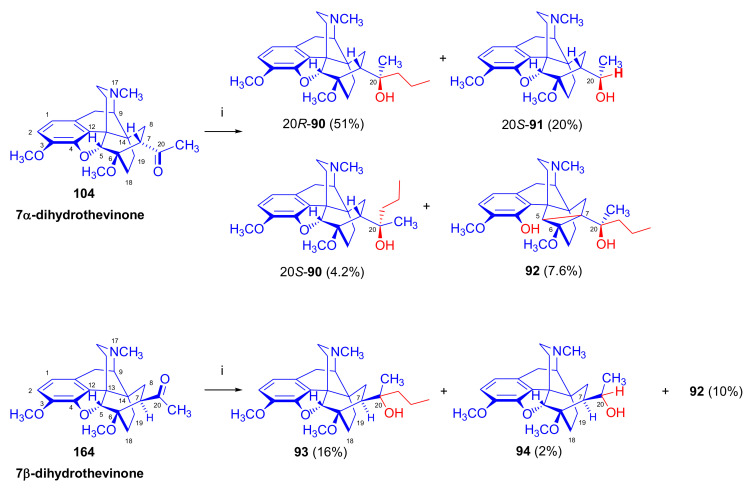
Grignard reaction of 7α-dihydrothevinone with *n*-propylmagnesium bromide. *Reagents and conditions*: (i): 6 equiv. *n*-propylmagnesium bromide, toluene-THF or toluene-diethylether, 2 h.

**Figure 32 molecules-27-02863-f032:**
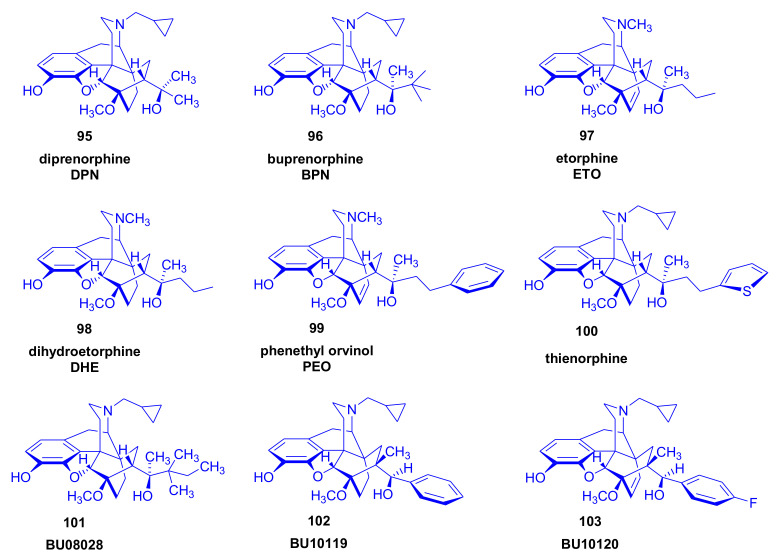
Structures of selected pharmacologically important orvinol derivatives.

**Figure 33 molecules-27-02863-f033:**
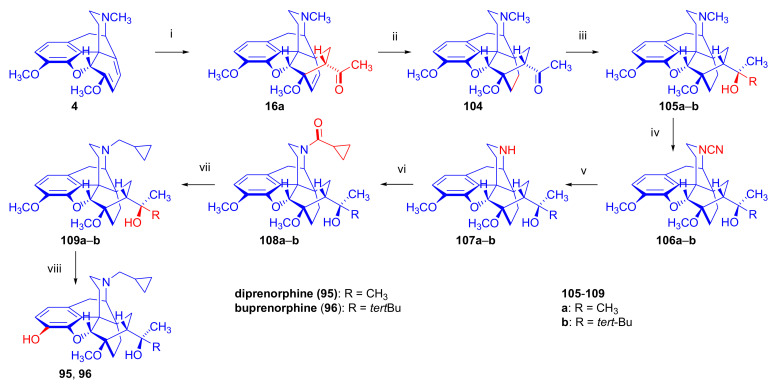
The first synthesis [74,97,111] of diprenorphine and buprenorphine. *Reagents and conditions*: (i): methyl vinyl ketone, reflux, 1 h, 93%; (ii): H_2_, 10% Pd-C, EtOH, 4 bar, 50 °C, 10h, 75%; (iii): methylmagnesium iodide or *tert*-butylmagnesium chloride, diethylether-benzene, reflux, 2 h; (iv): cyanogen bromide, CH_2_Cl_2_, room temperature, 16 h; (v): KOH, diethylene glycol, 165–170 °C, 75 min.; (vi): cyclopropylcarbonyl chloride, CH_2_Cl_2_, Et_3_N, 2 days; (vii): LiAlH_4_, THF, reflux, 4 h; and (viii): KOH, diethylene glycol, 210–220 °C, nitrogen atmosphere, 2 h.

**Figure 34 molecules-27-02863-f034:**
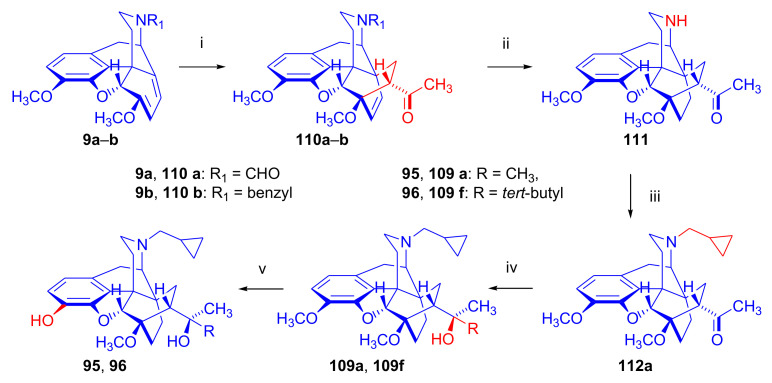
Synthesis of diprenorphine and buprenorphine from *N*^17^-benzyl-northebaine and *N*^17^-formyl-northebaine. *Reagents and conditions*: (i): methyl vinyl ketone, reflux, 1 h; (ii): (A) H_2_, Pd-C, EtOH, 60 °C, 5 bar, or (B) (R = CHO) HCl in EtOH, reflux, 4 h; (iii): cyclopropylmethyl bromide, NaHCO_3_, 90 °C, 20 h; (iv): methylmagnesium iodide or *tert*-butylmagnesium chloride, diethyl ether-toluene, 2 h; (v): KOH, diethylene glycol, 210 °C.

**Figure 35 molecules-27-02863-f035:**
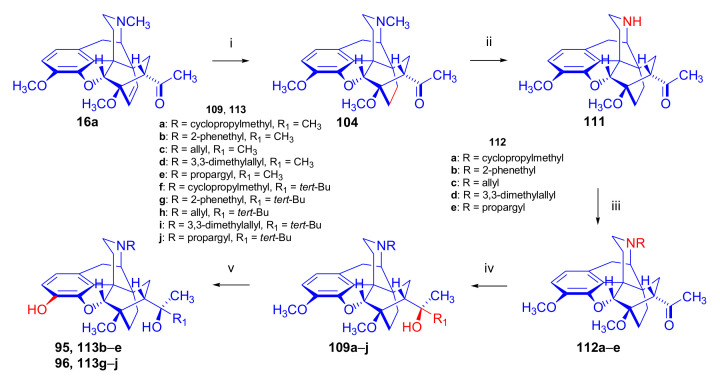
Synthesis of *N*^17^-substituted-diprenorphine and buprenorphine analogues. *Reagents and conditions*: (i): H_2_, Pd-C, toluene, 55 °C, 6 bar; (ii): (1) DEAD, benzene, reflux, 8 h, (2) pyridine hydrochloride; (iii): RBr, NaHCO_3_, 90–95 °C, 20 h; (iv): R_1_MgX, toluene-THF or diethyl toluene-ether; and (v): KOH, diethylene glycol, 210 °C.

**Figure 36 molecules-27-02863-f036:**
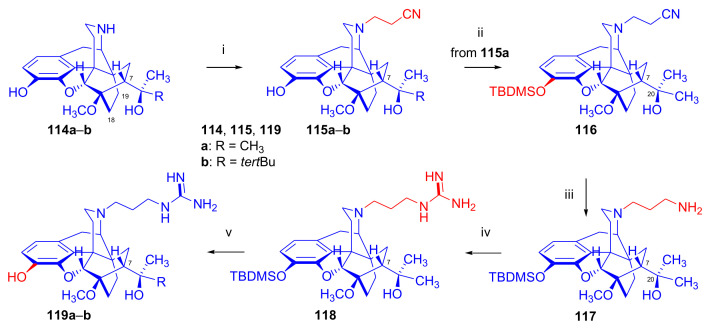
Synthesis of *N*^17^-[3-(2-carboxamidino)-amionopropyl]-nordiprenorphine and -norbuprenorphine. *Reagents and conditions*: (i): 3.5 equiv. acrylonitrile, EtOH, room temperature, overnight, 62%; (ii): TBDMSCl, imidazole, DMAP, DMF, room temperature, 2 h, 89%; (iii): LiAlH_4_, Et_2_O, room temperature, 4 h, 78%; (iv): 1*H*-pyrazole-1-carboxamidine hydrochloride, DIPEA, DMF, under nitrogen, room temperature, overnight, 76%; and (v): ammonium fluoride, MeOH, room temperature, overnight, 96%.

**Figure 37 molecules-27-02863-f037:**
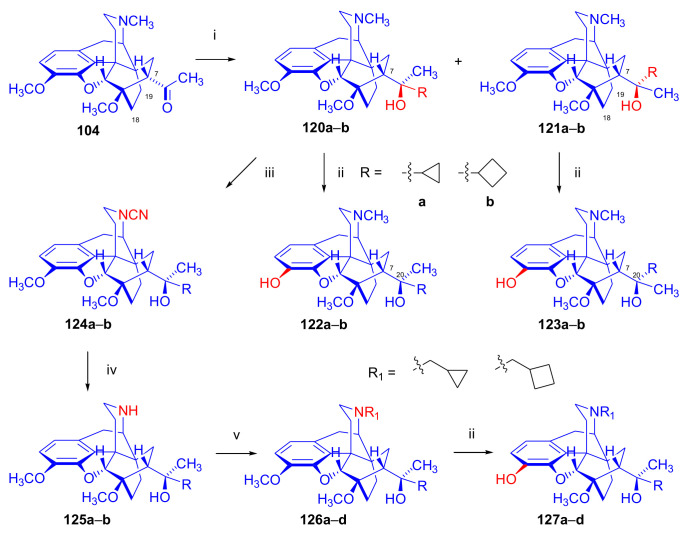
Synthesis of 20-cyclopropyl- and 20-cyclobutyl-substituted buprenorphine analogues. *Reagents and conditions*: (i): RMgBr, THF, reflux, 1.5 h; (ii): KOH, diethyleneglycol, 220 °C, 2 h; (iii): BrCN, CH_2_Cl_2_, reflux, 4 h; and (iv): KOH, diethylene glycol, 175 °C, 1.5 h; R_1_CH_2_Br, Mg(OAc)_2_, DMF, 100 °C, 2 h.

**Figure 38 molecules-27-02863-f038:**
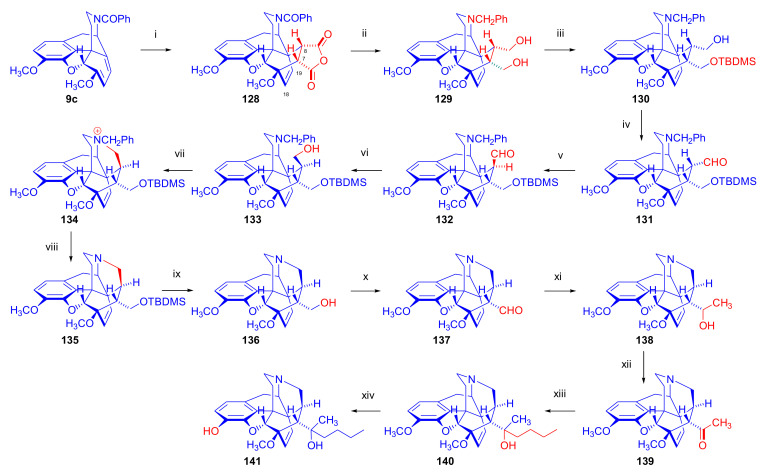
Synthesis of *N*^17^-*C*^8β^-methano-bridged homoetorphine. *Reagents and conditions*: (i): maleic anhydride, benzene, reflux, 7 h, 99%; (ii): LiAlH_4_, THF, reflux, 3 h, 64%; (iii): TBDMS-Cl, Et_3_N, DMAP, CH_2_Cl_2_, overnight, room temperature, 69%, and 8α-CH_2_OTBDMS regioisomer 28%; (iv): (COCl)_2_, DMSO, CH_2_Cl_2_, −78 °C to 0 °C, 97%; (v): silica gel, CHCl_3_, 2 days, 52%; (vi): NaBH_4_, MeOH, benzene, room temperature, 30 min, quant.; (vii): MsCl, Et_3_N, CH_2_Cl_2_, 0 °C, 5 min, 99%; (viii): NH_4_COOH, 5% Pd-C, MeOH, (1) room temperature, 30 min, (2) reflux, 45 s, 79%; (ix): 1M HCl, MeOH, room temperature, 40 min, 92%; (x): (COCl)_2_, DMSO, CH_2_Cl_2_-DMSO 8:2 (*v/v*), −78 °C to 0 °C, 77%; (xi): 2M MeMgI in Et_2_O, THF, room temperature, 30 min., 97%; (xii): (COCl)_2_, DMSO, CH_2_Cl_2_, −78 °C to 0 °C, 84%; (xiii): *n*BuMgBr, Et_2_O, 84%; and (xiv): propanethiol, NaH, DMF, reflux, 1 h, 77%.

**Figure 39 molecules-27-02863-f039:**
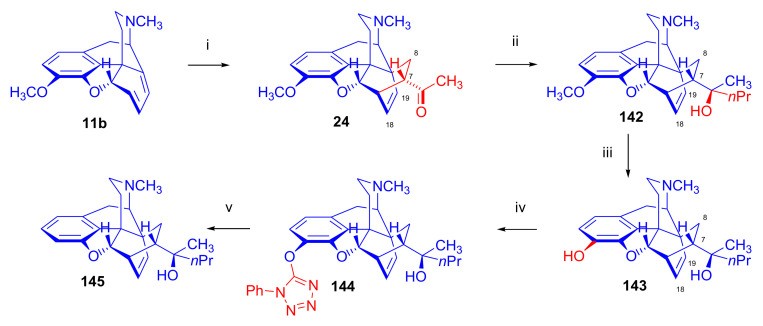
Synthesis of 6-demethoxy-3-deoxyetorphine from 6-demethoxythebaine. *Reagents and conditions*: (i): methyl vinyl ketone, reflux, (ii): *n*PrMgBr, (iii): KOH, diethylene glycol, 210 °C, (iv): 5-chloro-1-phenyl-1*H*-tetrazole, DMF, KO*t*Bu, nitrogen atmosphere, room temperature, 4 h, and (v): formic acid, 10% Pd-C, H_2_O, benzene, ethanol, reflux, 2 h.

**Figure 40 molecules-27-02863-f040:**
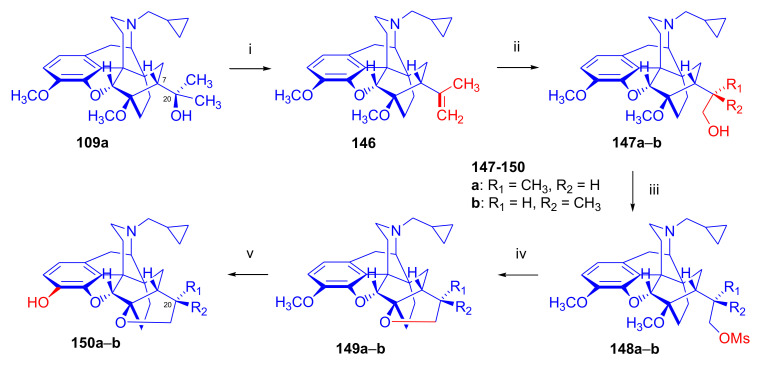
Synthesis of diprenorphine furanomorphides. *Reagents and conditions*: (i): formic acid, reflux, 12 min; (ii): (1) BH_3_·DMS, THF, (2) H_2_O_2_, 3M NaOH; (iii): MsCl, CH_2_Cl_2_, Et_3_N; (iv): LiAlH_4_, THF; and (v): *n*PrSNa.

**Figure 41 molecules-27-02863-f041:**
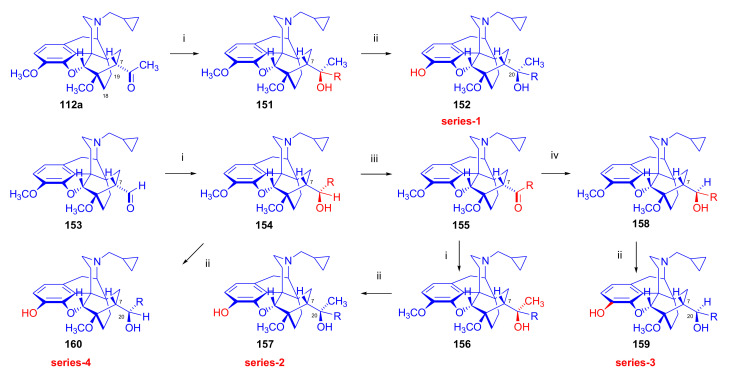
Synthesis of 20-substituted tertiary and secondary alcohol analogues of buprenorphine. *Reagents and conditions*: (i): RMgBr, THF, toluene, room temperature, 20 h; (ii): NaSPr, HMPA, 12 °C; (iii): (COCl)_2_, Et_3_N, DMSO, CH_2_Cl_2_, −78 °C; and (iv): LiAlH_4_, THF, room temperature, 1 h.

**Figure 42 molecules-27-02863-f042:**
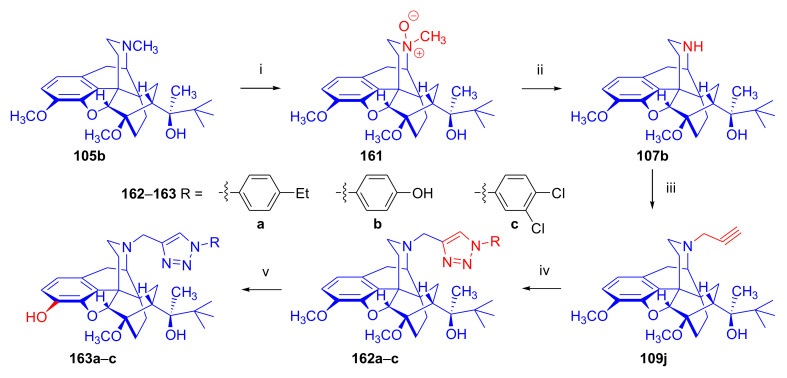
Synthesis of 1,2,3-triazole-tethered *N*^17^-norbuprenorphine derivatives. *Reagents and conditions*: (i): 33% H_2_O_2_, MeOH; (ii): stainless steel powder, 2-propanol, 50 °C, 8 h, 40%; (iii): 1.2 equiv. propargyl bromide, 1.2 equiv. NaH, THF, room temperature, 4 h, 80%; (iv): CuSO_4_·5H_2_O, sodium ascorbate, RN_3_, MeOH-H_2_O-CH_2_Cl_2_ 1:1:1 (*v/v/v*) room temperature, 5–15 min, 65–95%; and (v): *n*-heptanethiol, NaH, DMSO, 140 °C, 2.5 h, 45–75%.

**Figure 43 molecules-27-02863-f043:**
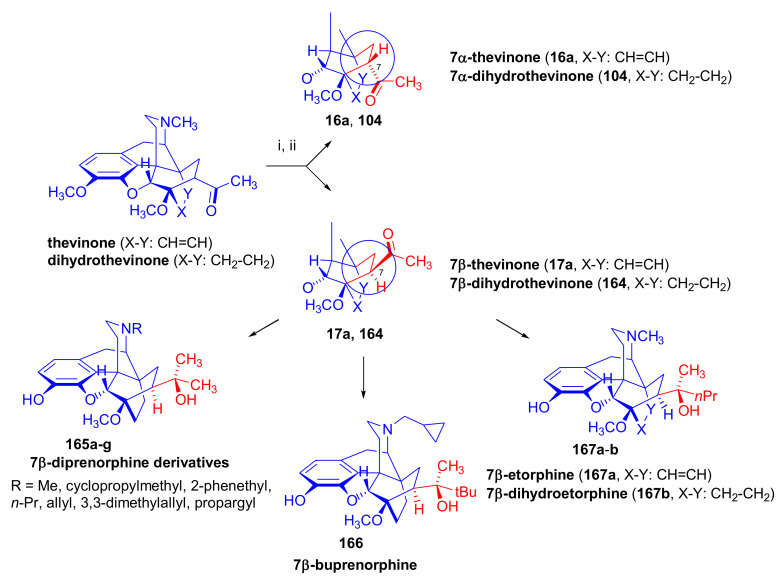
Separation of thevinone isomers and structures of 7β-diprenorphine analogues, 7β-buprenorphine, 7β-etorphine, and 7β-dihydroetorphine. *Reagents and conditions*: (i): isomerization; (ii): separation by fractional crystallization.

**Figure 44 molecules-27-02863-f044:**
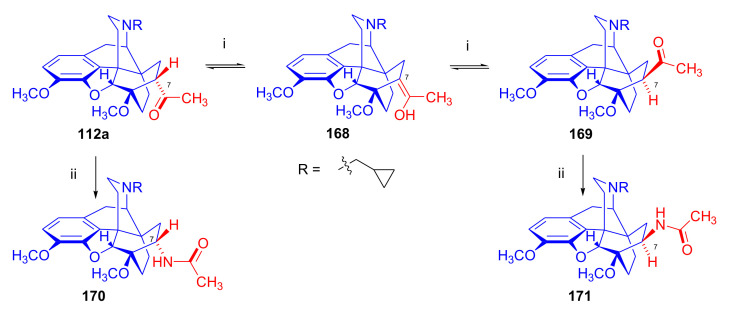
Epimerization and Schmidt reaction of *N*^17^-cyclopropymethyl-dihydronorthevinone. *Reagents and conditions*: (i and ii): 37% HClO_4_, NaN_3_, 70 °C, 16 h.

**Figure 45 molecules-27-02863-f045:**
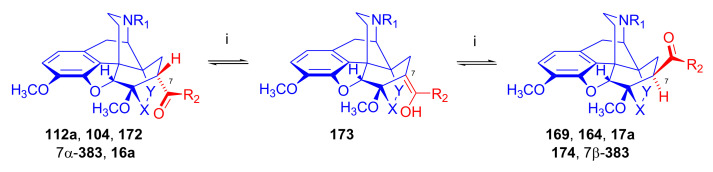
7α–7β epimerization of 6,14-ethenomorphinans induced by perchloric acid equilibrium-enolization [155].

**Figure 46 molecules-27-02863-f046:**
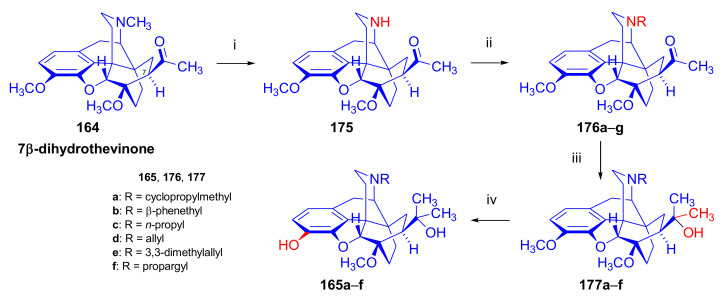
Synthesis of 7β-diprenorphine and its *N*^17^-substituted derivatives. *Reagents and conditions*: (i): (1) DEAD, benzene, reflux, 16 h, (2) pyridine hydrochloride, EtOH; (ii): RBr, NaHCO_3_, DMF, 90 °C, 20 h; (iii): methylmagnesium iodide, diethyl ether-toluene, 1 h; and (iv): KOH, diethylene glycol, 210 °C.

**Figure 47 molecules-27-02863-f047:**
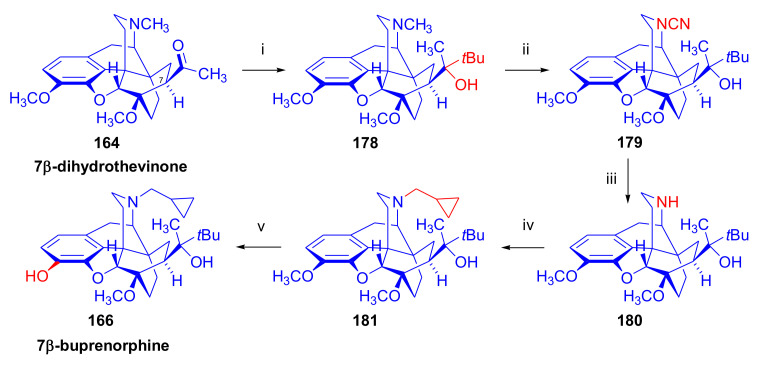
Synthesis of 7β-buprenorphine from 7β-dihydrothevinone. *Reagents and conditions*: (i): *tert*-butylmagnesium chloride, toluene-diethyl ether, 1.5 h; (ii): BrCN, CHCl_3_, room temperature, 20 h; (iii): KOH, diethylene glycol, 170 °C; (iv): cyclopropylmethyl bromide, NaHCO_3_, DMF, 90–95 °C, 20 h; and (v): KOH, diethylene glycol, 210–220 °C, 75 min.

**Figure 48 molecules-27-02863-f048:**
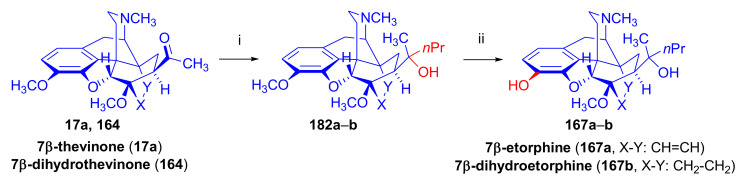
Synthesis of 7β-etorphine and 7β-dihydroetorphine. *Reagents and conditions*: (i): *n*-propylmagnesium bromide, toluene-diethyl ether, 2 h; (ii) KOH, diethylene glycol 210–220 °C, 75 min.

**Figure 49 molecules-27-02863-f049:**
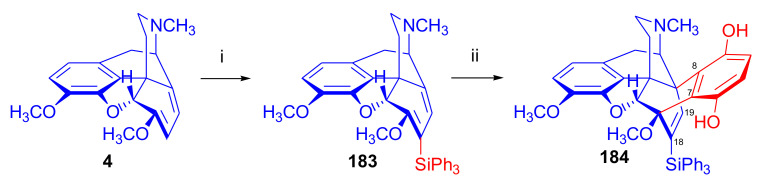
Synthesis of 18-triphenylsilyl-substituted 6,14-ethenomorphinan derivative. *Reagents and conditions*: (i): 2 equiv. BuLi, 2 equiv. TMEDA, 10 equiv. Ph_3_SiCl, 1. −78 °C, 30 min, 2. room temperature 12 h; (ii): 10 equiv. 1,4-benzoquinone, toluene, reflux, 4 h.

**Figure 50 molecules-27-02863-f050:**
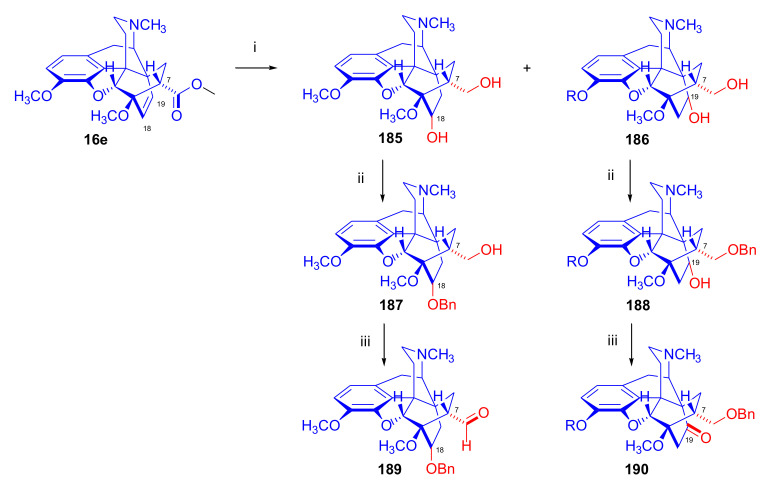
Preparation of 18- and 19-hydroxyl-substituted 6,14-ethenomorphinans and their benzyl derivatives. *Reagents and conditions*: (i): (1) BH_3_·Me_2_S, THF, −65–70 °C, (2) NaOH, 30% H_2_O_2_; (ii): PhCH_2_Br, NaH, DMF, −60 °C, 1h; and (iii): DMP, CH_2_Cl_2_, 3 h.

**Figure 51 molecules-27-02863-f051:**
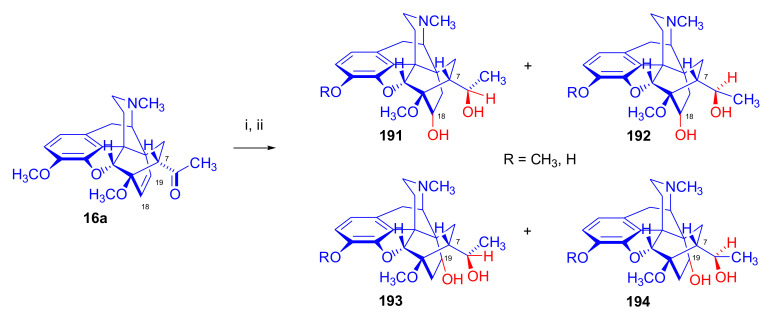
Hydroboration of thevinone. *Reagents and conditions*: (i): (1) BH_3_·Me_2_S, THF, −65–70 °C, (2) NaOH, 30% H_2_O_2_.

**Figure 52 molecules-27-02863-f052:**
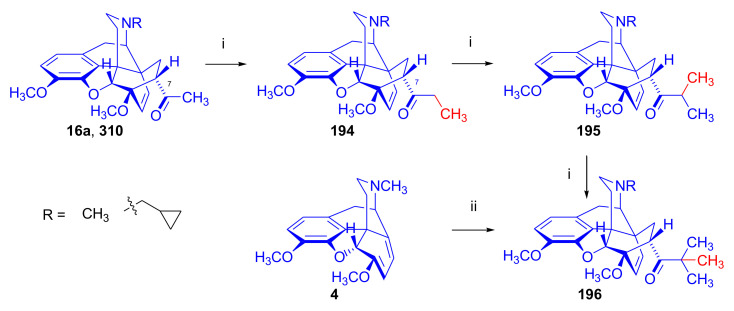
Methylation of the enolate of thevinone and its *N*^17^-cyclopropylmethyl analogue. *Reagents and conditions*: (i): (1) 1 equiv. *n*-butyl lithium, 1.5 equiv. diisopropylamine, THF,−78 °C, nitrogen, 1 h; (2) 1.1 equiv. substrate-ketone, THF, allowed to warm to room temperature; (3) 1.5 equiv. MeI, room temperature, 3 h; and (ii): *tert*-butyl vinyl ketone, toluene, reflux, 48 h [160].

**Figure 53 molecules-27-02863-f053:**
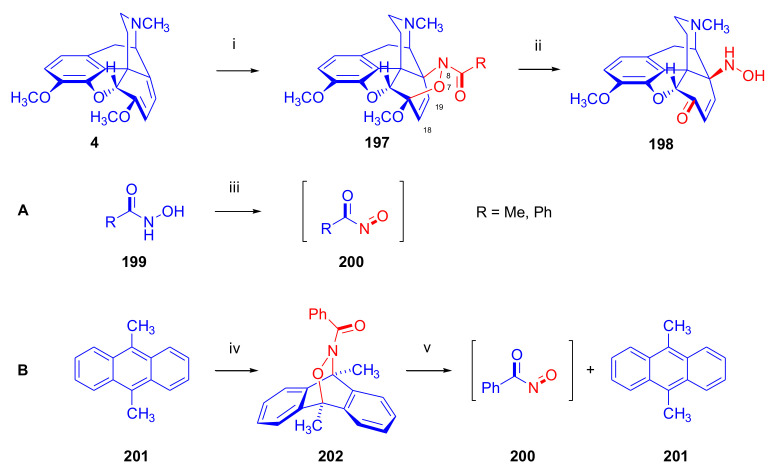
Hetero Diels–Alder reaction of thebaine with nitrosocarbonyl compounds. **A**. Formation of the nitroso reactive species. **B**. Diels–Alder reaction of 9,10-DMA with PhCONO and thermo retro-Diels–Alder reaction of the formed adduct. *Reagents and conditions*: (i): RCONHOH, Et_4_NIO_4_, aqueous AcOH-NaOAc (pH ca. 6), 0 °C; (ii): HCl, MeOH, H_2_O; (iii): 1.45 equiv. Et_4_NIO_4_, aqueous AcOH-NaOAc (pH ca. 6), 0 °C; (iv): benzohydroxamic acid, 1.45 equiv. Et_4_NIO_4_, aqueous AcOH-NaOAc (pH ca. 6), 0 °C; and (v): Δ, benzene.

**Figure 54 molecules-27-02863-f054:**
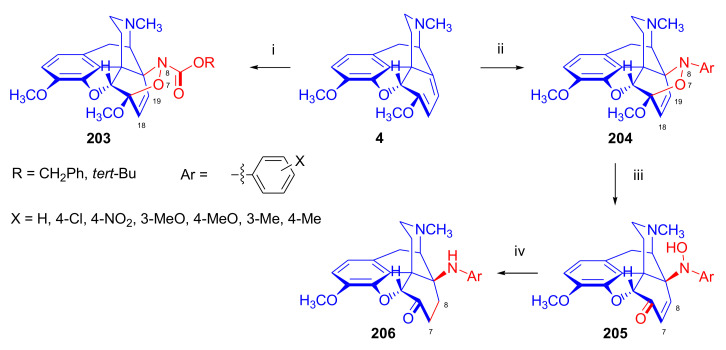
Cycloadducts of thebaine with nitrosoformates and nitrosoarenes. *Reagents and conditions*: (i): (A) ROCON_3_, Δ, DMSO; or (B) hydroxamic acids, Et_4_NIO_4_, CH_2_Cl_2_, = 0 °C, 1 h; (ii): ArNO, CHCl_3_, room temperature, 30 min; 65–97%; and (iii): 2M HCl, Δ, 90%; (iv): H_2_, 10% Pd-C, MeOH, room temperature, atmospheric pressure, 4 h.

**Figure 55 molecules-27-02863-f055:**
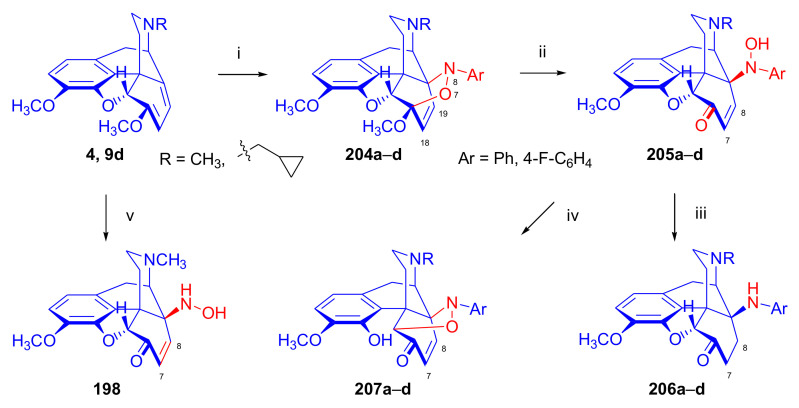
Synthesis of 14-(arylhydroxyamino)codeinone derivatives. *Reagents and conditions*: (i): nitrosobenzene or 4-fluoronitrosobenzene, CH_2_Cl_2_, room temperature, 5 min, 57–91%; (ii): 1M HCl, room temperature, 3 h, 55–91%; (iii): H_2_, Pd-C, MeOH, 3.8 bar, 2.5 h, 22–57%; (iv): NaOMe, MeOH, Δ, 22–68%; and (v): from **4**, (1) benzohydroxamic acid, Et_4_NIO_4_, CH_2_Cl_2_, (2) 3M HCl, 15 min, 47%.

**Figure 56 molecules-27-02863-f056:**
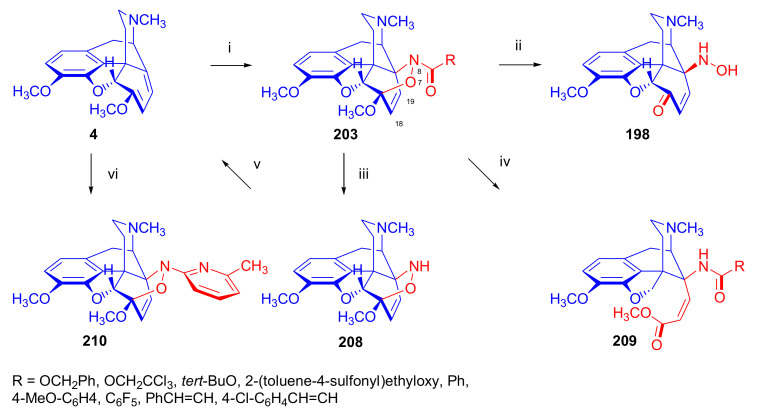
Synthesis of thebaine acyl nitroso HDA adducts. *Reagents and conditions*: (i): *N*-hydroxycarbamic esters, Et_4_NIO_4_ or BnMe_3_NIO_4_, CH_2_Cl_2_, 0 °C; (ii): from **203** R = *tert*Bu, HCl in MeOH, 90%; (iii): DBN, benzene, room temperature, 1 h, 56%; and (iv): two equiv. SmI_2_, THF, argon, 0 °C, 3 h, 77%; (v): attempted recrystallization.

**Figure 57 molecules-27-02863-f057:**
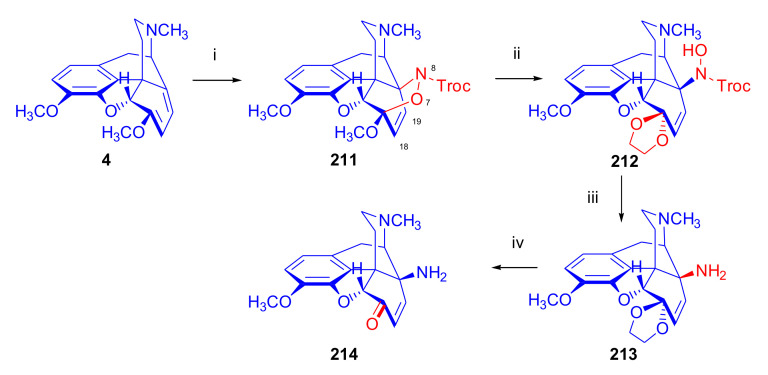
Synthesis of 14-aminocodeinone from thebaine. *Reagents and conditions*: (i): 2,2,2-trichloroethyl *N*-hydroxycarbamate, sodium periodate, EtOAc, 0.5 M sodium acetate (aq., pH 6), 0 °C, 1 h, 82%; (ii): 0.26 M hydrogen chloride in ethylene glycol, room temperature, 2 h, 95%; (iii): (NH_4_)_2_CO_3_, Zn powder, 70 °C, 1 h, 87%; and (iv): 6M HCl, MeOH-H_2_O 2:1 (*v/v*), reflux, 30 min, 80%.

**Figure 58 molecules-27-02863-f058:**
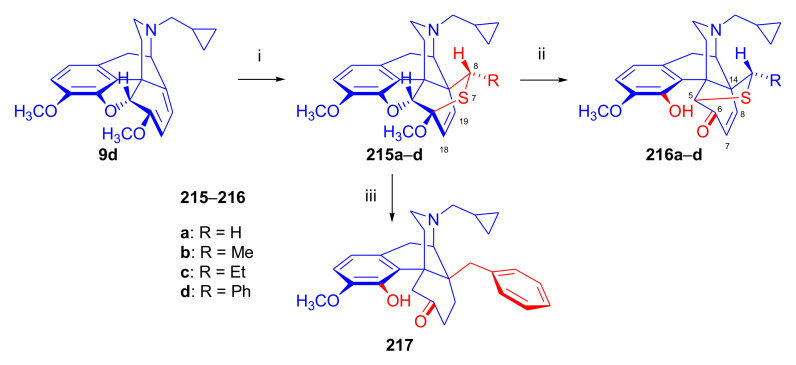
Synthesis of opiate antagonists with luteinizing hormone-stimulating properties. *Reagents and conditions*: (i): RCH_2_SS(O)CH_2_R, toluene, reflux, 1 h, 39–74%; (ii): 48% HBr, room temperature, 83–90%; and (iii): (1) Raney-Ni, MeOH, reflux, 23 h, (2) acetone, methanol, HCl, 45 °C, 10 min.

**Figure 59 molecules-27-02863-f059:**
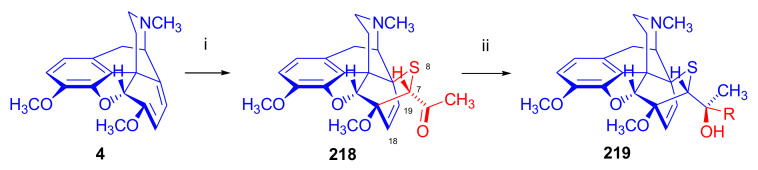
Synthesis of 20-alkyl-8-thiathevinols. *Reagents and conditions*: (i): (A) AcCH_2_SSO_3_Na, Et_3_N, EtOH-benzene 1:1 (v/v), room temperature, 160 h, 42% or (B) AcCH_2_SSO_2_Tol, CaCl_2_·2H_2_O, benzene, EtOH, 120 h, 75%; and (ii): RMgX, benzene or toluene, Et_2_O, reflux.

**Figure 60 molecules-27-02863-f060:**
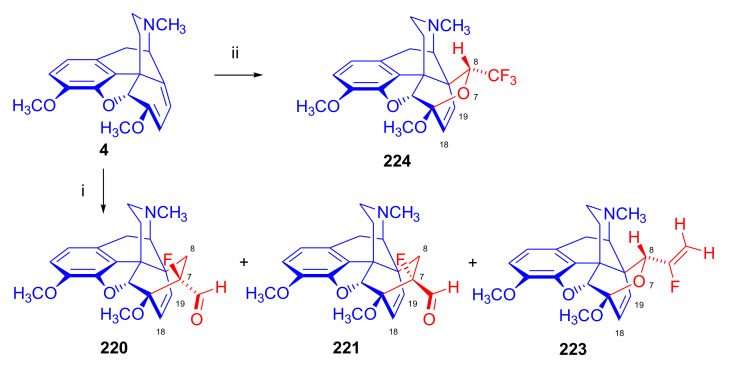
Reaction of thebaine with fluorine containing dienophiles. *Reagents and conditions*: (i): 2 equiv. 2-fluoroacrolein, benzene, 50 °C, 18 h; (ii): 10 equiv. trifluoroacetaldehyde, benzene, 50 °C, 24 h.

**Figure 61 molecules-27-02863-f061:**
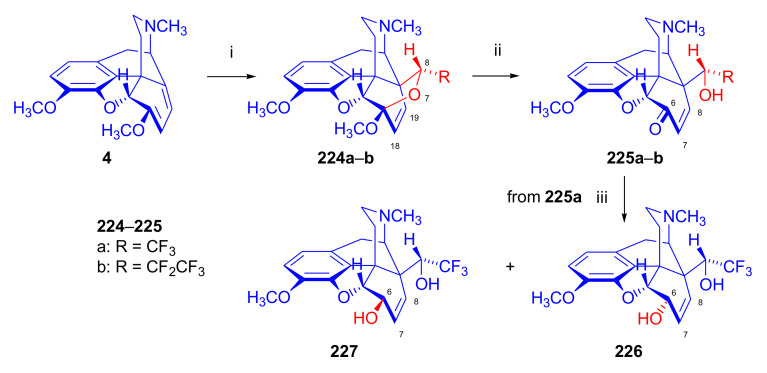
Reaction of thebaine with α-fluorinated aldehydes. *Reagents and conditions*: (i): 2,2,2-trifluoroacetaldehyde or 2,2,3,3-pentafluoropropanal, benzene, sealed tube, 50 °C, 24 h; (ii): cc. HCl, THF (1:10), 25 °C, 7 h; and (iii): NaBH_4_/CeCl_3_, methanol, 25 °C.

**Figure 62 molecules-27-02863-f062:**
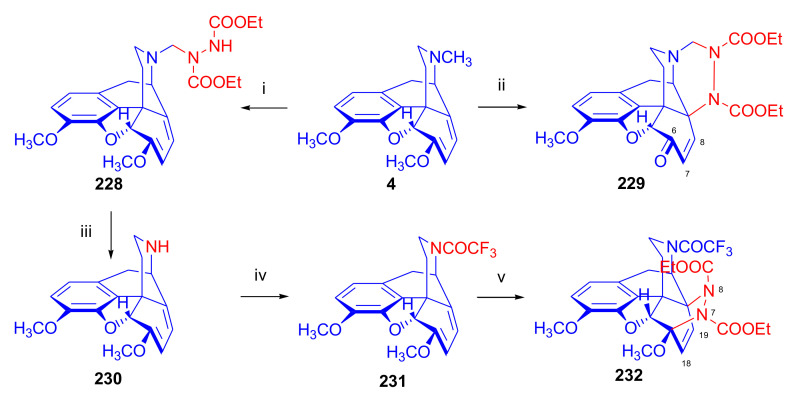
Reaction of thebaine and *N*^17^-trifluoroacetyl-northebaine with diethyl azodicarboxylate (DEAD). *Reagents and conditions*: (i): 1 equiv. DEAD, benzene, reflux, 16 h; (ii): (1) 2 equiv. DEAD, benzene, reflux, 2 h, (2) 2M HCl, Et_2_O, 1 h, 57%; (iii): saturated NH_4_Cl solution; (iv): trifluoroacetic anhydride, EtOAc, CHCl_3_, 25 °C, 45 min, 83%; and (v): 1.5 equiv. DEAD, benzene, reflux, 12 h, 91%.

**Figure 63 molecules-27-02863-f063:**
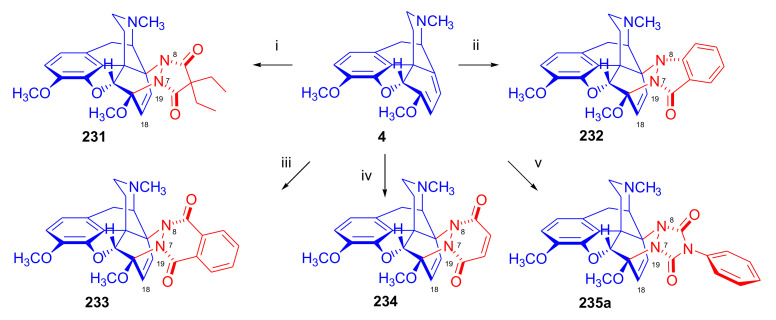
Diels–Alder adducts of thebaine with azaoxo-dienophiles. *Reagents and conditions*: (i) 4,4-diethyl-pirazolidine-3,5-dione, (A) oxidation with LTA, CH_2_Cl_2_, room temperature, 2 h; or (B) oxidation with *tert*-butylhypchlorite, acetone, (1) -70 °C, (2) room temperature (ii): 3-indazolone; (iii): 1,4-dihydrophthalazine-1,4-dione; (iv): 1,4-dihydopyridazine-3,5-dione; and (v): PTAD.

**Figure 64 molecules-27-02863-f064:**
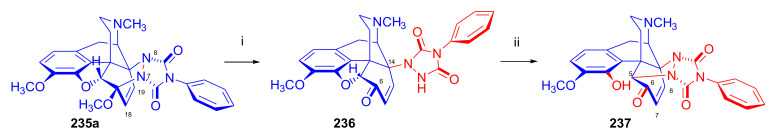
Rearrangement of thebaine-PTAD adduct. *Reagents and conditions*: (i): cc. hydrochloric acid, acetone, 50 °C; (ii): crystallization from hot ethanol or methanol.

**Figure 65 molecules-27-02863-f065:**
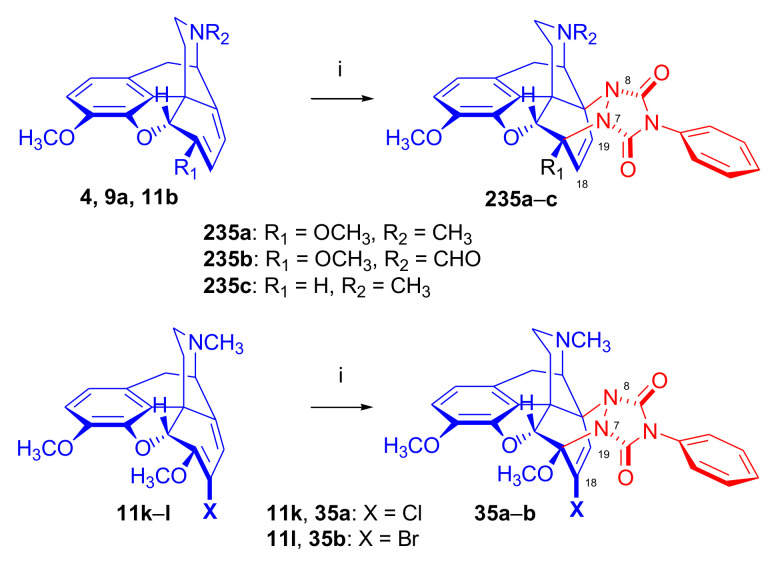
Reaction of 4-phenyl-4*H*-1,2,4-triazoline-3,5-dione with various morphinan-6,8-dienes. *Reagents and conditions*: (i): PTAD, acetone, room temperature, 20–90 min.

**Figure 66 molecules-27-02863-f066:**
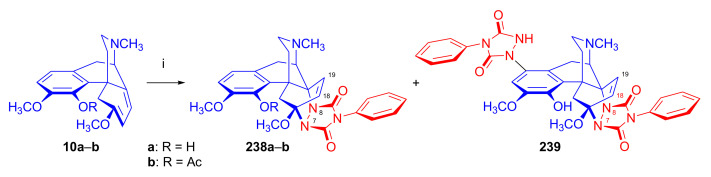
Reaction of opened E-ring morphinandienes with 4-phenyl-4*H*-1,2,4-triazoline-3,5-dione. *Reagents and conditions*: (i): PTAD, acetone, room temperature, 90 min.

**Figure 67 molecules-27-02863-f067:**
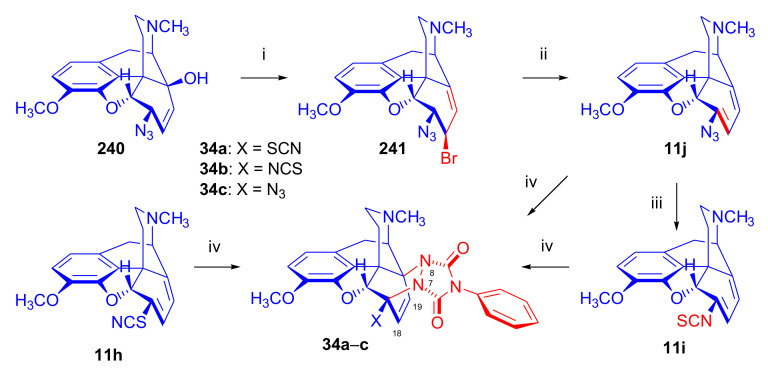
Synthesis of PTAD adducts of 6-substituted-6-demethoxythebaine derivatives. *Reagents and conditions*: (i). PBr_3_, CHCl_3_, 50 °C, 2 h; 77%; (ii): KO*t*Bu, ethanol, room temperature, 30 min, 75%; (iii): CS2, TPP, reflux, 2 h, 51%; and (iv): PTAD, acetone, room temperature, 20 min, 71–80%.

**Figure 68 molecules-27-02863-f068:**
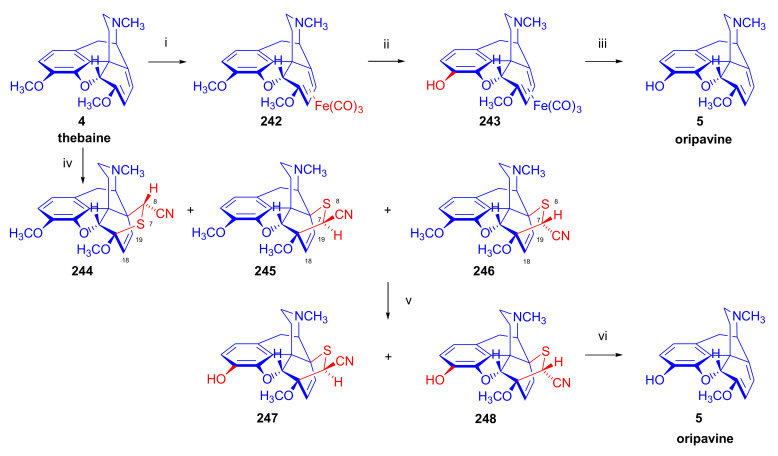
Conversion of thebaine to oripavine. *Reagents and conditions*: (i): iron pentacarbonyl, benzene, UV reactor, 40 °C, 48 h, 95%; (ii): 3-*O*-demethylation (**method A–D**: (**A**): BBr_3_, CH_2_Cl_2_, 0 °C, 20 min to room temperature, argon, 15 min, 83%; (**B**): BF_3_·SMe_2_, CH_2_Cl_2_, 0 °C, 4 h, room temperature, argon, 1.5 h, 83%; (**C**): MeSO_3_H, methionine, 50 °C, 28 h, 67%; (**D**): 1M 9-I-9-BBN in hexane, room temperature, 2 h, 70%; (iii): UV irradiation, acetonitrile, 40 °C, argon, 2.5 h, 35%; (iv): sodium *S*-(cyanomethyl)-sulfotioate, CaCl_2_·2H_2_O, Et_3_N, benzene-MeOH 1:1 (*v/v*), room temperature, 8 h, Σ yield (**244** + **245** + **246**) = 80%, product ratio: **244**:**245**:**246** = 1:4:3; (v): 3-*O*-demethylation (**method A–D**: (**A**): BBr_3_, CH_2_Cl_2_, 0 °C, 20 min to room temperature, argon, 15 min, 85%; (**B**): BF_3_·SMe_2_, CH_2_Cl_2_, 0 °C, 4 h, room temperature, argon, 2 h, 50%; (**C**): MeSO_3_H, methionine, 50 °C, 8 h, 51%; (**D**): 1M 9-I-9-BBN in hexane, room temperature, 4 h, 72%); and (vi): deprotection of the 6,8-diene system (**method A–B**: (A): 2,3-dimethylbutadiene, DMSO, BHT, 75 °C, 24 h, 65%, (B): (1) *m*CPBA, CH_2_Cl_2_, overnight, room temperature, argon, (2) EtOH, reflux, 2.5 h, 78%.

**Figure 69 molecules-27-02863-f069:**
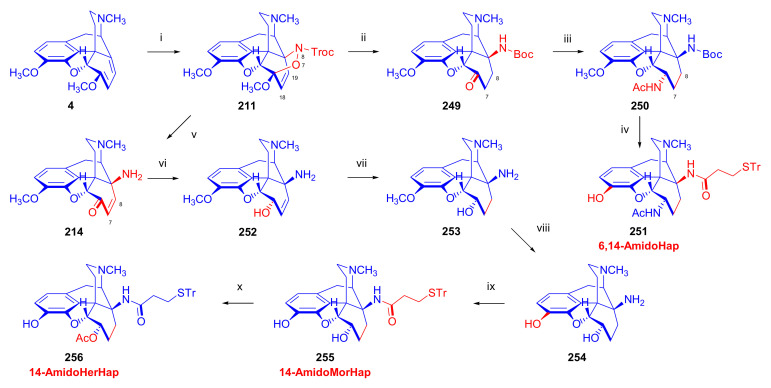
Synthesis of heroin vaccine haptens via HDA adduct of thebaine. *Reagents and conditions*: (i): 2,2,2-trichloroethyl *N*-hydroxycarbamate, NaIO_4_, 0.5M NaOAc (aq., pH 6), EtOAc, 25 °C, 95%; (ii): (1) H_2_, 10% Pd-C, AcOH, NaOAc, MeOH, 4.1 bar, RT, overnight; (2) Boc_2_O, NaHCO_3_, THF, H_2_O, 0 °C to 25 °C, 53%; (iii) (1) 2M NH_3_ in EtOH, Ti(OiPr)_4_, CH_2_Cl_2_, 25 °C 6 h; (2) NaBH_4_, EtOH, 25 °C, overnight; (3) Ac_2_O, Et_3_N, CH_2_Cl_2_, 40 °C, overnight, 81%; (iv): (1) BBr_3_, CH_2_Cl_2_, 1. −78 °C, 10 min, 2. 25 °C, 3 h, (2) (tritylthio)propanoic acid, TBTU, Et_3_N, 25 °C, overnight, 29%; (v): (1) 10-camphorsulfonic acid, ethylene glycol, CH_2_Cl_2_, RT, 2 h, (2) Zn, (NH_4_)_2_CO_3_, 100 °C, 5 h; (3) 6M HCl, MeOH-H_2_O 2:1 (*v/v*), reflux, 1 h, 47%; (vi): NaBH_4_, MeOH, RT, 3.5 h, quant.; (vii): H_2_, 5%Pd-C, MeOH, 3.4 bar, 2.5 h, 81%; (viii): BBr_3_, CH_2_Cl_2_, RT, 5 h, 84%; (ix): three equiv. (tritylthio)propanoic acid, three equiv. TBTU, three equiv. Et_3_N, DMF, RT, 3 h, 45%; and (x): Ac_2_O, DMAP, Et_3_N, CH_2_Cl_2_, RT, overnight, 39%.

**Figure 70 molecules-27-02863-f070:**
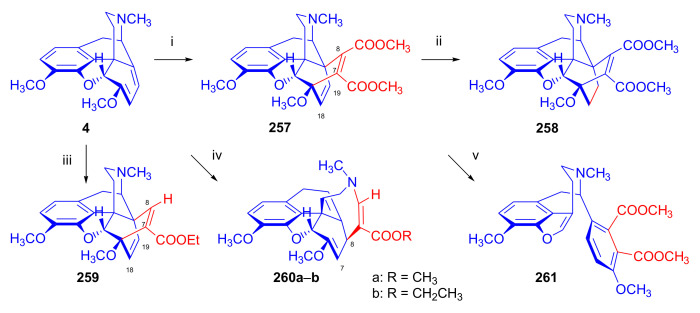
Reaction of thebaine with acetylenic dienophiles. *Reagents and conditions*: (i): dimethyl acetylenedicarboxylate (DMA, 1 equiv.), benzene, 50 °C, 1 h, 90%; (ii): H_2_, Pd-C, glacial acetic acid, 60 min; (iii): ethyl propiolate, 1.57 equiv., benzene, 50 °C, 8 h, 5.8%; (iv): ethyl propiolate or methyl propiolate, acetonitrile, room temperature, 30 min, quantitative yield; and (v): di-*n*-butylether, reflux, 10–15 min.

**Figure 71 molecules-27-02863-f071:**
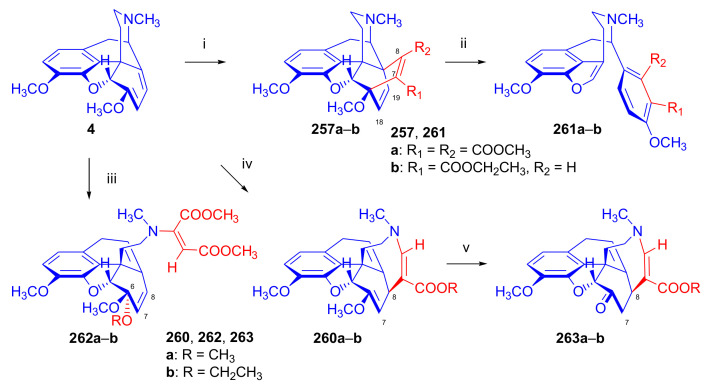
Reaction of thebaine with propargylic acid derivatives. *Reagents and conditions*: (i): DMAD or EP, benzene, 50 °C, 1 h; (ii): di-*n*-butylether, reflux, 10–15 min; (iii): DMAD, methanol, room temperature, 30 min, (iv): MP or EP, acetonitrile, room temperature, 30 min; and (v): 10% aqueous oxalic acid room temperature, 45 min.

**Figure 72 molecules-27-02863-f072:**
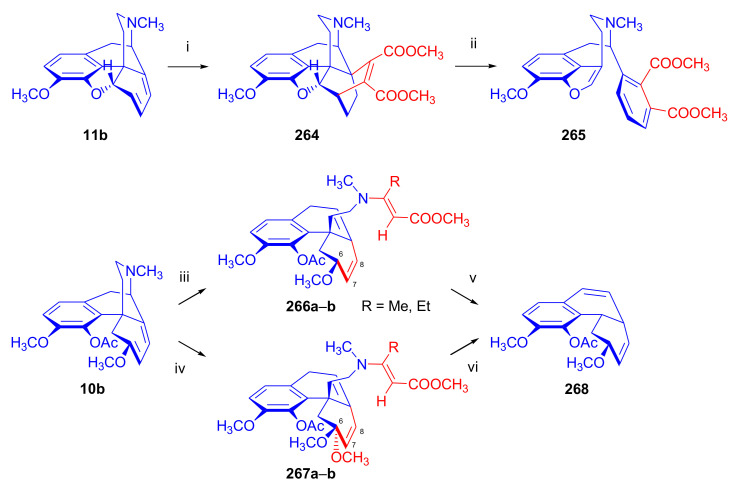
Reactions of 6-demethoxythebaine and 4-*O*-acetyl-β-dihydrothebaine with dimethyl acetylendicarboxylate and propargylic acid ethyl ester. *Reagents and conditions*: (i): MP or DMAD, benzene, reflux, 12 h, 72%; (ii): xylene, reflux, 10 min; (iii): MP or DMAD acetonitrile, room temperature, 30 min, or benzene, room temperature, 3h; (iv): MP or DMAD, methanol, room temperature, 5–10 min; (v): chloroform, 50 °C, 5 min; and (vi): thermal decomposition at 70 °C.

**Figure 73 molecules-27-02863-f073:**
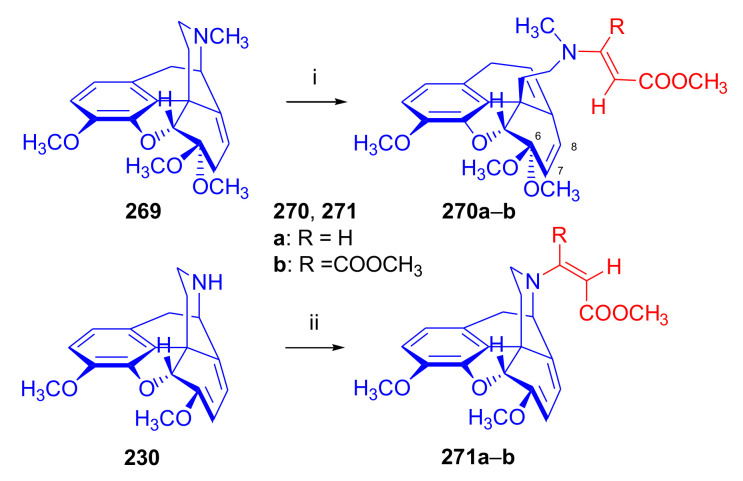
Reaction of neopinone dimethyl ketal and *N*^17^-northebaine with acetylenic dienophiles. *Reagents and conditions*: (i): MP or DMAD, acetonitrile, room temperature, 8–12 h, 48–53%; (ii): MP or DMAD, acetonitrile, room temperature, 30 min, 95–97%.

**Figure 74 molecules-27-02863-f074:**
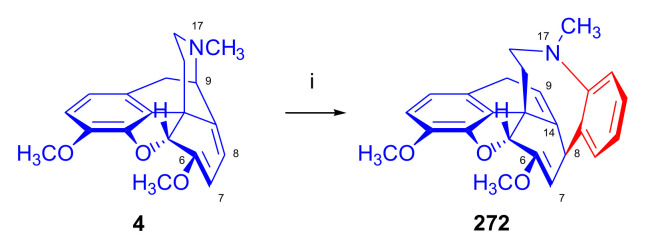
Reaction of thebaine with in situ generated benzyne. *Reagents and conditions*: (i): (1) antranilic acid, trichloroacetic acid, THF, isoamyl nitrite, 0 °C, 2 min, (2) 20 °C, 1.5 h, 3. dichloromethane, reflux, 30 min.

**Figure 75 molecules-27-02863-f075:**
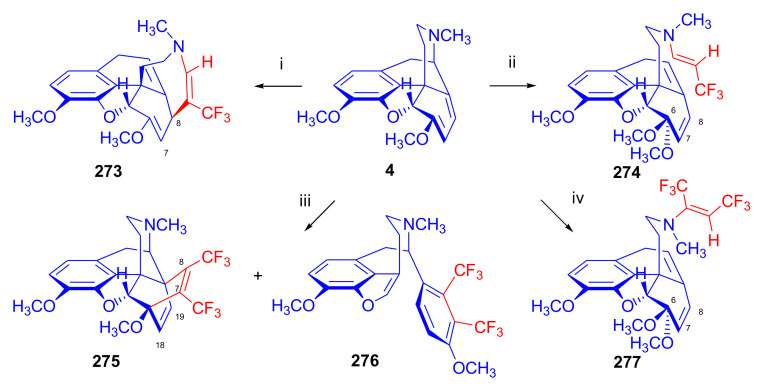
Reaction of thebaine with trifluoromethyl substituted acetylenic dienophiles. *Reagents and conditions*: (i): trifluoropropyne, sealed tube, room temperature, 18 h; (ii): trifluoropropyne, methanol, 18 h, room temperature; (iii): hexafluoro-2-butyne, benzene, 50 °C, 18 h; and (iv): hexafluoro-2-butyne, methanol, 50 °C, 18 h.

**Figure 76 molecules-27-02863-f076:**
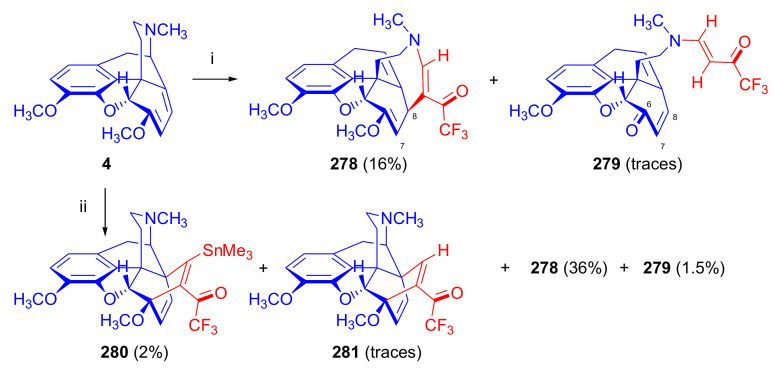
Reaction of thebaine with fluorinated acetylenes. *Reagents and conditions*: (i): trifluoroacetylacetylene, THF; (ii): 4-trimethylstannyl-1-trifluoroacetylacetylene.

**Figure 77 molecules-27-02863-f077:**
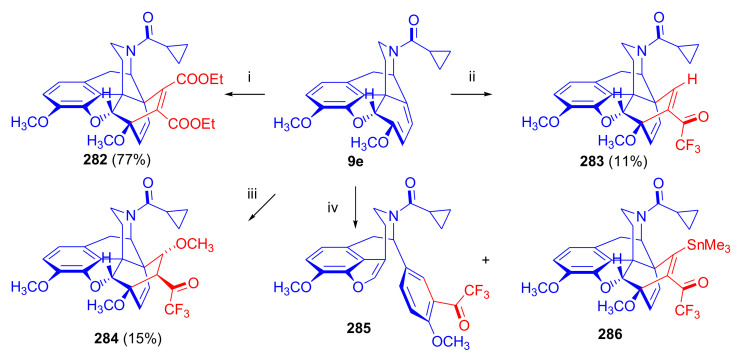
Reaction of *N*^17^-cyclopropylcarbonyl-northebaine with acetylenic dienophiles. *Reagents and conditions*: (i): diethyl acetylenedicarboxylate, acetonitrile, 50–55 °C, 77%; (ii): trifluoroacetylacetylene, THF, < 40 °C, 11%; and (iii) (1) trifluoroacetylacetylene, THF < 40 °C, (2) CH_3_OH, reflux, 1 h, 15%; (iv): 4-trimethylstannyl-1-trifluoroacetylacetylene, chlorobenzene, 90 °C, or THF, reflux.

**Figure 78 molecules-27-02863-f078:**
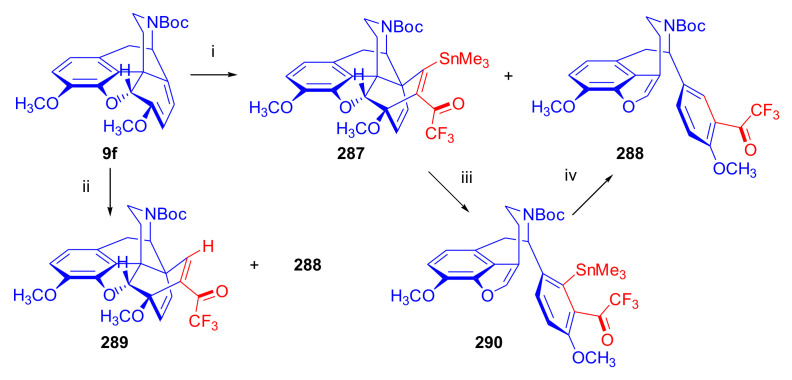
Reaction of *N*^17^-*tert*-butoxycarbonyl-northebaine with acetylenic dienophiles. *Reagents and conditions*: (i): 4-trimethylstannyl-1-trifluoroacetylacetylene, chlorobenzene, 65–75 °C, 4 h, or THF, reflux, 6.5 h; (ii): trifluoroacetylacetylene, THF, 5–20 °C, a few minutes.

**Figure 79 molecules-27-02863-f079:**
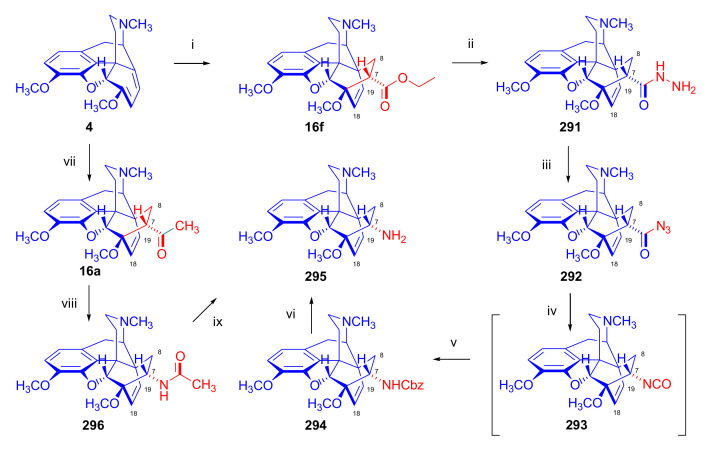
Synthesis of 7α-amino-6,14-*endo*-ethenotetrahydorothebaine. *Reagents and conditions*: (i): ethyl acrylate, reflux, 6 h; (ii): hydrazine hydrate, 2-ethoxyethanol, reflux, 8 h; (iii), sodium nitrite, 15% hydrochloric acid; (iv) and (v): benzyl alcohol, reflux, 10 min; (vi): HCl in EtOH, reflux, 1 h; (vii): methyl vinyl ketone, reflux, 1 h; (viii): sodium azide, 72% w/w perchloric acid, 65–75 °C, 5 h; and (ix): 5M HCl, steam-bath, 4 h.

**Figure 80 molecules-27-02863-f080:**
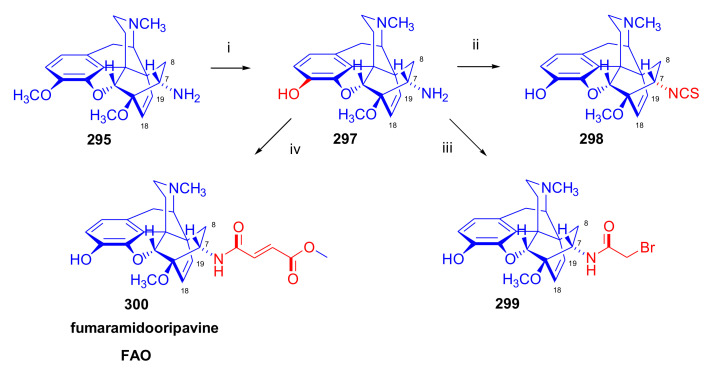
Synthesis of 7α-amino-6,14-ethenomorphinan derivatives as irreversible ligands for opioid receptors. *Reagents and conditions*: (i): BBr_3_, CHCl_3_, 15 min, 68%; (ii): thiophosgene, NaHCO_3_, CHCl_3_, H_2_O, 15 min, 63%; (iii): bromoacetic anhydride, NaHCO_3_, CHCl_3_, H_2_O, 15 min, 61%; and (iv): methylfumaroyl chloride, NaHCO_3_, CHCl_3_, H_2_O, 15 min, 82%.

**Figure 81 molecules-27-02863-f081:**
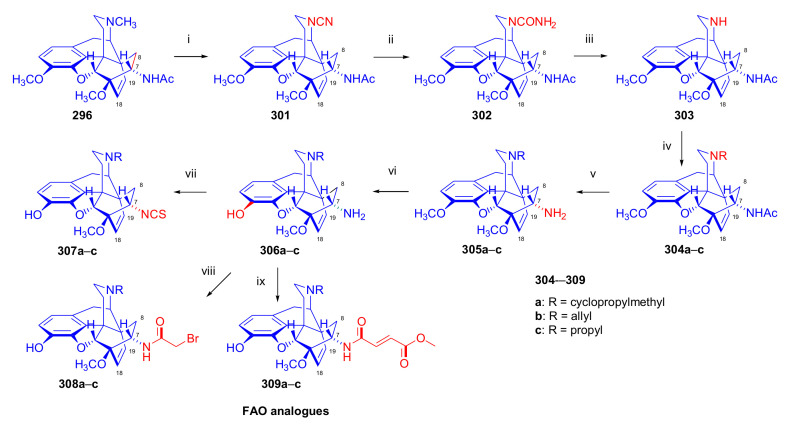
Synthesis of *N*^17^-substituted-7α-acylamino-6,14-ethenomorphinan derivatives. *Reagents and conditions*: (i): BrCN, CHCl_3_, 16 h, reflux; (ii): 1M HCl, 90 °C; (iii): NaNO_2_, 1M HCl, 4 h, 0 °C; (iv): RBr, Na_2_CO_3_, EtOH, 16–30 h, reflux or NaHCO_3_, DMF, 85 °C, 4 h; (v): 3M HCl, 110–115 °C; (vi): BBr_3_, CHCl_3_; (vii): 1.1 equiv. thiophosgene, NaHCO_3_, CHCl_3_, H_2_O, 20 min; (viii): bromoacetyl chloride, NaHCO_3_, CHCl_3_, H_2_O, 20 min; and (ix): methylfumaroyl chloride, NaHCO_3_, CHCl_3_, H_2_O, 20 min.

**Figure 82 molecules-27-02863-f082:**
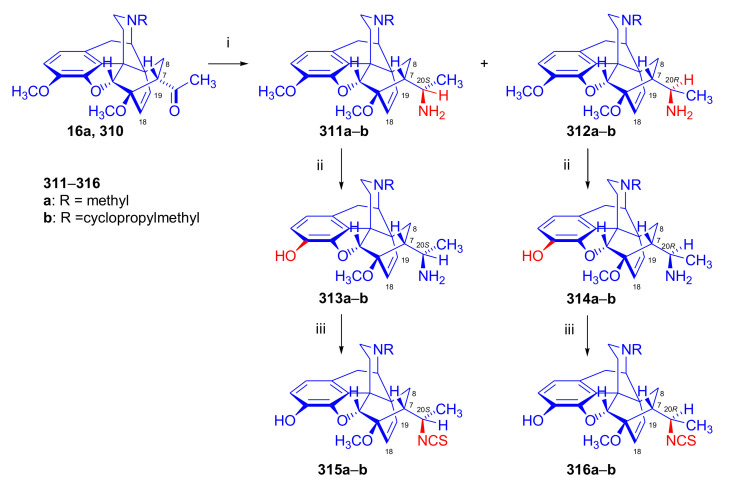
Synthesis of diastereomeric 20-isothiocyanato-6,14-ethenomorphinan derivatives. *Reagents and conditions*: (i): sodium cyanoborohydride, ammonium acetate, MeOH, 3 days; (ii): KOH, diethylene glycol, 210 °C, 140 min; and (iii): thiophosgene, NaHCO_3_, CH_2_Cl_2_, RT, 80 min.

**Figure 83 molecules-27-02863-f083:**
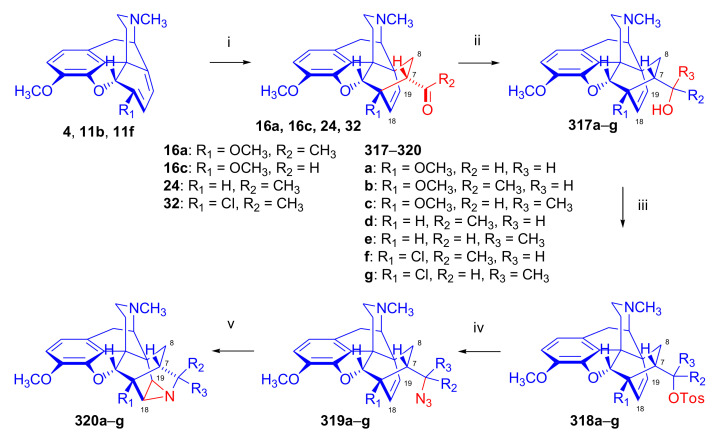
Synthesis of morphinans with a 4-azatetracyclo-[4.4.0.0^3,8^.0^2,4^]-decane ring system. *Reagents and conditions*: (i): acrolein or methyl vinyl ketone, toluene, reflux 24 h; (ii): 4.5 equiv. sodium borohydride, methanol, 0 °C, 30 min; (iii): 1.5 equiv. TosCl, pyridine, room temperature, one day to 6 days; and (iv,v): 5 equiv. sodium azide, H_2_O, 100 °C, 24 h.

**Figure 84 molecules-27-02863-f084:**
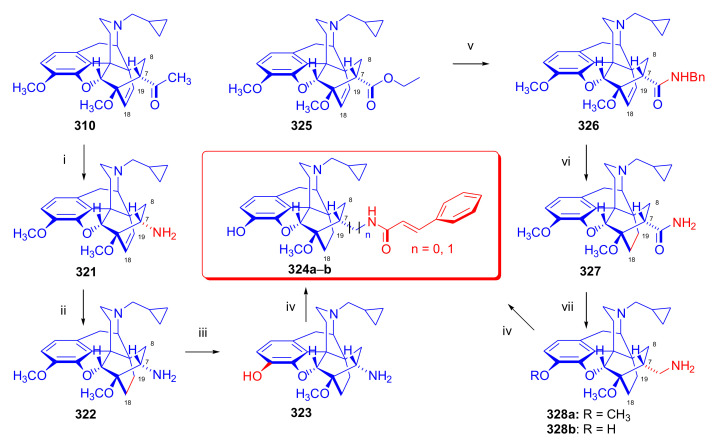
Synthesis of cinnamoyl derivatives of 7α-amino- and 7α-aminomethyl-*N*^17^-cyclopropylmethyl-6,14-*endo*-ethano-tetrahydoronororipavine. *Reagents and conditions*: (i): (1) HClO_4_, NaN_3_, (2) 5M HCl; (ii): H_2_, Pd-C, 2 bar, EtOH; (iii): KOH, diethylene glycol, reflux, 11 h; (iv): cinnamoyl chloride, CH_2_Cl_2_, NaHCO_3_; (v): (1) 2M hydrochloric acid, steam bath, 3 h; (2) oxalyl chloride, CH_2_Cl_2_, DMF, nitrogen atmosphere, RT, 1.5 h; (3) benzylamine, Et_3_N, CH_2_Cl_2_; (vi): (1) LiAlH_4_, THF, reflux, (2) H_2_, Pd-C, 45 °C, 2 bar; and (vii): KOH, diethylene glycol, 8 h, reflux.

**Figure 85 molecules-27-02863-f085:**
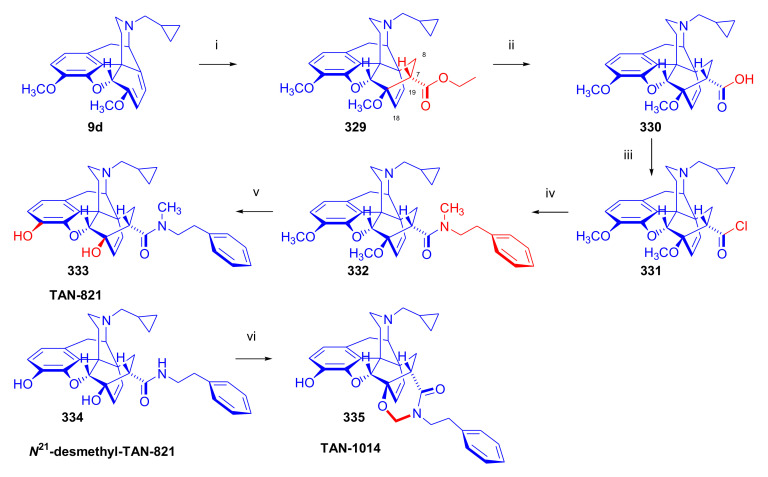
Synthesis of 6,14-ethenomorphinan ligands for the putative ε opioid receptor. *Reagents and conditions*: (i): ethyl acrylate, reflux, 15 h, 69%; (ii): 6M HCl, reflux, 7 h, 76%; (iii): oxalyl chloride, CHCl_3_, reflux, 2 h; (iv): *N*-methy-*N*-phenethylamine, Et_3_N, CHCl_3_, room temperature, 1 h; (v): 1M BBr_3_ in CH_2_Cl_2_, CHCl_2_, 0 °C, 1 h, 74%; and (vi): (CH_2_O)_n_, H_2_SO_4_, dioxane, reflux, 11 h, 61%.

**Figure 86 molecules-27-02863-f086:**
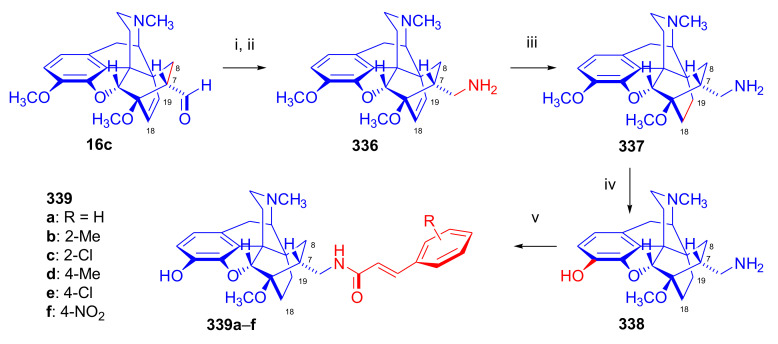
Synthesis of aryl substituted cinnamoyl derivatives of 7α-aminomethyl-*N*^17^-methyl-6,14-*endo*-ethanotetrahydoro-nororipavine. *Reagents and conditions*: (i): NH_2_OH·HCl, EtOH-H_2_O 1:1 (*v/v*), reflux, 6 h; (ii): LiAlH_4_, THF, reflux; (iii): H_2_, Pd-C, EtOH, 40 bar; (iv): BBr_3_, CH_2_Cl_2_, under nitrogen; and (v): substituted cinnamoyl chloride, Et_3_N, CH_2_Cl_2_, RT, overnight.

**Figure 87 molecules-27-02863-f087:**
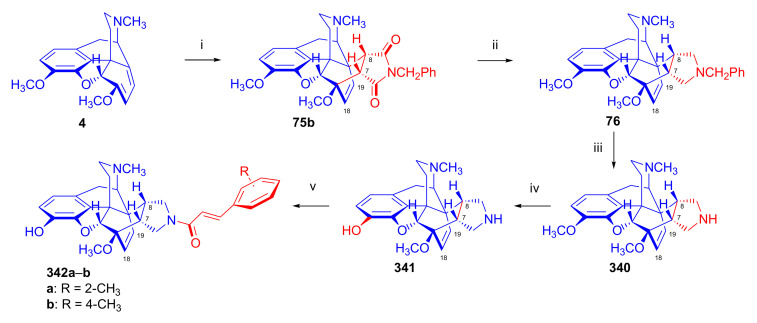
Synthesis of aryl substituted cinnamoyl derivatives of C^7^-C^8^-ring constrained 6,14-ethenomorphinan analogues. *Reagents and conditions*: (i): *N*-benzylmaleimide, toluene, reflux, 18 h; (ii): LiAlH_4_, THF, reflux, 16 h; (iii): H_2_, 10% Pd-C, EtOH, cc. HCl, 2.7 bar, 5 days; (iv): BBr_3_, CH_2_Cl_2_, 15 min; and (v): substituted cinnamoyl chloride, Et_3_N, CH_2_Cl_2_, RT, overnight.

**Figure 88 molecules-27-02863-f088:**
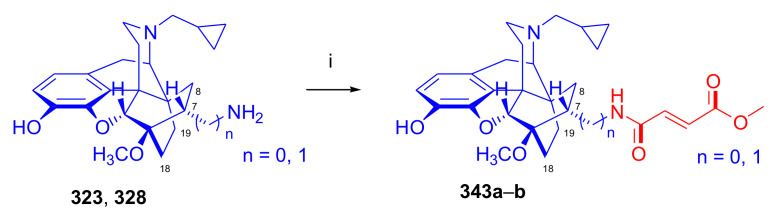
Synthesis of 7α-fumaroylamino-6,14-ethenomorphinan analogues. *Reagents and conditions*: (i): MeO_2_CCH=CHCOCl, Et_3_N, CH_2_Cl_2_, under nitrogen, 1 h, **343a**, 65%, **343b**, 45%.

**Figure 89 molecules-27-02863-f089:**
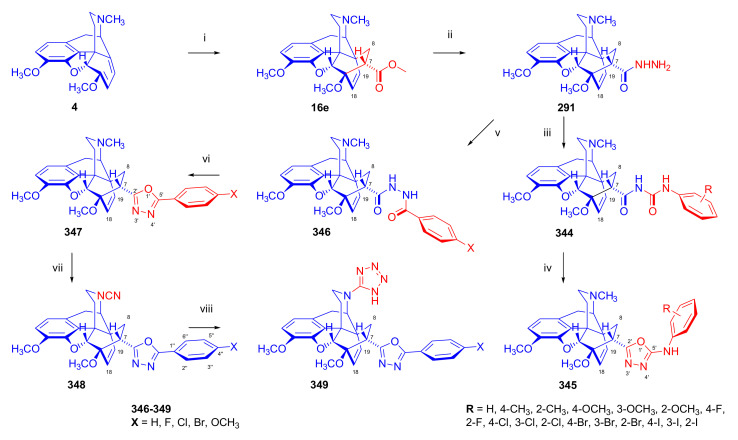
Synthesis of 7α-(1,3,4-oxadiazol-2-yl)arenamine- and 7α-(5-(halophenyl))-1,3,4-oxadiazole-substituted 6,14-ethenomorphinan derivatives. *Reagents and conditions*: (i): methyl acrylate, reflux, 6 h; (ii): hydrazine hydrate, 2-ethoxyethanol, reflux, 8 h; (iii): one equiv. aryl-isocyanate, toluene, 70 °C, 2 h; (iv): POCl_3_, 90 °C, 3 h; (v): one equiv. substituted benzoyl chlorides, toluene, reflux, 1 h; (vi): POCl_3_, reflux, 3 h; (vii): two equiv. cyanogen bromide, CHCl_3_, reflux, 24 h; and (viii): NaN_3_, NH_4_Cl, DMF, reflux, 4 h.

**Figure 90 molecules-27-02863-f090:**
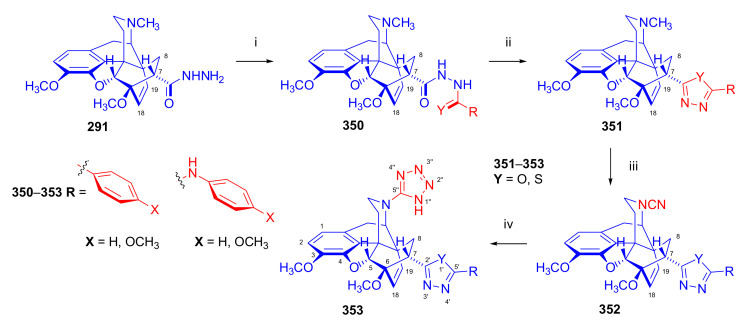
Synthesis of *N*^17^-(tetrazol-1*H*-5-yl)-7α-substituted (1,3,4-oxadiazol-2-yl) and (1,3,4-thiadiazol-2-yl) 6,14-ethenomorphinan derivatives. *Reagents and conditions*: (i): (A) benzoyl chloride or *p*-tolylchloride, or (B) phenylisocyanate or *p*-tolylisocyanate, or (C) phenyl isothiocyanate or *p*-tolylisothiocyanate, toluene, 70 °C, 2 h, 78–88%; (ii): POCl_3_ or H_3_PO_4_, 72–82% (iii): 2 equiv. BrCN, CHCl_3_, reflux, 24 h, 39–53%; and (iv) NaN_3_, NH_4_Cl, DMF, reflux, 4 h, 69–77%.

**Figure 91 molecules-27-02863-f091:**
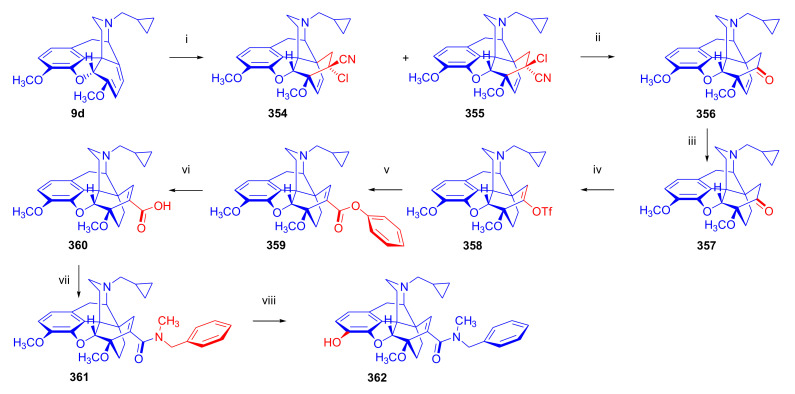
Synthesis of Δ^7,8^-7-carboxamido-6,14-ethenomorphinan derivatives. *Reagents and conditions*: (i): 2-chloroacrylonitrile, aromatic hydrocarbon, microwave irradiation, 180 °C, 10 bar, 30 min; (ii): 1M NaOH (aq), EtOH, 44%; (iii): H_2_, 5% Pd-C, EtOH, 60 °C, 12 h, 92%; (iv): *N*-phenyl-bis(trifluoromethanesulfonamide), KHMDS, toluene, -78 °C, 1 h, 99%; (v): 2,4,6-trichlorophenyl formate, palladium acetate, Xantphos, Et_3_N, 90%; (vi): (A) 1M NaOH (aq), THF, 60 °C, 7 h, or (B) 6M NaOH, 60 °C, 3 h, 60%; (vii): benzylmethylamine, THF, Et_3_N, DMAP, 50 °C, 3.5 h, 86%; and (viii): BBr_3_, CH_2_Cl_2_, room temperature, 1.5 h, quant.

**Figure 92 molecules-27-02863-f092:**
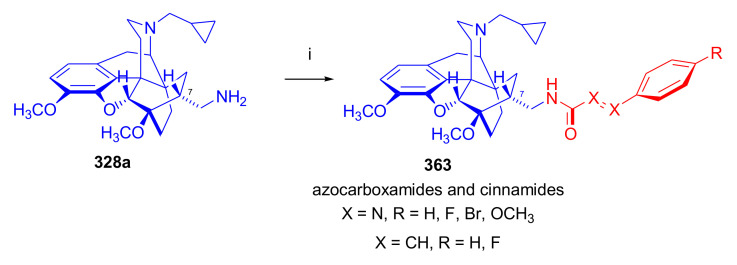
Synthesis of azocarboxamide and cinnamide type 6,14-ethenomorphinan derivatives. *Reagents and conditions*: (i) (A) substituted phenylazocarboxylate *tert*-butyl esters (R = H, F, Br, OCH_3_), K_2_CO_3_, or Et_3_N, EtOH, RT, 3–120 h, 42–87%; or (B) cinnamic acid chlorides (R = H, F), NaHCO_3_, CH_2_Cl_2_, RT, 28 h, 25–63%.

**Figure 93 molecules-27-02863-f093:**
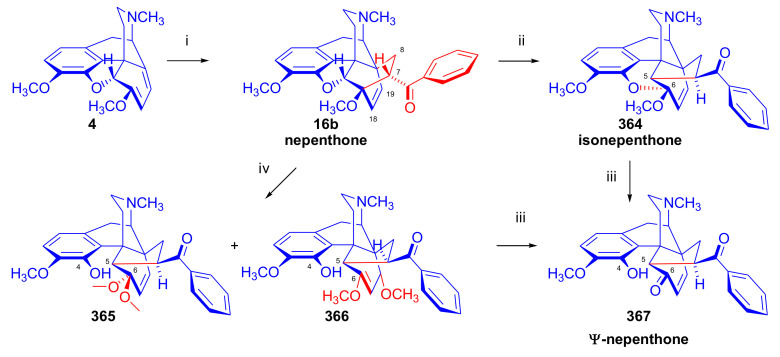
Synthesis of nepenthone and its rearranged derivatives. *Reagents and conditions*: (i): phenyl vinyl ketone, benzene, reflux, 2 h, 70%; (ii): NaOH, MeOH, reflux, 5–10 min., 75%; (iii): 5% HCl or AcOH; and (iv): NaOH, MeOH, reflux, 4 h.

**Figure 94 molecules-27-02863-f094:**
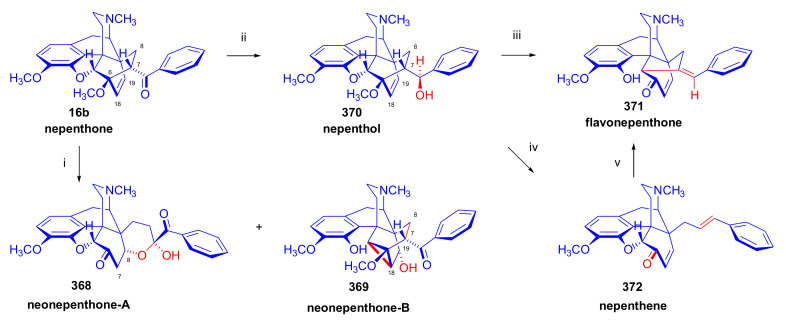
Acid-catalyzed rearrangement of nepenthone and its derivatives. *Reagents and conditions*: (i): 1:1 (*v/v*) cc. hydrochloric acid, glacial acetic acid, 100 °C, 6 h; (ii): aluminum isopropoxide, 2-propanol, reflux, 84%; or NaBH_4_, methanol, 30%; (iii): 1:1 (*v/v*) cc. hydrochloric acid, glacial acetic acid, 100 °C, 1 h, 84%; (iv): formic acid, reflux, 16 h, 76%; and (v): 1:1 (*v/v*) cc. hydrochloric acid, glacial acetic acid, 100 °C, 6 h.

**Figure 95 molecules-27-02863-f095:**
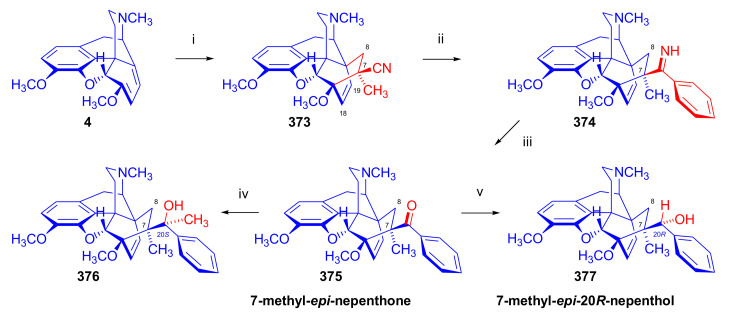
Synthesis of 7-methy-*epi*-nepenthone and its derivatives. *Reagents and conditions*: (i): methacrylonitrile, xylene, reflux, 40 h, 40%; (ii): one equiv. PhMgBr, benzene, reflux, 18 h, 65%; (iii): 50% AcOH (aqueous), 100 °C, 45 min, 76%; (iv): MeMgI, benzene, reflux, 3 h; and (v): sodium borohydride, 2-ethoxyethanol.

**Figure 96 molecules-27-02863-f096:**
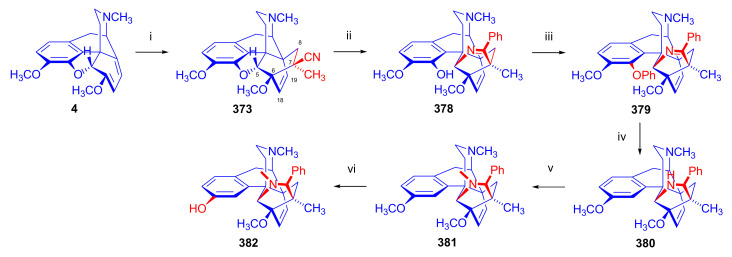
Synthesis of morphinan cyclic imines and pyrrolidines. *Reagents and conditions*: (i): methacrylonitrile, xylene, reflux, 40 h, 40%; (ii): PhMgBr, benzene; reflux, 18 h; (iii): C_6_H_5_Br, K_2_CO_3_, Cu, pyridine, reflux, 3 days; (iv): 10 eq. Na, NH_3_ (l), −50 °C, 2 h; (v): CH_2_O, NaCNBH_3_, CH_3_CN, DMSO; and (vi): PrSH, NaH, HMPA, 110 °C, 3 h, 66%.

**Figure 97 molecules-27-02863-f097:**
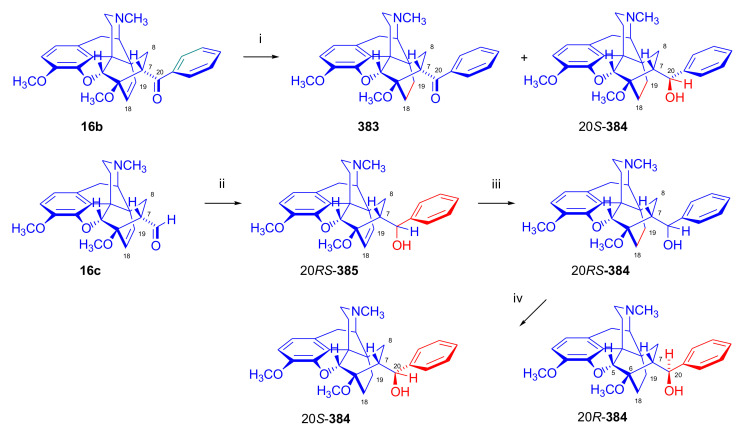
Catalytic hydrogenation of nepenthone under different conditions and synthesis of 20*R*-dihydronepenthol. *Reagents and conditions*: (i): H_2_, EtOH, catalyst (Pd-C, Pd-C + PtCl_4_, Pt-C or Rh-C), p (4–6 bar), t (25–64 °C), (ii): PhMgBr, toluene-diethyl ether; (iii): H_2_, EtOH, 3 bar, 55 °C; and (iv): fractional crystallization.

**Figure 98 molecules-27-02863-f098:**
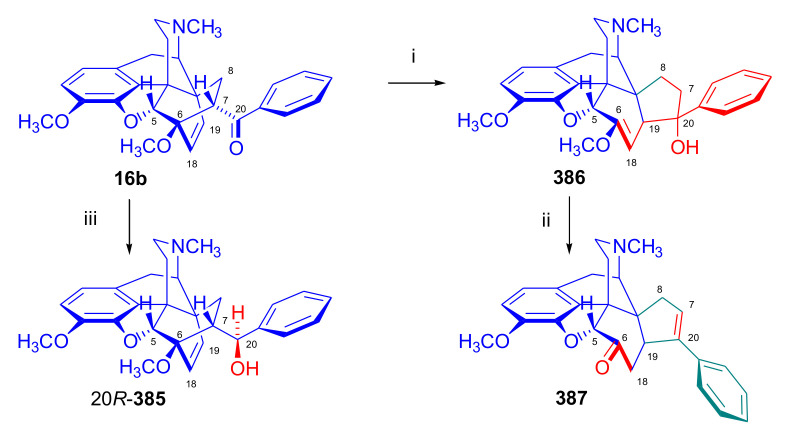
Reaction of nepenthone with chromium(II) reagents in aqueous medium. *Reagents and conditions*: (i): from nepenthone bitartrate (1 mmol), ethylenediaminetetraacetate (EDTA, Na_2_EDTA·H_2_O, 2.2 mmol), KOH (OH^−^: 3.34 mmol), water, [Cr(OAc)_2_·H_2_O]_2_ (1 mmol, 2 mmol CrII), pH = 5.2, argon, 18 h, 62%; (ii): 3% hydrochloric acid, reflux, 3 h, 96%; and (iii): from nepenthone bitartrate (1 mmol), iminodiacetate (IDA, 4 mmol), KOH (OH^−^: 5.57 mmol), water, [Cr(OAc)_2_·H_2_O]_2_ (1 mmol, 2 mmol CrII), pH = 6.1, argon, 18 h, 86%.

**Figure 99 molecules-27-02863-f099:**
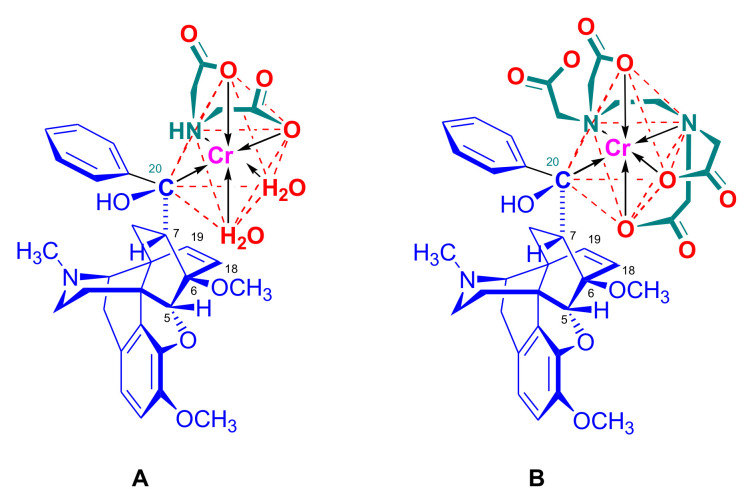
Structures of proposed organometallic complex intermediates formed from nepenthone in aqueous medium. **A**: [Cr(IDA)(H_2_O)_3_]-nepenthone complex. **B**: [Cr(EDTA)]^2−^-nepenthone complex.

**Figure 100 molecules-27-02863-f100:**
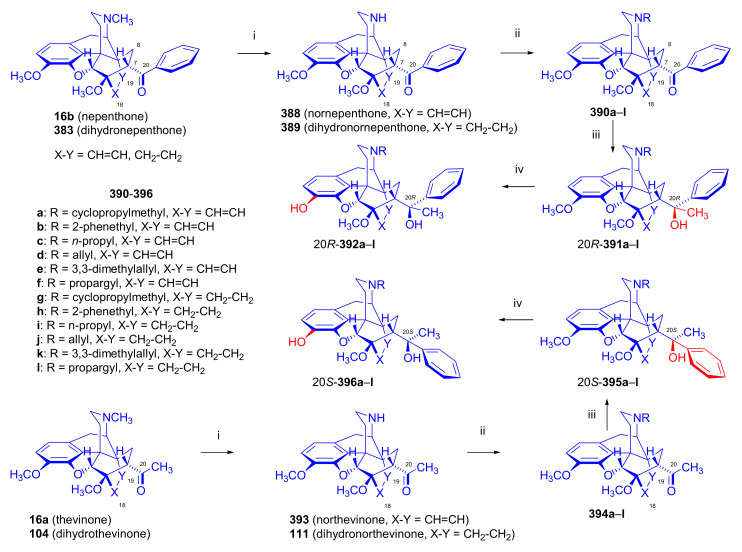
Synthesis of 20*R*- and 20*S*-phenyl-6-14-ethenomorphinan derivatives. *Reagents and conditions*: (i): (1) DEAD, benzene, reflux, 7 h, (2) pyridinium chloride, EtOH, 8 h, room temperature, 68–82%; (ii): RBr, NaHCO_3_, 90 °C, 20 h, 51–90%; (iii): for **390a–j**: MeMgI, toluene-THF, reflux, 1 h; 68–85% or for **394a–j**: PhMgBr, toluene-diethyl ether, reflux, 1 h, 67–79%; and (iv): KOH, diethylene glycol, nitrogen atmosphere, 210–220 °C, 42–65%.

**Figure 101 molecules-27-02863-f101:**
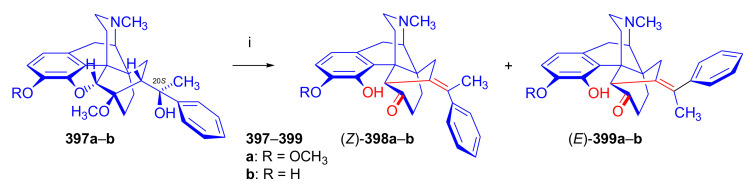
Synthesis of flavonepenthone analogues. *Reagents and conditions*: (i): 12M hydrochloric acid, 100 °C, 2 h.

**Figure 102 molecules-27-02863-f102:**
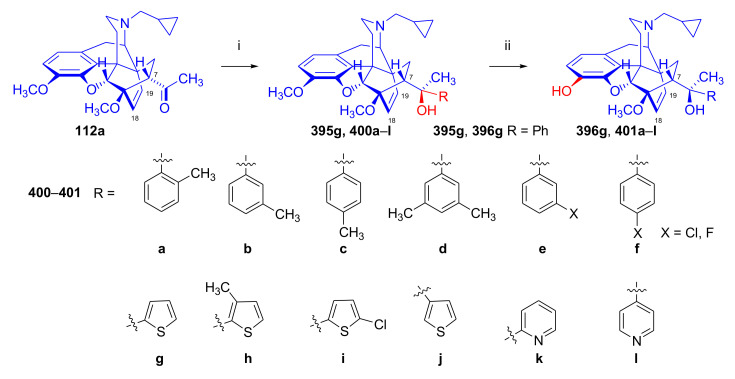
Synthesis of 20-aryl-thevinols and orvinols. *Reagents and conditions*: (i): for preparation of 20*S*-aryl-thevinols: arylmagnesium halides, toluene, room temperature, 22 h; for preparation of 20*R*-piridyl-thevinols: (1) pyridyl halide, *n*-BuLi, Et_2_O, −78 °C, under nitrogen, 10 min, (2) room temperature, 20 h; and (iii): 1-propanetiol, NaH, HMPA, 120 °C, 3 h.

**Figure 103 molecules-27-02863-f103:**
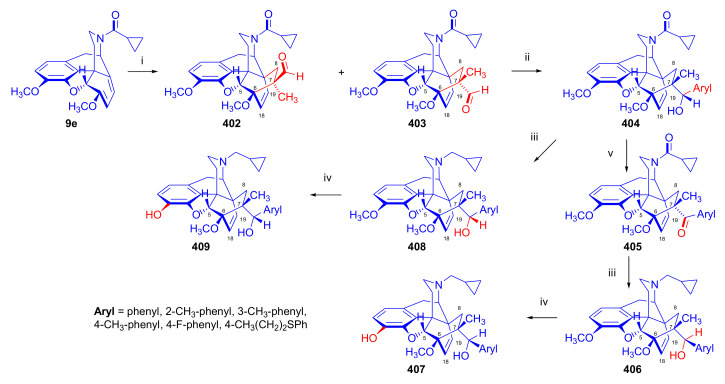
Synthesis of 7β-methyl orvinol analogues. *Reagents and conditions*: (i): metacrolein, LiBF_4_, room temperature, 16 h; (ii): two equiv. ArMgBr, three equiv. Bu_4_NBr, THF, reflux, 48 h; (iii): LiAlH_4_, THF; (iv): PrSNa, HMPA; and (v): oxalyl chloride, DMSO, CH_2_Cl_2_.

**Figure 104 molecules-27-02863-f104:**
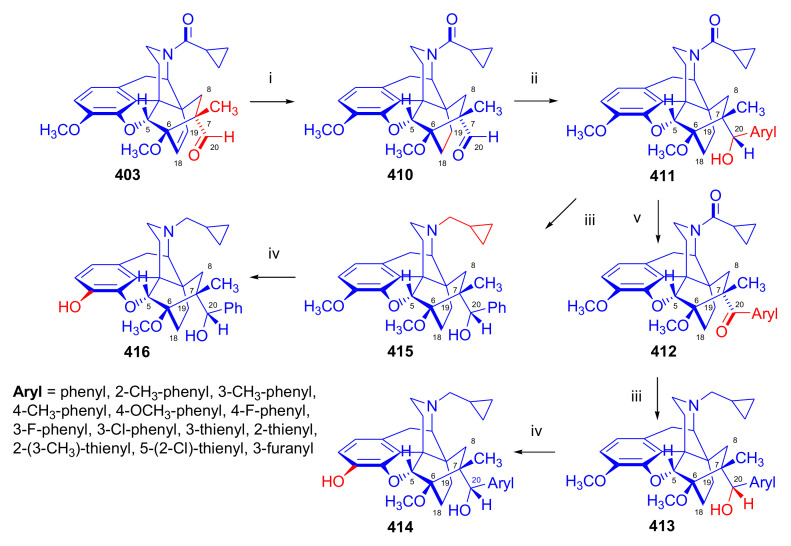
Synthesis of 7β-methyl-20-aryl-18,19-dihydro-orvinol analogues. *Reagents and conditions*: (i): H_2_, Pd-C, EtOH, 6.9 bar, 12 h; (ii): two equiv. ArMgBr, three equiv. Bu_4_NBr, THF, reflux, 48 h; (iii): LiAlH_4_, THF; (iv): PrSNa, HMPA; and (v): oxalyl chloride, DMSO, CH_2_Cl_2_.

**Figure 105 molecules-27-02863-f105:**
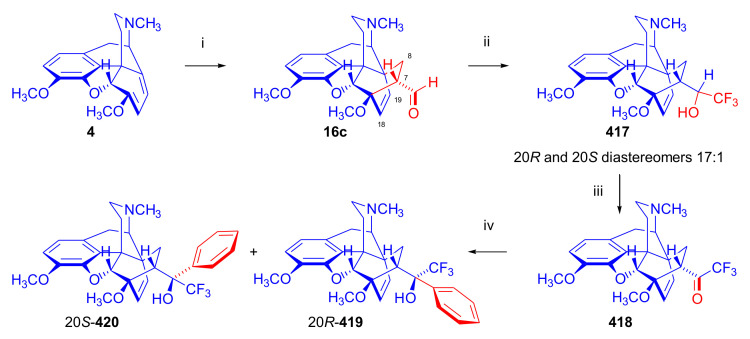
Synthesis of fluorinated 20*R*- and 20*S*-phenylthevinols. *Reagents and conditions*: (i): acroleine, benzene, reflux; (ii): (1) Me_3_SiCF_3_, TBAF, THF, (2) HCl/H_2_O; (iii): DMSO, oxalyl chloride, CH_2_Cl_2_, (1) -70 °C, 1.5 h, (2) Et_3_N, 20 °C, 10 min; and (iv): phenylmagnesium bromide, THF, 20 °C, 24 h.

**Figure 106 molecules-27-02863-f106:**
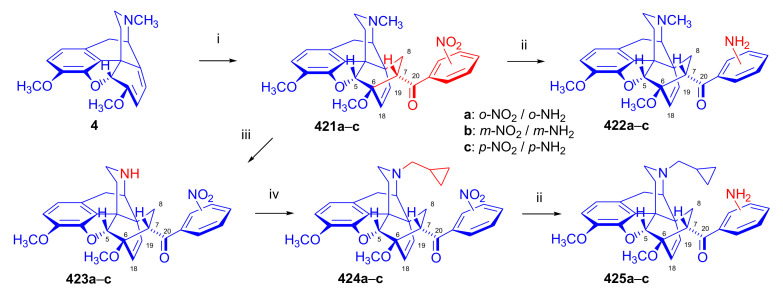
Synthesis of nepenthone analogues. *Reagents and conditions*: (i): (*o*-, *m*-, *p*-)-nitrophenyl vinyl ketones, toluene, reflux; (ii): Raney-Ni, 85% hydrazine hydrate, ethanol, 60 °C; (iii): (1) DEAD, benzene, reflux, (2) 1M hydrochloric acid, 70–80 °C; and (iv): cyclopropylmethyl bromide, K_2_CO_3_, DMF, 80–90 °C, 10 h.

**Figure 107 molecules-27-02863-f107:**
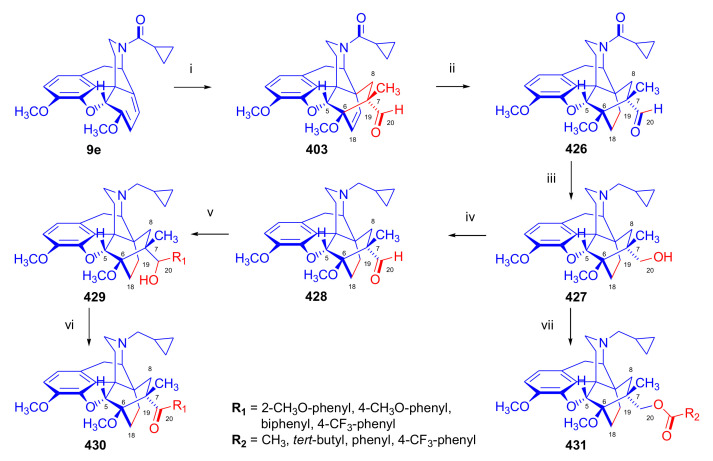
Synthesis of *N*^17^-cyclopropylmethyl-nornepenthone derivatives. *Reagents and conditions*: (i): metacrolein, LiBF_4_, CH_2_Cl_2_, room temperature, 28 h, 25%; (ii): H_2_, Pd-C, EtOAc, room temperature, 24 h, 90%, (iii): LiAlH_4_, THF, 0 °C, room temperature, 2 h, 80%; (iv): oxalyl chloride, DMSO, CH_2_Cl_2_, Et_3_N, −78 °C, 2 h, 95%; (v): 10 equiv. R_1_X, *n*-BuLi, *n*-hexane, THF, −78 °C, 2 h, 21–50%; (vi): Dess-Martin periodinane, CH_2_Cl_2_, 0 °C, 4 h, 17–80%; and (vii): R_2_COCl, CH_2_Cl_2_, Et_3_N, room temperature, 2 h, 18–83%.

**Figure 108 molecules-27-02863-f108:**
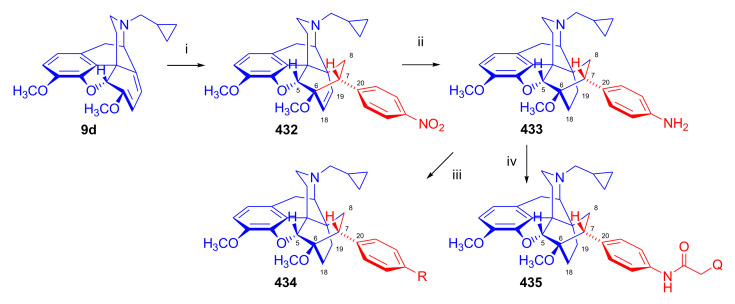
Synthesis of 7α-phenyl-6,14-ethenomorphinan derivatives. *Reagents and conditions*: (i): 4-nitrostyrene, xylene, reflux, 24 h, 78%; (ii): H_2_, Pd-C, 4.4 bar, 50 °C, 36 h, 84%; (iii): (A) RX, NaCO_3_, DMF, 30 °C, 4 h or (B) dibromoalkanes, Na_2_CO_3_, acetonitrile, 110 °C, microwave, or **C**: RX/RCOCl, Et_3_N, CH_2_Cl_2_, 70 °C, 2 h; and (iv): various acids, DIPEA, HATU, CH_2_Cl_2_, room temperature, 2 h.

**Figure 109 molecules-27-02863-f109:**
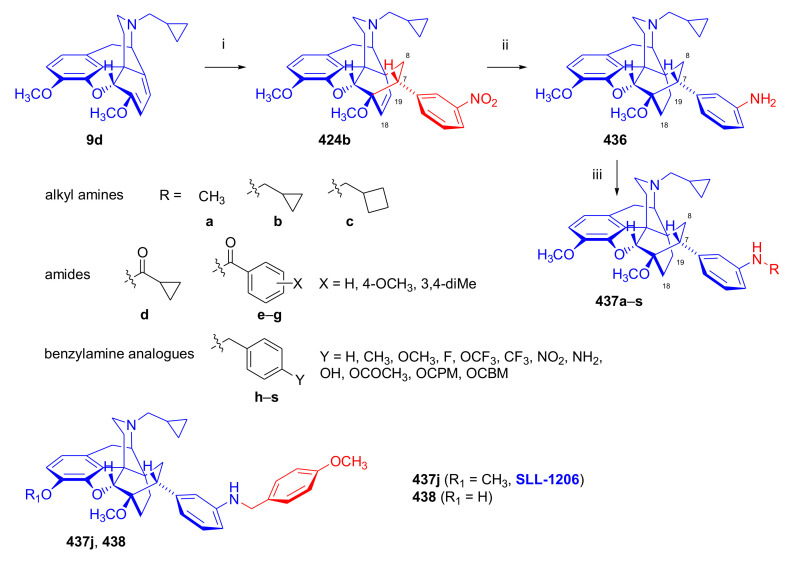
Synthesis of 7α-(*m*-substituted-phenyl)-*N*^17^-cyclopropylmethyl-6,14-etheno morphinans. *Reagents and conditions*: (i): 3-nitrostyrene, xylene, reflux, 30 h, 46%; (ii): H_2_, Pd-C, MeOH, 4.4 bar, 25 °C, 92%; and (iii): (A) alkyl amines: R = Me: (1): Cbz-Cl, DIPEA, CH_2_Cl_2_, 35 °C, (2): LiAlH_4_, THF, 0 °C, 47%; or (B) haloalkanes, Na_2_CO_3_, DMF, 80 °C, 15–78%; **C**-amides: carboxylic acid, HATU, DIPEA, or acid chlorides, Et_3_N, CH_2_Cl_2_, RT, 23–55%.

**Figure 110 molecules-27-02863-f110:**
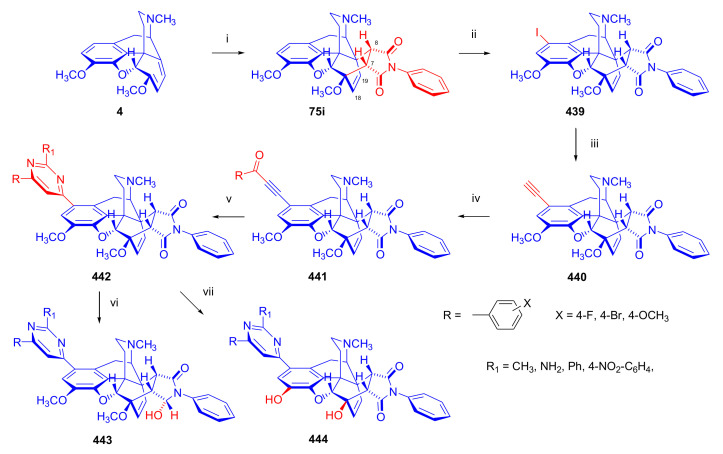
Synthesis of 1-pyrimidino-6,14-ethenomorphinans. *Reagents and conditions*: (i): *N*-phenylmaleimide, benzene, 80 °C, 15 min, 68%; (ii): NIS, TFA, 2 h, RT; (iii): (1) trimethylsilylacetylene, Pd[(PPh_3_)_2_]Cl_2_, PPh_3_, CuI, Et_3_N, DMF, 70 °C, 19 h, (2) Bu_4_NF, CH_2_Cl_2_, 15 min RT; (iv): Pd[(PPh_3_)_2_]Cl_2_, PPh_3_, CuI, Et_3_N, toluene, 70 °C, 8 h, 60–77%; (v): cyclocondensation with (A) acetamidine HCl, Na_2_CO_3_, acetonitrile, 8 h, reflux, or (B) guanidine HCl, Cs_2_CO_3_, acetonitrile, 8 h, reflux, or (C) benzamidine HCl or 4-nitrobenzamidine HCl, K_2_CO_3_, acetonitrile, 8 h, reflux; (vi): NaBH_4_, THF, 60 °C, 6 h, reflux, 60–80%; and (vii): 1M BBr_3_ in CHCl_3_, CHCl_3_, 3 h, room temperature, 50–90%.

**Figure 111 molecules-27-02863-f111:**
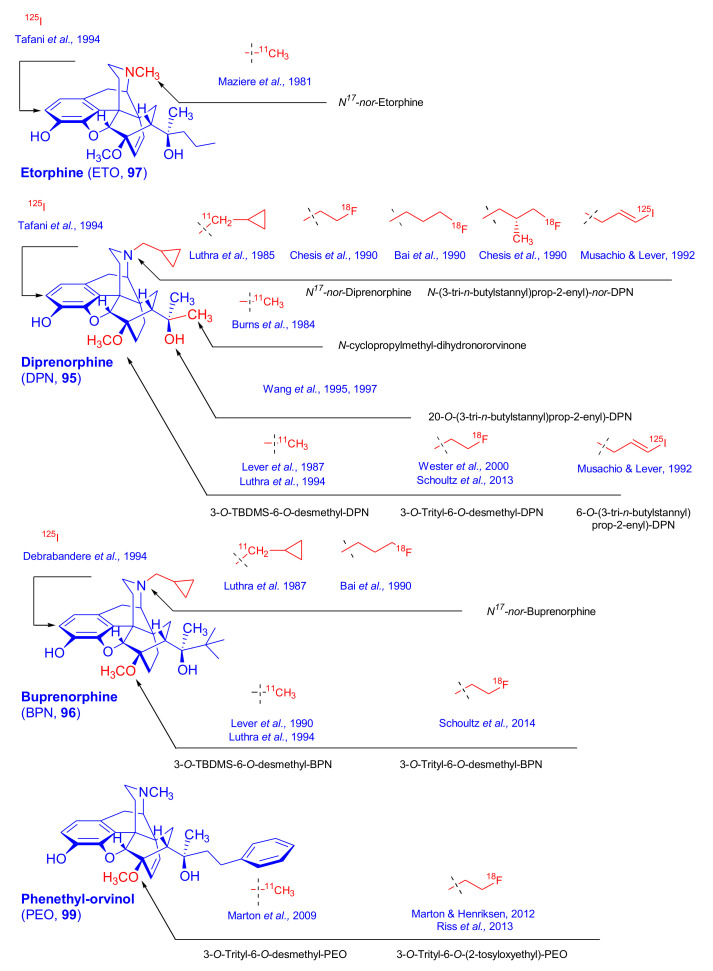
Strategies for the synthesis of radiolabeled orvinol derivatives.

**Figure 112 molecules-27-02863-f112:**
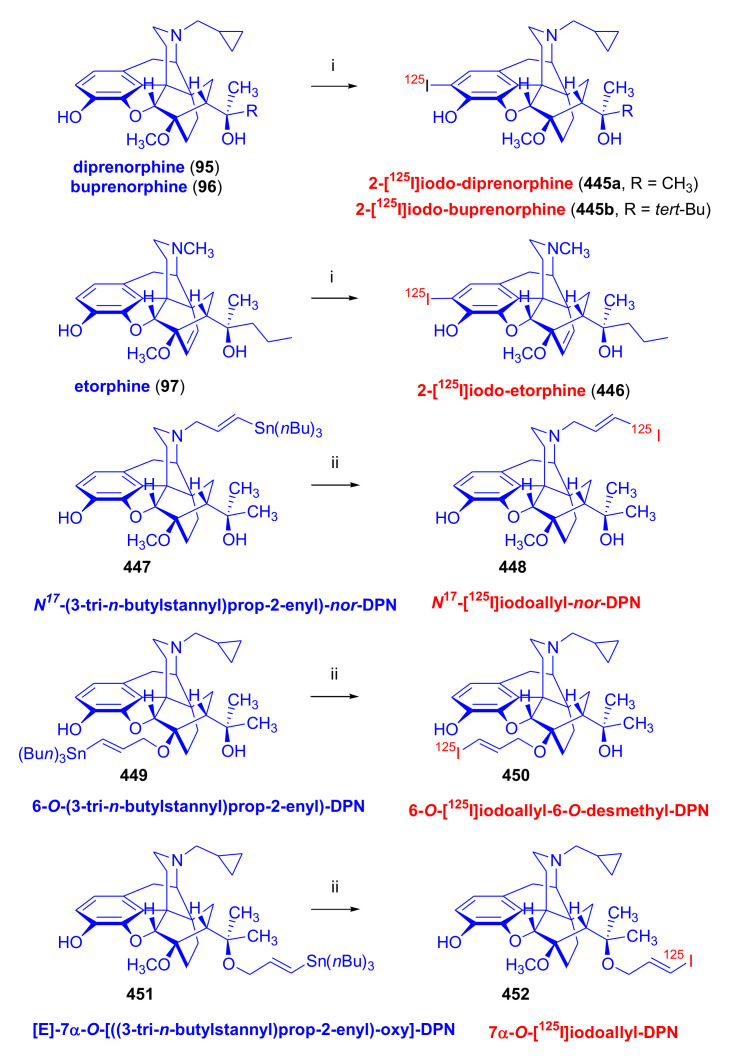
Synthesis of ^125^I-labeled orvinol derivatives. *Reagents and conditions*: (i) [^125^I]NaI, iodogen; (ii) [^125^I]NaI, chloramines-T, AcOH, MeOH, room temperature.

**Figure 113 molecules-27-02863-f113:**
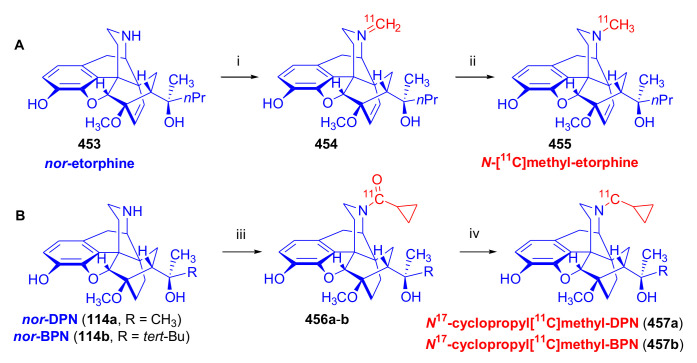
Synthesis of selected ^11^C-labeled orvinol derivatives. **A**. Synthesis of *N*-[^11^C]methyl-etorphine. **B**. Synthesis of *N*-cyclopropyl[^11^C]methyl-BPN and *N*-cyclopropyl[^11^C]methyl-BPN. *Reagents and conditions*: (i) H^11^CHO, AcOH, CH_3_CN, −5 °C; (ii): NaBH_3_CN, 70 °C, 5 min.; (iii): [1-carboxy-^11^C]cyclopropane carbonyl chloride, THF, −78 °C to RT, 2 min; and (iv): LiAlH_4_, THF, reflux, 5 min.

**Figure 114 molecules-27-02863-f114:**
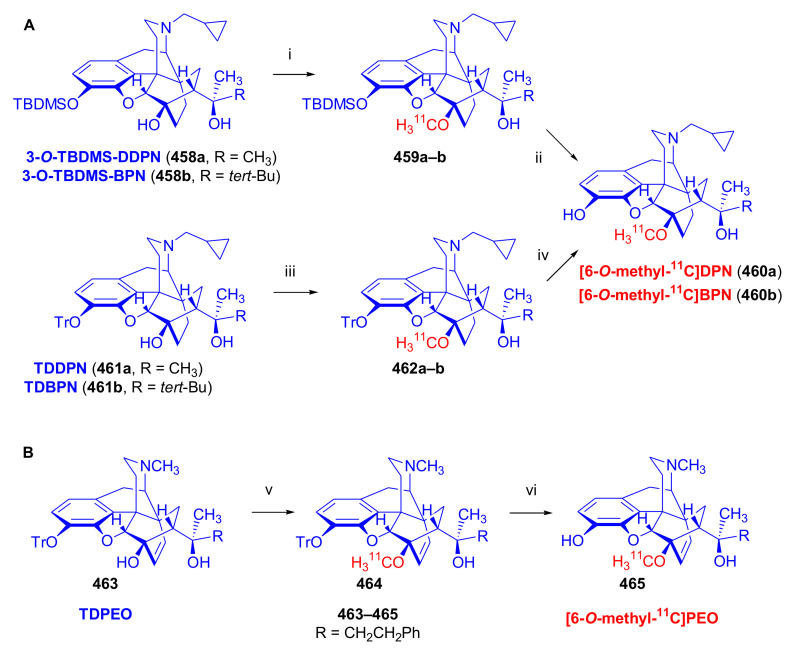
Synthesis of selected ^11^C-labeled orvinol derivatives. **A**. Synthesis of [6-*O*-methyl-^11^C]DPN and [6-*O*-methyl-^11^C]BPN. **B**. Synthesis of [6-*O*-methyl-^11^C]PEO. *Reagents and conditions*: (i): [^11^C]methyl iodide, NaH, DMF, 80 °C, 2 min; (ii): 1M HCl; (iii): [^11^C]methyl iodide, NaH, DMF, 95 °C, 5 min; (iv): 2M HCl, 95 °C, 2 min; (v): [^11^C]methyl iodide, NaH, DMF, 90 °C, 5 min; and (vi): 1M HCl, EtOH, 90 °C, 2 min.

**Figure 115 molecules-27-02863-f115:**
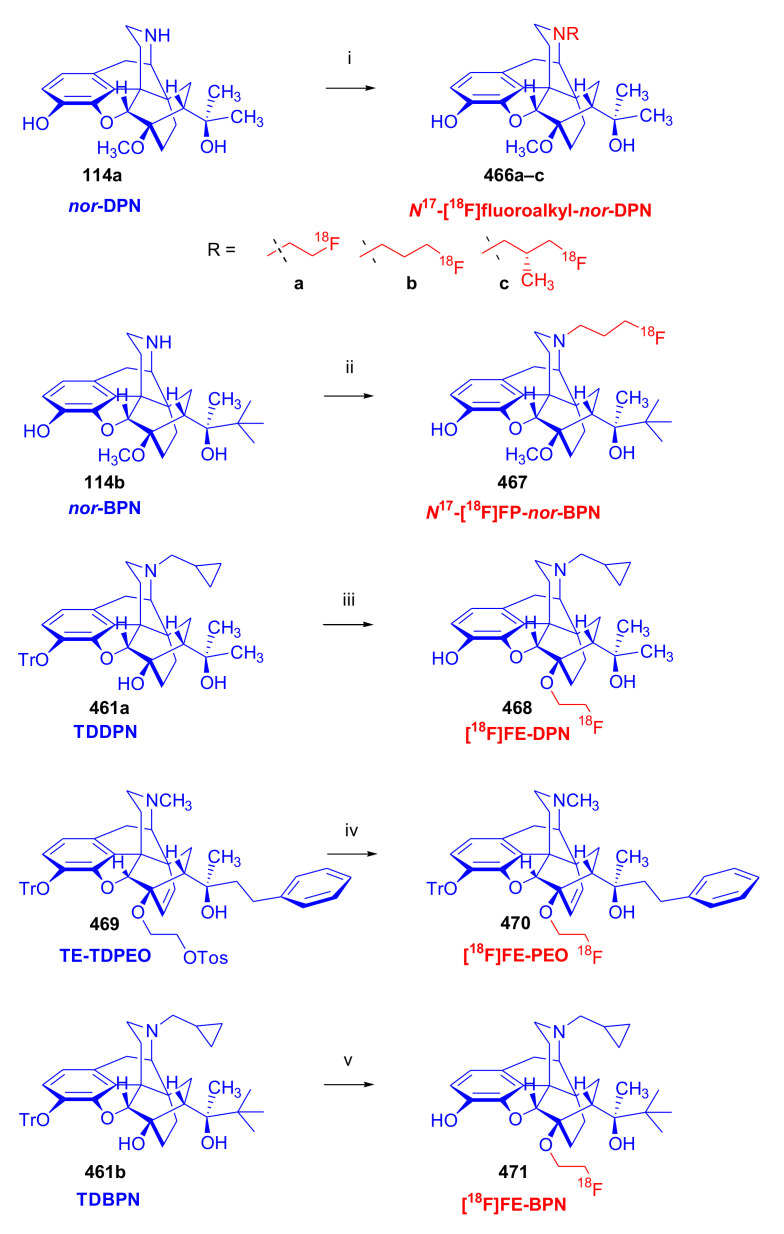
Synthesis of selected ^18^F-labeled orvinol derivatives. *Reagents and conditions*: (i): [^18^F]F(CH_2_)_n_I, NaH, 130 °C, 30 min; (ii): [^18^F]F(CH_2_)_3_I, NaHCO_3_, 110 °C, 20 min; (iii): (1) [^18^F]FEOTos, NaH, 100 °C, 5 min; (2) 2M HCl, 100 °C, 2 min.; (iv): (1) [K^+^ ⊂ K222]^18^F^−^, CH_3_CN, 90 °C, 10 min, (2) 1M HCl, in EtOH, 40 °C, 5 min; and (v): (1) [^18^F]FEOTos, 100 °C, 10 min, (2) 2M HCl, 40 °C, 5 min.

**Figure 116 molecules-27-02863-f116:**
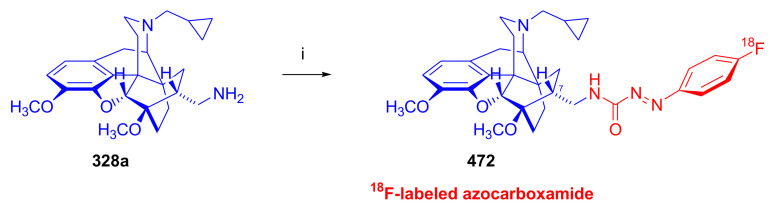
Synthesis of ^18^F-labeled buprenorphine-derived azocarboxamide. *Reagents and conditions*: (i) *tert*-butyl 4-[^18^F]fluoro-phenylazocarboxamide, CsCO_3_, EtOH, room temperature, 5 min.

**Table 1 molecules-27-02863-t001:** Definition of the substituents of morphinan-6,8-dienes depicted in Figure 2.

Comp.	Name	R_1_	R_2_	R_3_	R_4_	References
**4**	Thebaine	CH_3_	-	-	-	Berényi [38]
**5**	Oripavine	H	-	-	-	Hosztafi [39]
**9a**	*N*^17^-formyl-northebaine	CH_3_	CHO	-	-	Maat [40]
**9b**	*N*^17^-benzyl-northebaine	CH_3_	Bn	-	-	Fleischhacker [41]
**9c**	*N*^17^-benzoyl-northebaine	CH_3_	Bz	-	-	Maurer [42]
**9d**	*N*^17^-cyclopropylmethyl-northebaine	CH_3_	CPM	-	-	Lewis [43]
**9e**	*N*^17^-cyclopropylcarbonyl-northebaine	CH_3_	CPCO	-	-	Barton [44]
**9f**	*N*^17^-Boc-northebaine	CH_3_	Boc	-	-	Sandulenko [45]
**10a**	β-dihydrothebaine	H	CH_3_	OCH_3_	-	Bentley [46]
**10b**	4-*O*-acetyl-β-dihydrothebaine	Ac	CH_3_	OCH_3_	-	Hayakawa [47]
**10c**	4-*O*-phenyl-β-dihydrothebaine	Ph	CH_3_	OCH_3_	-	Ghosh [48]
**10d**	6-demethoxy-β-dihydrothebaine	H	CH_3_	H	-	Linders [49]
**10e**	4-*O*-acetyl-*N*^17^-formyl-6-demethoxy-β-dihydrothebaine	Ac	CHO	H	-	Linders [50]
**11a**	6-deoxythebaine	CH_3_	CH_3_	CH_3_	H	Knipmeyer [51]
**11b**	6-demethoxythebaine	CH_3_	CH_3_	H	H	Berényi [52]
**11c**	6-demethoxyoripavine	H	CH_3_	H	H	Berényi [53]
**11d**	*N*^17^-formyl-6-demethoxynorthebaine	CH_3_	CHO	H	H	Linders [54]
**11e**	6-fluoro-6-demethoxythebaine	CH_3_	CH_3_	F	H	Berényi [55]
**11f**	6-chloro-6-demethoxythebaine	CH_3_	CH_3_	Cl	H	Berényi [56]
**11g**	6-bromo-6-demethoxythebaine	CH_3_	CH_3_	Br	H	Berényi [56]
**11h**	6-thiocyanato-6-demethoxythebaine	CH_3_	CH_3_	SCN	H	Berényi [57]
**11i**	6-isothiocyanato-6-demethoxythebaine	CH_3_	CH_3_	NCS	H	Csutorás [58]
**11j**	6-azido-6-demethoxythebaine	CH_3_	CH_3_	N_3_	H	Csutorás [58]
**11k**	7-chloro-6-demethoxythebaine	CH_3_	CH_3_	H	Cl	Simon [59]
**11l**	7-bromo-6-demethoxythebaine	CH_3_	CH_3_	H	Br	Simon [59]
**11m**	7-thiocyanato-6-demethoxythebaine	CH_3_	CH_3_	H	SCN	Berényi [57]
**12a**	5β-methylthebaine	CH_3_	CH_3_	CH_3_	H	Boden [60]
**12b**	5β-ethylthebaine	CH_3_	CH_3_	Et	H	Woudenberg [61]
**12c**	5β-butylthebaine	CH_3_	CH_3_	Bu	H	Woudenberg [61]
**12d**	thebaine-5β-methanol	CH_3_	CH_3_	CH_2_OH	H	Woudenberg [62]
**12e**	10α-ethyl-5β-methylthebaine	CH_3_	CH_3_	CH_3_	Et	Woudenberg [61]
**12f**	5β,10α-diethylthebaine	CH_3_	CH_3_	Et	Et	Woudenberg [61]
**12g**	5β-trimethylsilylthebaine	CH_3_	CH_3_	SiMe_3_	H	Chen [63]
**13a**	5β-methyl-6-demethoxythebaine	CH_3_	H	-	-	Woudenberg [64]
**13b**	7-chloro-5β-methyl-6-demethoxythebaine	CH_3_	Cl	-	-	Woudenberg [65]
**13c**	7-methoxy-5β-methyl-6-demethoxythebaine	CH_3_	OCH_3_	-	-	Woudenberg [66]
**14a**	6-ethoxy-6-demethoxythebaine	CH_3_	CH_3_	EtO	-	Czakó [67]
**14b**	6-propoxy-6-demethoxythebaine	CH_3_	CH_3_	*n*PrO	-	Czakó [67]
**14c**	6-cyclopropylmethoxy-6-demethoxythebaine	CH_3_	CH_3_	CPMO	-	Czakó [67]
**14d**	6-phenyl-6-demethoxythebaine	CH_3_	CH_3_	Ph	-	Czakó [67]
**15a**	2′-aminothiazolo-[6,7:4′,5′]-morphinan-6,8-diene	NH_2_	-	-	-	Sipos [68]
**15b**	2′-methylaminothiazolo-[6,7:4′,5′]-morphinan-6,8-diene	CH_3_	-	-	-	Sipos [68]
**15c**	2′-phenylaminothiazolo-[6,7:4′,5′]-morphinan-6,8-diene	Ph	-	-	-	Sipos [68]

Abbreviations: Bn: benzyl group; Bz: benzoyl group; CPM: cyclopropylmethyl group; CPCO: cyclopropylcarbonyl group; Boc: *tert*-butoxycarbonyloxy group; Bu: *n*-butyl group; EtO: ethoxy group; *n*PrO: *n*-propoxy group; CPMO: cyclopropylmethoxy group; and Ph: phenyl group.

**Table 2 molecules-27-02863-t002:** Isomerization of 7-acyl-6,14-ethenomorphinans.

Starting 7-Acyl Compound	Pos.-7	X-Y	R_1_	R_2_	T [°C]	τ [h]	Product Ratio	
							7α	7β
**112a**	7α	CH_2_-CH_2_	CPM ^a^	CH_3_	70	16	67	33
**104**	7α	CH_2_-CH_2_	CH_3_	CH_3_	90	16	58	42
**172**	7α	CH_2_-CH_2_	CH_3_	*tert*-Bu ^b^	90	18	50	50
**383**	7α	CH_2_-CH_2_	CH_3_	Ph ^c^	70	4	56	44
**16a**	7α	CH=CH	CH_3_	CH_3_	70	4.5	67	17
**17a**	7β	CH=CH	CH_3_	CH_3_	70	4	83	17

^a^ CPM: cyclopropylmethyl group, ^b^
*tert*-Bu: *tert*-butly group, ^c^ Ph: phenyl group.

**Table 3 molecules-27-02863-t003:** Binding profile of selected 6,14-ethenomorphinan ligands at the human opioid receptors [22].

Ligand	Comp.	K_i_ [nM]				Action	References
		μ-OR	δ-OR	κ-OR	NOP		
morphine	**1**	2.06	>10,000	134	>10,000	agonist μ-OR	Valenzano, 2004 [271]
DPN	**95**	0.07	0.23	0.02	ND	antagonist	Raynor, 1994 [272]
BPN	**96**	1.5	6.1	2.5	77.4	partial μOR agonist, κOR antagonist, partial NOP agonist	Cami-Cobeci, 2011 [189]
5β-Me-BPN [a]	-	-	-	-	ND	inactive [a]	Aceto, 1984 [273]
PEO	**99**	0.18	5.1	0.12	ND	full agonist	Marton, 2009 [98]
FE-DPN	**468**	0.24	8.0	0.20	ND	antagonist	Schoultz, 2014 [274]
FE-BPN	**471**	0.24	2.1	0.12	ND	mixed agonist antagonist	Schoultz, 2014 [274]
FE-PEO	**470**	0.10	0.49	0.08	ND	full agonist	Schoultz, 2014 [274]
ETO	**97**	0.33	1.54	0.22	56.9	full agonist	Khroyan, 2011 [130]
ETO [b]	**97**	0.62 (4.26)			ND	agonist	Biyashev, 2002 [113]
7β-ETO [b]	**167a**	0.47 (11.5)			ND	agonist	Biyashev, 2002 [113]
5β-Me-ETO [c]	-	1.1	7.9	45.5 (κ_1_) 2670 (κ_2_) 12.7 (κ_3_)	ND	agonist	Maat, 1999 [275]
DHE	**98**	0.45	1.82	0.57	ND	agonist	Katsumata, 1995 [276]
DHE	**98**	0.10	1.5	0.74	120	full agonist	Olofsen, 2019 [126]
DHE [b]	**98**	0.78 (13.1)			ND	agonist	Biyashev, 2002 [113]
7β-DHE [b]	**167b**	1.61 (13.8)			ND	agonist	Biyashev, 2002 [113]
CYPR [d]	-	0.076	0.68	0.79	ND	antagonist	Smith, 1987 [277]
16-Me-CYPR [d]	-	1.77	0.73	59.6	ND	pure antagonist	Smith, 1987 [277]
thienorphine	**100**	0.22	0.69	0.14	ND	κOR agonist	Li, 2007 [278]
BU08028	**101**	2.14	1.59	5.63	8.46	mixed mOR/NOP agonist	Khroyan, 2011 [130]
BU127	**396g**	0.71	1.91	0.49	43.2	μOR and κOR antagonist, NOP agonist	Lewis, 2013 [279] Kumar, 2014 [114]
BU128	-	0.08	0.48	0.08	97	μOR and κOR antagonist, NOP agonist	Lewis, 2013 [278]
BU10112	-	0.17	0.40	0.04	79	μOR and κOR antagonist, NOP agonist	Lewis, 2013 [243]
BU10119	**102**	0.10	0.25	0.04	80	μOR and κOR antagonist, NOP agonist	Lewis, 2013 [279]
BU10120	**103**	0.16	0.47	0.05	34	μOR and κOR antagonist, NOP agonist	Lewis, 2013 [279]
SLL-020ACP [e]	**390a**	129.5	773.7	0.40	ND	selective κOR agonist	Li, 2017 [164]
“SMH-15a”	**414**	0.10	0.25	0.04	80	κOR antagonist	Cueva, 2015 [131]
SLL-039	**434**	321.3	133.7	0.47	ND	κOR agonist	Xiao, 2019 [266]
“LS-4j”	**435**	114	293	1.8	ND	agonist	Liu, 2021 [267]

[a] NIH 10,064, in vivo antinociceptive activity: hot plate, Nilsen, tail-flick, and inactive; [b] Wistar rat brain membrane preparations and [^3^H]naloxone, K_i_ [nM] without Na^+^ ions (100 mM Na^+^ present); [c] Hartley guinea pig membranes, [^3^H]DAMGO (μ), [^3^H]Cl-DPDPE (δ), [^3^H]U-69,593 (κ_1_), [^3^H]bremazocine (κ_2_), and [^3^H]NalbzOH (κ_3_); [d] K_e_ values in the mouse vas deferens; and [e] *N*^17^-cyclopropylmethyl-nornepenthone; ND: not determined.

**Table 4 molecules-27-02863-t004:** Radiosynthesis data of carbon-11 labeled orvinol derivatives.

Radioligand	No.	Decay-Corrected RCY [%]	Preparation Time [min.]	RCP [%]	Molar Activity (EOS) [GBq/μmol]	Reference
[*N*-methyl-^11^C]etorphine	**455**	12–16 *	n.d.	n.d.	0.03	[299] (1981)
[20-methyl-^11^C]thevinol	**-**	8 *	n.d.	n.d.	7.4	[300] (1984)
[*N*-cyclopropylmethyl-^11^C]DPN	**457a**	5–20	53	100	>1.7	[301] (1985)
[*N*-cyclopropylmethyl-^11^C]BPN	**457b**	20	57	100	6	[302] (1987)
[6-*O*-methyl-^11^C]DPN	**460a**	28	30	n.d.	64.4	[183] (1987)
[6-*O*-methyl-^11^C]DPN	**460a**	60–88	45	>99	15.5–23.8	[185] (1994)
[6-*O*-methyl-^11^C]DPN	**460a**	32	48	97	242.1	[306] (2014)
[6-*O*-methyl-^11^C]DPN	**460a**	22.3 ^a^	n.d.	93.5	370	[306] (2014)
[6-*O*-methyl-^11^C]DPN	**460a**	n.d.	40–50	>96	4–36	[308] (2000)
[6-*O*-methyl-^11^C]BPN	**460b**	23	24	97	41	[184] (1990)
[6-*O*-methyl-^11^C]BPN	**460b**	69–93	50	>99	12.8–21.2	[185] (1994)
[6-*O*-methyl-^11^C]BPN	**460b**	60	40	>97	7.6–20.9	[304] (1996)
[6-*O*-methyl-^11^C]PEO	**465**	41–73	n.d.	>99	60	[98] (2009)

*: Non decay corrected data; ^a^: via [^11^C]methyl triflate; n.d.: no data available.

**Table 5 molecules-27-02863-t005:** Radiosynthesis data of fluorine-18 labelled orvinol derivatives.

Radioligand	No.	Decay-Corrected RCY [%]	Preparation Time [min.]	RCP [%]	Molar Activity (EOS) [GBq/μmol]	Reference
*N*^17^-[^18^F]FE-*nor*-DPN	**466a**	0.3–0.6 *	n.d.	n.d.	4–10	[309] (1990)
*N*^17^-[^18^F]FP-*nor*-DPN	**466b**	4–6 *	n.d.	n.d.	31–67	[309] (1990)
*N*^17^-[^18^F]FP-*nor*-DPN	**466b**	15	100	n.d.	n.d.	[310] (1990)
*N*^17^-((*S*)-3-[^18^F]fluoro-2-methylpropyl)-*nor*-DPN	**466c**	4–6 *	n.d.	n.d.	31–67	[309] (1990)
[^18^F]FE-DPN	**468**	22	100	98	37	[308] (2000)
[^18^F]FE-DPN	**468**	18–34	100	n.d.	50–300	[274] (2014)
[^18^F]FE-PEO	**470**	35	42	>99	55–128	[177] (2012)
[^18^F]FE-PEO	**470**	28	90	>97	52–224	[314] (2013)
[^18^F]FE-PEO	**470**	18–34	100	n.d.	50–300	[274] (2014)
[^18^F]FE-BPN	**471**	18–34	100	n.d.	50–300	[274] (2014)

*: Non decay corrected data; n.d.: no data available.

## Data Availability

Not applicable.

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
