# Peer review of "Diels–Alder Adducts of Morphinan-6,8-Dienes and Their Transformations†"

_molecules, 2022, doi:10.3390/molecules27092863_

Round 1

Reviewer 1 Report

This review is a tour de force on morphinan-6,8-dienes, their Diels-alder chemistry, and subsequent synthetic manipulations of those D.A. adducts.  The amount of work that has gone into authoring this is just mind boggling.  I think it is organized well using morphinan-6,8-diene core structure type as the principal organizing component.  I think chemists who are interested in Diels-Alder chemistry and those interested in opioid receptor ligand synthesis will be interested in this paper and I recommend publication.  I did not read the radiochemistry section (3) since it is outside my area of expertise but I did find a number of typos in other sections listed below.

line

22  certain

30 reactions of

line 74 showing it to be

line 79 referred to as

line 94 importance for structure

lines 131-133 I think don't belong in this manuscript?

line 140 enol ether is 2 words

lines 150-155 refer to figure 4 here?

line 174 cut one this

line 175 leading to some confusion

line 203 correctly have

line 221 esters

line 289 improved

line 297 dienophiles

line 328 agonists

line 391 structure 48 is missing from Figure 16

line 567 examples

line 769 I think this should be bold font?

line 1996 thebaine

line 3308 may seem

line 3309 provided

Author Response

see attached file please

Reviewer 2 Report

The manuscript, "Diels-Alder Adducts of Morphinan-6,8-Dienes and their Transformations” from J. Marton and co-workers I very well written with highly important up to date content. I strongly recommend this manuscript for publication, however, after minor revision:

Line 55: The first reports….. – Please, revise this sentence.

Line 129: The structure of selected …. – please, revise this sentence.

Line 140: Diels-Alder (DA) reactions – further in the text Diels-Alder and DA is used alternately. Please, use only one form in the whole manuscript.

Line 329: in vitro and in vivo should be written in italics. Please check it through the manuscript.

Line 330: In GPI assay, 6-Halogenated-20-methylorvinols displayed 17-30 times more increased activity than in comparison to normorphine. according to the GPI investigations.

Line 552: Hromartka…… should be Hromatka

Line 784: The reference for Ki values should be added. Ki values for mu and kappa receptors are missing, the value for delta receptor is slightly different from that given in the Table 3.

Line 1004 and 1005: Please, add a short description of the results obtained for 20R- butylorvinol analogues in in vivo tests and add appropriate reference.

Line 1053: The diprenorphine furanomorphide …….  Please, add appropriate reference showing the described results.

Line 1085: Binding affinities of the synthesized ligands….. Please, add appropriate reference showing the described results.

Line 1123: The analgesic activities….  – please, describe which tests were used to examine the antinociceptive effect of compounds and add appropriate references.

Lines 1606 – 1610: Pharmacological characterization…. with activity less than 2% that of naloxone. - Please, add appropriate references showing the described results.

Line 1676: Pharmacological screening in mice….. Please, add appropriate reference showing the described results.

Line 1702: Agonist potencies… Please, add appropriate reference showing the described results.

Line 1904: Vaccination/immunization present 1904 a promising alternate approach for treating substance dependence - Please, add appropriate reference describing this approach.

Line 2096: Moiseev et al. - Please, add appropriate reference.

Lines 2233 – 2238: inappropriate reference is given.

Lines 2295 – 2302: please, add an appropriate reference in which the results of the mentioned biological test are given.

Lines 2306 – 2310: The synthesized compounds were tested in rat vas deferens preparation for inhibition of the electrically-stimulated RVD contractions. One of the prepared compounds…… please, add an appropriate reference in which the results of the mentioned biological test are given.

Lines 2331 – 2335: Although 17-Cylopropylmethyl-4,5α-epoxy-6β,21-epoxymethano-3-hydroxy-6,14-endo 2332 ethenomorphinan-7α-(N-phenethyl)carbox amide (335, TAN-1014, Figure 85) possess no agonistic effect on the rat vas deferens test……         please, add an appropriate reference in which the results of the mentioned biological test are given.

Lines  2354 – 2360: The oripavine derivatives (339a-f) have subnanomolar affinity (Ki = 0.14 – 0.25) for μ-ORs and also affinity in the nanomolar……..  please, add an appropriate reference in which the results of the mentioned biological test are given.

Lines 2376 – 2382: Biochemical characterization of the conformationally….. please add an appropriate reference in which the results of the mentioned biological test are given.

Lines 2382 – 2384: Compared to the 7α-(cinnamoyl)amionomethyl derivatives (339b, 339d), the constrained analogues (342a-b) had considerably higher efficacy as demonstrated by an analgesia test. - Please, add a short description of the results of mentioned tests and add appropriate reference.

Lines 2394 – 2400: Both compounds (343a-b) showed high affinity for all three classical OR subtypes; the Ki values…… please, add an appropriate reference in which the results of the mentioned biological test are given.

Lines 2438 – 2442 - The analgesic activity of the target 6,14-ethenomorphinan…… please, add an appropriate reference in which the results of the mentioned biological test are given.

Lines 2481 – 2488: Competitive binding against subtype selective radioligands… please, add an appropriate reference in which the results of the mentioned biological test are given.

Lines 2498 – 2504: The highest binding affinities (kind of assays) were found for….. please, add an appropriate reference for this section.

Lines 2509 – 2514: On the basis of radioligand binding studies with human……. please, add an appropriate reference at the end of this section.

Lines 2720 – 2729: The prepared ligands were evaluated in competitive binding assays….. please, add an appropriate references in which the results of the mentioned biological tests are given.

Lines 2771 – 2785: please, add an appropriate references for this section.

Lines 2813 – 2821: please, add an appropriate references for this section.

Lines 2906 – 2915: please, add an appropriate references for this section.

 Lines 2934 – 2936 - The lead compound of 2934 the series was the highly selective and potent…. Please, add an appropriate reference at the end of this sentence.

Lines 2961 – 2968: . Binding affinities….. please, add appropriate references for this section.

Lines 3005 – 3011: The analgesic activity of selected compounds…  ….. please, add appropriate references at the end of this section.

Lines 3066 – 3069: In another report, the 6-O-[ 123I]iodoallyl-6-O-desmethyl- diprenorphine ([123I]-O-IA-DPN, 450) was utilized in ex vivo autoradiographic studies, where the tracer uptake is measured post mortem. - please, add appropriate reference at the end of this sentence.

Line 3150: Please, add an appropriate reference after Luthra et al.

Lines 3213 – 3221: The binding properties of N17-(3-[ 18F]fluoropropyl)-nor-buprenorphine and N17-(3-[ 18F]fluoropropyl)-diprenorphine were investigated for ORs by means of ex vivo distribution in rodent brain and PET studies in baboons……  Please, add proper references.

Lines 3268 – 3276 – The in vitro and in vivo characteristics of the………. Please, revise this section and add appropriate references.

Author Response

see attached file please
